# A Semi-Supervised Clustering Approach For Graph Learning with Neural Networks

## Abstract

We propose a semi-supervised approach that combines any unsupervised clustering objective and supervised objective for end-to-end training any neural networks to improve node classification in attributed graphs, particularly when training labels are sparse. Our framework formulates node classification as semi-supervised inference of neural network models of attributed graphs with cluster structure. We use this framework to understand how neural networks for graph clustering can jointly cluster node attributes and graph structure, despite graph clustering objectives explicitly considering only graph structure and cluster assignments. Our framework also enables neural network architectures such as transformers and multilayer perceptrons to learn on graphs without positional encodings and without spectral or message passing layers found in graph neural networks. We evaluate our framework on six real-world attributed graph datasets.

## 1 Introduction

In graph learning, a common task is to identify groups of similar nodes, known as *classes* or *clusters* (Wu et al., 2022; Liu et al., 2023; Daneshfar et al., 2024). In a social network, for example, where connections between people are represented as edges in a graph, node classes or clusters can represent groups of people that are members of the same community (Bedi & Sharma, 2016). Graphs can also have attributes associated with each node, such as the age and dietary preferences of each person in a social network. Many different notions of communities in the same social network can be considered meaningful, and how the both attributes of individuals and their connections in the network give rise to these communities can be complex. Analogously, many different clusterings of the same attributed graph can be considered meaningful, and how both the attributes of nodes and their adjacencies give rise to these clusterings can be complex. Graph neural networks (GNNs) (Wu et al., 2022; Corso et al., 2024) are artificial neural networks designed to learn complex functions of node attributes and adjacencies. Where sufficient training labels are available, GNNs can be trained supervised and *end-to-end* for node classification (Kipf & Welling, 2016a). When no training labels are available, GNNs have recently been adapted for end-to-end unsupervised clustering (Shchur & Günnemann, 2019; Tsitsulin et al., 2023; Blöcker et al., 2024). However, unsupervised clustering may yield cluster assignments that do not align with the node labels of interest in real-world graphs. While it can be prohibitively expensive to obtain sufficient training labels for supervised node classification, it can be practical to label just a handful of examples for each label. In this work we propose a semi-supervised approach that combines unsupervised clustering objectives and supervised objectives for training neural networks to improve node classification in attributed graphs, particularly when training labels are sparse.

### 1.1 Research Gaps and Contributions

We summarize the following research gaps we identify and our contributions to each.

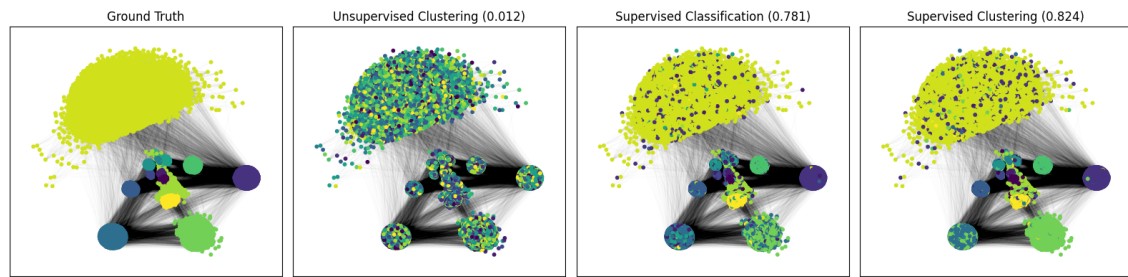

Figure 1: Visualization of node labellings comparing an unsupervised clustering, a semi-supervised node classification, and a semi-supervised clustering of the Coauthor CS dataset (Shchur et al., 2018), selected among 10 runs of random splits of 2 training nodes per cluster, 50 validation nodes, and 1000 test nodes. The unsupervised clustering labelling was selected from the run and configuration with the lowest training loss. The supervised classification and supervised clustering labellings were selected from the runs that yielded the highest validation set Matthews correlation coefficient (MCC). MCCs for each node labelling is displayed in parentheses.

*Research Gap 1:* While self-supervised deep graph representation learning methods (Veličković et al., 2019; Hassani & Khasahmadi, 2020; Zhu et al., 2020; Devvrit et al., 2022; Thakoor et al., 2022) have enabled deep graph clustering (Wang et al., 2023; Liu et al., 2023) with a two-step approach, Tsitsulin et al. (2023) recently identified a research gap in graph neural networks for *end-to-end* unsupervised graph clustering. They proposed a method to bridge the gap between traditional graph clustering objectives and graph neural networks, upon which other methods (Blöcker et al., 2024; Hansen & Bianchi, 2023) have been proposed. We identify a similar research gap, also identified by a recent review (Daneshfar et al., 2024), in graph neural networks for end-to-end *semi-supervised* graph clustering.

**Contribution 1:** We propose a reformulation of semi-supervised node classification on attributed graphs in the very sparse label setting as semi-supervised attributed graph clustering. Our experiments show that training with semi-supervised objectives consistently outperform a purely supervised objective. Figure 1 visualizes our approach on a real-world attributed graph dataset.

*Research Gap 2:* In unsupervised clustering on real-world attributed graphs, the loss landscape of clusters can be complex with many candidate solutions that can be considered meaningful (Peixoto, 2021; Blöcker & Scholtes, 2024).

**Contribution 2:** We propose a semi-supervised framework for evaluating existing end-to-end unsupervised deep graph clustering methods such as NOCD (Shchur & Günnemann, 2019), DMoN (Tsitsulin et al., 2023), Neuromap (Blöcker et al., 2024), and contribute an initial evaluation of these methods on six real-world attributed graph datasets. We additionally contribute a fully neural stochastic block model which we evaluate against.

*Research Gap 3:* We identify a contradiction where attributed graph clustering methods such as NOCD (Shchur & Günnemann, 2019), DMoN (Tsitsulin et al., 2023), Neuromap (Blöcker et al., 2024) optimize objectives that are functions of cluster assignments and graph structure, but not of node attributes. Constructing Bayesian network representations of their conditional dependencies, we identify a further contradiction that the explicit or *implicit* generative models that these methods infer are only generative models of graph structure and not of node attributes. On the other hand, we observe that MLPs have been shown effectively optimize unsupervised graph clustering objectives end-to-end (Shchur & Günnemann, 2019; Blöcker et al., 2024).

**Contribution 3:** We propose an alternative answer to the question posed in Shchur & Günnemann (2019): "Do we really need a graph neural network?" by proposing two simple ways existing methods can be adapted to learn explicit or *implicit* generative models both of graph structure and node attributes. If graph neural networks are used to optimize graph clustering objectives, an additional neural network module can be integrated to generate node attributes form cluster assignments. Alternatively, optimizing graph clustering objectives with neural networks of *only* node attributes learns a generative model akin to the *neural-prior stochastic block model* (Duranthon & Zdeborová, 2023b), a recently proposed generative model of graphs with cluster structure from node attributes.

*Research Gap 4:* Adapting other neural networks such as multilayer perceptrons and transformers (Vaswani et al., 2017) typically involve incorporating a combination of the spectral and/or message passing layers found in graph neural networks (Chen et al., 2021; Rampasek et al., 2022; Qarkaxhija et al., 2024; Buterez et al., 2024) and/or additional encodings informative of graph structure (Ying et al., 2021; Kim et al., 2022; Ma et al., 2023; Huang et al., 2024).

**Contribution 4:** We propose an alternative approach for neural networks such as transformers and multi-layer perceptrons to learn on attributed graphs without positional or structural encodings and without spectral or message passing layers found in graph neural networks. We evaluate this ability of the framework on real-world attributed graph datasets and demonstrate that it improves learning on architectures such as transformers and MLPs.

## 2 SEMI-SUPERVISED DEEP GRAPH CLUSTERING END-TO-END

**Setting:** Consider a node-attributed graph $G := (V, E)$ of $n$ nodes with node set $V$ and edge set $E$. Nodes have attributes captured in a node attribute matrix $\boldsymbol{X}$ and edges (equivalently, node adjacencies) are represented by an adjacency matrix $\boldsymbol{A}$. Each node is additionally assigned to one or multiple clusters from a set $S$ of cardinality $d_S$ of which $(d_S)_{\text{obs}} \leq d_S$ are observed. The node cluster assignments are represented with a label matrix $\boldsymbol{S}$ of shape $n \times d_S$. Nodes and clusters are indexed with zero-based integers. Without loss of generality it is assumed in this work that the labels of the first $s$ nodes belonging to the first $(d_S)_{\text{obs}}$ clusters are observed, which can be used for training. $\boldsymbol{S}$ is defined in generality to allow for various forms of cluster assignments, such as hard or soft assignments (Yu et al., 2005), and overlapping or non-overlapping clusters (Shchur & Günnemann, 2019). The primary learning task of interest is to infer the node label matrix $\boldsymbol{S}$. We primarily consider the setting of sparse training labels, i.e. $s \ll n$, though our proposed framework proposed does not rely on this assumption.

The formulations in this section primarily consider unweighted and undirected graphs without edge and graph attributes for both brevity and to match the setting of the experiments on real-world attributed graph datasets that are also unweighted and undirected. In appendix B.1 we discuss extensions to weighted and directed graphs with edge and graph attributes.

### 2.1 GRAPH NEURAL NETWORKS WITHOUT *Attributed* GRAPH GENERATIVE MODELS

Our first contribution is to identify the generative models that graph neural networks for semi-supervised node classification and unsupervised graph clustering infer. By constructing their Bayesian network representations, we show how these graph neural networks should not learn generative models of (node-)*attributed* graphs.

#### 2.1.1 EXPLICIT AND *Implicit* GENERATIVE MODELS

Graph clustering functions such as NOCD (Shchur & Günnemann, 2019), DMoN (Tsitsulin et al., 2023), Neuromap (Blöcker et al., 2024) and graph pooling functions such as MinCutPool (Bianchi et al., 2020) and

DiffPool (Ying et al., 2018) optimize objectives that can generally be expressed

$$l_{\text{clustering}} \approx c\left(\widehat{\boldsymbol{S}}, \boldsymbol{A}\right) \tag{1}$$

where $l_{\text{clustering}}$ is the resulting training loss and $c$ is the clustering objective function. For the clustering objective $c\left(\widehat{\boldsymbol{S}}, \boldsymbol{A}\right)$ of NOCD, DMoN, Neuromap, and similar methods, we propose that there exists a function $\widehat{\boldsymbol{A}} \sim f\left(\widehat{\boldsymbol{S}}\right)$ that enables the existence a graph reconstruction objective $L_E\left(\widehat{\boldsymbol{A}}, \boldsymbol{A}\right)$ such that

$$\arg\min_{\widehat{\boldsymbol{A}}}\left(L_E\left(f\left(\widehat{\boldsymbol{S}}\right), \boldsymbol{A}\right)\right) \approx \arg\min_{\widehat{\boldsymbol{S}}}\left(c\left(\widehat{\boldsymbol{S}}, \boldsymbol{A}\right)\right) \tag{2}$$

We refer to this function $\widehat{\boldsymbol{A}} \sim f\left(\widehat{\boldsymbol{S}}\right)$ as the explicit or *implicit* generative model of the graph structure of the clustering objective. For DiffPool and NOCD, the generative model is explicit (see background section A). For DMoN, Neuromap, and similar methods, this generative model is *implicit*. The implicit generative models of modularity maximization and Infomap optimizing the map equation (Smiljanić et al., 2023) – the discrete combinatorial optimization objectives relaxed by DMoN and Neuromap, respectively – and their equivalence to stochastic block models have been shown by Peixoto & Kirkley (2022). We hypothesise that this can be extended to the graph clustering functions of DMoN, Neuromap, and similar methods, and empirically evaluate this in our experiments.

### 2.1.2 BAYESIAN NETWORKS REVEALING GENERATIVE MODELS

Using this interpretation of generative models of graph clustering functions, we construct and analyse Bayesian network representations of graph neural networks for (semi-)supervised node classification and unsupervised graph clustering in figure 2. This reveals the generative models inferred in each case:

- Graph neural networks $\text{GNN}_{(\boldsymbol{X},\boldsymbol{A})\to\widehat{\boldsymbol{S}}}$ for (semi-)supervised node classification learn generative models of labels *from* node attributes and graph structure.
- Graph neural networks for unsupervised graph clustering $\text{GAE}_\text{o}$ use $\text{GNN}_{(\boldsymbol{X},\boldsymbol{A})\to\widehat{\boldsymbol{S}}}$ encoders to infer generative models $\widehat{\boldsymbol{A}} \sim f\left(\widehat{\boldsymbol{S}}\right)$ of graph structure from cluster assignments, but *not* of node attributes.

This also reveals an equivalence between graph neural networks for unsupervised graph clustering and graph autoencoders (Kipf & Welling, 2016b).

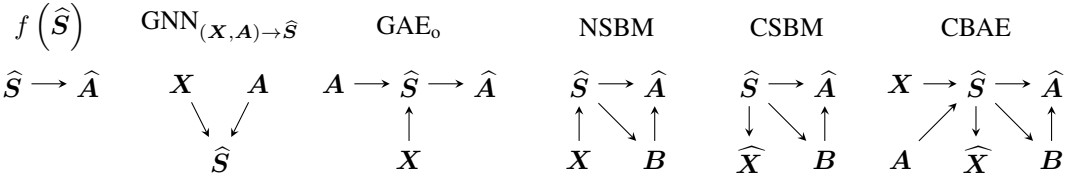

Figure 2: Bayesian network representations of a graph neural network for (semi-)supervised node classification ($\text{GNN}_{(\boldsymbol{X},\boldsymbol{A})\to\widehat{\boldsymbol{S}}}$), a class of graph neural network autoencoders for unsupervised graph clustering ($\text{GAE}_\text{o}$) that only optimize graph reconstruction objectives or unsupervised clustering objectives, and our proposed NSBM, CSBM, and CBAE architectures.

## 2.2 CONDITIONAL DEPENDENCIES OF ATTRIBUTED GRAPH GENERATIVE MODELS WITH CLUSTER STRUCTURE

To construct models that learn generative models of *attributed* graphs with cluster structure, we make the contribution of identifying the following requirements of node-attributed generative models of graphs with cluster structure and their representations as Bayesian networks:

- $\widehat{A}$ depends on $\widehat{S}$, represented by directed edge from $\widehat{S}$ to $\widehat{A}$ in a Bayesian network.
- Either $\widehat{X}$ depends on $\widehat{S}$ or $\widehat{S}$ depends on $X$, represented by the existence of a directed edge between them in a Bayesian network.
- Conditional independence between $X$ (or $\widehat{X}$) and $\widehat{A}$ given $\widehat{S}$.

Requiring conditional independence between $X$ (or $\widehat{X}$) and $\widehat{A}$ given $\widehat{S}$ ensures that the cluster structure of the attributed graph contains all the necessary information to infer both node attributes and graph structure.

We identify two generative models of attributed graphs with cluster structure that satisfy the conditions proposed above – the neural-prior stochastic block model (Duranthon & Zdeborová, 2023b) and the contextual stochastic block model (Deshpande et al., 2018) – as evidenced by their Bayesian network representations in figure 3 in appendix A. In the following sections, we propose fully neural network variants of these models, the NSBM and the CSBM, respectively.

### 2.3 TRANSFORMERS AND MLPS LEARN GRAPHS WITH NEURAL(-PRIOR) SBMS

The first fully end-to-end neural network generative model of attributed graphs with cluster structure that we propose is the NSBM, constructed as follows:

$$\widehat{A} \sim \text{NSBM}\left(X\right) = \text{SBM}_f\left(\widehat{S}\right) \tag{3}$$

where

$$\widehat{S} \coloneqq \text{NN}_{X \to \widehat{S}}\left(X\right)$$

As we constructed in section 2.1.1, inspired by Peixoto & Kirkley (2022), we propose that $\text{SBM}_f = f$ to be any explicit or *implicit* stochastic block model of $\widehat{S}$, such that $L_{\text{E}}\left(\widehat{A}, A\right)$ can represent the graph clustering functions of DMoN (Ying et al., 2018), MinCutPool (Bianchi et al., 2020), DMoN (Tsitsulin et al., 2023), Neuromap (Blöcker et al., 2024), and similar approaches.

We also contribute a construction of a fully neural stochastic block model $\text{SBM}_{\text{NN}}$, constructed as follows:

$$\widehat{A} \sim \text{SBM}_{\text{NN}}\left(\widehat{S}\right) = \text{SBM}_{\text{B}}\left(\widehat{S}, B\right) \tag{4}$$

where $\text{SBM}_{\text{B}}$ is a Bernoulli SBM (see equation 12, appendix A) and

$$B \coloneqq \phi\left(Z^T Z\right)$$
$$Z \coloneqq \text{NN}_{\widehat{S} \to (Z)}\left(\widehat{S}\right)$$

$\phi$ is any function that ensures that the values of $B$ are valid probabilities. Its Bayesian network representation is shown in figure 2. Learning a block matrix $B$ can enable more diverse cluster structures (Peixoto, 2019;

Pei et al., 2019; Scholkemper & Schaub, 2023) to align with observed node labels. The neural networks denoted NN can be any neural network. In particular, $\text{NN}_{\boldsymbol{X}\to(\widehat{\boldsymbol{S}},\boldsymbol{B})}$ in the NSBM can be any neural network function of node attributes. We focus on the cases where $\text{NN}_{\boldsymbol{X}\to(\widehat{\boldsymbol{S}},\boldsymbol{B})}$ is a transformer (Vaswani et al., 2017) or a multilayer perceptron (MLP).

### 2.3.1 INFERRING ATTRIBUTED STOCHASTIC BLOCK MODELS BY TRAINING NEURAL NETWORKS END-TO-END

Using the NSBM, we contribute a framework for neural networks such as transformers and MLPs to learn on attributed graphs without positional or structural encodings (Ying et al., 2021; Kim et al., 2022; Ma et al., 2023; Huang et al., 2024) and without spectral or message passing layers found in graph neural networks (Chen et al., 2021; Rampasek et al., 2022; Qarkaxhija et al., 2024; Buterez et al., 2024).

Where $\text{NN}_{\boldsymbol{X}\to\widehat{\boldsymbol{S}}}(\boldsymbol{X})$ in equation 3 is a transformer or MLP, the NSBM can be trained end-to-end by optimizing reconstruction losses of its predictions of $\boldsymbol{S}$ and $\boldsymbol{A}$ from their observed values.

$$l_{\text{NSBM}} \coloneqq L_S\left(\widehat{\boldsymbol{S}}_{0:s,:(d_S)_{\text{obs}}}, \boldsymbol{S}_{0:s,:(d_S)_{\text{obs}}}\right) + L_E\left(\widehat{\boldsymbol{A}}, \boldsymbol{A}\right) + L_{\text{regularization}} \tag{5}$$

$L_{\text{regularization}}$ is an optional regularization term such as in DiffPool, MinCutPool, and DMoN (see appendix A). As we constructed in section 2.1.1 $L_E\left(\widehat{\boldsymbol{A}}, \boldsymbol{A}\right)$ can be replaced by the clustering objectives of DMoN, Neuromap, and similar methods. As $\widehat{\boldsymbol{A}}$ is reconstructed from $\widehat{\boldsymbol{S}}$, we can choose $\widehat{\boldsymbol{S}}$ to be of size $n \times (d_S)_{\text{max}}$ where $(d_S)_{\text{max}} > (d_S)_{\text{obs}}$ particularly when $(d_S)_{\text{obs}}$ is small to enable a less lossy reconstruction of $\boldsymbol{A}$, and to allow for unseen clusters to be inferred (see appendix B.2.1 and A).

Using a $\text{SBM}_f$ function for the NSBM such as such as DiffPool, NOCD, MinCutPool, DMoN, Neuromap, or $\text{SBM}_{\text{NN}}$, a transformer or MLP-based NSBM makes *no changes* to the architecture of a transformer or MLP, respectively, apart from its training loss, while enabling it to learn on attributed graphs without positional or structural encodings, spectral or message passing layers.

### 2.4 GRAPH NEURAL NETWORKS WITH ATTRIBUTE RECONSTRUCTION LEARN CONTEXTUAL SBMs

The second fully end-to-end neural network generative model of attributed graphs with cluster structure that we propose is the CSBM, constructed as follows:

$$\widehat{\boldsymbol{X}}, \widehat{\boldsymbol{A}} \sim \text{CSBM}\left(\widehat{\boldsymbol{S}}\right) \coloneqq \text{NN}_{\widehat{\boldsymbol{S}}\to\widehat{\boldsymbol{X}}}\left(\widehat{\boldsymbol{S}}\right), \text{SBM}_f\left(\widehat{\boldsymbol{S}}\right) \tag{6}$$

Unlike NSBM, CSBM has $\widehat{\boldsymbol{S}}$ as an independent variable and hence requires a graph neural network encoder. We therefore construct the following CBAE autoencoder which infers the CSBM.

$$\widehat{\boldsymbol{X}}, \widehat{\boldsymbol{A}} \sim \text{CBAE} \coloneqq \text{CSBM}\left(\text{GNN}_{(\boldsymbol{X},\boldsymbol{A})\to\widehat{\boldsymbol{S}}}(\boldsymbol{X},\boldsymbol{A})\right) \tag{7}$$

Its Bayesian network representation is shown in figure 2.

### 2.4.1 RECONSTRUCTING ATTRIBUTES LEARNS GENERATIVE MODELS OF ATTRIBUTED GRAPHS

Like the NSBM, the CBAE is also trained end-to-end by optimizing the reconstruction losses of its predictions.

$$l_{\text{CBAE}} \coloneqq L_E\left(\widehat{\boldsymbol{A}}, \boldsymbol{A}\right) + L_V\left(\widehat{\boldsymbol{X}}, \boldsymbol{X}\right) + L_{\text{regularization}} \tag{8}$$

Depending on the attributes observed, the attribute reconstruction objective $L_V$ can a combination of regression objectives such as mean squared error or mean absolute error, and classification objectives such as cross-entropy. As we identified in section 2.1.1, without the attribute reconstruction we propose for the CBAE, graph neural networks for unsupervised graph clustering should learn generative models only of graph structure. This highlights the importance of attribute reconstruction for learning on graphs with attributes with graph neural networks.

## 2.5 Reconstructing Semi-Supervised Node Classification

By reinterpreting the reconstruction objectives of the NSBM (equation 5) and CBAE (equation 8) as semi-supervised clustering objectives, we contribute a framework where any neural network or graph neural network can utilize any unsupervised clustering objective, reconstruction objective, or self-supervised objective to learn on attributed graphs with cluster structure in a semi-supervised manner.

$$l_{\text{NSBM}} := L_{\text{supervised}} \left( \widehat{\boldsymbol{S}}_{0:s,:(d_S)_{\text{obs}}}, \boldsymbol{S}_{0:s,:(d_S)_{\text{obs}}} \right) + (L_{\text{unsupervised}})_{\text{NSBM}} \tag{9}$$

$$l_{\text{CBAE}} := L_{\text{supervised}} \left( \widehat{\boldsymbol{S}}_{0:s,:(d_S)_{\text{obs}}}, \boldsymbol{S}_{0:s,:(d_S)_{\text{obs}}} \right) + (L_{\text{unsupervised}})_{\text{CBAE}} \tag{10}$$

where $L_S = L_{\text{supervised}}$ is a supervised classification objective such as cross-entropy (see appendix A), and

$$(L_{\text{unsupervised}})_{\text{NSBM}} = L_E + L_{\text{regularization}}$$
$$(L_{\text{unsupervised}})_{\text{CBAE}} = L_E + L_V + L_{\text{regularization}}$$

are unsupervised objectives. As we constructed in section 2.1.1, $L_E$ can be replaced with any unsupervised clustering objective such NOCD, DMoN, Neuromap, and similar methods, and $L_{\text{regularization}}$ is an optional regularization term such as in DiffPool, MinCutPool, and DMoN (see appendix A and E.4.1).

While we formulate semi-supervised node classification and semi-supervised graph clustering equivalently in setting of our proposed methodology, we propose two distinctions between the two: by the methods used to solve the problem, and by the sparsity of the labels. For semi-supervised node classification training losses are purely supervised while for semi-supervised graph clustering training loss combine supervised and unsupervised objectives. In our experiments (section 3), we explore the two label sparsity settings, one with labels designed to be sufficient for supervised training, and one with labels designed to be sparse for semi-supervised training where semi-supervised graph clustering objectives should outperform purely supervised objectives. By constructing this semi-supervised framework we contribute (1) a way for any unsupervised clustering objective, reconstruction objective, or self-supervised objective to be used alongside any supervised objective to improve the performance of any neural network or graph neural network on node classification, particularly when labels are sparse, and (2) a way to evaluate the performance of graph clustering objectives on real-world attributed graph datasets in the setting of semi-supervised node classification.

## 3 Experiments

In our experiments we contribute (1) an evaluation of our proposed NSBM and CBAE in the setting of semi-supervised attributed graph clustering to improve the performance of any neural network or graph neural network on multi-class node classification. (2) a framework for evaluating the performance of attributed graph clustering objectives in the setting of semi-supervised node classification and an initial benchmark of existing attributed graph clustering methods adapted to end-to-end semi-supervised attributed graph clustering.

Our experiments focus on adapting multi-class classification to semi-supervised graph clustering on 6 real-world attributed graph datasets – Coauthor CS, Coauthor Physics, Amazon Computers, and Amazon Photo

from Shchur et al. (2018), and `roman-empire` and `amazon-ratings` from Platonov et al. (2023b). The cardinality of the set of labels $d_S$ is known and a non-zero number of training nodes per label is given.

To adapt these datasets to a semi-supervised clustering, we sparsify the labels by randomly selecting a subset of labels to be observed and the rest to be unobserved. Specifically, for each prescribed training, validation, and test split for each dataset, we randomly select 2 training nodes per label from the training split, and 50 validation nodes from the validation split. The test split remains the same as the original split.

Following the proposed approach in equation 9, semi-supervised clustering objectives combine the supervised cross-entropy objective $L_S$ in equation 16 with the unsupervised objectives $L_E$, $L_{\mathrm{regularization}}$, and $L_V$ in equation 17. We evaluate the performance of **4** unsupervised graph clustering objectives: NOCD (Shchur & Günnemann, 2019), DMoN (Tsitsulin et al., 2023), Neuromap (Blöcker et al., 2024), and our proposed SBM$_{\mathrm{NN}}$, and **2** unsupervised attribute reconstruction objectives: DMoN (Tsitsulin et al., 2023) and (unit) L2 regularization. Shchur & Günnemann (2019) showed that L2 regularization was beneficial, and we evaluate the case of unit L2 regularization which corresponds to an exact variational and description length objective (Graves, 2011), motivated by the interpretation of these model as generative models. $L_V$ is mean squared error loss. The performance of semi-supervised objectives over purely supervised objectives is evaluated on **4** neural network architectures, two graph neural networks – GCN (Kipf & Welling, 2016a) and GraphSAGE (Hamilton et al., 2017) – and two "non-graph" neural networks – MLP and Transformer.

Our main results are summarized in table 1 for label sparsified versions of the datasets, and in appendix F.2 for the default splits of the datasets. Semi-supervised objectives consistently outperform a purely supervised objective on label sparsified versions of the datasets and the performance of semi-supervised objectives over purely supervised objectives is significantly more pronounced when labels are sparse. Figure 1 and the additional visualizations in appendix F.1 support these findings. The best performing graph neural network on 3 of 6 datasets use the attribute reconstruction in our CBAE approach, also significantly more pronounced when labels are sparse (see appendix F.3 for comparing attribute reconstruction on default splits of the datasets).

Though graph neural networks such as GCN and GraphSAGE consistently produce the best performance on these datasets, transformers and MLPs trained with semi-supervised objectives under our NSBM framework both outperform a purely supervised objective on most datasets and perform competitively with graph neural networks on some datasets such as the Coauthor CS dataset. MLPs perform consistently better than transformers on these datasets. Results from Tsitsulin et al. (2023) and Blöcker et al. (2024) support these findings for unsupervised graph clustering showing that on datasets where k-means node clustering (only) on features outperforms SBM inference or Infomap (only) on the graph structure, MLPs outperformed graph neural networks. It has also been empirically observed and studied theoretically how graph neural networks use graphs when detrimentally in cases where graph structure should be ignored in favour of attributes (Bechler-Speicher et al., 2024).

Across architectures, the unsupervised graph clustering and regularization objectives of DMoN produced the best performance on 4 of 6 datasets, while transformers consistently benefited most from unit L2 regularization. Additional results in appendix F.4 echo that the SBM$_{\mathrm{NN}}$ function we designed and NOCD did not perform as well as implicit objectives such as DMoN. While we highlight the importance of designing architectures by uncovering and analysing their conditional dependencies, implicit models such as DMoN could provide a more expressive for learning on graphs. Notably we show that unsupervised graph clustering objectives with regularization outperform regularization alone, ablating the effect of regularization. Despite the small number of validation nodes in the sparse label setting, early stopping and model selection on the validation set still produces the best performance on the test set.

In the appendix, we include additional experimental details and results and discuss possible extensions of our framework to settings such as an unknown number of labels as well as unbalanced and unseen labels.

Table 1: Summary results comparing semi-supervised graph clustering (where $L_E$ is not "None") compared with semi-supervised node classification (where $L_E$ is "None") for 4 neural network architectures – Transformers, MultiLayer Perceptrons (MLPs), and Graph neural networks GCN and GraphSAGE – evaluated on 6 label sparsified real-world attributed graph datasets. The better result for each architecture is highlighted in **bold** and the best result across all architectures is underlined. The $L_E$ and $L_{\text{regularization}}$ columns indicate which unsupervised clustering and regularization losses where used in addition to a supervised cross-entropy loss for training. A ✗ or ✓ in the $L_V$ column indicates if attributes were reconstructed, and a ✗ or ✓ in the $L_S$ column indicates if a supervised cross-entropy loss was used for training. The "ES" column indicates if training loss or validation MCC was used as an early stopping criterion.

| Model | $f$ | $L_{\text{regularization}}$ | $L_V$ | $\text{NN}_{\boldsymbol{S \to X}}$ | ES | MCC | Accuracy |
|---|---|---|---|---|---|---|---|
| **Amazon Computers** | | | | | | | |
| GCN | None | None | ✗ | None | Loss | $0.502 \pm 0.096$ | $0.569 \pm 0.101$ |
| GCN | DMoN | DMoN | ✓ | MLP | Loss | $\mathbf{0.507 \pm 0.065}$ | $\mathbf{0.574 \pm 0.065}$ |
| GraphSAGE | None | DMoN | ✗ | None | Loss | $0.466 \pm 0.105$ | $0.522 \pm 0.114$ |
| GraphSAGE | DMoN | None | ✗ | None | MCC | $\mathbf{0.484 \pm 0.110}$ | $\mathbf{0.550 \pm 0.121}$ |
| MLP | None | DMoN | N/A | None | MCC | $\mathbf{0.222 \pm 0.034}$ | $\mathbf{0.297 \pm 0.041}$ |
| MLP | DMoN | DMoN | N/A | None | MCC | $0.207 \pm 0.058$ | $0.297 \pm 0.066$ |
| Transformer | None | None | N/A | None | Loss | $0.000 \pm 0.000$ | $0.075 \pm 0.051$ |
| Transformer | NOCD | L2 | N/A | None | Loss | $\mathbf{0.106 \pm 0.093}$ | $\mathbf{0.177 \pm 0.105}$ |
| **Amazon Photo** | | | | | | | |
| GCN | None | None | ✗ | None | MCC | $\mathbf{0.620 \pm 0.078}$ | $0.653 \pm 0.087$ |
| GCN | DMoN | None | ✗ | None | MCC | $0.618 \pm 0.093$ | $\mathbf{0.659 \pm 0.098}$ |
| GraphSAGE | None | DMoN | ✗ | None | MCC | $\mathbf{0.626 \pm 0.089}$ | $\mathbf{0.671 \pm 0.086}$ |
| GraphSAGE | DMoN | DMoN | ✗ | None | MCC | $0.571 \pm 0.102$ | $0.618 \pm 0.105$ |
| MLP | None | DMoN | N/A | None | MCC | $\mathbf{0.399 \pm 0.052}$ | $\mathbf{0.474 \pm 0.044}$ |
| MLP | DMoN | None | N/A | None | MCC | $0.387 \pm 0.019$ | $0.461 \pm 0.016$ |
| Transformer | None | L2 | N/A | None | MCC | $0.016 \pm 0.024$ | $0.130 \pm 0.079$ |
| Transformer | NOCD | L2 | N/A | None | Loss | $\mathbf{0.155 \pm 0.138}$ | $\mathbf{0.232 \pm 0.137}$ |
| **Coauthor CS** | | | | | | | |
| GCN | None | DMoN | ✗ | None | MCC | $0.748 \pm 0.052$ | $0.770 \pm 0.050$ |
| GCN | DMoN | None | ✗ | None | MCC | $\mathbf{0.756 \pm 0.045}$ | $\mathbf{0.778 \pm 0.042}$ |
| GraphSAGE | None | DMoN | ✗ | None | MCC | $0.758 \pm 0.042$ | $0.778 \pm 0.041$ |
| GraphSAGE | DMoN | DMoN | ✗ | None | MCC | $\mathbf{0.764 \pm 0.037}$ | $\mathbf{0.784 \pm 0.037}$ |
| MLP | None | None | N/A | None | MCC | $0.602 \pm 0.061$ | $0.639 \pm 0.059$ |
| MLP | NOCD | L2 | N/A | None | Loss | $\mathbf{0.710 \pm 0.046}$ | $\mathbf{0.738 \pm 0.044}$ |
| Transformer | None | L2 | N/A | None | MCC | $0.476 \pm 0.120$ | $0.516 \pm 0.130$ |
| Transformer | NOCD | L2 | N/A | None | Loss | $\mathbf{0.677 \pm 0.241}$ | $\mathbf{0.699 \pm 0.245}$ |
| **Coauthor Physics** | | | | | | | |
| GCN | None | None | ✗ | None | MCC | $0.816 \pm 0.046$ | $0.870 \pm 0.039$ |
| GCN | DMoN | None | ✗ | None | MCC | $\mathbf{0.830 \pm 0.042}$ | $\mathbf{0.882 \pm 0.030}$ |
| GraphSAGE | None | None | ✗ | None | MCC | $\mathbf{0.795 \pm 0.061}$ | $\mathbf{0.856 \pm 0.048}$ |
| GraphSAGE | DMoN | DMoN | ✓ | Transformer | MCC | $0.791 \pm 0.040$ | $0.853 \pm 0.026$ |
| MLP | None | None | N/A | None | MCC | $0.493 \pm 0.120$ | $0.613 \pm 0.146$ |

Table 1: Summary results on label-sparsified real-world attributed graph datasets.

| Model | $f$ | $L_{\text{regularization}}$ | $L_V$ | $\text{NN}_{S \to X}$ | ES | MCC | Accuracy |
|---|---|---|---|---|---|---|---|
| MLP | NOCD | L2 | N/A | None | Loss | **0.622 ± 0.090** | **0.725 ± 0.064** |
| Transformer | None | L2 | N/A | None | MCC | 0.490 ± 0.107 | **0.592 ± 0.134** |
| Transformer | NOCD | L2 | N/A | None | MCC | **0.495 ± 0.265** | 0.586 ± 0.310 |
| **amazon-ratings** | | | | | | | |
| GCN | None | L2 | ✗ | None | MCC | -0.003 ± 0.030 | 0.252 ± 0.093 |
| GCN | NOCD | None | ✓ | MLP | Loss | **0.031 ± 0.010** | **0.344 ± 0.006** |
| GraphSAGE | None | L2 | ✗ | None | MCC | **0.002 ± 0.004** | **0.032 ± 0.076** |
| GraphSAGE | NOCD | L2 | ✗ | None | MCC | 0.001 ± 0.005 | 0.031 ± 0.085 |
| MLP | None | None | N/A | None | Loss | 0.000 ± 0.000 | **0.368 ± 0.000** |
| MLP | NOCD | L2 | N/A | None | MCC | **0.000 ± 0.001** | 0.063 ± 0.134 |
| Transformer | None | None | N/A | None | Loss | **0.000 ± 0.000** | **0.358 ± 0.032** |
| Transformer | NOCD | None | N/A | None | Loss | 0.000 ± 0.000 | 0.321 ± 0.062 |
| **roman-empire** | | | | | | | |
| GCN | None | DMoN | ✗ | None | Loss | **0.090 ± 0.005** | **0.196 ± 0.006** |
| GCN | DMoN | DMoN | ✗ | None | Loss | 0.090 ± 0.006 | 0.195 ± 0.006 |
| GraphSAGE | None | DMoN | ✗ | None | Loss | 0.214 ± 0.184 | 0.301 ± 0.139 |
| GraphSAGE | DMoN | DMoN | ✓ | MLP | Loss | **0.327 ± 0.019** | **0.388 ± 0.014** |
| MLP | None | DMoN | N/A | None | MCC | 0.222 ± 0.034 | 0.240 ± 0.035 |
| MLP | DMoN | DMoN | N/A | None | MCC | **0.234 ± 0.043** | **0.261 ± 0.041** |
| Transformer | None | L2 | N/A | None | MCC | -0.000 ± 0.026 | **0.022 ± 0.050** |
| Transformer | NOCD | DMoN | N/A | None | MCC | **0.004 ± 0.012** | 0.013 ± 0.041 |

## 4 CONCLUSION

In this work we propose and evaluate a semi-supervised clustering approach for graph learning with neural networks. Our framework combines enables any unsupervised graph clustering objective to improve the performance of any neural network or graph neural network on multi-class node classification in the setting of semi-supervised graph clustering. The perspective we introduce helps us to understand how current graph neural network methods for graph clustering jointly cluster node attributes and graph structure, despite clustering objectives explicitly considering only node attributes and graph structure. Our framework also introduces an alternative graph learning approach, enabling such as neural networks such as transformers and multi-layer perceptrons to learn on graphs without positional or structural encodings and without spectral or message passing layers found in graph neural networks. The results of our experiments demonstrate that our approach is complementary to existing semi-supervised node classification approaches on six real-world node-attributed graph datasets, improving learning when training labels are sparse.

We propose that the adaptation evaluation of our approach for link prediction and other edge-level tasks, extending the approach for edge and graph-level learning tasks, and further investigating the impact of attribute reconstruction for graph neural network methods for graph clustering could be interesting avenues for future work. Theoretical analysis of the expressivity of the alternative graph learning approach for non-graph neural networks is also an open question which we hope to explore with the community. We discuss potential future work further in appendix B.2.

## 5 REPRODUCIBILITY STATEMENT

All source code and documentation to reproduce results is made available in the supplementary zip file. All dataset sources are detailed in the appendix, and the source code includes the necessary code to fetch the datasets. Experiments were run on a cluster with virtual machines each with 16 vCPUs and 128 GB RAM, a single NVIDIA L40 GPU with 48 GB VRAM, and 100 GB disk space. The presented experiments require approximately 1 year on a single virtual machine of the cluster to run, and the source code is designed to be run on a cluster with multiple virtual machines in parallel.

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

## A  ADDITIONAL BACKGROUND

In this section we provide additional background on the neural-prior stochastic block model, the contextual stochastic block model, Bayesian networks, and end-to-end semi-supervised node classification and unsupervised graph clustering with graph neural networks.

### A.1  NODE-ATTRIBUTED STOCHASTIC BLOCK MODELS

Attributed graphs with cluster structure can be generated by attributed variants of a class of random graph models called stochastic block models. Stochastic block models (SBMs) are generative models of random graphs with cluster structure. A prediction $\widehat{A}$ of $A$ is generated from predicted assignments of nodes to clusters $\widehat{S}$, and a block matrix $B$, which can be fixed or inferred from data.

$$\widehat{A} \sim \text{SBM}\left(\widehat{S}, B\right) \tag{11}$$

Each element $B_{a,b}$ of the block matrix defines the probability that a node in cluster $a$ connects to a node in cluster $b$. A prediction of $\widehat{S}$ of $S$ and, if learned, $B$ can be inferred by minimizing the reconstruction error between $\widehat{A}$ and $A$. $B$ and other variables that are considered never directed observed are always written in this work without a $\widehat{\phantom{x}}$.

One variant of the SBM can be constructed such that each edge $\widehat{A}_{i,j} \sim \text{Bernoulli}(W_{i,j})$ is drawn independently element-wise from a Bernoulli distribution

$$\widehat{A} \sim \text{SBM}_{\text{B}}\left(\widehat{S}, B\right) \coloneqq \text{Bernoulli}(W) \coloneqq \text{Bernoulli}\left(\sigma\left(\widehat{S}B\widehat{S}^T\right)\right) \tag{12}$$

where $\sigma$ is an activation function applied element-wise that ensures, when necessary, that $W$ is a matrix of valid edge probabilities e.g. sigmoid. For the case when $S$ is normalized such that cluster assignments for each node sums to 1, $\sigma$ can simply be an identity map.

This work builds on two node-attributed variants of the SBM, the *neural-prior stochastic block model* (Duranthon & Zdeborová, 2023b) and the *contextual stochastic block model* (Deshpande et al., 2018).

### A.1.1 NEURAL-PRIOR STOCHASTIC BLOCK MODEL

A generative model of $\widehat{S}$ and $\widehat{A}$ from $X$ has been proposed by Duranthon & Zdeborová (2023b), termed the neural-prior stochastic block model, which can be expressed as

$$\widehat{A} \sim \text{NSBM}_{\text{o}}(X) \coloneqq \text{SBM}\left(\text{NN}_{X \to \widehat{S}}(X), B\right) \tag{13}$$

where $\widehat{S} = \text{NN}_{X \to \widehat{S}}(X)$ and $\text{NN}_{X \to \widehat{S}}$ can be any neural network function of $X$ that generates a matrix $\widehat{S}$, composed with a stochastic block model SBM. This neural-prior stochastic block model is denoted $\text{NSBM}_{\text{o}}$ to distinguish it from its variant proposed later in this work.

### A.1.2 CONTEXTUAL STOCHASTIC BLOCK MODEL

A generative model of $\widehat{X}$ and $\widehat{A}$ from $\widehat{S}$ has been proposed by Deshpande et al. (2018), termed the *contextual stochastic block model*, later studied and developed in works such as Dreveton et al. (2023); Duranthon & Zdeborová (2023a), and can expressed as

$$\widehat{X}, \widehat{A} \sim \text{CSBM}_{\text{o}}\left(\widehat{S}, B\right) \coloneqq P_{\widehat{S} \to \widehat{X}}\left(\widehat{S}\right), \text{SBM}\left(\widehat{S}, B\right) \tag{14}$$

such that $\widehat{A}$ is generated by an SBM as before, and $\widehat{X}$ is sampled from a distribution $P_{\widehat{S} \to \widehat{X}}\left(\widehat{S}\right)$. This contextual-prior stochastic block model is denoted $\text{CSBM}_{\text{o}}$ to distinguish it from its variant proposed later in this work.

## A.2 GENERATIVE MODELS AS BAYESIAN NETWORKS

The dependence of variables in generative models such as the stochastic block model (SBM) and two of its node-attributed variants, $\text{NSBM}_{\text{o}}$ and $\text{CSBM}_{\text{o}}$ can be visualized as directed acyclic graphical models – Bayesian networks – as shown in figure 2. Bayesian networks (Kitson et al., 2023) can be helpful to visualize conditional dependencies and are used in section 2 to build upon the generative models introduced in this section. A directed edge from a variable $a$ to a variable $b$ in a Bayesian network represents a dependence of $b$ on $a$. The Bayesian networks in figure 3 show that $\text{NSBM}_{\text{o}}$ and $\text{CSBM}_{\text{o}}$ are two possible Bayesian networks that generate graph structures $\widehat{A}$ from clusters $\widehat{S}$ and with $A$ and $X$ (or $\widehat{X}$) conditionally independent given $\widehat{S}$. There are, however, other generative models connecting $X$ and $A$ (or their predictions) and $\widehat{S}$, such as graph neural networks for (semi-supervised) node classification.

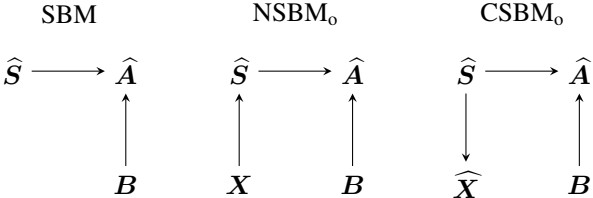

Figure 3: Bayesian network representations of the stochastic block model (SBM) and two of its node attributed variants, NSBM$_\text{o}$ and CSBM$_\text{o}$.

The Bayesian network representations of graph neural networks with a purely supervised node classification objective illustrate that $\boldsymbol{A}$ and $\boldsymbol{X}$ are conditionally independent but made conditionally dependent given $\boldsymbol{S}$. This is the opposite case of the generative model of attributed graphs with cluster structure proposed, where $\widehat{\boldsymbol{A}}$ and $\boldsymbol{X}$ (or $\widehat{\boldsymbol{X}}$ are conditionally dependent, but made conditionally independent given $\widehat{\boldsymbol{S}}$.

We refer the reader to surveys of Bayesian networks (Kitson et al., 2023) for further background on Bayesian networks.

### A.3 SEMI-SUPERVISED NODE CLASSIFICATION

In the case where $\boldsymbol{S}$ is (partially) observed, a graph neural network for (semi-)supervised node classification can be interpreted as a generative model of labels from node attributes and adjacencies.

$$\widehat{\boldsymbol{S}} \coloneqq \text{GNN}_{(\boldsymbol{X}, \boldsymbol{A}) \to \widehat{\boldsymbol{S}}}(\boldsymbol{X}, \boldsymbol{A}). \tag{15}$$

The cross entropy between observed (training) labels and predicted labels is commonly used as loss function for training (Kipf & Welling, 2016a; Tang & Liao, 2022; Zhao et al., 2024), which can be interpreted as a negative log-likelihood loss for the generative model (Miao et al., 2024).

$$l_{\text{supervised}} \coloneqq L_S\left(\widehat{\boldsymbol{S}}_{0:s,:}, \boldsymbol{S}_{0:s,:}\right) \coloneqq L_{\text{cross-entropy}}\left(\widehat{\boldsymbol{S}}_{0:s,:}, \boldsymbol{S}_{0:s,:}\right) \tag{16}$$

where $\widehat{\boldsymbol{S}}$ and $\boldsymbol{S}$ are of the same shape and $0:s$ represents the indexed training split. It is assumed that $(d_S)_{\text{obs}} = d_S$, i.e. all clusters are observed in the training split so that $(d_S)_{\text{obs}} \coloneqq (d_S)_{\text{obs}}$ and $\widehat{\boldsymbol{S}}$ and $\boldsymbol{S}$ are of the same shape. Its Bayesian network is visualized in figure 2. As $\boldsymbol{A}$ is an independent variable, these graph neural networks are not generative models of graphs.

### A.4 UNSUPERVISED CLUSTERING

In the case where $\boldsymbol{S}$ is not observed, $\boldsymbol{S}$ can be predicted by inferring a stochastic block model using graph neural networks. Graph neural networks for unsupervised clustering can be written as neural network functions identical to equation 15, while distinguished from node classification models by their training with unsupervised loss functions. This work proposes that unsupervised losses can be functions of $\widehat{\boldsymbol{S}}$ and, optionally, any combination of $\widehat{\boldsymbol{X}}$ and $\widehat{\boldsymbol{A}}$ such that

$$l_{\text{unsupervised}} \coloneqq L_{\text{clustering}}\left(\widehat{\boldsymbol{S}}, \boldsymbol{X}, \boldsymbol{A}\right) \coloneqq L_E\left(\widehat{\boldsymbol{A}}, \boldsymbol{A}\right) + L_V\left(\widehat{\boldsymbol{X}}, \boldsymbol{X}\right) + L_{\text{regularization}} \tag{17}$$

In architectures such as DiffPool (Ying et al., 2018), NOCD (Shchur & Günnemann, 2019), MinCutPool (Bianchi et al., 2020), DMoN (Tsitsulin et al., 2023), and TVGNN (Hansen & Bianchi, 2023), $L_V\left(\widehat{\boldsymbol{X}}, \boldsymbol{X}\right) = 0$ and the unsupervised clustering loss only depends on $\boldsymbol{S}$ and $\boldsymbol{A}$.

For example, DiffPool employs "entropy regularization" $L_{\text{regularization}} := -\frac{1}{n}\sum_{i=1}^{n}\sum_{a=1}^{d_S} S_{i,a}\log\left(S_{i,a}\right)$ to encourage cluster assignments that are close to one-hot vectors, and reconstructs $\boldsymbol{A}$ as $\widehat{\boldsymbol{A}} := \widehat{\boldsymbol{S}}\widehat{\boldsymbol{S}}^T$, computing $L_E$ as the Frobenius norm of the element-wise difference between $\widehat{\boldsymbol{A}}$ and $\boldsymbol{A}$, which can be interpreted as a reconstruction negative log-likelihood of an SBM with weighted edges and an identity block matrix $\boldsymbol{B} = \boldsymbol{I}$.

Extending the perspective of Peixoto & Kirkley (2022), other clustering approaches including spectral clustering (MinCutPool (Bianchi et al., 2020)) and modularity (DMoN (Tsitsulin et al., 2023)) can be interpreted as also inferring an *implicit* stochastic block model.

NOCD (Shchur & Günnemann, 2019) employs a "Bernoulli-Poisson" model, equivalent to variants of random dot product graphs and latent space models (Athreya et al., 2018), and a special case of the SBM$_\text{B}$ formulation in equation 12 with a fixed identity block matrix $\boldsymbol{B} = \boldsymbol{I}$ and a custom activation

$$\boldsymbol{W} := \sigma_{\text{NOCD}}\left(\widehat{\boldsymbol{S}}\widehat{\boldsymbol{S}}^T\right) := 1 - \exp\left(\text{ReLu}\left(\widehat{\boldsymbol{S}}\widehat{\boldsymbol{S}}^T\right)\right) \tag{18}$$

Sampling a set of positive edges $E_+$ and a set of negative edges $E_-$ of equal cardinality $|E_+| = |E_-| = m$ from $\boldsymbol{A}$, the negative log-likelihood $L_E$ of reconstructing $\boldsymbol{A}$ from $\widehat{\boldsymbol{A}}$ can be approximated as the binary cross entropy between the set of sampled edges $E_{\text{sampled}} := E_+ \cup E_-$ and their corresponding probabilities in $\boldsymbol{W}$, a common balanced link prediction objective (Li et al., 2023).

$$L_E := -\frac{1}{m}\sum_{(i,j)\in E_{\text{sampled}}}\left(A_{i,j}\log\left(W_{i,j}\right) + \left(1 - A_{i,j}\right)\log\left(1 - W_{i,j}\right)\right) \tag{19}$$

where $W_{i,j} = \sigma_{\text{NOCD}}\left((\boldsymbol{S})_{i:} \odot \boldsymbol{S}_{j:}\right)$ and $\odot$ represents the element-wise or Hadamard product. This approach to computing the reconstruction loss of $\boldsymbol{A}$ can enable reduced computational costs by calculating only a subset of entries of $\boldsymbol{W}$. This work adapts this loss for generative models later proposed, as described in the section 2.

Methods such as DiffPool and NOCD that explicitly reconstruct a graph can be interpreted as graph autoencoders (GAE$_\text{o}$) (Kipf & Welling, 2016b), with an encoder generating $\widehat{\boldsymbol{S}}$ according to equation 15, and a general decoder reconstructing $\boldsymbol{A}$ as $\widehat{\boldsymbol{A}} := \text{decoder}\left(\boldsymbol{S}\right)$. A Bayesian network representation of these models in figure 2 shows how $\boldsymbol{X}$ is treated as conditional evidence (Grover et al., 2019), and the ability of models to learn complex conditional dependencies between $\boldsymbol{S}$ and $\boldsymbol{A}$ and between $\boldsymbol{S}$ and $\boldsymbol{X}$ is limited.

# B  EXTENDING THE FRAMEWORK

In this section we describe extensions of our proposed framework to model other graph types and discuss potential extensions to other graph learning tasks.

## B.1  GRAPH TYPES

Our proposed framework can be extended to model other graph types such as weighted graphs, directed graphs, and graphs with edge and graph attributes.

### B.1.1  WEIGHTED GRAPHS

To model weighted graphs, a different distribution SBM $\left(\widehat{\boldsymbol{S}}, \boldsymbol{B}\right) = P_{\boldsymbol{W}\rightarrow\widehat{\boldsymbol{A}}}\left(\boldsymbol{W}\right)$ (and corresponding reconstruction loss $L_E$ ) modelling continuous edge weights can be used (instead of the Bernoulli distribution in

equation 12). For example, for a Gaussian distribution with zero mean and unit variance, the reconstruction loss can be the mean squared error between $\widehat{A}$ and $A$. For stochastic generation of $\widehat{A}$, $\widehat{S}$ (and, optionally, $B$) can be generated with a stochastic neural network, e.g. with (Monte Carlo) dropout (Gal & Ghahramani, 2016) (see appendix B.2.3 for details).

### B.1.2 Directed Graphs

An SBM can be made directed if its block matrix is directed, which can be fixed or learned. The following subsection extending the framework to edge and graph attributes describes how a directed block matrix can be learned for the proposed NSBM and CSBM.

### B.1.3 Edge and Graph Attributes

To model node attributes $X_V$, edge attributes $X_E$, and graph attributes $X_G$, the NSBM and CSBM we propose can be extended as follows. The graph neural network encoder of the autoencoders can be a neural network that is able to learn with edge attributes such as SAT (Chen et al., 2022), and a global pooling operation such as sum, mean, or max pooling can be incorporated to produce graph attributes.

$U$ can be interpreted as analogous to the node attributes in the NSBM and CSBM. $\widehat{A}$ and $X$ or $(\widehat{X})$ are conditionally independent given $\widehat{S}$ but conditionally dependent given $\widehat{\mathbf{X}}_E$.

The NSBM can be extended to model weighted and directed graphs with edge and graph attributes as follows.

$$\underset{n \times u}{U_V} := \text{NN}_{X_V \to U_V}\left(\underset{n \times d_V}{X_V}\right)$$

$$\underset{1 \times u}{U_G} := \text{NN}_{X_G \to U_G}\left(\underset{1 \times d_G}{X_G}\right)$$

$$\underset{(n+1) \times u}{U} := \left[\begin{array}{c} U_V \\ U_G \end{array}\right]$$

$$\underset{(n+1) \times d_S}{\widehat{S}} := \text{NN}_{U \to \widehat{S}}\left(U\right)$$

$$\underset{(n+1) \times d_S}{Z_\uparrow}, \underset{(n+1) \times d_S}{Z_\downarrow} := \text{NN}_{\widehat{S} \to (Z_\uparrow, Z_\downarrow)}\left(\widehat{S}\right)$$

$$\underset{d_S \times d_S}{B} := \phi\left(Z_\downarrow^T Z_\uparrow\right)$$

$$\underset{n \times n}{\widehat{A}} \sim \text{SBM}\left(\widehat{S}_{:n,:}, B\right)$$

$$\underset{d_E}{\left(\widehat{\mathbf{X}}_E\right)_{i,j,:}} := \widehat{A}_{ij}\,\text{NN}_{U \to \widehat{\mathbf{X}}_E}\left(\underset{u}{(U)_{i,:}} \odot \underset{u}{(U)_{j,:}}\right)$$

where SBM broadly represents any suitable stochastic block model including weighted variants discussed earlier in this section.

In this approach, transformed graph attributes are treated as transformed attributes of a virtual node.

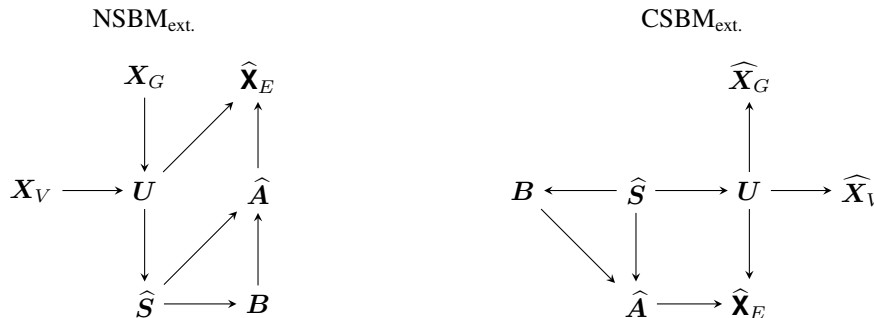

Figure 4: Bayesian networks of the NSBM and CSBM extended to generate graphs with node, edge, and graph attributes.

The CSBM can also be extended to model weighted and directed graphs with edge and graph attributes as follows.

$$\underset{n \times u}{\boldsymbol{U}} \coloneqq \mathrm{NN}_{\widehat{\boldsymbol{S}} \to \boldsymbol{U}} \left( \widehat{\boldsymbol{S}} \right)$$

$$\underset{n \times d_X}{\widehat{\boldsymbol{X}}_V} \coloneqq \mathrm{NN}_{\boldsymbol{U} \to \widehat{\boldsymbol{X}}_V} \left( \boldsymbol{U} \right)$$

$$\underset{d_G}{\widehat{\boldsymbol{X}}_G} \coloneqq \mathrm{NN}_{\boldsymbol{U} \to \widehat{\boldsymbol{X}}_G} \left( \sum_{i}^{n} (\boldsymbol{U})_{i,:} \right)$$

$$\underset{n \times k}{\boldsymbol{Z}_\uparrow}, \underset{n \times k}{\boldsymbol{Z}_\downarrow} \coloneqq \mathrm{NN}_{\widehat{\boldsymbol{S}} \to (\boldsymbol{Z}_\uparrow, \boldsymbol{Z}_\downarrow)} \left( \widehat{\boldsymbol{S}} \right)$$

$$\underset{k \times k}{\boldsymbol{B}} \coloneqq \phi \left( \boldsymbol{Z}_\downarrow^T \boldsymbol{Z}_\uparrow \right)$$

$$\underset{n \times n}{\widehat{\boldsymbol{A}}} \sim \mathrm{SBM} \left( \widehat{\boldsymbol{S}}, \boldsymbol{B} \right)$$

$$\underset{d_E}{\left( \widehat{\boldsymbol{X}}_E \right)_{i,j,:}} \coloneqq \widehat{A}_{ij} \, \mathrm{NN}_{\boldsymbol{U} \to \widehat{\boldsymbol{X}}_E} \left( \underset{u}{(\boldsymbol{U})_{i,:}} \odot \underset{u}{(\boldsymbol{U})_{j,:}} \right)$$

## B.2 OTHER GRAPH LEARNING TASKS

Here we discuss potential extensions of our proposed framework to other graph learning tasks such as semi-supervised graph clustering with unseen and unbalanced clusters, edge and graph-level tasks, and stochastic generation of attributed graphs with cluster structure.

### B.2.1 UNSEEN AND UNBALANCED CLUSTERS

When unknown, the cardinality $d_S$ of the set of node labels $S$ can be learned by setting an upper bound estimate $(d_S)_{\mathrm{max}} \geq (d_S)_{\mathrm{obs}}$. One heuristic for choosing this is upper bound is $\sqrt{n}$, found from empirical studies of real-world graphs (Ghasemian et al., 2019). The first $(d_S)_{\mathrm{obs}}$ rows of $\widehat{\boldsymbol{S}}$ can be trained to align with supervised training labels, while the remaining $(d_S)_{\mathrm{max}} - (d_S)_{\mathrm{obs}}$ learn unseen labels. Implicit regularization

of neural networks (Razin et al., 2022) and optional explicit regularization can learn compressed representations, learning a predicted $\widehat{d_S} < (d_S)_{\max}$ as demonstrated in the unsupervised clustering experiments in Blöcker et al. (2024).

Methods such as NOCD (Shchur & Günnemann, 2019), Neuromap (Blöcker et al., 2024), and the $\text{SBM}_{\text{NN}}$ we propose avoid assuming a balanced clustering, and can be used in place of methods such as MinCutPool (Bianchi et al., 2020), DMoN (Tsitsulin et al., 2023), and ACCPool (Hansen & Bianchi, 2023), in the setting of unbalanced clusters.

### B.2.2 EDGE AND GRAPH-LEVEL TASKS

The graph structure reconstruction objectives we study and propose in our framework is compatible with common link prediction objectives (see appendix A). The extension of our framework to link prediction is straightforward, as the reconstruction loss $L_E$ can be replaced with a link prediction loss such as the binary cross entropy loss in equation 19.

The extensions to model edge and graph attributes we discussed in the previous section can be used adapt the framework to edge and graph-level tasks such as edge classification and regression, and graph classification and regression by optimizing their respective reconstruction losses.

### B.2.3 STOCHASTIC GENERATION OF ATTRIBUTE GRAPHS WITH CLUSTER STRUCTURE

While we study and develop our framework using common neural network that are designed to be deterministic at test time, the proposed NSBM and CBAE generative models can be extended to sample from learned distributions at test time. As all neural networks in our experiments were trained with dropout, Monte Carlo dropout (Gal & Ghahramani, 2016) can be used to sample from the learned distribution at test time.

The extensions to edge and graph attributes we discussed in the previous section (appendix B.1) can be used to adapt the NSBM and CSBM to generate graphs with edge and graph attributes.

Our framework is also designed to be compatible with other deep generative models (Zhu et al., 2022), and the NSBM and CBAE with objectives such as variational learning, normalizing flows, and score matching can be used to learn and generate attributed graphs with cluster structure.

## C SCALABILITY AND COMPUTATIONAL COMPLEXITY

Due to the generality of our proposed framework, the computational complexity of the proposed NSBM and CSBM and their respective autoencoders is not fixed, but depends on the neural network modules, clustering methods, and regularization methods used. For example, if prototypical transformers with quadratic complexity in the number of nodes are used as neural networks of the NSBM, the computational complexity of the NSBM will also be quadratic in the number of nodes. However, if an MLP is used, the computational complexity of the NSBM will be linear in the number of nodes, unless a clustering method of quadratic complexity such as DiffPool (Ying et al., 2018) is used.

While we primarily focus on the full-batch setting, which is unamenable to large-scale graphs, the proposed framework can in principle be extended to allow mini-batching by using methods such as Lim et al. (2021) and by combining any (graph) neural network and clustering method that allows for mini-batching. We leave the exploration of mini-batching for future work.

# D  OTHER RELATED WORK

## D.1  SEMI-SUPERVISED NODE CLASSIFICATION

Semi-supervised node classification (Kipf & Welling, 2016a) studies the problem of classifying nodes in the setting of sparse training labels. The inductive bias provided by the observed graph can be effectively utilized by methods such as label propagation (Xiaojin & Zoubin, 2002; Li et al., 2019; Lin et al., 2020; Rossi et al., 2022).

Since the seminal work of semi-supervised node classification with graph convolutional networks (Kipf & Welling, 2016a), several later works study a setting of sparser training labels. Li et al. (2019) combines label propagation and graph filters proposing a "generalized label propagation" framework, while Lin et al. (2020) studied how metric learning can improve the performance for graph convolutional networks and label propagation frameworks. Several other later works adopt the perspective of Bayesian inference, from contrastive learning (Wan et al., 2021) to Bayesian graph neural networks (Pal et al., 2020), while others such as Dai et al. (2021) consider a similar problem of noisy labels.

This work provides an alternative and complementary approach to semi-supervised classification with sparse training labels, compatible with any end-to-end neural network architecture.

## D.2  GENERATIVE MODELS OF ATTRIBUTED GRAPHS WITH CLUSTER STRUCTURE

Generative models of graphs with cluster structure and their inference methods can be broadly categorized into deep learning and non-deep learning methods. Numerous non-neural network-based variants of the stochastic block model have been developed to model different notions of cluster structures (Peixoto, 2018; 2022), along with various inference algorithms (Peixoto, 2020; Mao & Zhang, 2023).

Attributed stochastic block models include the contextual stochastic block model (Deshpande et al., 2018) and the neural-prior stochastic block model (Duranthon & Zdeborová, 2023b). The contextual stochastic block model generates both node attributes and graph structure from node clusters (see appendix A.1.2), and prior works have proposed various distributions to model the generation of node attributes and adjacencies from node clusters including the exponential family of distributions (Dreveton et al., 2023).

Generative models of graphs with cluster structure that incorporate neural networks in their generative processes include the aforementioned neural-prior stochastic block model (see appendix A.1.1) and the feature-first block model (Tray & Kontoyiannis, 2021). The Feature-first Block Model (Tray & Kontoyiannis, 2021) is a similar model to the neural-prior stochastic block model (Duranthon & Zdeborová, 2023b) proposed with an inference algorithm proposed consisting a two-part Markov chain Monte Carlo approach with Metropolis Hastings.

In this work we propose alternative fully neural variants of the contextual and neural stochastic block model and a corresponding deep learning-based inference framework that composes common neural network modules, providing both a perspective to understand current graph neural network approaches for graph clustering and an alternative deep learning approach for graph learning that enables graph learning with transformers and multilayer perceptrons.

## D.3  SEMI-SUPERVISED DEEP GRAPH CLUSTERING

The application of the neural-prior stochastic block model and contextual stochastic block model to semi-supervised clustering been explored in synthetic data, inferred by an approximate message passing and belief propagation scheme (Duranthon & Zdeborová, 2023b;a). This work proposes a fully neural version of the

contextual stochastic block model, and formulates the inference of the proposes NSBM and CSBM variant as neural network learning with gradient descent.

Graph neural network approaches for semi-supervised clustering have been explored in He et al. (2022c;b;a). Leveraging a supervised clustering objective based on graph kernel methods (Tian et al., 2019), He et al. (2022c) proposed a semi-supervised node clustering framework SSSNet with graph neural networks for signed graphs, later adapted for directed graphs He et al. (2022b;a)). SSSNet is trained with an unsupervised clustering loss together with two supervised losses proposed by Tian et al. (2019): a supervised cross-entropy classification loss and a third "triplet" graph kernel-based loss. DIGRAC (He et al., 2022b) and MSGNN He et al. (2022a) extended the framework for semi-supervised clustering on synthetic directed and signed directed graphs, respectively. DIGRAC studied on synthetic graphs generated from directed stochastic block models, without attributes as an ablation study, comparing self-supervised vs supervised methods. MSGNN evaluated various graph neural network architectures for directed and signed graphs using this three-part semi-supervised loss.

Our contributions are complementary, as we draw connections with other end-to-end graph clustering approaches, propose a positional encoding-free learning setting for transformer and multilayer perceptron without graph neural network layers, and position semi-supervised clustering as inference of (neural) attributed graph generative models. While the frameworks are compatible in principle, we did not include the unsupervised graph clustering losses in He et al. (2022c) and He et al. (2022b) for the following reasons: The unsupervised clustering loss proposed in He et al. (2022c) was designed for distinguishing friends and enemies and is unsuitable for unsigned graphs (He et al., 2022c). The experiments were unable to incorporate the open source implementations of the unsupervised probabilistic imbalance clustering loss in DIGRAC (He et al., 2022b; 2023) due to memory issues.

We refer the reader to Shukla et al. (2019) for a survey of semi-supervised graph clustering methods, including neural network-based methods. The survey highlights the graph neural networks for semi-supervised clustering as a promising direction for future research, and the proposed framework in this work is a step in this direction.

## D.4 DEEP ATTRIBUTED GRAPH CLUSTERING

Methods for deep attributed graph clustering can be categorized into end-to-end deep attributed graph clustering methods and multi-step deep attributed graph clustering methods (Wang et al., 2023; Liu et al., 2023) End-to-end deep attributed graph clustering methods learn node embeddings and cluster them in a single step, while multi-step deep attributed graph clustering methods first learn node embeddings and then cluster them with a separate clustering method.

End-to-end deep attributed graph clustering methods include NOCD (Shchur & Günnemann, 2019), DMoN (Tsitsulin et al., 2023), and Neuromap (**?**).

Multi-step deep attributed graph clustering methods include Veličković et al. (2019); Zhu et al. (2020); Hassani & Khasahmadi (2020); Thakoor et al. (2022); Devvrit et al. (2022) that focus on self-supervised representation learning that enables a two-step classification or clustering approach by first learning node embeddings and then classifying these embeddings with a separate classifier or clustering them using non-neural network clustering methods such as k-means.

Our work focuses on end-to-end deep attributed graph clustering methods that learn node embeddings and cluster them in a single step. However, as our framework is compatible with any end-to-end unsupervised / self-supervised graph learning objective, it can in principle be used with end-to-end self-supervised graph representation learning methods. We leave the exploration of this for future work.

We refer the reader to Wang et al. (2023) and Liu et al. (2023) and for recent overviews and surveys of deep graph clustering methods.

## D.5 LEARNING ON GRAPHS WITH "PURE" TRANSFORMERS AND MLPS

Transformers (Vaswani et al., 2017) are set-to-set and sequence-to-sequence (Yun et al., 2020) artificial neural networks that employ attention mechanisms to model relationships between input sets or sequences. Graph transformers (Rampasek et al., 2022) mix graph neural network layers with attention layers and use additional positional encodings informative of graph structure (Huang et al., 2024) to learn on graphs. Transformers without graph neural network layers can also learn on graphs using these positional encodings informative of graph structure (Ying et al., 2021; Kim et al., 2022; Ma et al., 2023). Conversely, Buterez et al. (2024) showed how Transformers without positional encodings can use adjacency matrices as attention masks to learn on graphs effectively, an approach that can be interpreted as using graph attention (Veličković et al., 2018). In this work we propose a learning method for Transformers and MLPs that neither uses graph neural network layers nor positional encodings.

Blöcker et al. (2024) and Shchur & Günnemann (2019) have evaluated the performance MLPs for unsupervised attributed graph clustering, equivalent to MLP-based NSBMs in our framework for the purely unsupervised setting. Although Shchur & Günnemann (2019) reported significantly poorer relative unsupervised clustering performance of MLPs relative to GCNs when optimizing the NOCD clustering objective, the more recent evaluation by Blöcker et al. (2024) on more recent datasets (including datasets used in our experiments) and methods demonstrated that MLPs perform similarly to GNNs with different clustering objectives. Notably the results by Blöcker et al. (2024) also demonstrated that while MLPs had significantly poorer performance versus GNNs optimizing the NOCD clustering objective on certain datasets, MLPs optimizing other clustering objectives on those datasets perform similarly to GNNs. Our work provides a complementary perspective to understand why MLPs should perform similarly to GNNs when optimizing graph clustering objectives for attributed graph clustering, and introduces a framework for evaluating end-to-end graph clustering objectives on real-world attributed graphs in the semi-supervised setting. We propose that the variability of MLP performance due to different clustering objectives could be an interesting avenue for future work.

## D.6 GRAPH AUTOENCODERS WITH ATTRIBUTE RECONSTRUCTION

While prototypical graph autoencoder (Kipf & Welling, 2016b) were designed to only reconstruct graph structure, a notable exception developed in later works is the family of graph autoencoders for deep graph generation that includes GraphVAE (Simonovsky & Komodakis, 2018) that reconstruct both node attributes and graph structure (Zhu et al., 2022). Wang et al. (2022) proposed and evaluated a specific graph autoencoder architecture that reconstructs both node attributes and graph structure for attributed graph clustering and link prediction on the Planetoid datasets Cora, Citeseer, and Pubmed (Yang et al., 2016). Conversely, Hou et al. (2022) explored graph autoencoders that reconstruct attributes *instead* of graph structure for node classification and graph classification.

Our work introduces a perspective to understand the importance of reconstructing both node attributes and graph structure for learning on attributed graphs with graph neural networks, and introduces a general framework that can utilize any graph autoencoder that reconstructs both node attributes and graph structure for semi-supervised graph clustering.

## D.7 CLUSTERING *for* GRAPH NEURAL NETWORKS

Graph neural networks can be designed to implicitly and/or explicitly utilize graph cluster structures in their learning. Implicit optimization of clustering objectives by graph neural networks have been studied in

works such as Hansen & Bianchi (2023) which studied how a local quadratic variation partitioning objective is minimized by message passing graph neural networks such as graph convolutional networks (Kipf & Welling, 2016a).

Explicit uses of clustering objectives include pooling (Ying et al., 2018; Bianchi et al., 2020; Hansen & Bianchi, 2023) and batch sampling (Chiang et al., 2019) for graph neural networks. Clustering-based pooling approaches in aforementioned works such as DiffPool (Ying et al., 2018), MinCutPool (Bianchi et al., 2020), and TVGNN (Hansen & Bianchi, 2023) use explicit clustering objectives to pool graphs for graph-level tasks such as graph classification and graph regression. Clustering-based subgraph methods include Chiang et al. (2019) that propose using low-memory complexity graph clustering algorithms suited to large graphs to sample and batch large graphs, reducing memory requirements for graph neural networks. Finkelshtein et al. (2024) studied a similar approach that leveraged community detection to sparsify large graphs.

Mehta et al. (2019) incorporated the stochastic block model into the design of variational graph autoencoders (Kipf & Welling, 2016b) to model sparse graphs. Yang et al. (2020) study a connection between graph attention networks (Veličković et al., 2018) and the stochastic block model, framing graph attention networks as semi-amortized variational inference of stochastic block models. Our work provides a complementary perspective to understand how neural networks can infer attributed stochastic block models. We propose end-to-end fully neural variants of attributed stochastic block models that can be used with any neural network architecture. Our framework describes how any graph neural network can infer these attributed stochastic block models with attribute reconstruction, and how non-graph neural networks can learn on attributed graphs without positional encodings or graph neural network layers such as message passing or spectral convolutions by inferring these attributed stochastic block models.

# E ADDITIONAL EXPERIMENTAL DETAILS

In this section, we provide additional details on the datasets, architectures, and experimental setup used in the experiments in this work.

## E.1 DATASET DETAILS

We sourced the real-world attributed graph multi-class node classification datasets in our experiments from the following code and data repositories using the following libraries.

- Coauthor CS, Coauthor Physics, Amazon Computer, and Amazon Photo from Shchur et al. (2018) using the geometric deep learning library PyTorch Geometric (Fey & Lenssen, 2019).

- `roman-empire` and `amazon-ratings` from Platonov et al. (2023b) using PyTorch Geometric.

Each undirected edge between nodes $i$ and $j$ in the undirected graph datasets is represented with two directed edges, one from node $i$ to $j$ and another from node $j$ to $i$, following the approach in PyTorch Geometric (Fey & Lenssen, 2019). Node attributes are normalized so that the attributes for each node sum to 1.

10 splits for each dataset `roman-empire` and `amazon-ratings` from Platonov et al. (2023b) were made public. 1 split for each dataset `ogbn-arxiv` and `ogbn-products` from the Open Graph Benchmark (Hu et al., 2021) was made public. The splits for the Coauthor CS, Coauthor Physics, Amazon Computer, and Amazon Photo datasets from Shchur et al. (2018) were not made public, but the work proposes 10 random splits of 20 training nodes per class, 500 validation nodes, and 1000 test nodes.

We create a label-sparsified version of each dataset by selecting 2 training nodes per class and 50 validation nodes from the splits described above. We do not modify the test labels.

We run each experiment configuration 10 times on each dataset and its label-sparsified version and report the mean and standard deviation of the results. For the `roman-empire` and `amazon-ratings` datasets, we use a different public split for each run. For the Coauthor CS, Coauthor Physics, Amazon Computer, and Amazon Photo datasets we use a random split with the same number of training nodes per class, validation nodes, and test nodes for each run.

Table 2: Summary statistics of real-world attributed graph datasets used in experiments.

| Dataset | Nodes | Edges | Node Attributes | Classes |
|---|---|---|---|---|
| Coauthor CS | 18,333 | 163,788 | 6,805 | 15 |
| Coauthor Physics | 34,493 | 495,924 | 8,415 | 5 |
| Amazon Computers | 13,752 | 491,722 | 767 | 10 |
| Amazon Photo | 7,650 | 238,162 | 745 | 8 |
| roman-empire | 22,662 | 32,927 | 300 | 18 |
| amazon-ratings | 24,492 | 93,050 | 300 | 5 |
| ogbn-arxiv | 169,343 | 1,166,243 | 128 | 40 |
| ogbn-products | 2,449,029 | 61,859,140 | 100 | 47 |

## E.2 ARCHITECTURE DETAILS

The transformer architecture used is a prototypical 6 layer encoder-only transformer (Vaswani et al., 2017), adopting its default PyTorch (Ansel et al., 2024) with 8 attention heads, 512 embedding dimensions and 2048 hidden dimensions in its feedforward block, with a bias in its linear and layernorm layers. MLP is equivalent to the feedforward block in the transformer. Both use a dropout rate of 0.1.

GCN and GraphSAGE are the graph neural network architectures used in experiments, adopting its implementations in PyTorch Geometric (Fey & Lenssen, 2019). They use 3 layers, 256 hidden dimensions, and a dropout rate of 0.5.

All models were trained with $s = 100$ cluster dimensions. This was chosen to be of the same order of magnitude as the square root of the number of nodes of the median dataset size, following empirical studies of the upper bound of the number of clusters in real-world graphs (Ghasemian et al., 2019). As graph structure $\boldsymbol{A}$ is reconstructed from $\widehat{\boldsymbol{S}}$, the number of clusters $s$ is chosen to be large enough so that $\widehat{\boldsymbol{A}} = \mathrm{SBM}_f\left(\widehat{\boldsymbol{S}}\right)$ is not too lossy a reconstruction of $\boldsymbol{A}$, while being small enough to limit the computational complexity of the model.

As evidenced by the additional experimental results in section F, having the number of clusters $s$ larger than the number of classes does not significantly affect the performance of a purely supervised model.

## E.3 HYPERPARAMETER SETTINGS

For a controlled experiment under a limited compute budget, hyperparameters such as learning rates or the number of layers were not tuned. Architectural hyperparameters are detailed above in the architecture details. We used the same learning rate of 0.01 for all parameters, for all experiments, optimized with AdamW (Loshchilov & Hutter, 2019). Where $L_{\mathrm{regularization}}$ is "L2" weight decay is set to 1.0 and otherwise weight decay is set to 0.01.

### E.4 UNSUPERVISED CLUSTERING OBJECTIVES

In this section we provide additional details on the unsupervised clustering objectives used in the experiments.

#### E.4.1 $L_E$ OBJECTIVES

We evaluated 4 unsupervised clustering objectives – NOCD (Shchur & Günnemann, 2019), DMoN (Tsitsulin et al., 2023), Neuromap (Blöcker et al., 2024), and our proposed SBM$_{\text{NN}}$ – as $c \approx L_E$ objectives (see section 2.1.1) in the experiments.

As NOCD can be interpreted as a special case of the SBM (see section A), we implement NOCD simply as the case where the SBM has a fixed identity block matrix $\boldsymbol{B} = \boldsymbol{I}$. Instead of the custom activation function in the NOCD implementation to generate valid edge probabilities, we use the sigmoid activation function which matches a typical binary cross entropy from logits loss used in link prediction objectives (Zhang & Chen, 2018; Li et al., 2023; Qarkaxhija et al., 2024). We argue that these implementations are equivalent.

"L2" is $l_2$ regularization on the weights of a neural network, motivated by the interpretation of these neural networks as generative models (Graves, 2011).

DMoN comprises two unsupervised objectives – a spectral modularity objective and a "collapse" regularization objective. We use its spectral modularity objective as an $L_E$ objectives and its collapse regularization objective as an $L_{\text{regularization}}$ objective. We chose DMoN's objectives as representative $L_E$ and $L_{\text{regularization}}$ objectives as they were designed to jointly clustering node attributes and adjacencies (unsupervised) using graph neural networks.

DMoN's spectral modularity $L_E$ objective is calculated as

$$(L_E)_{\text{DMoN}} := -\frac{1}{2m}\text{Tr}\left(\boldsymbol{S}^T\left(\boldsymbol{A} - \frac{\boldsymbol{d} \otimes \boldsymbol{d}}{2m}\right)\boldsymbol{S}\right) \tag{20}$$

where $m = |E|$ is the number of edges in the graph and $\boldsymbol{d}$ is a degree vector representing the sum of the rows of the adjacency matrix $\boldsymbol{A}$, and $\boldsymbol{d} \otimes \boldsymbol{d}$ is the outer product of $\boldsymbol{d}$ with itself. The implementation of DMoN (Tsitsulin et al., 2023) was sourced from PyTorch Geometric (Fey & Lenssen, 2019).

Neurommap's (Blöcker et al., 2024) codelength $L_E$ objective is calculated as

$$(L_E)_{\text{Neuromap}} = q\log_2 q - (\boldsymbol{q}_{\text{m}}\log_2\boldsymbol{q}_{\text{m}})\underset{s}{\boldsymbol{1}} - (\boldsymbol{m}_{\text{exit}}\log_2\boldsymbol{m}_{\text{exit}})\underset{s}{\boldsymbol{1}} - (\boldsymbol{p}\log_2\boldsymbol{p})\underset{n}{\boldsymbol{1}} + (\boldsymbol{p}_{\text{m}}\log_2\boldsymbol{p}\text{m})\underset{s}{\boldsymbol{1}} \tag{21}$$

$\underset{k}{\boldsymbol{1}}$ is the $k$-dimensional vector of ones, and logarithms are applied component-wise. Each of the terms is defined as follows.

$$q = 1 - \text{tr}(\boldsymbol{C}) \quad \boldsymbol{q}_{\text{m}} = \boldsymbol{C}\underset{s}{\boldsymbol{1}} - \text{diag}(\boldsymbol{C}) \quad \boldsymbol{m}_{\text{exit}} = (\boldsymbol{1}^\top\boldsymbol{C})^\top - \text{diag}(\boldsymbol{C}) \quad \boldsymbol{p}_{\text{m}} = \boldsymbol{q}_{\text{m}} + \underset{s}{\boldsymbol{1}}^\top\boldsymbol{C}$$

where

$$\underset{s \times s}{\boldsymbol{C}} = \boldsymbol{S}^\top\boldsymbol{F}\boldsymbol{S} \tag{22}$$

and

$$\underset{n \times n}{\boldsymbol{F}} = \frac{\alpha}{w_{\text{tot}}}\boldsymbol{A} + (1 - \alpha)\text{diag}(\boldsymbol{p})\boldsymbol{T} \tag{23}$$

The transition matrix $\boldsymbol{T}$ is calculated using the graph's total weight $w_{\text{tot}} = \sum_{i \in V}\sum_{j \in V} w_{ij}$ and the vector of weighted node in-degrees $\boldsymbol{d}^{\text{in}}$ (as Neuromap is defined for directed graphs) as

$$T_{ij} = \begin{cases} \dfrac{w_{ij}}{\sum_{j \in V} w_{ij}} & \text{if } \sum_{j \in V} w_{ij} > 0, \\ 0 & \text{otherwise,} \end{cases} \qquad d_j^{\text{in}} = \sum_{i \in V} w_{ij}.$$

and the vector $\boldsymbol{p}$ of node visit rates is calculated using smart teleportation Lambiotte & Rosvall (2012) and the power iteration method which iteratively updates the visit rates $\boldsymbol{p}^{(t)}$ (until convergence) as

$$\boldsymbol{p}^{(t+1)} \leftarrow \frac{\alpha}{w_{\text{tot}}} \boldsymbol{d}^{\text{in}} + (1 - \alpha) \, \boldsymbol{p}^{(t)} \boldsymbol{T} \tag{24}$$

where $\boldsymbol{p}^{(0)} = \boldsymbol{d}^{\text{in}}$ and $\alpha$ is a parameter chosen proportionally to the nodes' in-degrees. The implementation of Neuromap (Blöcker et al., 2024) was sourced from its official repository at `https://github.com/chrisbloecker/neuromap`.

For the $\text{SBM}_{\text{NN}}$ we propose, we evaluated two neural networks as $\text{NN}_{\widehat{\boldsymbol{S}} \to \widehat{\boldsymbol{z}}}$ in equation 4 to generate $\boldsymbol{B}$ – a transformer and an MLP with the same architectures and hyperparameter settings as the transformers and MLPs detailed above.

### E.4.2    $L_V$ OBJECTIVES

For the "GCN" and "GraphSAGE" models that refer to CBAE autoencoders inferring a CSBM according to equation 7 we evaluated two choices of $\text{NN}_{\widehat{\boldsymbol{S}} \to \widehat{\boldsymbol{X}}}$ – a transformer and an MLP. We set the transformer and MLP architectures and hyperparameter settings the same as the transformers and MLPs detailed above. $L_V$ was then calculated as the mean squared error of the reconstruction $\widehat{\boldsymbol{X}}$ of $\boldsymbol{X}$.

### E.4.3    REGULARIZATION OBJECTIVES

We evaluated two $L_{\text{regularization}}$ objectives – "L2" and "DMoN" (Tsitsulin et al., 2023).

The collapse regularization $L_{\text{regularization}}$ objective of DMoN aims to discourage the trivial partition of assigning all nodes to the same cluster and can be expressed as

$$(L_{\text{regularization}})_{\text{DMoN}} := \frac{\sqrt{s}}{n} \left\| \sum_i (\boldsymbol{S}_{:,i}) \right\|_2 - 1 \tag{25}$$

where $\|\cdot\|_2$ is the $l_2$ vector norm. The implementation of DMoN (Tsitsulin et al., 2023) was sourced from PyTorch Geometric (Fey & Lenssen, 2019).

L2 regularization was calculated as the $l_2$ norm of all weights in the neural network. Shchur & Günnemann (2019) showed that small L2 regularization was beneficial, and we evaluate the case of unit L2 regularization which corresponds to an exact variational and description length objective (Graves, 2011), motivated by the interpretation of these model as generative models (see section B.2.3).

### E.5    EVALUATION METRICS

The experiments focused on semi-supervised *multi-class* node classification for real-world node-attributed graphs. Two evaluation metrics are used: accuracy and Matthews correlation coefficient (Gösgens et al., 2021; Platonov et al., 2023a). Both metrics are evaluated on the test set for each dataset and split. Accuracy alone is often standard as a sole evaluation metric for multi-class classification (as done in Yang et al. (2016); Kipf & Welling (2016a); Shchur et al. (2018); Platonov et al. (2023b); Hu et al. (2021)).

We also employed the Matthews correlation coefficient as an additional evaluation metric to provide a more comprehensive evaluation due to its theoretical and empirical (Gösgens et al., 2021; Platonov et al., 2023a) advantages in evaluating classifications, and its use in graph learning with encoding-free transformers without graph neural network layers such as Buterez et al. (2024). Its advantages are evident in the experiments, such as in evaluating performance on the `amazon-ratings` dataset (see table 1). Despite some models

achieving an accuracy of roughly 0.3 and other models achieving near zero accuracy, the Matthews correlation coefficient was zero across most models, indicating that these models were not predicting any better than random.

Other clustering evaluation metrics such as modularity or graph conductance (as used in Tsitsulin et al. (2023) for unsupervised clustering) were not used in the experiments as they are defined for non-attributed graphs.

### E.6 EARLY STOPPING AND MODEL SELECTION WITHOUT A VALIDATION SET

While early stopping and model selection on the validation set is standard practice for supervised learning (Li et al., 2020; Bai et al., 2021; Ji et al., 2021), for unsupervised learning a validation set is unavailable, and early stopping and model selection on the training loss has been explored for unsupervised graph clustering (Shchur & Günnemann, 2019; Blöcker et al., 2024). As our framework positions semi-supervised clustering as from both supervised and unsupervised clustering perspectives, we evaluated early stopping and model selection on the training loss and on the validation Matthews correlation coefficient (MCC). Despite the small validation set size (50 nodes) on label sparsified splits, we found that early stopping and model selection on the validation MCC produced the best generalization performance on the test set for both default and label sparsified splits of the datasets.

### E.7 SOFTWARE USED

Experiments leveraged PyTorch (Ansel et al., 2024) and PyTorch Geometric (PyG) (Fey & Lenssen, 2019) libraries for graph deep learning, and scikit learn for evaluation metrics (Pedregosa et al., 2011). Compiling results into tables utilized the library pandas (The pandas development team)., and visualizations utilized the libraries cuGraph (Team, 2023), NetworkX (Hagberg et al., 2008), and Matplotlib (Hunter, 2007). In particular, visualizations used the ForceAtlas2 (Jacomy et al., 2014) algorithm implementation by the GPU accelerated cuGraph library. All experiments were run on a SLURM (Yoo et al., 2003) cluster with Apptainer (formerly Singularity) (Kurtzer et al., 2017) using the PyG container NVIDIA NGC container registry which contains all of the software dependencies above.

# F ADDITIONAL RESULTS

## F.1 VISUALISATIONS OF GRAPH LABELLINGS

### F.1.1 AMAZON-RATINGS DATASET, DEFAULT SPLIT WITH DEFAULT TRAIN NODES PER CLASS AND DEFAULT VALIDATION NODES.

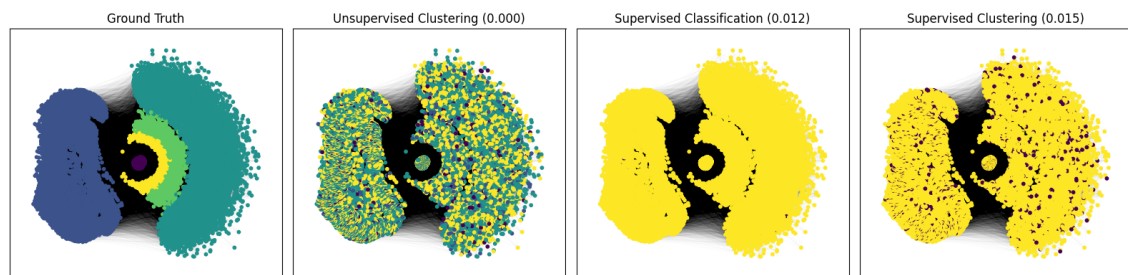

Figure 5: Visualisations of graph labellings for the amazon-ratings dataset, default split with default train nodes per class and default validation nodes. Model selection based on training loss.

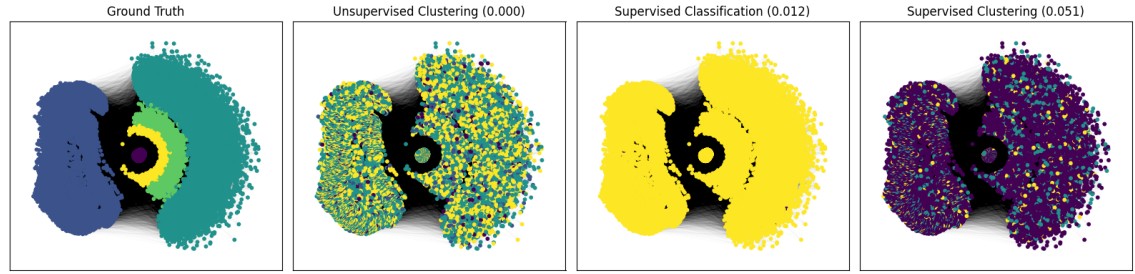

Figure 6: Visualisations of graph labellings for the amazon-ratings dataset, default split with default train nodes per class and default validation nodes. Model selection based on training set MCC.

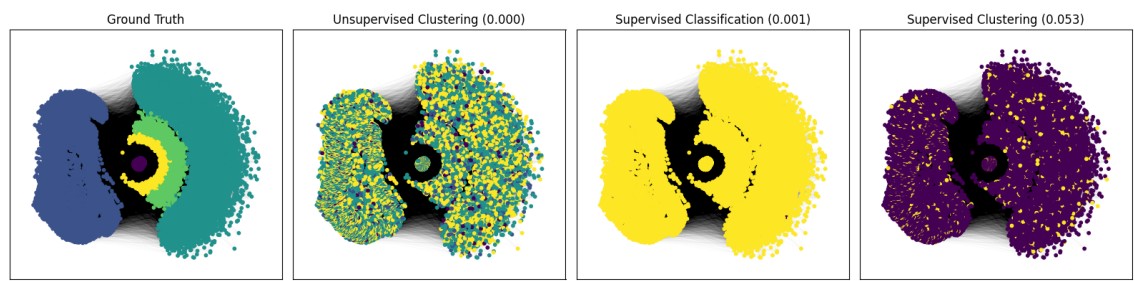

Figure 7: Visualisations of graph labellings for the amazon-ratings dataset, default split with default train nodes per class and default validation nodes. Model selection based on validation set MCC.

### F.1.2 AMAZON-RATINGS DATASET, SPARSE SPLIT WITH 2 TRAIN NODES PER CLASS AND 50 VALIDATION NODES.

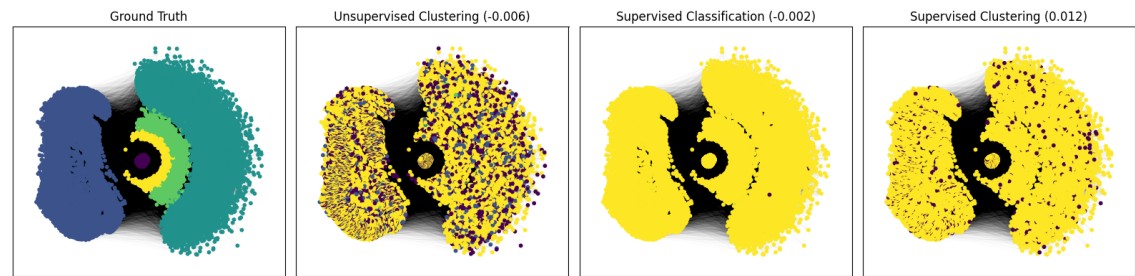

Figure 8: Visualisations of graph labellings for the amazon-ratings dataset, sparse split with 2 train nodes per class and 50 validation nodes. Model selection based on training loss.

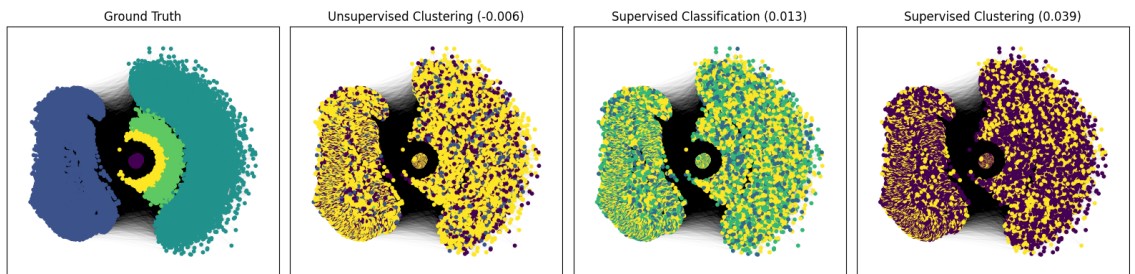

Figure 9: Visualisations of graph labellings for the amazon-ratings dataset, sparse split with 2 train nodes per class and 50 validation nodes. Model selection based on training set MCC.

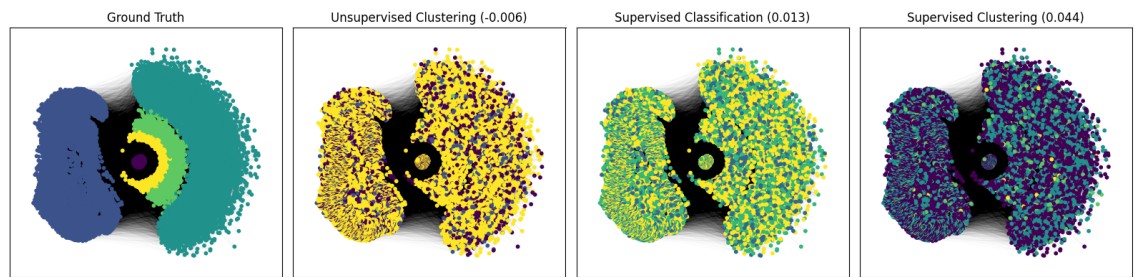

Figure 10: Visualisations of graph labellings for the amazon-ratings dataset, sparse split with 2 train nodes per class and 50 validation nodes. Model selection based on validation set MCC.

### F.1.3 COAUTHOR CS DATASET, DEFAULT SPLIT WITH 20 TRAIN NODES PER CLASS AND 500 VALIDATION NODES.

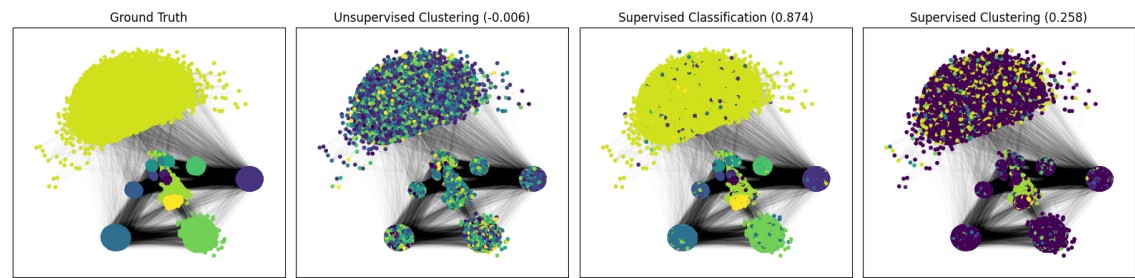

Figure 11: Visualisations of graph labellings for the Coauthor CS dataset, default split with 20 train nodes per class and 500 validation nodes. Model selection based on training loss.

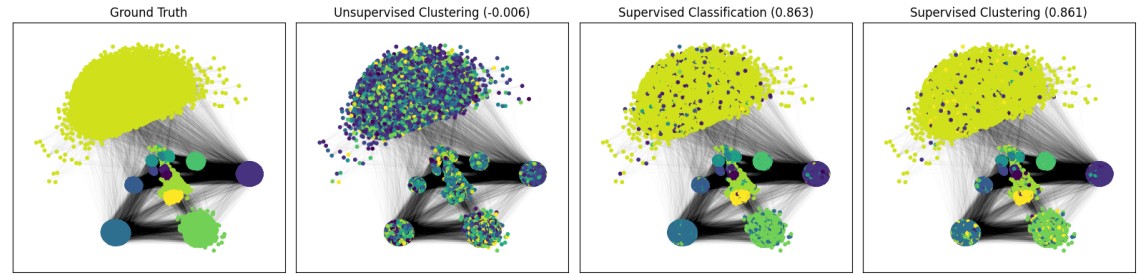

Figure 12: Visualisations of graph labellings for the Coauthor CS dataset, default split with 20 train nodes per class and 500 validation nodes. Model selection based on training set MCC.

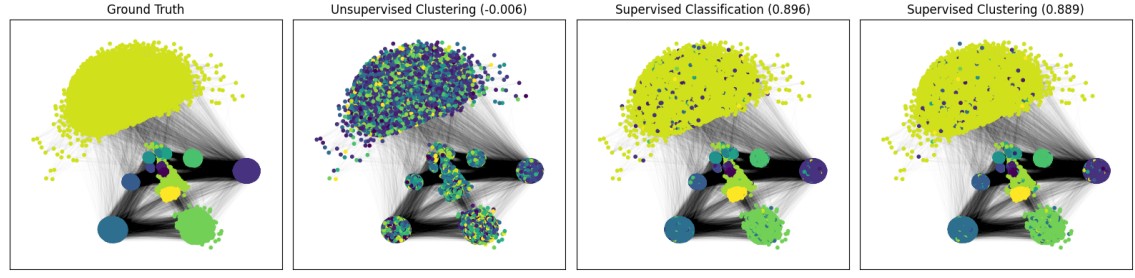

Figure 13: Visualisations of graph labellings for the Coauthor CS dataset, default split with 20 train nodes per class and 500 validation nodes. Model selection based on validation set MCC.

### F.1.4 COAUTHOR CS DATASET, SPARSE SPLIT WITH 2 TRAIN NODES PER CLASS AND 50 VALIDATION NODES.

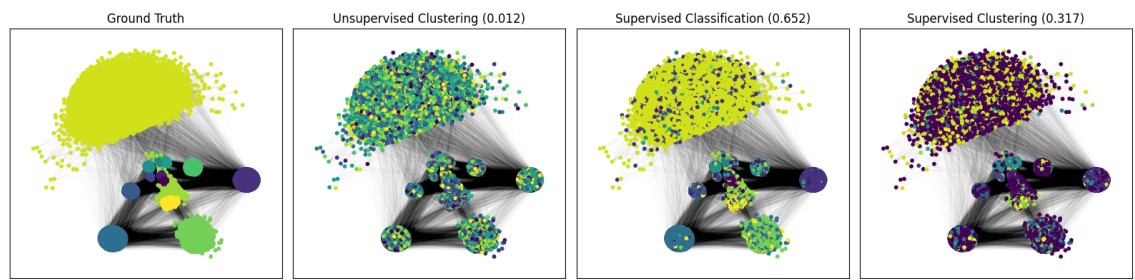

Figure 14: Visualisations of graph labellings for the Coauthor CS dataset, sparse split with 2 train nodes per class and 50 validation nodes. Model selection based on training loss.

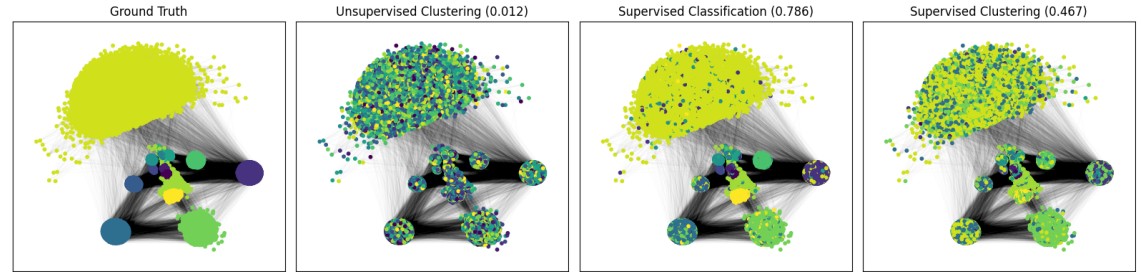

Figure 15: Visualisations of graph labellings for the Coauthor CS dataset, sparse split with 2 train nodes per class and 50 validation nodes. Model selection based on training set MCC.

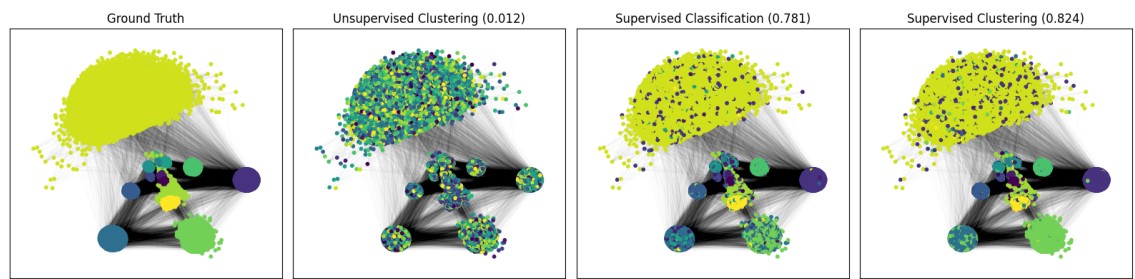

Figure 16: Visualisations of graph labellings for the Coauthor CS dataset, sparse split with 2 train nodes per class and 50 validation nodes. Model selection based on validation set MCC.

F.1.5  AMAZON COMPUTERS DATASET, DEFAULT SPLIT WITH 20 TRAIN NODES PER CLASS AND 500 VALIDATION NODES.

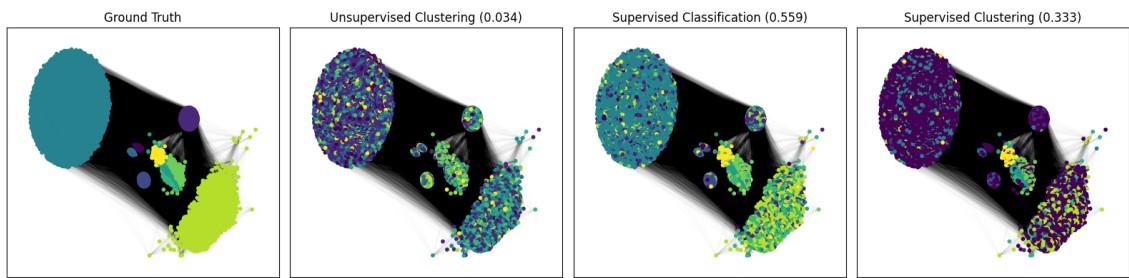

Figure 17: Visualisations of graph labellings for the Amazon Computers dataset, default split with 20 train nodes per class and 500 validation nodes. Model selection based on training loss.

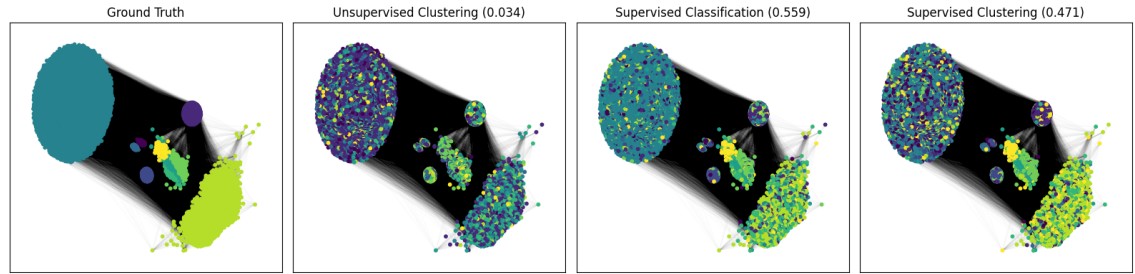

Figure 18: Visualisations of graph labellings for the Amazon Computers dataset, default split with 20 train nodes per class and 500 validation nodes. Model selection based on training set MCC.

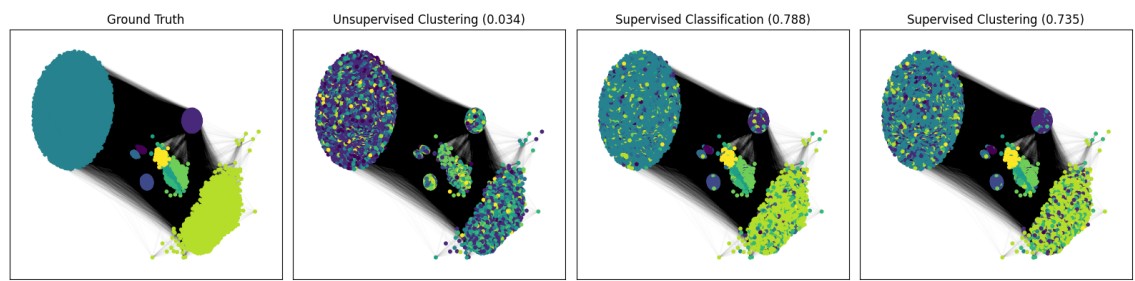

Figure 19: Visualisations of graph labellings for the Amazon Computers dataset, default split with 20 train nodes per class and 500 validation nodes. Model selection based on validation set MCC.

### F.1.6 AMAZON COMPUTERS DATASET, SPARSE SPLIT WITH 2 TRAIN NODES PER CLASS AND 50 VALIDATION NODES.

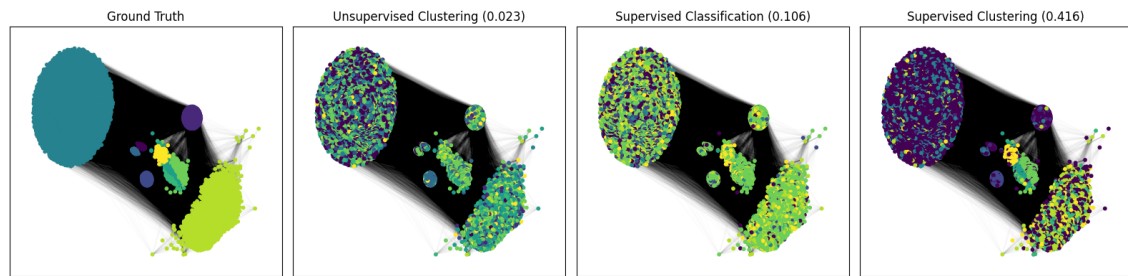

Figure 20: Visualisations of graph labellings for the Amazon Computers dataset, sparse split with 2 train nodes per class and 50 validation nodes. Model selection based on training loss.

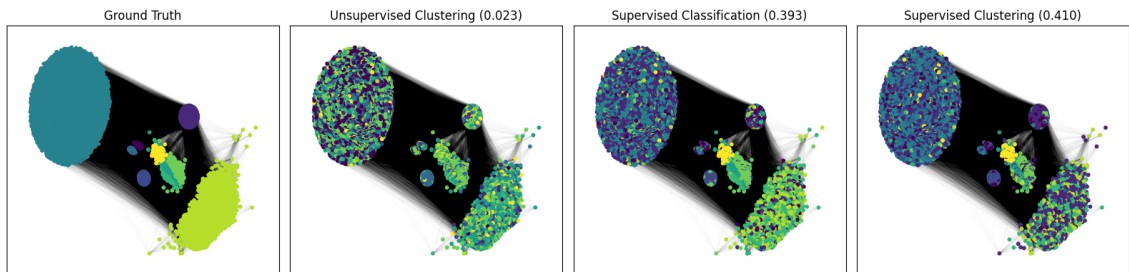

Figure 21: Visualisations of graph labellings for the Amazon Computers dataset, sparse split with 2 train nodes per class and 50 validation nodes. Model selection based on training set MCC.

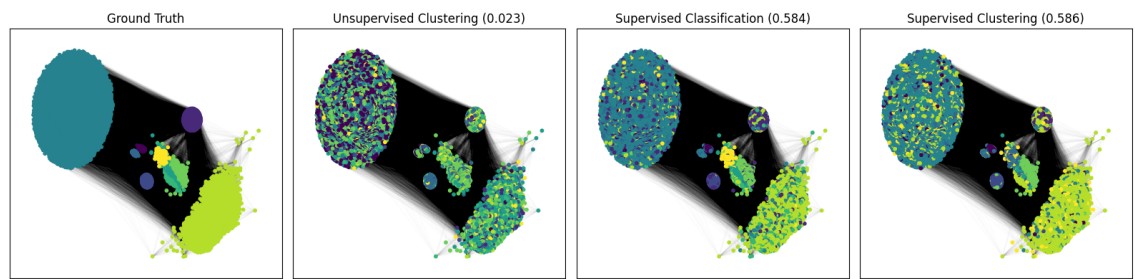

Figure 22: Visualisations of graph labellings for the Amazon Computers dataset, sparse split with 2 train nodes per class and 50 validation nodes. Model selection based on validation set MCC.

### F.1.7 AMAZON PHOTO DATASET, DEFAULT SPLIT WITH 20 TRAIN NODES PER CLASS AND 500 VALIDATION NODES.

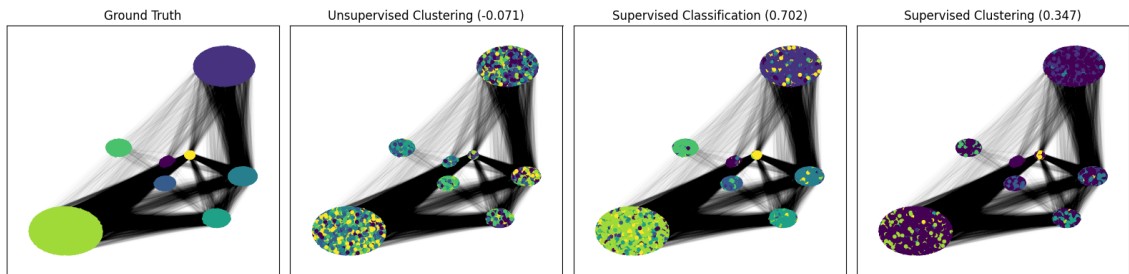

Figure 23: Visualisations of graph labellings for the Amazon Photo dataset, default split with 20 train nodes per class and 500 validation nodes. Model selection based on training loss.

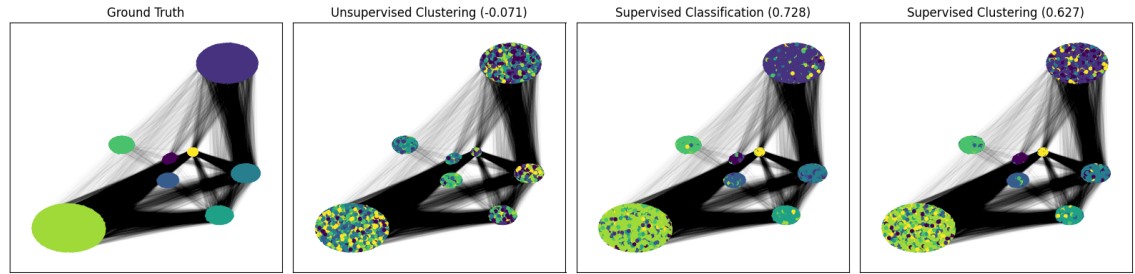

Figure 24: Visualisations of graph labellings for the Amazon Photo dataset, default split with 20 train nodes per class and 500 validation nodes. Model selection based on training set MCC.

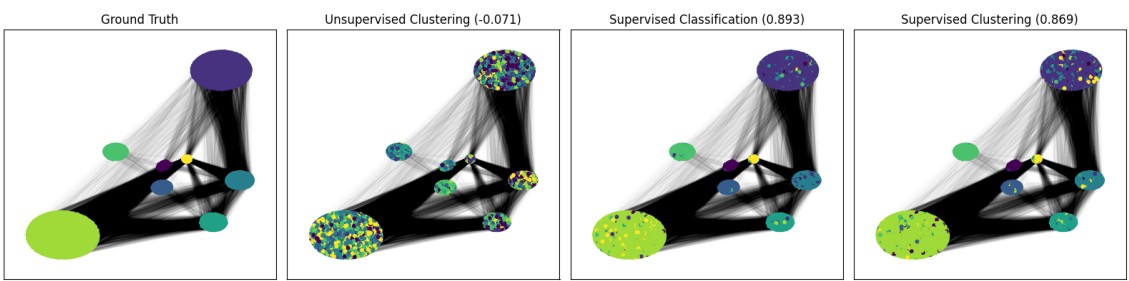

Figure 25: Visualisations of graph labellings for the Amazon Photo dataset, default split with 20 train nodes per class and 500 validation nodes. Model selection based on validation set MCC.

### F.1.8 AMAZON PHOTO DATASET, SPARSE SPLIT WITH 2 TRAIN NODES PER CLASS AND 50 VALIDATION NODES.

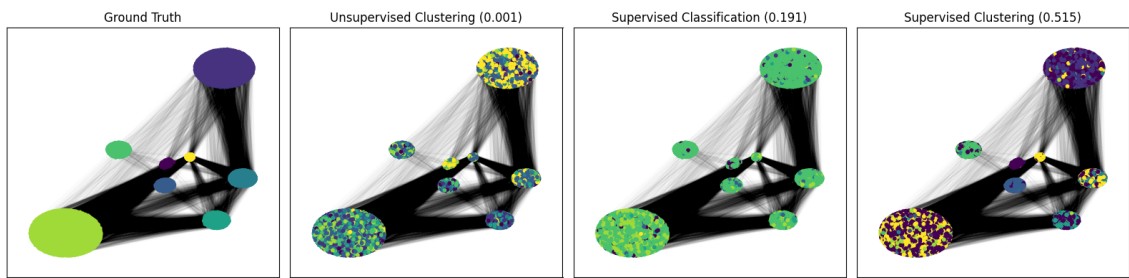

Figure 26: Visualisations of graph labellings for the Amazon Photo dataset, sparse split with 2 train nodes per class and 50 validation nodes. Model selection based on training loss.

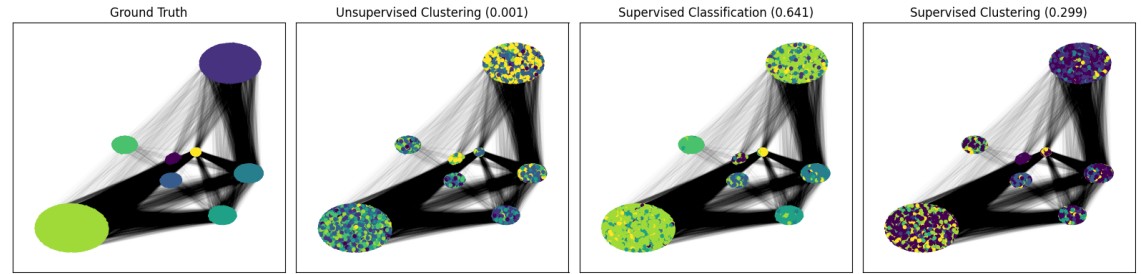

Figure 27: Visualisations of graph labellings for the Amazon Photo dataset, sparse split with 2 train nodes per class and 50 validation nodes. Model selection based on training set MCC.

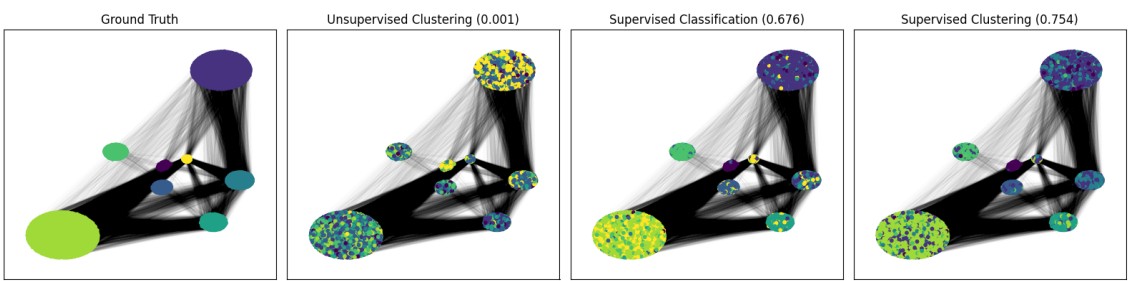

Figure 28: Visualisations of graph labellings for the Amazon Photo dataset, sparse split with 2 train nodes per class and 50 validation nodes. Model selection based on validation set MCC.

### F.1.9 COAUTHOR PHYSICS DATASET, DEFAULT SPLIT WITH 20 TRAIN NODES PER CLASS AND 500 VALIDATION NODES.

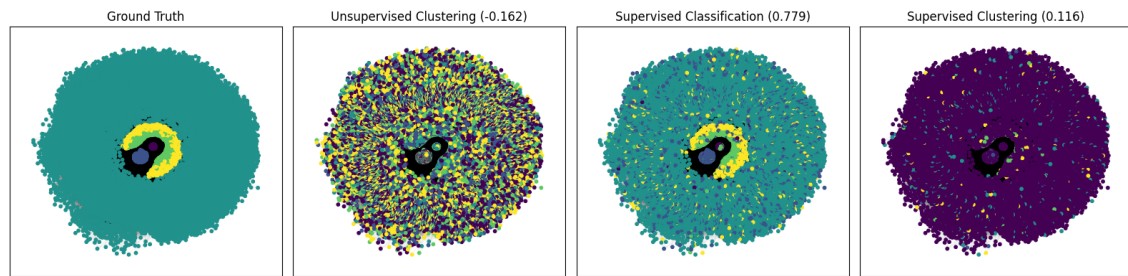

Figure 29: Visualisations of graph labellings for the Coauthor Physics dataset, default split with 20 train nodes per class and 500 validation nodes. Model selection based on training loss.

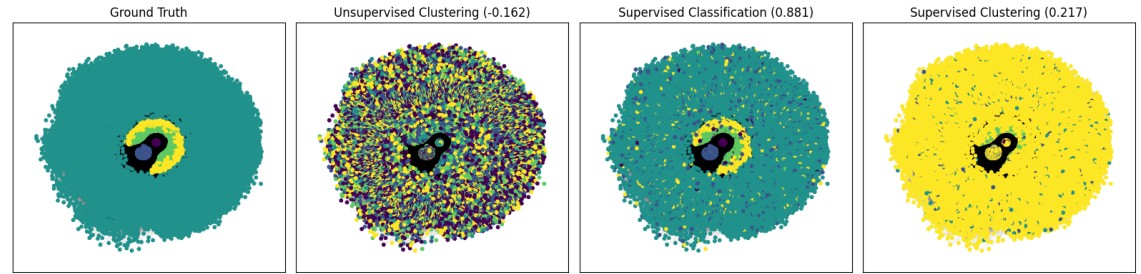

Figure 30: Visualisations of graph labellings for the Coauthor Physics dataset, default split with 20 train nodes per class and 500 validation nodes. Model selection based on training set MCC.

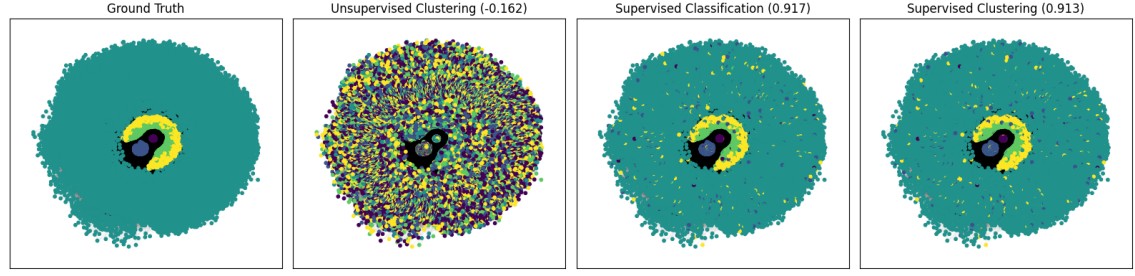

Figure 31: Visualisations of graph labellings for the Coauthor Physics dataset, default split with 20 train nodes per class and 500 validation nodes. Model selection based on validation set MCC.

### F.1.10 COAUTHOR PHYSICS DATASET, SPARSE SPLIT WITH 2 TRAIN NODES PER CLASS AND 50 VALIDATION NODES.

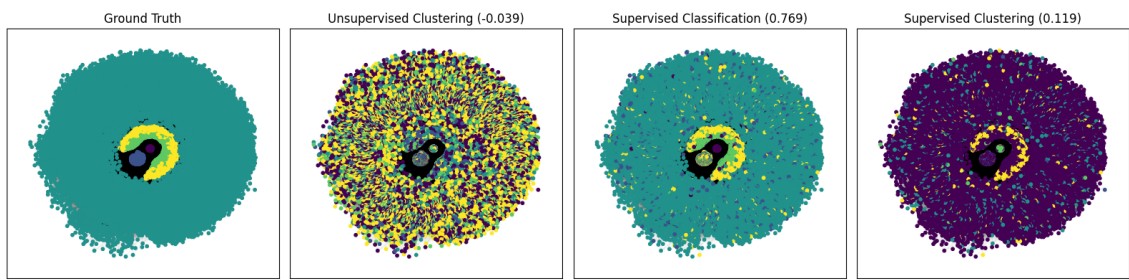

Figure 32: Visualisations of graph labellings for the Coauthor Physics dataset, sparse split with 2 train nodes per class and 50 validation nodes. Model selection based on training loss.

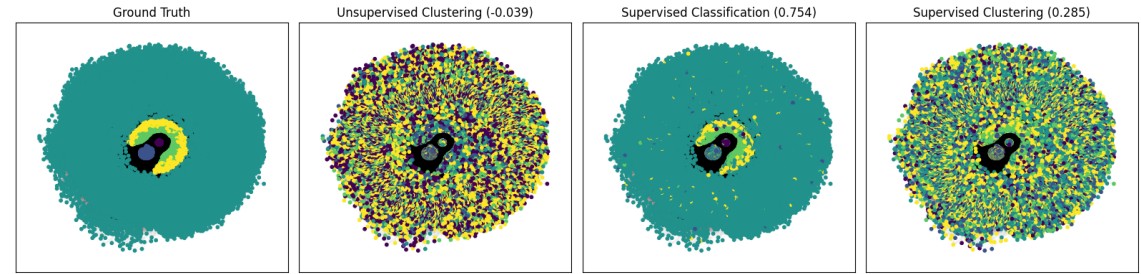

Figure 33: Visualisations of graph labellings for the Coauthor Physics dataset, sparse split with 2 train nodes per class and 50 validation nodes. Model selection based on training set MCC.

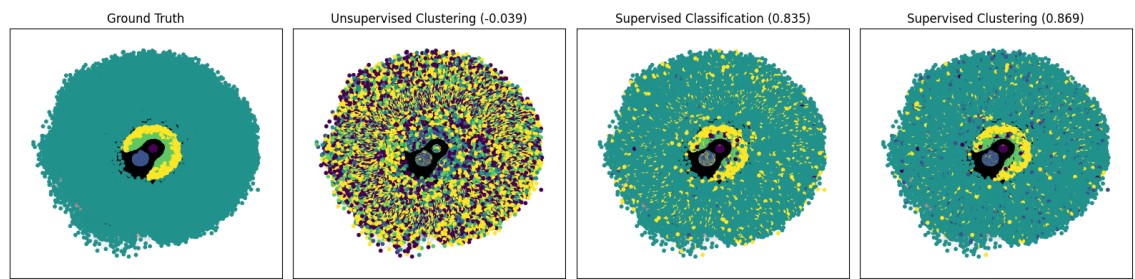

Figure 34: Visualisations of graph labellings for the Coauthor Physics dataset, sparse split with 2 train nodes per class and 50 validation nodes. Model selection based on validation set MCC.

### F.1.11 ROMAN-EMPIRE DATASET, DEFAULT SPLIT WITH DEFAULT TRAIN NODES PER CLASS AND DEFAULT VALIDATION NODES.

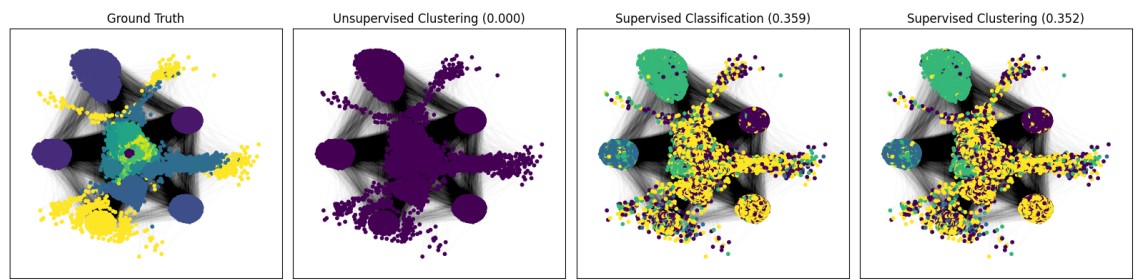

Figure 35: Visualisations of graph labellings for the roman-empire dataset, default split with default train nodes per class and default validation nodes. Model selection based on training loss.

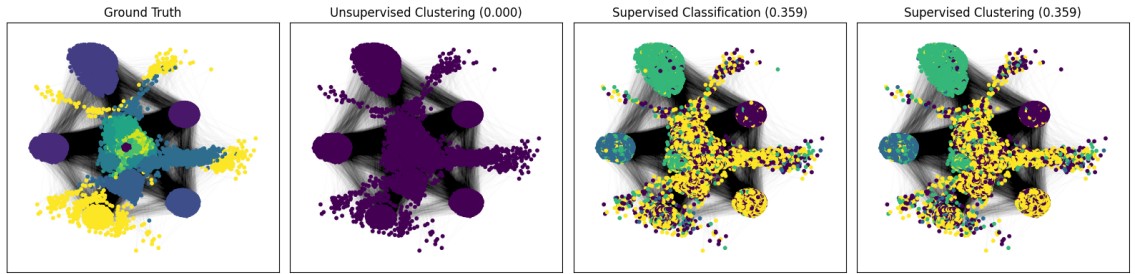

Figure 36: Visualisations of graph labellings for the roman-empire dataset, default split with default train nodes per class and default validation nodes. Model selection based on training set MCC.

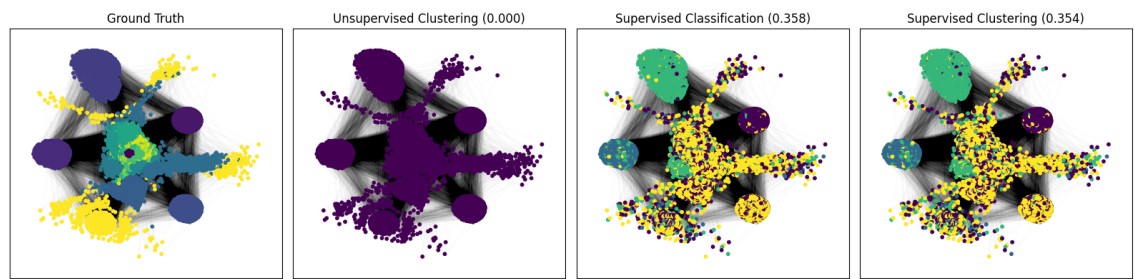

Figure 37: Visualisations of graph labellings for the roman-empire dataset, default split with default train nodes per class and default validation nodes. Model selection based on validation set MCC.

F.1.12    ROMAN−EMPIRE DATASET, SPARSE SPLIT WITH 2 TRAIN NODES PER CLASS AND 50 VALIDATION NODES.

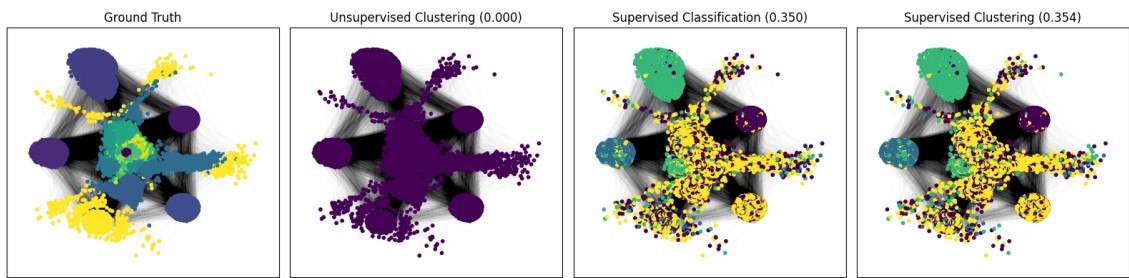

Figure 38: Visualisations of graph labellings for the roman-empire dataset, sparse split with 2 train nodes per class and 50 validation nodes. Model selection based on training loss.

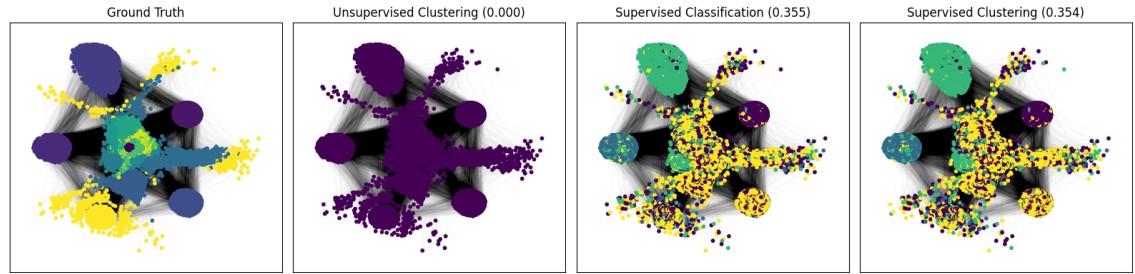

Figure 39: Visualisations of graph labellings for the roman-empire dataset, sparse split with 2 train nodes per class and 50 validation nodes. Model selection based on training set MCC.

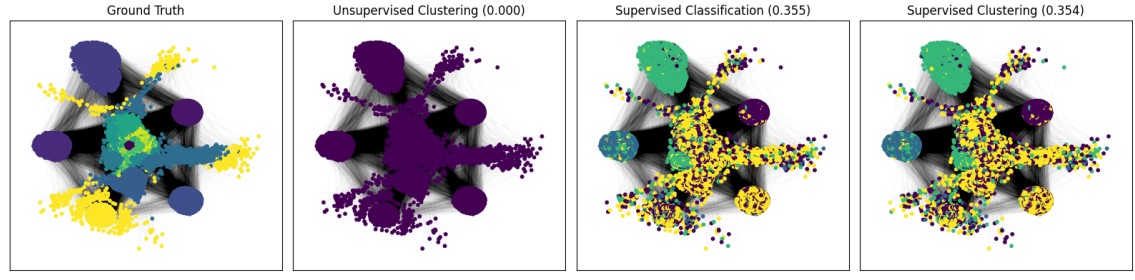

Figure 40: Visualisations of graph labellings for the roman-empire dataset, sparse split with 2 train nodes per class and 50 validation nodes. Model selection based on validation set MCC.

### F.2 Additional Summary Results

In this section we present additional summary results for both default and sparse label splits of each dataset. We compare early stopping and model selection based on training loss, training set MCC, and validation set MCC.

#### F.2.1 AMAZON-RATINGS DATASET, DEFAULT SPLIT WITH DEFAULT TRAIN NODES PER CLASS AND DEFAULT VALIDATION NODES.

Table 3: Comparing (semi-)supervised node classification with semi-supervised graph clustering. Model selection based on training loss.

| Model | $f$ | $L_{\text{regularization}}$ | $L_V$ | $\text{NN}_{S \to X}$ | ES | MCC | Accuracy |
|---|---|---|---|---|---|---|---|
| GCN | None | None | ✗ | None | Loss | **0.001 ± 0.004** | 0.368 ± 0.000 |
| GCN | DMoN | None | ✓ | MLP | Loss | 0.001 ± 0.006 | **0.368 ± 0.000** |
| GraphSAGE | None | None | ✗ | None | Loss | **0.000 ± 0.000** | **0.368 ± 0.000** |
| GraphSAGE | DMoN | None | ✗ | None | Loss | 0.000 ± 0.000 | 0.368 ± 0.000 |
| Transformer | None | None | N/A | None | Loss | **0.000 ± 0.000** | 0.354 ± 0.043 |
| Transformer | DMoN | DMoN | N/A | None | Loss | 0.000 ± 0.000 | **0.358 ± 0.032** |
| MLP | None | None | N/A | None | Loss | **0.000 ± 0.000** | **0.368 ± 0.000** |
| MLP | DMoN | DMoN | N/A | None | Loss | 0.000 ± 0.000 | 0.368 ± 0.000 |

Table 4: Comparing (semi-)supervised node classification with semi-supervised graph clustering. Model selection based on training set MCC.

| Model | $f$ | $L_{\text{regularization}}$ | $L_V$ | $\text{NN}_{S \to X}$ | ES | MCC | Accuracy |
|---|---|---|---|---|---|---|---|
| GCN | None | DMoN | ✗ | None | MCC | 0.001 ± 0.003 | 0.027 ± 0.085 |
| GCN | NOCD | None | ✗ | None | Loss | **0.028 ± 0.009** | **0.331 ± 0.020** |
| GraphSAGE | None | None | ✗ | None | Loss | 0.000 ± 0.000 | **0.368 ± 0.000** |
| GraphSAGE | $\text{SBM}_{\text{NN}}$ | L2 | ✗ | None | MCC | **0.000 ± 0.001** | 0.023 ± 0.074 |
| Transformer | None | None | N/A | None | Loss | **0.000 ± 0.000** | **0.354 ± 0.043** |
| Transformer | NOCD | None | N/A | None | Loss | 0.000 ± 0.000 | 0.308 ± 0.052 |
| MLP | None | L2 | N/A | None | MCC | **0.002 ± 0.005** | 0.077 ± 0.124 |
| MLP | Neuromap | L2 | N/A | None | MCC | 0.001 ± 0.004 | **0.090 ± 0.140** |

Table 5: Comparing (semi-)supervised node classification with semi-supervised graph clustering. Model selection based on validation set MCC.

| Model | $f$ | $L_{\text{regularization}}$ | $L_V$ | $\text{NN}_{\boldsymbol{S}\to\boldsymbol{X}}$ | ES | MCC | Accuracy |
|---|---|---|---|---|---|---|---|
| GCN | None | None | ✗ | None | MCC | $0.000 \pm 0.000$ | $0.000 \pm 0.000$ |
| GCN | NOCD | None | ✗ | None | Loss | $\mathbf{0.028 \pm 0.009}$ | $\mathbf{0.331 \pm 0.020}$ |
| GraphSAGE | None | L2 | ✗ | None | MCC | $\mathbf{-0.000 \pm 0.001}$ | $0.027 \pm 0.085$ |
| GraphSAGE | Neuromap | DMoN | ✓ | MLP | MCC | $-0.001 \pm 0.002$ | $0.005 \pm 0.015$ |
| Transformer | None | None | N/A | None | Loss | $\mathbf{0.000 \pm 0.000}$ | $\mathbf{0.354 \pm 0.043}$ |
| Transformer | NOCD | None | N/A | None | Loss | $0.000 \pm 0.000$ | $0.308 \pm 0.052$ |
| MLP | None | L2 | N/A | None | MCC | $\mathbf{0.002 \pm 0.005}$ | $\mathbf{0.077 \pm 0.124}$ |
| MLP | DMoN | L2 | N/A | None | MCC | $0.000 \pm 0.004$ | $0.036 \pm 0.075$ |

### F.2.2 AMAZON-RATINGS DATASET, SPARSE SPLIT WITH 2 TRAIN NODES PER CLASS AND 50 VALIDATION NODES.

Table 6: Comparing (semi-)supervised node classification with semi-supervised graph clustering. Model selection based on training loss.

| Model | $f$ | $L_{\text{regularization}}$ | $L_V$ | $\text{NN}_{\boldsymbol{S}\to\boldsymbol{X}}$ | ES | MCC | Accuracy |
|---|---|---|---|---|---|---|---|
| GCN | None | None | ✗ | None | Loss | $-0.001 \pm 0.001$ | $0.368 \pm 0.000$ |
| GCN | DMoN | DMoN | ✓ | MLP | Loss | $\mathbf{0.004 \pm 0.008}$ | $\mathbf{0.368 \pm 0.000}$ |
| GraphSAGE | None | None | ✗ | None | Loss | $\mathbf{0.000 \pm 0.000}$ | $\mathbf{0.368 \pm 0.000}$ |
| GraphSAGE | DMoN | DMoN | ✗ | None | Loss | $0.000 \pm 0.000$ | $0.368 \pm 0.000$ |
| Transformer | None | None | N/A | None | Loss | $\mathbf{0.000 \pm 0.000}$ | $0.358 \pm 0.032$ |
| Transformer | DMoN | DMoN | N/A | None | Loss | $0.000 \pm 0.000$ | $\mathbf{0.368 \pm 0.000}$ |
| MLP | None | None | N/A | None | Loss | $\mathbf{0.000 \pm 0.000}$ | $\mathbf{0.368 \pm 0.000}$ |
| MLP | DMoN | None | N/A | None | Loss | $0.000 \pm 0.000$ | $0.358 \pm 0.032$ |

Table 7: Comparing (semi-)supervised node classification with semi-supervised graph clustering. Model selection based on training set MCC.

| Model | $f$ | $L_{\text{regularization}}$ | $L_V$ | $\text{NN}_{\boldsymbol{S}\to\boldsymbol{X}}$ | ES | MCC | Accuracy |
|---|---|---|---|---|---|---|---|
| GCN | None | L2 | ✗ | None | MCC | $-0.003 \pm 0.030$ | $0.252 \pm 0.093$ |
| GCN | NOCD | None | ✓ | MLP | Loss | $\mathbf{0.031 \pm 0.010}$ | $\mathbf{0.344 \pm 0.006}$ |
| GraphSAGE | None | L2 | ✗ | None | MCC | $\mathbf{0.002 \pm 0.004}$ | $\mathbf{0.032 \pm 0.076}$ |
| GraphSAGE | $\text{SBM}_{\text{NN}}$ | None | ✗ | None | MCC | $-0.001 \pm 0.002$ | $0.026 \pm 0.081$ |
| Transformer | None | None | N/A | None | Loss | $\mathbf{0.000 \pm 0.000}$ | $\mathbf{0.358 \pm 0.032}$ |
| Transformer | NOCD | None | N/A | None | Loss | $0.000 \pm 0.000$ | $0.321 \pm 0.062$ |
| MLP | None | None | N/A | None | Loss | $0.000 \pm 0.000$ | $\mathbf{0.368 \pm 0.000}$ |
| MLP | NOCD | L2 | N/A | None | MCC | $\mathbf{0.000 \pm 0.001}$ | $0.063 \pm 0.134$ |

### F.2.3 COAUTHOR CS DATASET, DEFAULT SPLIT WITH 20 TRAIN NODES PER CLASS AND 500 VALIDATION NODES.

Table 8: Comparing (semi-)supervised node classification with semi-supervised graph clustering. Model selection based on training loss.

| Model | $f$ | $L_{\text{regularization}}$ | $L_V$ | NN$_{S \rightarrow X}$ | ES | MCC | Accuracy |
|---|---|---|---|---|---|---|---|
| GCN | None | None | ✗ | None | Loss | **0.875 ± 0.014** | **0.889 ± 0.013** |
| GCN | NOCD | DMoN | ✓ | Transformer | Loss | 0.319 ± 0.129 | 0.262 ± 0.159 |
| GraphSAGE | None | None | ✗ | None | Loss | **0.887 ± 0.007** | **0.899 ± 0.006** |
| GraphSAGE | NOCD | DMoN | ✗ | None | Loss | 0.217 ± 0.027 | 0.153 ± 0.022 |
| Transformer | None | None | N/A | None | Loss | **0.038 ± 0.042** | **0.100 ± 0.089** |
| Transformer | DMoN | DMoN | N/A | None | Loss | 0.014 ± 0.025 | 0.027 ± 0.017 |
| MLP | None | DMoN | N/A | None | Loss | 0.348 ± 0.033 | 0.284 ± 0.055 |
| MLP | DMoN | None | N/A | None | Loss | **0.420 ± 0.050** | **0.365 ± 0.078** |

Table 9: Comparing (semi-)supervised node classification with semi-supervised graph clustering. Model selection based on training set MCC.

| Model | $f$ | $L_{\text{regularization}}$ | $L_V$ | NN$_{S \rightarrow X}$ | ES | MCC | Accuracy |
|---|---|---|---|---|---|---|---|
| GCN | None | None | ✗ | None | Loss | **0.875 ± 0.014** | **0.889 ± 0.013** |
| GCN | DMoN | DMoN | ✗ | None | Loss | 0.867 ± 0.013 | 0.881 ± 0.012 |
| GraphSAGE | None | DMoN | ✗ | None | Loss | **0.810 ± 0.024** | **0.824 ± 0.025** |
| GraphSAGE | DMoN | DMoN | ✓ | MLP | Loss | 0.768 ± 0.020 | 0.782 ± 0.021 |
| Transformer | None | L2 | N/A | None | MCC | **0.803 ± 0.035** | **0.823 ± 0.032** |
| Transformer | DMoN | L2 | N/A | None | MCC | 0.783 ± 0.063 | 0.805 ± 0.058 |
| MLP | None | None | N/A | None | Loss | **0.859 ± 0.013** | **0.874 ± 0.012** |
| MLP | DMoN | None | N/A | None | Loss | 0.420 ± 0.050 | 0.365 ± 0.078 |

Table 10: Comparing (semi-)supervised node classification with semi-supervised graph clustering. Model selection based on validation set MCC.

| Model | $f$ | $L_{\text{regularization}}$ | $L_V$ | NN$_{S \rightarrow X}$ | ES | MCC | Accuracy |
|---|---|---|---|---|---|---|---|
| GCN | None | None | ✗ | None | MCC | **0.876 ± 0.015** | **0.889 ± 0.014** |
| GCN | DMoN | None | ✓ | Transformer | MCC | 0.875 ± 0.013 | 0.889 ± 0.012 |
| GraphSAGE | None | None | ✗ | None | MCC | **0.889 ± 0.008** | **0.901 ± 0.007** |
| GraphSAGE | DMoN | None | ✗ | None | MCC | 0.881 ± 0.015 | 0.894 ± 0.014 |
| Transformer | None | L2 | N/A | None | MCC | **0.803 ± 0.035** | **0.823 ± 0.032** |
| Transformer | DMoN | L2 | N/A | None | MCC | 0.783 ± 0.063 | 0.805 ± 0.058 |
| MLP | None | None | N/A | None | MCC | **0.862 ± 0.011** | **0.877 ± 0.011** |
| MLP | DMoN | None | N/A | None | MCC | 0.839 ± 0.013 | 0.856 ± 0.013 |

### F.2.4 COAUTHOR CS DATASET, SPARSE SPLIT WITH 2 TRAIN NODES PER CLASS AND 50 VALIDATION NODES.

Table 11: Comparing (semi-)supervised node classification with semi-supervised graph clustering. Model selection based on training loss.

| Model | $f$ | $L_{\text{regularization}}$ | $L_V$ | NN$_{\boldsymbol{S \to X}}$ | ES | MCC | Accuracy |
|---|---|---|---|---|---|---|---|
| GCN | None | None | ✗ | None | Loss | **0.742 ± 0.055** | **0.763 ± 0.054** |
| GCN | NOCD | None | ✗ | None | Loss | 0.424 ± 0.086 | 0.430 ± 0.089 |
| GraphSAGE | None | None | ✗ | None | Loss | **0.727 ± 0.069** | **0.748 ± 0.067** |
| GraphSAGE | NOCD | DMoN | ✓ | Transformer | Loss | 0.296 ± 0.061 | 0.287 ± 0.059 |
| Transformer | None | DMoN | N/A | None | Loss | 0.002 ± 0.005 | 0.070 ± 0.065 |
| Transformer | DMoN | DMoN | N/A | None | Loss | **0.013 ± 0.031** | **0.075 ± 0.064** |
| MLP | None | None | N/A | None | Loss | **0.566 ± 0.084** | **0.589 ± 0.101** |
| MLP | DMoN | DMoN | N/A | None | Loss | 0.475 ± 0.079 | 0.497 ± 0.084 |

Table 12: Comparing (semi-)supervised node classification with semi-supervised graph clustering. Model selection based on training set MCC.

| Model | $f$ | $L_{\text{regularization}}$ | $L_V$ | NN$_{\boldsymbol{S \to X}}$ | ES | MCC | Accuracy |
|---|---|---|---|---|---|---|---|
| GCN | None | None | ✗ | None | Loss | **0.742 ± 0.055** | **0.763 ± 0.054** |
| GCN | DMoN | None | ✓ | MLP | Loss | 0.668 ± 0.051 | 0.690 ± 0.049 |
| GraphSAGE | None | None | ✗ | None | Loss | **0.727 ± 0.069** | **0.748 ± 0.067** |
| GraphSAGE | DMoN | None | ✗ | None | Loss | 0.637 ± 0.029 | 0.654 ± 0.027 |
| Transformer | None | L2 | N/A | None | MCC | 0.476 ± 0.120 | 0.516 ± 0.130 |
| Transformer | DMoN | L2 | N/A | None | MCC | **0.518 ± 0.104** | **0.552 ± 0.112** |
| MLP | None | None | N/A | None | Loss | **0.566 ± 0.084** | **0.589 ± 0.101** |
| MLP | DMoN | None | N/A | None | Loss | 0.467 ± 0.067 | 0.493 ± 0.077 |

### F.2.5 AMAZON COMPUTERS DATASET, DEFAULT SPLIT WITH 20 TRAIN NODES PER CLASS AND 500 VALIDATION NODES.

Table 13: Comparing (semi-)supervised node classification with semi-supervised graph clustering. Model selection based on training loss.

| Model | $f$ | $L_{\text{regularization}}$ | $L_V$ | $\text{NN}_{S \rightarrow X}$ | ES | MCC | Accuracy |
|---|---|---|---|---|---|---|---|
| GCN | None | None | ✗ | None | Loss | **0.716 ± 0.050** | **0.763 ± 0.045** |
| GCN | NOCD | None | ✓ | MLP | Loss | 0.587 ± 0.069 | 0.626 ± 0.082 |
| GraphSAGE | None | None | ✗ | None | Loss | **0.740 ± 0.021** | **0.781 ± 0.020** |
| GraphSAGE | NOCD | DMoN | ✗ | None | Loss | 0.290 ± 0.044 | 0.225 ± 0.086 |
| Transformer | None | None | N/A | None | Loss | **0.000 ± 0.000** | 0.087 ± 0.061 |
| Transformer | DMoN | DMoN | N/A | None | Loss | 0.000 ± 0.000 | **0.131 ± 0.140** |
| MLP | None | None | N/A | None | Loss | **0.547 ± 0.031** | **0.612 ± 0.032** |
| MLP | DMoN | DMoN | N/A | None | Loss | 0.431 ± 0.023 | 0.473 ± 0.029 |

Table 14: Comparing (semi-)supervised node classification with semi-supervised graph clustering. Model selection based on training set MCC.

| Model | $f$ | $L_{\text{regularization}}$ | $L_V$ | $\text{NN}_{S \rightarrow X}$ | ES | MCC | Accuracy |
|---|---|---|---|---|---|---|---|
| GCN | None | DMoN | ✗ | None | Loss | **0.723 ± 0.036** | **0.770 ± 0.032** |
| GCN | DMoN | DMoN | ✗ | None | Loss | 0.709 ± 0.041 | 0.756 ± 0.038 |
| GraphSAGE | None | None | ✗ | None | Loss | **0.740 ± 0.021** | **0.781 ± 0.020** |
| GraphSAGE | DMoN | None | ✗ | None | Loss | 0.728 ± 0.022 | 0.772 ± 0.021 |
| Transformer | None | L2 | N/A | None | MCC | 0.031 ± 0.050 | **0.234 ± 0.095** |
| Transformer | DMoN | L2 | N/A | None | MCC | **0.048 ± 0.037** | 0.193 ± 0.106 |
| MLP | None | None | N/A | None | Loss | **0.547 ± 0.031** | **0.612 ± 0.032** |
| MLP | DMoN | DMoN | N/A | None | Loss | 0.431 ± 0.023 | 0.473 ± 0.029 |

Table 15: Comparing (semi-)supervised node classification with semi-supervised graph clustering. Model selection based on validation set MCC.

| Model | $f$ | $L_{\text{regularization}}$ | $L_V$ | $\text{NN}_{S \rightarrow X}$ | ES | MCC | Accuracy |
|---|---|---|---|---|---|---|---|
| GCN | None | DMoN | ✗ | None | Loss | **0.723 ± 0.036** | **0.770 ± 0.032** |
| GCN | DMoN | DMoN | ✗ | None | Loss | 0.709 ± 0.041 | 0.756 ± 0.038 |
| GraphSAGE | None | DMoN | ✗ | None | MCC | **0.751 ± 0.020** | **0.794 ± 0.018** |
| GraphSAGE | DMoN | None | ✗ | None | MCC | 0.732 ± 0.028 | 0.777 ± 0.026 |
| Transformer | None | L2 | N/A | None | MCC | 0.031 ± 0.050 | **0.234 ± 0.095** |
| Transformer | DMoN | L2 | N/A | None | MCC | **0.048 ± 0.037** | 0.193 ± 0.106 |
| MLP | None | None | N/A | None | MCC | **0.586 ± 0.026** | **0.656 ± 0.024** |
| MLP | DMoN | None | N/A | None | MCC | 0.553 ± 0.027 | 0.616 ± 0.026 |

### F.2.6 AMAZON COMPUTERS DATASET, SPARSE SPLIT WITH 2 TRAIN NODES PER CLASS AND 50 VALIDATION NODES.

Table 16: Comparing (semi-)supervised node classification with semi-supervised graph clustering. Model selection based on training loss.

| Model | $f$ | $L_{\text{regularization}}$ | $L_V$ | NN$_{S \to X}$ | ES | MCC | Accuracy |
|---|---|---|---|---|---|---|---|
| GCN | None | DMoN | ✗ | None | Loss | **0.521 ± 0.064** | **0.580 ± 0.069** |
| GCN | NOCD | DMoN | ✓ | MLP | Loss | 0.297 ± 0.070 | 0.271 ± 0.085 |
| GraphSAGE | None | DMoN | ✗ | None | Loss | **0.466 ± 0.105** | **0.522 ± 0.114** |
| GraphSAGE | NOCD | DMoN | ✓ | Transformer | Loss | 0.210 ± 0.075 | 0.162 ± 0.096 |
| Transformer | None | None | N/A | None | Loss | **0.000 ± 0.000** | 0.075 ± 0.051 |
| Transformer | DMoN | None | N/A | None | Loss | 0.000 ± 0.000 | **0.133 ± 0.138** |
| MLP | None | DMoN | N/A | None | Loss | **0.192 ± 0.035** | **0.278 ± 0.048** |
| MLP | DMoN | DMoN | N/A | None | Loss | 0.189 ± 0.042 | 0.260 ± 0.070 |

Table 17: Comparing (semi-)supervised node classification with semi-supervised graph clustering. Model selection based on training set MCC.

| Model | $f$ | $L_{\text{regularization}}$ | $L_V$ | NN$_{S \to X}$ | ES | MCC | Accuracy |
|---|---|---|---|---|---|---|---|
| GCN | None | DMoN | ✗ | None | Loss | **0.521 ± 0.064** | **0.580 ± 0.069** |
| GCN | DMoN | DMoN | ✓ | MLP | Loss | 0.507 ± 0.065 | 0.574 ± 0.065 |
| GraphSAGE | None | DMoN | ✗ | None | Loss | **0.466 ± 0.105** | **0.522 ± 0.114** |
| GraphSAGE | DMoN | DMoN | ✗ | None | Loss | 0.453 ± 0.092 | 0.504 ± 0.100 |
| Transformer | None | L2 | N/A | None | MCC | -0.001 ± 0.030 | 0.172 ± 0.101 |
| Transformer | NOCD | L2 | N/A | None | Loss | **0.106 ± 0.093** | **0.177 ± 0.105** |
| MLP | None | DMoN | N/A | None | Loss | **0.192 ± 0.035** | **0.278 ± 0.048** |
| MLP | DMoN | DMoN | N/A | None | Loss | 0.189 ± 0.042 | 0.260 ± 0.070 |

## F.2.7 AMAZON PHOTO DATASET, DEFAULT SPLIT WITH 20 TRAIN NODES PER CLASS AND 500 VALIDATION NODES.

Table 18: Comparing (semi-)supervised node classification with semi-supervised graph clustering. Model selection based on training loss.

| Model | $f$ | $L_{\text{regularization}}$ | $L_V$ | $\text{NN}_{S \to X}$ | ES | MCC | Accuracy |
|---|---|---|---|---|---|---|---|
| GCN | None | DMoN | ✗ | None | Loss | **0.799 ± 0.075** | **0.819 ± 0.081** |
| GCN | NOCD | DMoN | ✓ | Transformer | Loss | 0.706 ± 0.089 | 0.722 ± 0.105 |
| GraphSAGE | None | DMoN | ✗ | None | Loss | **0.816 ± 0.052** | **0.838 ± 0.055** |
| GraphSAGE | NOCD | DMoN | ✓ | Transformer | Loss | 0.314 ± 0.054 | 0.251 ± 0.058 |
| Transformer | None | None | N/A | None | Loss | **0.000 ± 0.000** | 0.149 ± 0.060 |
| Transformer | DMoN | DMoN | N/A | None | Loss | 0.000 ± 0.000 | **0.154 ± 0.072** |
| MLP | None | None | N/A | None | Loss | **0.680 ± 0.021** | **0.720 ± 0.020** |
| MLP | DMoN | DMoN | N/A | None | Loss | 0.585 ± 0.053 | 0.623 ± 0.054 |

Table 19: Comparing (semi-)supervised node classification with semi-supervised graph clustering. Model selection based on training set MCC.

| Model | $f$ | $L_{\text{regularization}}$ | $L_V$ | $\text{NN}_{S \to X}$ | ES | MCC | Accuracy |
|---|---|---|---|---|---|---|---|
| GCN | None | DMoN | ✗ | None | Loss | 0.799 ± 0.075 | 0.819 ± 0.081 |
| GCN | DMoN | DMoN | ✓ | Transformer | Loss | **0.840 ± 0.031** | **0.862 ± 0.028** |
| GraphSAGE | None | DMoN | ✗ | None | Loss | 0.816 ± 0.052 | 0.838 ± 0.055 |
| GraphSAGE | DMoN | DMoN | ✓ | Transformer | Loss | **0.830 ± 0.026** | **0.853 ± 0.024** |
| Transformer | None | L2 | N/A | None | MCC | **0.093 ± 0.032** | **0.198 ± 0.054** |
| Transformer | NOCD | L2 | N/A | None | Loss | 0.071 ± 0.113 | 0.155 ± 0.110 |
| MLP | None | None | N/A | None | Loss | **0.680 ± 0.021** | **0.720 ± 0.020** |
| MLP | DMoN | None | N/A | None | Loss | 0.618 ± 0.048 | 0.652 ± 0.052 |

Table 20: Comparing (semi-)supervised node classification with semi-supervised graph clustering. Model selection based on validation set MCC.

| Model | $f$ | $L_{\text{regularization}}$ | $L_V$ | $\text{NN}_{S \to X}$ | ES | MCC | Accuracy |
|---|---|---|---|---|---|---|---|
| GCN | None | None | ✗ | None | MCC | 0.804 ± 0.069 | 0.826 ± 0.071 |
| GCN | DMoN | DMoN | ✓ | Transformer | Loss | **0.840 ± 0.031** | **0.862 ± 0.028** |
| GraphSAGE | None | None | ✗ | None | Loss | **0.830 ± 0.059** | 0.850 ± 0.062 |
| GraphSAGE | DMoN | None | ✓ | MLP | Loss | 0.828 ± 0.030 | **0.851 ± 0.027** |
| Transformer | None | L2 | N/A | None | MCC | **0.093 ± 0.032** | 0.198 ± 0.054 |
| Transformer | DMoN | L2 | N/A | None | MCC | 0.085 ± 0.068 | **0.244 ± 0.058** |
| MLP | None | None | N/A | None | MCC | **0.698 ± 0.024** | 0.739 ± 0.023 |
| MLP | DMoN | None | N/A | None | MCC | 0.697 ± 0.031 | **0.740 ± 0.027** |

### F.2.8 AMAZON PHOTO DATASET, SPARSE SPLIT WITH 2 TRAIN NODES PER CLASS AND 50 VALIDATION NODES.

Table 21: Comparing (semi-)supervised node classification with semi-supervised graph clustering. Model selection based on training loss.

| Model | $f$ | $L_{\text{regularization}}$ | $L_V$ | $\text{NN}_{S \rightarrow X}$ | ES | MCC | Accuracy |
|---|---|---|---|---|---|---|---|
| GCN | None | DMoN | ✗ | None | Loss | $\mathbf{0.644 \pm 0.088}$ | $\mathbf{0.679 \pm 0.088}$ |
| GCN | NOCD | DMoN | ✓ | MLP | Loss | $0.374 \pm 0.044$ | $0.365 \pm 0.039$ |
| GraphSAGE | None | None | ✗ | None | Loss | $\mathbf{0.640 \pm 0.068}$ | $\mathbf{0.675 \pm 0.072}$ |
| GraphSAGE | NOCD | None | ✓ | Transformer | Loss | $0.389 \pm 0.107$ | $0.402 \pm 0.110$ |
| Transformer | None | DMoN | N/A | None | Loss | $\mathbf{0.000 \pm 0.000}$ | $0.105 \pm 0.078$ |
| Transformer | DMoN | None | N/A | None | Loss | $0.000 \pm 0.000$ | $\mathbf{0.170 \pm 0.070}$ |
| MLP | None | DMoN | N/A | None | Loss | $\mathbf{0.235 \pm 0.063}$ | $0.288 \pm 0.062$ |
| MLP | DMoN | None | N/A | None | Loss | $0.235 \pm 0.049$ | $\mathbf{0.301 \pm 0.061}$ |

Table 22: Comparing (semi-)supervised node classification with semi-supervised graph clustering. Model selection based on training set MCC.

| Model | $f$ | $L_{\text{regularization}}$ | $L_V$ | $\text{NN}_{S \rightarrow X}$ | ES | MCC | Accuracy |
|---|---|---|---|---|---|---|---|
| GCN | None | DMoN | ✗ | None | Loss | $0.644 \pm 0.088$ | $0.679 \pm 0.088$ |
| GCN | DMoN | DMoN | ✗ | None | Loss | $\mathbf{0.700 \pm 0.075}$ | $\mathbf{0.736 \pm 0.068}$ |
| GraphSAGE | None | None | ✗ | None | Loss | $0.640 \pm 0.068$ | $0.675 \pm 0.072$ |
| GraphSAGE | DMoN | None | ✗ | None | Loss | $\mathbf{0.667 \pm 0.068}$ | $\mathbf{0.702 \pm 0.070}$ |
| Transformer | None | L2 | N/A | None | MCC | $0.016 \pm 0.024$ | $0.130 \pm 0.079$ |
| Transformer | NOCD | L2 | N/A | None | Loss | $\mathbf{0.155 \pm 0.138}$ | $\mathbf{0.232 \pm 0.137}$ |
| MLP | None | DMoN | N/A | None | Loss | $\mathbf{0.235 \pm 0.063}$ | $0.288 \pm 0.062$ |
| MLP | DMoN | None | N/A | None | Loss | $0.235 \pm 0.049$ | $\mathbf{0.301 \pm 0.061}$ |

### F.2.9 COAUTHOR PHYSICS DATASET, DEFAULT SPLIT WITH 20 TRAIN NODES PER CLASS AND 500 VALIDATION NODES.

Table 23: Comparing (semi-)supervised node classification with semi-supervised graph clustering. Model selection based on training loss.

| Model | $f$ | $L_{\text{regularization}}$ | $L_V$ | NN$_{S \to X}$ | ES | MCC | Accuracy |
|---|---|---|---|---|---|---|---|
| GCN | None | None | ✗ | None | Loss | **0.891 ± 0.015** | **0.925 ± 0.012** |
| GCN | NOCD | DMoN | ✓ | Transformer | Loss | 0.298 ± 0.114 | 0.363 ± 0.138 |
| GraphSAGE | None | None | ✗ | None | Loss | **0.899 ± 0.011** | **0.931 ± 0.008** |
| GraphSAGE | NOCD | DMoN | ✗ | None | Loss | 0.150 ± 0.053 | 0.212 ± 0.024 |
| Transformer | None | None | N/A | None | Loss | 0.032 ± 0.100 | 0.243 ± 0.200 |
| Transformer | DMoN | None | N/A | None | Loss | **0.034 ± 0.108** | **0.244 ± 0.194** |
| MLP | None | None | N/A | None | Loss | **0.820 ± 0.026** | **0.872 ± 0.023** |
| MLP | DMoN | DMoN | N/A | None | Loss | 0.230 ± 0.072 | 0.263 ± 0.076 |

Table 24: Comparing (semi-)supervised node classification with semi-supervised graph clustering. Model selection based on training set MCC.

| Model | $f$ | $L_{\text{regularization}}$ | $L_V$ | NN$_{S \to X}$ | ES | MCC | Accuracy |
|---|---|---|---|---|---|---|---|
| GCN | None | None | ✗ | None | Loss | **0.891 ± 0.015** | **0.925 ± 0.012** |
| GCN | DMoN | None | ✗ | None | Loss | 0.703 ± 0.031 | 0.750 ± 0.029 |
| GraphSAGE | None | None | ✗ | None | Loss | **0.899 ± 0.011** | **0.931 ± 0.008** |
| GraphSAGE | DMoN | None | ✗ | None | Loss | 0.565 ± 0.043 | 0.615 ± 0.030 |
| Transformer | None | L2 | N/A | None | MCC | **0.751 ± 0.092** | 0.821 ± 0.074 |
| Transformer | DMoN | L2 | N/A | None | MCC | 0.750 ± 0.073 | **0.829 ± 0.051** |
| MLP | None | None | N/A | None | Loss | **0.820 ± 0.026** | **0.872 ± 0.023** |
| MLP | DMoN | None | N/A | None | Loss | 0.228 ± 0.097 | 0.248 ± 0.047 |

Table 25: Comparing (semi-)supervised node classification with semi-supervised graph clustering. Model selection based on validation set MCC.

| Model | $f$ | $L_{\text{regularization}}$ | $L_V$ | NN$_{S \to X}$ | ES | MCC | Accuracy |
|---|---|---|---|---|---|---|---|
| GCN | None | DMoN | ✗ | None | MCC | **0.909 ± 0.012** | **0.938 ± 0.008** |
| GCN | DMoN | None | ✗ | None | MCC | 0.907 ± 0.008 | 0.936 ± 0.006 |
| GraphSAGE | None | DMoN | ✗ | None | MCC | **0.905 ± 0.006** | **0.935 ± 0.005** |
| GraphSAGE | DMoN | None | ✗ | None | MCC | 0.904 ± 0.007 | 0.934 ± 0.005 |
| Transformer | None | L2 | N/A | None | MCC | **0.751 ± 0.092** | 0.821 ± 0.074 |
| Transformer | DMoN | L2 | N/A | None | MCC | 0.750 ± 0.073 | **0.829 ± 0.051** |
| MLP | None | None | N/A | None | MCC | **0.819 ± 0.023** | **0.874 ± 0.016** |
| MLP | NOCD | L2 | N/A | None | Loss | 0.744 ± 0.198 | 0.826 ± 0.103 |

### F.2.10 COAUTHOR PHYSICS DATASET, SPARSE SPLIT WITH 2 TRAIN NODES PER CLASS AND 50 VALIDATION NODES.

Table 26: Comparing (semi-)supervised node classification with semi-supervised graph clustering. Model selection based on training loss.

| Model | $f$ | $L_{\text{regularization}}$ | $L_V$ | $\text{NN}_{S \to X}$ | ES | MCC | Accuracy |
|---|---|---|---|---|---|---|---|
| GCN | None | None | ✗ | None | Loss | $\mathbf{0.796 \pm 0.058}$ | $\mathbf{0.859 \pm 0.041}$ |
| GCN | NOCD | None | ✗ | None | Loss | $0.391 \pm 0.104$ | $0.434 \pm 0.122$ |
| GraphSAGE | None | None | ✗ | None | Loss | $\mathbf{0.802 \pm 0.047}$ | $\mathbf{0.864 \pm 0.034}$ |
| GraphSAGE | NOCD | DMoN | ✗ | None | Loss | $0.213 \pm 0.060$ | $0.278 \pm 0.030$ |
| Transformer | None | None | N/A | None | Loss | $\mathbf{0.023 \pm 0.081}$ | $0.159 \pm 0.124$ |
| Transformer | DMoN | None | N/A | None | Loss | $0.023 \pm 0.074$ | $\mathbf{0.199 \pm 0.154}$ |
| MLP | None | None | N/A | None | Loss | $\mathbf{0.488 \pm 0.060}$ | $\mathbf{0.604 \pm 0.089}$ |
| MLP | DMoN | DMoN | N/A | None | Loss | $0.293 \pm 0.106$ | $0.408 \pm 0.163$ |

Table 27: Comparing (semi-)supervised node classification with semi-supervised graph clustering. Model selection based on training set MCC.

| Model | $f$ | $L_{\text{regularization}}$ | $L_V$ | $\text{NN}_{S \to X}$ | ES | MCC | Accuracy |
|---|---|---|---|---|---|---|---|
| GCN | None | None | ✗ | None | Loss | $\mathbf{0.796 \pm 0.058}$ | $\mathbf{0.859 \pm 0.041}$ |
| GCN | DMoN | None | ✗ | None | Loss | $0.487 \pm 0.062$ | $0.552 \pm 0.060$ |
| GraphSAGE | None | DMoN | ✗ | None | MCC | $\mathbf{0.757 \pm 0.068}$ | $\mathbf{0.820 \pm 0.064}$ |
| GraphSAGE | DMoN | None | ✗ | None | Loss | $0.209 \pm 0.073$ | $0.350 \pm 0.054$ |
| Transformer | None | L2 | N/A | None | Loss | $0.356 \pm 0.119$ | $0.443 \pm 0.164$ |
| Transformer | NOCD | L2 | N/A | None | Loss | $\mathbf{0.542 \pm 0.124}$ | $\mathbf{0.632 \pm 0.126}$ |
| MLP | None | None | N/A | None | Loss | $\mathbf{0.488 \pm 0.060}$ | $\mathbf{0.604 \pm 0.089}$ |
| MLP | DMoN | None | N/A | None | Loss | $0.302 \pm 0.114$ | $0.396 \pm 0.145$ |

### F.2.11 ROMAN-EMPIRE DATASET, DEFAULT SPLIT WITH DEFAULT TRAIN NODES PER CLASS AND DEFAULT VALIDATION NODES.

Table 28: Comparing (semi-)supervised node classification with semi-supervised graph clustering. Model selection based on training loss.

| Model | $f$ | $L_{\text{regularization}}$ | $L_V$ | $NN_{S \to X}$ | ES | MCC | Accuracy |
|---|---|---|---|---|---|---|---|
| GCN | None | None | ✗ | None | Loss | $0.087 \pm 0.005$ | $0.191 \pm 0.006$ |
| GCN | DMoN | DMoN | ✗ | None | Loss | $\mathbf{0.088 \pm 0.007}$ | $\mathbf{0.193 \pm 0.007}$ |
| GraphSAGE | None | None | ✗ | None | Loss | $0.285 \pm 0.150$ | $0.355 \pm 0.113$ |
| GraphSAGE | DMoN | None | ✓ | MLP | Loss | $\mathbf{0.318 \pm 0.034}$ | $\mathbf{0.381 \pm 0.027}$ |
| Transformer | None | None | N/A | None | Loss | $\mathbf{0.000 \pm 0.000}$ | $0.139 \pm 0.000$ |
| Transformer | DMoN | DMoN | N/A | None | Loss | $0.000 \pm 0.000$ | $\mathbf{0.140 \pm 0.000}$ |
| MLP | None | DMoN | N/A | None | Loss | $\mathbf{0.081 \pm 0.102}$ | $\mathbf{0.183 \pm 0.062}$ |
| MLP | DMoN | None | N/A | None | Loss | $0.061 \pm 0.099$ | $0.173 \pm 0.065$ |

Table 29: Comparing (semi-)supervised node classification with semi-supervised graph clustering. Model selection based on training set MCC.

| Model | $f$ | $L_{\text{regularization}}$ | $L_V$ | $NN_{S \to X}$ | ES | MCC | Accuracy |
|---|---|---|---|---|---|---|---|
| GCN | None | None | ✗ | None | Loss | $0.087 \pm 0.005$ | $0.191 \pm 0.006$ |
| GCN | DMoN | DMoN | ✓ | MLP | Loss | $\mathbf{0.089 \pm 0.002}$ | $\mathbf{0.199 \pm 0.002}$ |
| GraphSAGE | None | None | ✗ | None | Loss | $0.285 \pm 0.150$ | $0.355 \pm 0.113$ |
| GraphSAGE | DMoN | DMoN | ✓ | MLP | Loss | $\mathbf{0.323 \pm 0.019}$ | $\mathbf{0.385 \pm 0.014}$ |
| Transformer | None | L2 | N/A | None | MCC | $0.012 \pm 0.038$ | $0.022 \pm 0.068$ |
| Transformer | NOCD | L2 | N/A | None | MCC | $\mathbf{0.013 \pm 0.039}$ | $\mathbf{0.030 \pm 0.065}$ |
| MLP | None | None | N/A | None | MCC | $0.226 \pm 0.037$ | $0.251 \pm 0.038$ |
| MLP | DMoN | DMoN | N/A | None | MCC | $\mathbf{0.245 \pm 0.043}$ | $\mathbf{0.262 \pm 0.051}$ |

Table 30: Comparing (semi-)supervised node classification with semi-supervised graph clustering. Model selection based on validation set MCC.

| Model | $f$ | $L_{\text{regularization}}$ | $L_V$ | $NN_{S \to X}$ | ES | MCC | Accuracy |
|---|---|---|---|---|---|---|---|
| GCN | None | DMoN | ✗ | None | Loss | $0.087 \pm 0.005$ | $0.192 \pm 0.005$ |
| GCN | DMoN | None | ✗ | None | Loss | $\mathbf{0.089 \pm 0.005}$ | $\mathbf{0.193 \pm 0.004}$ |
| GraphSAGE | None | None | ✗ | None | Loss | $0.285 \pm 0.150$ | $0.355 \pm 0.113$ |
| GraphSAGE | DMoN | DMoN | ✓ | MLP | Loss | $\mathbf{0.323 \pm 0.019}$ | $\mathbf{0.385 \pm 0.014}$ |
| Transformer | None | L2 | N/A | None | MCC | $0.012 \pm 0.038$ | $0.022 \pm 0.068$ |
| Transformer | NOCD | L2 | N/A | None | MCC | $\mathbf{0.013 \pm 0.039}$ | $\mathbf{0.030 \pm 0.065}$ |
| MLP | None | None | N/A | None | MCC | $0.226 \pm 0.037$ | $0.251 \pm 0.038$ |
| MLP | DMoN | DMoN | N/A | None | MCC | $\mathbf{0.245 \pm 0.043}$ | $\mathbf{0.262 \pm 0.051}$ |

### F.2.12 ROMAN-EMPIRE DATASET, SPARSE SPLIT WITH 2 TRAIN NODES PER CLASS AND 50 VALIDATION NODES.

Table 31: Comparing (semi-)supervised node classification with semi-supervised graph clustering. Model selection based on training loss.

| Model | $f$ | $L_{\text{regularization}}$ | $L_V$ | NN$_{S \rightarrow X}$ | ES | MCC | Accuracy |
|---|---|---|---|---|---|---|---|
| GCN | None | None | ✗ | None | Loss | $0.089 \pm 0.003$ | $\mathbf{0.195 \pm 0.005}$ |
| GCN | DMoN | DMoN | ✗ | None | Loss | $\mathbf{0.090 \pm 0.006}$ | $0.195 \pm 0.006$ |
| GraphSAGE | None | DMoN | ✗ | None | Loss | $0.214 \pm 0.184$ | $0.301 \pm 0.139$ |
| GraphSAGE | DMoN | DMoN | ✓ | MLP | Loss | $\mathbf{0.327 \pm 0.019}$ | $\mathbf{0.388 \pm 0.014}$ |
| Transformer | None | None | N/A | None | Loss | $\mathbf{0.000 \pm 0.000}$ | $\mathbf{0.139 \pm 0.000}$ |
| Transformer | DMoN | DMoN | N/A | None | Loss | $0.000 \pm 0.000$ | $0.139 \pm 0.001$ |
| MLP | None | DMoN | N/A | None | Loss | $0.109 \pm 0.098$ | $0.196 \pm 0.058$ |
| MLP | DMoN | None | N/A | None | Loss | $\mathbf{0.112 \pm 0.081}$ | $\mathbf{0.202 \pm 0.049}$ |

Table 32: Comparing (semi-)supervised node classification with semi-supervised graph clustering. Model selection based on training set MCC.

| Model | $f$ | $L_{\text{regularization}}$ | $L_V$ | NN$_{S \rightarrow X}$ | ES | MCC | Accuracy |
|---|---|---|---|---|---|---|---|
| GCN | None | DMoN | ✗ | None | Loss | $\mathbf{0.090 \pm 0.005}$ | $0.196 \pm 0.006$ |
| GCN | DMoN | DMoN | ✓ | MLP | Loss | $0.089 \pm 0.003$ | $\mathbf{0.198 \pm 0.002}$ |
| GraphSAGE | None | DMoN | ✗ | None | Loss | $0.214 \pm 0.184$ | $0.301 \pm 0.139$ |
| GraphSAGE | DMoN | DMoN | ✓ | MLP | Loss | $\mathbf{0.327 \pm 0.019}$ | $\mathbf{0.388 \pm 0.014}$ |
| Transformer | None | None | N/A | None | Loss | $0.000 \pm 0.000$ | $\mathbf{0.139 \pm 0.000}$ |
| Transformer | NOCD | DMoN | N/A | None | MCC | $\mathbf{0.004 \pm 0.012}$ | $0.013 \pm 0.041$ |
| MLP | None | DMoN | N/A | None | MCC | $0.222 \pm 0.034$ | $0.240 \pm 0.035$ |
| MLP | DMoN | DMoN | N/A | None | MCC | $\mathbf{0.234 \pm 0.043}$ | $\mathbf{0.261 \pm 0.041}$ |

## F.3 NO ATTRIBUTE RECONSTRUCTION VS ATTRIBUTE RECONSTRUCTION

In this section we present results an ablation of attribute reconstruction for semi-supervised graph clustering with graph neural networks GCN and GraphSAGE.

## F.3.1 AMAZON-RATINGS DATASET, DEFAULT SPLIT WITH DEFAULT TRAIN NODES PER CLASS AND DEFAULT VALIDATION NODES.

Table 33: Ablating attribute reconstruction for semi-supervised graph clustering with graph neural networks GCN and GraphSAGE. Model selection based on training loss.

| Model | $f$ | $L_{\text{regularization}}$ | $L_V$ | $\text{NN}_{S \to X}$ | ES | MCC | Accuracy |
|---|---|---|---|---|---|---|---|
| GCN | DMoN | None | ✗ | None | Loss | **0.001 ± 0.002** | 0.368 ± 0.000 |
| GCN | DMoN | None | ✓ | MLP | Loss | 0.001 ± 0.006 | **0.368 ± 0.000** |
| GraphSAGE | DMoN | None | ✗ | None | Loss | **0.000 ± 0.000** | **0.368 ± 0.000** |
| GraphSAGE | DMoN | None | ✓ | MLP | Loss | 0.000 ± 0.000 | 0.368 ± 0.000 |

Table 34: Ablating attribute reconstruction for semi-supervised graph clustering with graph neural networks GCN and GraphSAGE. Model selection based on training set MCC.

| Model | $f$ | $L_{\text{regularization}}$ | $L_V$ | $\text{NN}_{S \to X}$ | ES | MCC | Accuracy |
|---|---|---|---|---|---|---|---|
| GCN | NOCD | None | ✗ | None | Loss | 0.028 ± 0.009 | 0.331 ± 0.020 |
| GCN | NOCD | DMoN | ✓ | MLP | Loss | **0.030 ± 0.007** | **0.336 ± 0.025** |
| GraphSAGE | SBM$_{\text{NN}}$ | L2 | ✗ | None | MCC | **0.000 ± 0.001** | **0.023 ± 0.074** |
| GraphSAGE | Neuromap | DMoN | ✓ | Transformer | MCC | -0.000 ± 0.000 | 0.023 ± 0.073 |

Table 35: Ablating attribute reconstruction for semi-supervised graph clustering with graph neural networks GCN and GraphSAGE. Model selection based on validation set MCC.

| Model | $f$ | $L_{\text{regularization}}$ | $L_V$ | $\text{NN}_{S \to X}$ | ES | MCC | Accuracy |
|---|---|---|---|---|---|---|---|
| GCN | NOCD | None | ✗ | None | Loss | 0.028 ± 0.009 | 0.331 ± 0.020 |
| GCN | NOCD | None | ✓ | MLP | Loss | **0.032 ± 0.008** | **0.337 ± 0.020** |
| GraphSAGE | SBM$_{\text{NN}}$ | L2 | ✗ | None | MCC | **0.000 ± 0.001** | **0.023 ± 0.074** |
| GraphSAGE | Neuromap | DMoN | ✓ | MLP | MCC | -0.001 ± 0.002 | 0.005 ± 0.015 |

F.3.2 AMAZON-RATINGS DATASET, SPARSE SPLIT WITH 2 TRAIN NODES PER CLASS AND 50 VALIDATION NODES.

Table 36: Ablating attribute reconstruction for semi-supervised graph clustering with graph neural networks GCN and GraphSAGE. Model selection based on training loss.

| Model | $f$ | $L_{\text{regularization}}$ | $L_V$ | NN$_{S \rightarrow X}$ | ES | MCC | Accuracy |
|---|---|---|---|---|---|---|---|
| GCN | DMoN | None | ✗ | None | Loss | -0.003 ± 0.004 | 0.368 ± 0.000 |
| GCN | DMoN | DMoN | ✓ | MLP | Loss | **0.004 ± 0.008** | **0.368 ± 0.000** |
| GraphSAGE | DMoN | DMoN | ✗ | None | Loss | **0.000 ± 0.000** | **0.368 ± 0.000** |
| GraphSAGE | DMoN | None | ✓ | MLP | Loss | 0.000 ± 0.000 | 0.368 ± 0.000 |

Table 37: Ablating attribute reconstruction for semi-supervised graph clustering with graph neural networks GCN and GraphSAGE. Model selection based on training set MCC.

| Model | $f$ | $L_{\text{regularization}}$ | $L_V$ | NN$_{S \rightarrow X}$ | ES | MCC | Accuracy |
|---|---|---|---|---|---|---|---|
| GCN | NOCD | None | ✗ | None | Loss | 0.030 ± 0.008 | 0.339 ± 0.005 |
| GCN | NOCD | None | ✓ | MLP | Loss | **0.031 ± 0.010** | **0.344 ± 0.006** |
| GraphSAGE | SBM$_{\text{NN}}$ | None | ✗ | None | MCC | -0.001 ± 0.002 | 0.026 ± 0.081 |
| GraphSAGE | Neuromap | None | ✓ | MLP | MCC | **0.000 ± 0.000** | **0.037 ± 0.116** |

Table 38: Ablating attribute reconstruction for semi-supervised graph clustering with graph neural networks GCN and GraphSAGE. Model selection based on validation set MCC.

| Model | $f$ | $L_{\text{regularization}}$ | $L_V$ | NN$_{S \rightarrow X}$ | ES | MCC | Accuracy |
|---|---|---|---|---|---|---|---|
| GCN | NOCD | DMoN | ✗ | None | Loss | 0.028 ± 0.008 | 0.338 ± 0.006 |
| GCN | NOCD | None | ✓ | MLP | Loss | **0.031 ± 0.010** | **0.344 ± 0.006** |
| GraphSAGE | NOCD | L2 | ✗ | None | MCC | **0.001 ± 0.005** | 0.031 ± 0.085 |
| GraphSAGE | Neuromap | None | ✓ | MLP | MCC | 0.000 ± 0.000 | **0.037 ± 0.116** |

### F.3.3 COAUTHOR CS DATASET, DEFAULT SPLIT WITH 20 TRAIN NODES PER CLASS AND 500 VALIDATION NODES.

Table 39: Ablating attribute reconstruction for semi-supervised graph clustering with graph neural networks GCN and GraphSAGE. Model selection based on training loss.

| Model | $f$ | $L_{\text{regularization}}$ | $L_V$ | $\text{NN}_{\boldsymbol{S} \to \boldsymbol{X}}$ | ES | MCC | Accuracy |
|---|---|---|---|---|---|---|---|
| GCN | NOCD | DMoN | ✗ | None | Loss | $0.252 \pm 0.026$ | $0.183 \pm 0.025$ |
| GCN | NOCD | DMoN | ✓ | Transformer | Loss | $\mathbf{0.319 \pm 0.129}$ | $\mathbf{0.262 \pm 0.159}$ |
| GraphSAGE | NOCD | DMoN | ✗ | None | Loss | $\mathbf{0.217 \pm 0.027}$ | $0.153 \pm 0.022$ |
| GraphSAGE | NOCD | DMoN | ✓ | Transformer | Loss | $0.215 \pm 0.060$ | $\mathbf{0.157 \pm 0.054}$ |

Table 40: Ablating attribute reconstruction for semi-supervised graph clustering with graph neural networks GCN and GraphSAGE. Model selection based on training set MCC.

| Model | $f$ | $L_{\text{regularization}}$ | $L_V$ | $\text{NN}_{\boldsymbol{S} \to \boldsymbol{X}}$ | ES | MCC | Accuracy |
|---|---|---|---|---|---|---|---|
| GCN | DMoN | DMoN | ✗ | None | Loss | $0.867 \pm 0.013$ | $0.881 \pm 0.012$ |
| GCN | DMoN | None | ✓ | Transformer | Loss | $\mathbf{0.875 \pm 0.012}$ | $\mathbf{0.888 \pm 0.011}$ |
| GraphSAGE | DMoN | None | ✗ | None | Loss | $\mathbf{0.812 \pm 0.035}$ | $\mathbf{0.825 \pm 0.035}$ |
| GraphSAGE | DMoN | DMoN | ✓ | MLP | Loss | $0.768 \pm 0.020$ | $0.782 \pm 0.021$ |

Table 41: Ablating attribute reconstruction for semi-supervised graph clustering with graph neural networks GCN and GraphSAGE. Model selection based on validation set MCC.

| Model | $f$ | $L_{\text{regularization}}$ | $L_V$ | $\text{NN}_{\boldsymbol{S} \to \boldsymbol{X}}$ | ES | MCC | Accuracy |
|---|---|---|---|---|---|---|---|
| GCN | DMoN | DMoN | ✗ | None | Loss | $0.867 \pm 0.013$ | $0.881 \pm 0.012$ |
| GCN | DMoN | None | ✓ | Transformer | MCC | $\mathbf{0.875 \pm 0.013}$ | $\mathbf{0.889 \pm 0.012}$ |
| GraphSAGE | DMoN | None | ✗ | None | MCC | $\mathbf{0.881 \pm 0.015}$ | $\mathbf{0.894 \pm 0.014}$ |
| GraphSAGE | DMoN | DMoN | ✓ | Transformer | MCC | $0.877 \pm 0.013$ | $0.890 \pm 0.012$ |

### F.3.4 COAUTHOR CS DATASET, SPARSE SPLIT WITH 2 TRAIN NODES PER CLASS AND 50 VALIDATION NODES.

Table 42: Ablating attribute reconstruction for semi-supervised graph clustering with graph neural networks GCN and GraphSAGE. Model selection based on training loss.

| Model | $f$ | $L_{\text{regularization}}$ | $L_V$ | $\text{NN}_{\boldsymbol{S} \to \boldsymbol{X}}$ | ES | MCC | Accuracy |
|---|---|---|---|---|---|---|---|
| GCN | NOCD | None | ✗ | None | Loss | $0.424 \pm 0.086$ | $0.430 \pm 0.089$ |
| GCN | NOCD | None | ✓ | Transformer | Loss | $\mathbf{0.461 \pm 0.080}$ | $\mathbf{0.468 \pm 0.082}$ |
| GraphSAGE | NOCD | DMoN | ✗ | None | Loss | $\mathbf{0.327 \pm 0.042}$ | $\mathbf{0.307 \pm 0.048}$ |
| GraphSAGE | NOCD | DMoN | ✓ | Transformer | Loss | $0.296 \pm 0.061$ | $0.287 \pm 0.059$ |

Table 43: Ablating attribute reconstruction for semi-supervised graph clustering with graph neural networks GCN and GraphSAGE. Model selection based on training set MCC.

| Model | $f$ | $L_{\text{regularization}}$ | $L_V$ | $\text{NN}_{\boldsymbol{S} \to \boldsymbol{X}}$ | ES | MCC | Accuracy |
|---|---|---|---|---|---|---|---|
| GCN | DMoN | DMoN | ✗ | None | Loss | $\mathbf{0.707 \pm 0.066}$ | $\mathbf{0.729 \pm 0.061}$ |
| GCN | DMoN | None | ✓ | MLP | Loss | $0.668 \pm 0.051$ | $0.690 \pm 0.049$ |
| GraphSAGE | DMoN | None | ✗ | None | Loss | $0.637 \pm 0.029$ | $0.654 \pm 0.027$ |
| GraphSAGE | DMoN | None | ✓ | Transformer | Loss | $\mathbf{0.675 \pm 0.042}$ | $\mathbf{0.695 \pm 0.043}$ |

Table 44: Ablating attribute reconstruction for semi-supervised graph clustering with graph neural networks GCN and GraphSAGE. Model selection based on validation set MCC.

| Model | $f$ | $L_{\text{regularization}}$ | $L_V$ | $\text{NN}_{\boldsymbol{S} \to \boldsymbol{X}}$ | ES | MCC | Accuracy |
|---|---|---|---|---|---|---|---|
| GCN | DMoN | None | ✗ | None | MCC | $\mathbf{0.756 \pm 0.045}$ | $\mathbf{0.778 \pm 0.042}$ |
| GCN | DMoN | DMoN | ✓ | Transformer | MCC | $0.754 \pm 0.040$ | $0.776 \pm 0.040$ |
| GraphSAGE | DMoN | DMoN | ✗ | None | MCC | $\mathbf{0.764 \pm 0.037}$ | $\mathbf{0.784 \pm 0.037}$ |
| GraphSAGE | DMoN | DMoN | ✓ | Transformer | MCC | $0.735 \pm 0.045$ | $0.757 \pm 0.043$ |

F.3.5 AMAZON COMPUTERS DATASET, DEFAULT SPLIT WITH 20 TRAIN NODES PER CLASS AND 500 VALIDATION NODES.

Table 45: Ablating attribute reconstruction for semi-supervised graph clustering with graph neural networks GCN and GraphSAGE. Model selection based on training loss.

| Model | $f$ | $L_{\text{regularization}}$ | $L_V$ | $\text{NN}_{\boldsymbol{S} \to \boldsymbol{X}}$ | ES | MCC | Accuracy |
|---|---|---|---|---|---|---|---|
| GCN | DMoN | DMoN | ✗ | None | Loss | **0.709 ± 0.041** | **0.756 ± 0.038** |
| GCN | NOCD | None | ✓ | MLP | Loss | 0.587 ± 0.069 | 0.626 ± 0.082 |
| GraphSAGE | NOCD | DMoN | ✗ | None | Loss | **0.290 ± 0.044** | **0.225 ± 0.086** |
| GraphSAGE | NOCD | DMoN | ✓ | Transformer | Loss | 0.250 ± 0.042 | 0.175 ± 0.039 |

Table 46: Ablating attribute reconstruction for semi-supervised graph clustering with graph neural networks GCN and GraphSAGE. Model selection based on training set MCC.

| Model | $f$ | $L_{\text{regularization}}$ | $L_V$ | $\text{NN}_{\boldsymbol{S} \to \boldsymbol{X}}$ | ES | MCC | Accuracy |
|---|---|---|---|---|---|---|---|
| GCN | DMoN | DMoN | ✗ | None | Loss | **0.709 ± 0.041** | **0.756 ± 0.038** |
| GCN | DMoN | None | ✓ | Transformer | Loss | 0.706 ± 0.037 | 0.754 ± 0.030 |
| GraphSAGE | DMoN | None | ✗ | None | Loss | **0.728 ± 0.022** | **0.772 ± 0.021** |
| GraphSAGE | DMoN | DMoN | ✓ | Transformer | Loss | 0.714 ± 0.024 | 0.758 ± 0.024 |

Table 47: Ablating attribute reconstruction for semi-supervised graph clustering with graph neural networks GCN and GraphSAGE. Model selection based on validation set MCC.

| Model | $f$ | $L_{\text{regularization}}$ | $L_V$ | $\text{NN}_{\boldsymbol{S} \to \boldsymbol{X}}$ | ES | MCC | Accuracy |
|---|---|---|---|---|---|---|---|
| GCN | DMoN | DMoN | ✗ | None | Loss | **0.709 ± 0.041** | **0.756 ± 0.038** |
| GCN | DMoN | None | ✓ | Transformer | Loss | 0.706 ± 0.037 | 0.754 ± 0.030 |
| GraphSAGE | DMoN | None | ✗ | None | MCC | **0.732 ± 0.028** | **0.777 ± 0.026** |
| GraphSAGE | DMoN | DMoN | ✓ | Transformer | Loss | 0.714 ± 0.024 | 0.758 ± 0.024 |

### F.3.6 AMAZON COMPUTERS DATASET, SPARSE SPLIT WITH 2 TRAIN NODES PER CLASS AND 50 VALIDATION NODES.

Table 48: Ablating attribute reconstruction for semi-supervised graph clustering with graph neural networks GCN and GraphSAGE. Model selection based on training loss.

| Model | $f$ | $L_{\text{regularization}}$ | $L_V$ | $\text{NN}_{S \to X}$ | ES | MCC | Accuracy |
|---|---|---|---|---|---|---|---|
| GCN | NOCD | None | ✗ | None | Loss | **0.328 ± 0.094** | **0.302 ± 0.147** |
| GCN | NOCD | DMoN | ✓ | MLP | Loss | 0.297 ± 0.070 | 0.271 ± 0.085 |
| GraphSAGE | NOCD | DMoN | ✗ | None | Loss | **0.215 ± 0.102** | **0.190 ± 0.124** |
| GraphSAGE | NOCD | DMoN | ✓ | Transformer | Loss | 0.210 ± 0.075 | 0.162 ± 0.096 |

Table 49: Ablating attribute reconstruction for semi-supervised graph clustering with graph neural networks GCN and GraphSAGE. Model selection based on training set MCC.

| Model | $f$ | $L_{\text{regularization}}$ | $L_V$ | $\text{NN}_{S \to X}$ | ES | MCC | Accuracy |
|---|---|---|---|---|---|---|---|
| GCN | DMoN | None | ✗ | None | Loss | 0.476 ± 0.086 | 0.529 ± 0.097 |
| GCN | DMoN | DMoN | ✓ | MLP | Loss | **0.507 ± 0.065** | **0.574 ± 0.065** |
| GraphSAGE | DMoN | DMoN | ✗ | None | Loss | **0.453 ± 0.092** | **0.504 ± 0.100** |
| GraphSAGE | DMoN | None | ✓ | Transformer | Loss | 0.389 ± 0.061 | 0.446 ± 0.075 |

Table 50: Ablating attribute reconstruction for semi-supervised graph clustering with graph neural networks GCN and GraphSAGE. Model selection based on validation set MCC.

| Model | $f$ | $L_{\text{regularization}}$ | $L_V$ | $\text{NN}_{S \to X}$ | ES | MCC | Accuracy |
|---|---|---|---|---|---|---|---|
| GCN | DMoN | None | ✗ | None | Loss | 0.476 ± 0.086 | 0.529 ± 0.097 |
| GCN | DMoN | DMoN | ✓ | MLP | Loss | **0.507 ± 0.065** | **0.574 ± 0.065** |
| GraphSAGE | DMoN | None | ✗ | None | MCC | **0.484 ± 0.110** | **0.550 ± 0.121** |
| GraphSAGE | NOCD | DMoN | ✓ | Transformer | MCC | 0.391 ± 0.122 | 0.455 ± 0.139 |

F.3.7 AMAZON PHOTO DATASET, DEFAULT SPLIT WITH 20 TRAIN NODES PER CLASS AND 500 VALIDATION NODES.

Table 51: Ablating attribute reconstruction for semi-supervised graph clustering with graph neural networks GCN and GraphSAGE. Model selection based on training loss.

| Model | $f$ | $L_{\text{regularization}}$ | $L_V$ | $\text{NN}_{S \to X}$ | ES | MCC | Accuracy |
|---|---|---|---|---|---|---|---|
| GCN | DMoN | DMoN | ✗ | None | Loss | **0.812 ± 0.058** | **0.833 ± 0.062** |
| GCN | NOCD | DMoN | ✓ | Transformer | Loss | 0.706 ± 0.089 | 0.722 ± 0.105 |
| GraphSAGE | NOCD | DMoN | ✗ | None | Loss | 0.304 ± 0.062 | 0.241 ± 0.067 |
| GraphSAGE | NOCD | DMoN | ✓ | Transformer | Loss | **0.314 ± 0.054** | **0.251 ± 0.058** |

Table 52: Ablating attribute reconstruction for semi-supervised graph clustering with graph neural networks GCN and GraphSAGE. Model selection based on training set MCC.

| Model | $f$ | $L_{\text{regularization}}$ | $L_V$ | $\text{NN}_{S \to X}$ | ES | MCC | Accuracy |
|---|---|---|---|---|---|---|---|
| GCN | DMoN | DMoN | ✗ | None | Loss | 0.812 ± 0.058 | 0.833 ± 0.062 |
| GCN | DMoN | DMoN | ✓ | Transformer | Loss | **0.840 ± 0.031** | **0.862 ± 0.028** |
| GraphSAGE | DMoN | None | ✗ | None | Loss | 0.794 ± 0.062 | 0.815 ± 0.067 |
| GraphSAGE | DMoN | DMoN | ✓ | Transformer | Loss | **0.830 ± 0.026** | **0.853 ± 0.024** |

Table 53: Ablating attribute reconstruction for semi-supervised graph clustering with graph neural networks GCN and GraphSAGE. Model selection based on validation set MCC.

| Model | $f$ | $L_{\text{regularization}}$ | $L_V$ | $\text{NN}_{S \to X}$ | ES | MCC | Accuracy |
|---|---|---|---|---|---|---|---|
| GCN | DMoN | DMoN | ✗ | None | MCC | 0.790 ± 0.088 | 0.809 ± 0.093 |
| GCN | DMoN | DMoN | ✓ | Transformer | Loss | **0.840 ± 0.031** | **0.862 ± 0.028** |
| GraphSAGE | DMoN | DMoN | ✗ | None | MCC | 0.794 ± 0.065 | 0.815 ± 0.070 |
| GraphSAGE | DMoN | None | ✓ | MLP | Loss | **0.828 ± 0.030** | **0.851 ± 0.027** |

### F.3.8  AMAZON PHOTO DATASET, SPARSE SPLIT WITH 2 TRAIN NODES PER CLASS AND 50 VALIDATION NODES.

Table 54: Ablating attribute reconstruction for semi-supervised graph clustering with graph neural networks GCN and GraphSAGE. Model selection based on training loss.

| Model | $f$ | $L_{\text{regularization}}$ | $L_V$ | $\text{NN}_{S \rightarrow X}$ | ES | MCC | Accuracy |
|---|---|---|---|---|---|---|---|
| GCN | NOCD | DMoN | ✗ | None | Loss | $0.333 \pm 0.112$ | $0.311 \pm 0.127$ |
| GCN | NOCD | DMoN | ✓ | MLP | Loss | $\mathbf{0.374 \pm 0.044}$ | $\mathbf{0.365 \pm 0.039}$ |
| GraphSAGE | NOCD | DMoN | ✗ | None | Loss | $0.252 \pm 0.082$ | $0.205 \pm 0.088$ |
| GraphSAGE | NOCD | None | ✓ | Transformer | Loss | $\mathbf{0.389 \pm 0.107}$ | $\mathbf{0.402 \pm 0.110}$ |

Table 55: Ablating attribute reconstruction for semi-supervised graph clustering with graph neural networks GCN and GraphSAGE. Model selection based on training set MCC.

| Model | $f$ | $L_{\text{regularization}}$ | $L_V$ | $\text{NN}_{S \rightarrow X}$ | ES | MCC | Accuracy |
|---|---|---|---|---|---|---|---|
| GCN | DMoN | DMoN | ✗ | None | Loss | $\mathbf{0.700 \pm 0.075}$ | $\mathbf{0.736 \pm 0.068}$ |
| GCN | DMoN | DMoN | ✓ | Transformer | Loss | $0.657 \pm 0.068$ | $0.698 \pm 0.060$ |
| GraphSAGE | DMoN | None | ✗ | None | Loss | $\mathbf{0.667 \pm 0.068}$ | $\mathbf{0.702 \pm 0.070}$ |
| GraphSAGE | DMoN | None | ✓ | Transformer | Loss | $0.594 \pm 0.110$ | $0.639 \pm 0.099$ |

Table 56: Ablating attribute reconstruction for semi-supervised graph clustering with graph neural networks GCN and GraphSAGE. Model selection based on validation set MCC.

| Model | $f$ | $L_{\text{regularization}}$ | $L_V$ | $\text{NN}_{S \rightarrow X}$ | ES | MCC | Accuracy |
|---|---|---|---|---|---|---|---|
| GCN | DMoN | None | ✗ | None | MCC | $0.618 \pm 0.093$ | $0.659 \pm 0.098$ |
| GCN | DMoN | DMoN | ✓ | Transformer | Loss | $\mathbf{0.657 \pm 0.068}$ | $\mathbf{0.698 \pm 0.060}$ |
| GraphSAGE | DMoN | DMoN | ✗ | None | MCC | $0.571 \pm 0.102$ | $0.618 \pm 0.105$ |
| GraphSAGE | DMoN | None | ✓ | Transformer | Loss | $\mathbf{0.594 \pm 0.110}$ | $\mathbf{0.639 \pm 0.099}$ |

F.3.9 COAUTHOR PHYSICS DATASET, DEFAULT SPLIT WITH 20 TRAIN NODES PER CLASS AND 500 VALIDATION NODES.

Table 57: Ablating attribute reconstruction for semi-supervised graph clustering with graph neural networks GCN and GraphSAGE. Model selection based on training loss.

| Model | $f$ | $L_{\text{regularization}}$ | $L_V$ | $\text{NN}_{S \to X}$ | ES | MCC | Accuracy |
|-------|-----|------|-------|-------|-----|-----|----------|
| GCN | NOCD | DMoN | ✗ | None | Loss | $0.188 \pm 0.068$ | $0.249 \pm 0.064$ |
| GCN | NOCD | DMoN | ✓ | Transformer | Loss | $\mathbf{0.298 \pm 0.114}$ | $\mathbf{0.363 \pm 0.138}$ |
| GraphSAGE | NOCD | DMoN | ✗ | None | Loss | $0.150 \pm 0.053$ | $0.212 \pm 0.024$ |
| GraphSAGE | NOCD | DMoN | ✓ | Transformer | Loss | $\mathbf{0.189 \pm 0.064}$ | $\mathbf{0.243 \pm 0.047}$ |

Table 58: Ablating attribute reconstruction for semi-supervised graph clustering with graph neural networks GCN and GraphSAGE. Model selection based on training set MCC.

| Model | $f$ | $L_{\text{regularization}}$ | $L_V$ | $\text{NN}_{S \to X}$ | ES | MCC | Accuracy |
|-------|-----|------|-------|-------|-----|-----|----------|
| GCN | DMoN | None | ✗ | None | Loss | $\mathbf{0.703 \pm 0.031}$ | $\mathbf{0.750 \pm 0.029}$ |
| GCN | DMoN | DMoN | ✓ | MLP | Loss | $0.686 \pm 0.022$ | $0.733 \pm 0.022$ |
| GraphSAGE | DMoN | None | ✗ | None | Loss | $0.565 \pm 0.043$ | $0.615 \pm 0.030$ |
| GraphSAGE | DMoN | None | ✓ | Transformer | Loss | $\mathbf{0.646 \pm 0.069}$ | $\mathbf{0.689 \pm 0.068}$ |

Table 59: Ablating attribute reconstruction for semi-supervised graph clustering with graph neural networks GCN and GraphSAGE. Model selection based on validation set MCC.

| Model | $f$ | $L_{\text{regularization}}$ | $L_V$ | $\text{NN}_{S \to X}$ | ES | MCC | Accuracy |
|-------|-----|------|-------|-------|-----|-----|----------|
| GCN | DMoN | None | ✗ | None | MCC | $\mathbf{0.907 \pm 0.008}$ | $\mathbf{0.936 \pm 0.006}$ |
| GCN | DMoN | DMoN | ✓ | Transformer | MCC | $0.902 \pm 0.014$ | $0.933 \pm 0.010$ |
| GraphSAGE | DMoN | None | ✗ | None | MCC | $\mathbf{0.904 \pm 0.007}$ | $\mathbf{0.934 \pm 0.005}$ |
| GraphSAGE | DMoN | DMoN | ✓ | MLP | MCC | $0.899 \pm 0.006$ | $0.931 \pm 0.004$ |

### F.3.10 COAUTHOR PHYSICS DATASET, SPARSE SPLIT WITH 2 TRAIN NODES PER CLASS AND 50 VALIDATION NODES.

Table 60: Ablating attribute reconstruction for semi-supervised graph clustering with graph neural networks GCN and GraphSAGE. Model selection based on training loss.

| Model | $f$ | $L_{\text{regularization}}$ | $L_V$ | NN$_{S \to X}$ | ES | MCC | Accuracy |
|-------|-----|------------------------------|-------|----------------|-----|-----|----------|
| GCN | NOCD | None | ✗ | None | Loss | $\mathbf{0.391 \pm 0.104}$ | $0.434 \pm 0.122$ |
| GCN | NOCD | None | ✓ | Transformer | Loss | $0.366 \pm 0.156$ | $\mathbf{0.452 \pm 0.143}$ |
| GraphSAGE | NOCD | DMoN | ✗ | None | Loss | $0.213 \pm 0.060$ | $0.278 \pm 0.030$ |
| GraphSAGE | NOCD | DMoN | ✓ | Transformer | Loss | $\mathbf{0.261 \pm 0.117}$ | $\mathbf{0.327 \pm 0.100}$ |

Table 61: Ablating attribute reconstruction for semi-supervised graph clustering with graph neural networks GCN and GraphSAGE. Model selection based on training set MCC.

| Model | $f$ | $L_{\text{regularization}}$ | $L_V$ | NN$_{S \to X}$ | ES | MCC | Accuracy |
|-------|-----|------------------------------|-------|----------------|-----|-----|----------|
| GCN | DMoN | None | ✗ | None | Loss | $0.487 \pm 0.062$ | $0.552 \pm 0.060$ |
| GCN | DMoN | None | ✓ | Transformer | Loss | $\mathbf{0.636 \pm 0.081}$ | $\mathbf{0.704 \pm 0.074}$ |
| GraphSAGE | DMoN | None | ✗ | None | Loss | $0.209 \pm 0.073$ | $0.350 \pm 0.054$ |
| GraphSAGE | DMoN | None | ✓ | Transformer | Loss | $\mathbf{0.450 \pm 0.095}$ | $\mathbf{0.541 \pm 0.069}$ |

Table 62: Ablating attribute reconstruction for semi-supervised graph clustering with graph neural networks GCN and GraphSAGE. Model selection based on validation set MCC.

| Model | $f$ | $L_{\text{regularization}}$ | $L_V$ | NN$_{S \to X}$ | ES | MCC | Accuracy |
|-------|-----|------------------------------|-------|----------------|-----|-----|----------|
| GCN | DMoN | None | ✗ | None | MCC | $\mathbf{0.830 \pm 0.042}$ | $\mathbf{0.882 \pm 0.030}$ |
| GCN | DMoN | None | ✓ | Transformer | MCC | $0.761 \pm 0.074$ | $0.832 \pm 0.057$ |
| GraphSAGE | DMoN | None | ✗ | None | MCC | $\mathbf{0.792 \pm 0.047}$ | $\mathbf{0.855 \pm 0.034}$ |
| GraphSAGE | DMoN | DMoN | ✓ | Transformer | MCC | $0.791 \pm 0.040$ | $0.853 \pm 0.026$ |

F.3.11 `ROMAN-EMPIRE` DATASET, DEFAULT SPLIT WITH DEFAULT TRAIN NODES PER CLASS AND DEFAULT VALIDATION NODES.

Table 63: Ablating attribute reconstruction for semi-supervised graph clustering with graph neural networks GCN and GraphSAGE. Model selection based on training loss.

| Model | $f$ | $L_{\text{regularization}}$ | $L_V$ | NN$_{S \to X}$ | ES | MCC | Accuracy |
|---|---|---|---|---|---|---|---|
| GCN | DMoN | DMoN | ✗ | None | Loss | $0.088 \pm 0.007$ | $0.193 \pm 0.007$ |
| GCN | DMoN | DMoN | ✓ | MLP | Loss | $\mathbf{0.089 \pm 0.002}$ | $\mathbf{0.199 \pm 0.002}$ |
| GraphSAGE | DMoN | None | ✗ | None | Loss | $0.177 \pm 0.186$ | $0.274 \pm 0.141$ |
| GraphSAGE | DMoN | None | ✓ | MLP | Loss | $\mathbf{0.318 \pm 0.034}$ | $\mathbf{0.381 \pm 0.027}$ |

Table 64: Ablating attribute reconstruction for semi-supervised graph clustering with graph neural networks GCN and GraphSAGE. Model selection based on training set MCC.

| Model | $f$ | $L_{\text{regularization}}$ | $L_V$ | NN$_{S \to X}$ | ES | MCC | Accuracy |
|---|---|---|---|---|---|---|---|
| GCN | DMoN | DMoN | ✗ | None | Loss | $0.088 \pm 0.007$ | $0.193 \pm 0.007$ |
| GCN | DMoN | DMoN | ✓ | MLP | Loss | $\mathbf{0.089 \pm 0.002}$ | $\mathbf{0.199 \pm 0.002}$ |
| GraphSAGE | NOCD | None | ✗ | None | Loss | $0.213 \pm 0.046$ | $0.224 \pm 0.047$ |
| GraphSAGE | DMoN | DMoN | ✓ | MLP | Loss | $\mathbf{0.323 \pm 0.019}$ | $\mathbf{0.385 \pm 0.014}$ |

Table 65: Ablating attribute reconstruction for semi-supervised graph clustering with graph neural networks GCN and GraphSAGE. Model selection based on validation set MCC.

| Model | $f$ | $L_{\text{regularization}}$ | $L_V$ | NN$_{S \to X}$ | ES | MCC | Accuracy |
|---|---|---|---|---|---|---|---|
| GCN | DMoN | None | ✗ | None | Loss | $0.089 \pm 0.005$ | $0.193 \pm 0.004$ |
| GCN | DMoN | DMoN | ✓ | MLP | Loss | $\mathbf{0.089 \pm 0.002}$ | $\mathbf{0.199 \pm 0.002}$ |
| GraphSAGE | NOCD | None | ✗ | None | Loss | $0.213 \pm 0.046$ | $0.224 \pm 0.047$ |
| GraphSAGE | DMoN | DMoN | ✓ | MLP | Loss | $\mathbf{0.323 \pm 0.019}$ | $\mathbf{0.385 \pm 0.014}$ |

### F.3.12 ROMAN-EMPIRE DATASET, SPARSE SPLIT WITH 2 TRAIN NODES PER CLASS AND 50 VALIDATION NODES.

Table 66: Ablating attribute reconstruction for semi-supervised graph clustering with graph neural networks GCN and GraphSAGE. Model selection based on training loss.

| Model | $f$ | $L_{\text{regularization}}$ | $L_V$ | $\text{NN}_{S \to X}$ | ES | MCC | Accuracy |
|---|---|---|---|---|---|---|---|
| GCN | DMoN | DMoN | ✗ | None | Loss | $\mathbf{0.090 \pm 0.006}$ | $0.195 \pm 0.006$ |
| GCN | DMoN | DMoN | ✓ | MLP | Loss | $0.089 \pm 0.003$ | $\mathbf{0.198 \pm 0.002}$ |
| GraphSAGE | DMoN | None | ✗ | None | Loss | $0.213 \pm 0.183$ | $0.301 \pm 0.139$ |
| GraphSAGE | DMoN | DMoN | ✓ | MLP | Loss | $\mathbf{0.327 \pm 0.019}$ | $\mathbf{0.388 \pm 0.014}$ |

Table 67: Ablating attribute reconstruction for semi-supervised graph clustering with graph neural networks GCN and GraphSAGE. Model selection based on training set MCC.

| Model | $f$ | $L_{\text{regularization}}$ | $L_V$ | $\text{NN}_{S \to X}$ | ES | MCC | Accuracy |
|---|---|---|---|---|---|---|---|
| GCN | DMoN | DMoN | ✗ | None | Loss | $\mathbf{0.090 \pm 0.006}$ | $0.195 \pm 0.006$ |
| GCN | DMoN | DMoN | ✓ | MLP | Loss | $0.089 \pm 0.003$ | $\mathbf{0.198 \pm 0.002}$ |
| GraphSAGE | DMoN | None | ✗ | None | Loss | $0.213 \pm 0.183$ | $0.301 \pm 0.139$ |
| GraphSAGE | DMoN | DMoN | ✓ | MLP | Loss | $\mathbf{0.327 \pm 0.019}$ | $\mathbf{0.388 \pm 0.014}$ |

Table 68: Ablating attribute reconstruction for semi-supervised graph clustering with graph neural networks GCN and GraphSAGE. Model selection based on validation set MCC.

| Model | $f$ | $L_{\text{regularization}}$ | $L_V$ | $\text{NN}_{S \to X}$ | ES | MCC | Accuracy |
|---|---|---|---|---|---|---|---|
| GCN | DMoN | DMoN | ✗ | None | Loss | $\mathbf{0.090 \pm 0.006}$ | $0.195 \pm 0.006$ |
| GCN | DMoN | None | ✓ | MLP | Loss | $0.089 \pm 0.005$ | $\mathbf{0.198 \pm 0.004}$ |
| GraphSAGE | DMoN | None | ✗ | None | Loss | $0.213 \pm 0.183$ | $0.301 \pm 0.139$ |
| GraphSAGE | DMoN | DMoN | ✓ | MLP | Loss | $\mathbf{0.327 \pm 0.019}$ | $\mathbf{0.388 \pm 0.014}$ |

### F.4 COMPARISON OF POOLING METHODS AND REGULARISATION

In this section we present results comparing graph clustering and regularization objectives and the effect of ablating regularization objectives. We compare clustering with regularization, clustering without regularization, and regularization without clustering for each neural network.

Our results show that unsupervised graph clustering objectives with regularization outperform regularization alone, ablating the effect of regularization.

### F.4.1 AMAZON-RATINGS DATASET, DEFAULT SPLIT WITH DEFAULT TRAIN NODES PER CLASS AND DEFAULT VALIDATION NODES.

Table 69: Comparing graph clustering and regularization objectives with an ablation study of regularization. The ablation compares clustering with clusteirng with regularization, clustering without regularization, and regularization without clustering for each neural network. Model selection based on training loss.

| Model | $f$ | $L_{\text{regularization}}$ | $L_V$ | $NN_{S \to X}$ | ES | MCC | Accuracy |
|---|---|---|---|---|---|---|---|
| GCN | None | None | ✗ | None | Loss | $0.001 \pm 0.004$ | $0.368 \pm 0.000$ |
| GCN | None | DMoN | ✗ | None | Loss | $-0.001 \pm 0.002$ | $0.368 \pm 0.000$ |
| GCN | None | L2 | ✗ | None | Loss | $0.000 \pm 0.000$ | $0.368 \pm 0.000$ |
| GCN | DMoN | None | ✓ | MLP | Loss | $0.001 \pm 0.006$ | $\mathbf{0.368 \pm 0.000}$ |
| GCN | DMoN | DMoN | ✓ | MLP | Loss | $-0.000 \pm 0.004$ | $0.368 \pm 0.000$ |
| GCN | DMoN | L2 | ✗ | None | Loss | $0.000 \pm 0.000$ | $0.368 \pm 0.000$ |
| GCN | NOCD | None | ✗ | None | Loss | $0.028 \pm 0.009$ | $0.331 \pm 0.020$ |
| GCN | NOCD | DMoN | ✗ | None | Loss | $\mathbf{0.029 \pm 0.010}$ | $0.333 \pm 0.020$ |
| GCN | NOCD | L2 | ✗ | None | Loss | $0.000 \pm 0.000$ | $0.368 \pm 0.000$ |
| GCN | Neuromap | None | ✗ | None | Loss | $0.000 \pm 0.000$ | $0.368 \pm 0.000$ |
| GCN | Neuromap | DMoN | ✗ | None | Loss | $0.000 \pm 0.000$ | $0.368 \pm 0.000$ |
| GCN | Neuromap | L2 | ✗ | None | Loss | $0.000 \pm 0.000$ | $0.368 \pm 0.000$ |
| GCN | SBM$_{NN}$ | None | ✗ | None | Loss | $0.020 \pm 0.014$ | $0.346 \pm 0.010$ |
| GCN | SBM$_{NN}$ | DMoN | ✗ | None | Loss | $0.021 \pm 0.014$ | $0.346 \pm 0.006$ |
| GCN | SBM$_{NN}$ | L2 | ✗ | None | Loss | $0.000 \pm 0.000$ | $0.368 \pm 0.000$ |
| GraphSAGE | None | None | ✗ | None | Loss | $\mathbf{0.000 \pm 0.000}$ | $\mathbf{0.368 \pm 0.000}$ |
| GraphSAGE | None | DMoN | ✗ | None | Loss | $0.000 \pm 0.000$ | $0.368 \pm 0.000$ |
| GraphSAGE | None | L2 | ✗ | None | Loss | $0.000 \pm 0.000$ | $0.368 \pm 0.000$ |
| GraphSAGE | DMoN | None | ✗ | None | Loss | $0.000 \pm 0.000$ | $0.368 \pm 0.000$ |
| GraphSAGE | DMoN | DMoN | ✗ | None | Loss | $0.000 \pm 0.000$ | $0.368 \pm 0.000$ |
| GraphSAGE | DMoN | L2 | ✗ | None | Loss | $0.000 \pm 0.000$ | $0.368 \pm 0.000$ |
| GraphSAGE | NOCD | None | ✓ | MLP | Loss | $0.000 \pm 0.000$ | $0.354 \pm 0.043$ |
| GraphSAGE | NOCD | DMoN | ✗ | None | Loss | $0.000 \pm 0.000$ | $0.368 \pm 0.000$ |
| GraphSAGE | NOCD | L2 | ✗ | None | Loss | $0.000 \pm 0.000$ | $0.368 \pm 0.000$ |
| GraphSAGE | Neuromap | None | ✗ | None | Loss | $0.000 \pm 0.000$ | $0.368 \pm 0.000$ |
| GraphSAGE | Neuromap | DMoN | ✗ | None | Loss | $0.000 \pm 0.000$ | $0.368 \pm 0.000$ |
| GraphSAGE | Neuromap | L2 | ✗ | None | Loss | $0.000 \pm 0.000$ | $0.368 \pm 0.000$ |
| GraphSAGE | SBM$_{NN}$ | None | ✓ | MLP | Loss | $0.000 \pm 0.000$ | $0.368 \pm 0.000$ |
| GraphSAGE | SBM$_{NN}$ | DMoN | ✗ | None | Loss | $0.000 \pm 0.000$ | $0.368 \pm 0.000$ |
| GraphSAGE | SBM$_{NN}$ | L2 | ✗ | None | Loss | $0.000 \pm 0.000$ | $0.368 \pm 0.000$ |
| Transformer | None | None | N/A | None | Loss | $\mathbf{0.000 \pm 0.000}$ | $0.354 \pm 0.043$ |
| Transformer | None | DMoN | N/A | None | Loss | $0.000 \pm 0.000$ | $0.358 \pm 0.032$ |
| Transformer | None | L2 | N/A | None | Loss | $0.000 \pm 0.000$ | $\mathbf{0.368 \pm 0.000}$ |
| Transformer | DMoN | None | N/A | None | Loss | $0.000 \pm 0.000$ | $0.368 \pm 0.000$ |
| Transformer | DMoN | DMoN | N/A | None | Loss | $0.000 \pm 0.000$ | $0.358 \pm 0.032$ |
| Transformer | DMoN | L2 | N/A | None | Loss | $0.000 \pm 0.000$ | $0.368 \pm 0.000$ |
| Transformer | NOCD | None | N/A | None | Loss | $0.000 \pm 0.000$ | $0.308 \pm 0.052$ |
| Transformer | NOCD | DMoN | N/A | None | Loss | $0.000 \pm 0.000$ | $0.301 \pm 0.060$ |
| Transformer | NOCD | L2 | N/A | None | Loss | $0.000 \pm 0.000$ | $0.368 \pm 0.000$ |
| Transformer | Neuromap | None | N/A | None | Loss | $0.000 \pm 0.000$ | $0.270 \pm 0.122$ |
| Transformer | Neuromap | DMoN | N/A | None | Loss | $0.000 \pm 0.000$ | $0.241 \pm 0.143$ |
| Transformer | Neuromap | L2 | N/A | None | Loss | $0.000 \pm 0.000$ | $0.368 \pm 0.000$ |

Table 69: Comparing graph clustering and regularization objectives for each neural network.

| Model | $f$ | $L_{\text{regularization}}$ | $L_V$ | NN$_{S \to X}$ | ES | MCC | Accuracy |
|---|---|---|---|---|---|---|---|
| Transformer | SBM$_{NN}$ | None | N/A | None | Loss | $0.000 \pm 0.000$ | $0.284 \pm 0.046$ |
| Transformer | SBM$_{NN}$ | DMoN | N/A | None | Loss | $0.000 \pm 0.000$ | $0.314 \pm 0.058$ |
| Transformer | SBM$_{NN}$ | L2 | N/A | None | Loss | $0.000 \pm 0.000$ | $0.368 \pm 0.000$ |
| MLP | None | None | N/A | None | Loss | $\mathbf{0.000 \pm 0.000}$ | $\mathbf{0.368 \pm 0.000}$ |
| MLP | None | DMoN | N/A | None | Loss | $0.000 \pm 0.000$ | $0.368 \pm 0.000$ |
| MLP | None | L2 | N/A | None | Loss | $0.000 \pm 0.000$ | $0.368 \pm 0.000$ |
| MLP | DMoN | None | N/A | None | Loss | $0.000 \pm 0.000$ | $0.368 \pm 0.000$ |
| MLP | DMoN | DMoN | N/A | None | Loss | $0.000 \pm 0.000$ | $0.368 \pm 0.000$ |
| MLP | DMoN | L2 | N/A | None | Loss | $0.000 \pm 0.000$ | $0.368 \pm 0.000$ |
| MLP | NOCD | None | N/A | None | Loss | $0.000 \pm 0.000$ | $0.301 \pm 0.060$ |
| MLP | NOCD | DMoN | N/A | None | Loss | $0.000 \pm 0.000$ | $0.304 \pm 0.056$ |
| MLP | NOCD | L2 | N/A | None | Loss | $0.000 \pm 0.000$ | $0.368 \pm 0.000$ |
| MLP | Neuromap | None | N/A | None | Loss | $0.000 \pm 0.000$ | $0.306 \pm 0.093$ |
| MLP | Neuromap | DMoN | N/A | None | Loss | $0.000 \pm 0.000$ | $0.314 \pm 0.058$ |
| MLP | Neuromap | L2 | N/A | None | Loss | $0.000 \pm 0.000$ | $0.368 \pm 0.000$ |
| MLP | SBM$_{NN}$ | None | N/A | None | Loss | $0.000 \pm 0.000$ | $0.288 \pm 0.042$ |
| MLP | SBM$_{NN}$ | DMoN | N/A | None | Loss | $0.000 \pm 0.000$ | $0.274 \pm 0.035$ |
| MLP | SBM$_{NN}$ | L2 | N/A | None | Loss | $0.000 \pm 0.000$ | $0.304 \pm 0.056$ |

Table 70: Comparing graph clustering and regularization objectives with an ablation study of regularization. The ablation compares clustering with clusteirng with regularization, clustering without regularization, and regularization without clustering for each neural network. Model selection based on training set MCC.

| Model | $f$ | $L_{\text{regularization}}$ | $L_V$ | NN$_{S \to X}$ | ES | MCC | Accuracy |
|---|---|---|---|---|---|---|---|
| GCN | None | None | ✗ | None | Loss | $0.001 \pm 0.004$ | $\mathbf{0.368 \pm 0.000}$ |
| GCN | None | DMoN | ✗ | None | MCC | $0.001 \pm 0.003$ | $0.027 \pm 0.085$ |
| GCN | None | L2 | ✗ | None | Loss | $0.000 \pm 0.000$ | $0.368 \pm 0.000$ |
| GCN | DMoN | None | ✓ | Transformer | MCC | $0.010 \pm 0.019$ | $0.119 \pm 0.130$ |
| GCN | DMoN | DMoN | ✗ | None | MCC | $0.000 \pm 0.000$ | $0.000 \pm 0.000$ |
| GCN | DMoN | L2 | ✗ | None | MCC | $-0.003 \pm 0.034$ | $0.269 \pm 0.074$ |
| GCN | NOCD | None | ✗ | None | Loss | $0.028 \pm 0.009$ | $0.331 \pm 0.020$ |
| GCN | NOCD | DMoN | ✗ | None | Loss | $\mathbf{0.029 \pm 0.010}$ | $0.333 \pm 0.020$ |
| GCN | NOCD | L2 | ✗ | None | MCC | $0.003 \pm 0.005$ | $0.108 \pm 0.173$ |
| GCN | Neuromap | None | ✗ | None | Loss | $0.000 \pm 0.000$ | $0.368 \pm 0.000$ |
| GCN | Neuromap | DMoN | ✗ | None | Loss | $0.000 \pm 0.000$ | $0.368 \pm 0.000$ |
| GCN | Neuromap | L2 | ✗ | None | Loss | $0.000 \pm 0.000$ | $0.368 \pm 0.000$ |
| GCN | SBM$_{NN}$ | None | ✓ | Transformer | Loss | $0.018 \pm 0.017$ | $0.355 \pm 0.014$ |
| GCN | SBM$_{NN}$ | DMoN | ✗ | None | Loss | $0.021 \pm 0.014$ | $0.346 \pm 0.006$ |
| GCN | SBM$_{NN}$ | L2 | ✗ | None | Loss | $0.000 \pm 0.000$ | $0.368 \pm 0.000$ |
| GraphSAGE | None | None | ✗ | None | Loss | $0.000 \pm 0.000$ | $\mathbf{0.368 \pm 0.000}$ |
| GraphSAGE | None | DMoN | ✗ | None | Loss | $0.000 \pm 0.000$ | $0.368 \pm 0.000$ |
| GraphSAGE | None | L2 | ✗ | None | Loss | $0.000 \pm 0.000$ | $0.368 \pm 0.000$ |
| GraphSAGE | DMoN | None | ✗ | None | Loss | $0.000 \pm 0.000$ | $0.368 \pm 0.000$ |
| GraphSAGE | DMoN | DMoN | ✗ | None | Loss | $0.000 \pm 0.000$ | $0.368 \pm 0.000$ |
| GraphSAGE | DMoN | L2 | ✗ | None | Loss | $0.000 \pm 0.000$ | $0.368 \pm 0.000$ |

Continued on next page

Table 70: Comparing graph clustering and regularization objectives for each neural network.

| Model | $f$ | $L_{\text{regularization}}$ | $L_V$ | NN$_{S \to X}$ | ES | MCC | Accuracy |
|---|---|---|---|---|---|---|---|
| GraphSAGE | NOCD | None | ✗ | None | Loss | $0.000 \pm 0.000$ | $0.368 \pm 0.000$ |
| GraphSAGE | NOCD | DMoN | ✗ | None | Loss | $0.000 \pm 0.000$ | $0.368 \pm 0.000$ |
| GraphSAGE | NOCD | L2 | ✗ | None | Loss | $0.000 \pm 0.000$ | $0.368 \pm 0.000$ |
| GraphSAGE | Neuromap | None | ✗ | None | Loss | $0.000 \pm 0.000$ | $0.368 \pm 0.000$ |
| GraphSAGE | Neuromap | DMoN | ✓ | Transformer | MCC | $-0.000 \pm 0.000$ | $0.023 \pm 0.073$ |
| GraphSAGE | Neuromap | L2 | ✗ | None | Loss | $0.000 \pm 0.000$ | $0.368 \pm 0.000$ |
| GraphSAGE | SBM$_{\text{NN}}$ | None | ✗ | None | Loss | $0.000 \pm 0.000$ | $0.368 \pm 0.000$ |
| GraphSAGE | SBM$_{\text{NN}}$ | DMoN | ✗ | None | Loss | $0.000 \pm 0.000$ | $0.348 \pm 0.042$ |
| GraphSAGE | SBM$_{\text{NN}}$ | L2 | ✗ | None | MCC | $\mathbf{0.000 \pm 0.001}$ | $0.023 \pm 0.074$ |
| Transformer | None | None | N/A | None | Loss | $\mathbf{0.000 \pm 0.000}$ | $0.354 \pm 0.043$ |
| Transformer | None | DMoN | N/A | None | Loss | $0.000 \pm 0.000$ | $0.358 \pm 0.032$ |
| Transformer | None | L2 | N/A | None | Loss | $0.000 \pm 0.000$ | $\mathbf{0.368 \pm 0.000}$ |
| Transformer | DMoN | None | N/A | None | Loss | $0.000 \pm 0.000$ | $0.368 \pm 0.000$ |
| Transformer | DMoN | DMoN | N/A | None | Loss | $0.000 \pm 0.000$ | $0.358 \pm 0.032$ |
| Transformer | DMoN | L2 | N/A | None | Loss | $0.000 \pm 0.000$ | $0.368 \pm 0.000$ |
| Transformer | NOCD | None | N/A | None | Loss | $0.000 \pm 0.000$ | $0.308 \pm 0.052$ |
| Transformer | NOCD | DMoN | N/A | None | Loss | $0.000 \pm 0.000$ | $0.301 \pm 0.060$ |
| Transformer | NOCD | L2 | N/A | None | Loss | $0.000 \pm 0.000$ | $0.368 \pm 0.000$ |
| Transformer | Neuromap | None | N/A | None | Loss | $0.000 \pm 0.000$ | $0.270 \pm 0.122$ |
| Transformer | Neuromap | DMoN | N/A | None | Loss | $0.000 \pm 0.000$ | $0.241 \pm 0.143$ |
| Transformer | Neuromap | L2 | N/A | None | Loss | $0.000 \pm 0.000$ | $0.368 \pm 0.000$ |
| Transformer | SBM$_{\text{NN}}$ | None | N/A | None | Loss | $0.000 \pm 0.000$ | $0.284 \pm 0.046$ |
| Transformer | SBM$_{\text{NN}}$ | DMoN | N/A | None | Loss | $0.000 \pm 0.000$ | $0.314 \pm 0.058$ |
| Transformer | SBM$_{\text{NN}}$ | L2 | N/A | None | Loss | $0.000 \pm 0.000$ | $0.368 \pm 0.000$ |
| MLP | None | None | N/A | None | Loss | $0.000 \pm 0.000$ | $\mathbf{0.368 \pm 0.000}$ |
| MLP | None | DMoN | N/A | None | Loss | $0.000 \pm 0.000$ | $0.368 \pm 0.000$ |
| MLP | None | L2 | N/A | None | MCC | $\mathbf{0.002 \pm 0.005}$ | $0.077 \pm 0.124$ |
| MLP | DMoN | None | N/A | None | Loss | $0.000 \pm 0.000$ | $0.368 \pm 0.000$ |
| MLP | DMoN | DMoN | N/A | None | Loss | $0.000 \pm 0.000$ | $0.368 \pm 0.000$ |
| MLP | DMoN | L2 | N/A | None | MCC | $0.000 \pm 0.004$ | $0.036 \pm 0.075$ |
| MLP | NOCD | None | N/A | None | MCC | $0.000 \pm 0.000$ | $0.023 \pm 0.074$ |
| MLP | NOCD | DMoN | N/A | None | MCC | $-0.002 \pm 0.005$ | $0.061 \pm 0.132$ |
| MLP | NOCD | L2 | N/A | None | Loss | $0.000 \pm 0.000$ | $0.368 \pm 0.000$ |
| MLP | Neuromap | None | N/A | None | Loss | $0.000 \pm 0.000$ | $0.306 \pm 0.093$ |
| MLP | Neuromap | DMoN | N/A | None | Loss | $0.000 \pm 0.000$ | $0.314 \pm 0.058$ |
| MLP | Neuromap | L2 | N/A | None | MCC | $0.001 \pm 0.004$ | $0.090 \pm 0.140$ |
| MLP | SBM$_{\text{NN}}$ | None | N/A | None | Loss | $0.000 \pm 0.000$ | $0.288 \pm 0.042$ |
| MLP | SBM$_{\text{NN}}$ | DMoN | N/A | None | Loss | $0.000 \pm 0.000$ | $0.274 \pm 0.035$ |
| MLP | SBM$_{\text{NN}}$ | L2 | N/A | None | Loss | $0.000 \pm 0.000$ | $0.304 \pm 0.056$ |

Table 71: Comparing graph clustering and regularization objectives with an ablation study of regularization. The ablation compares clustering with clusteirng with regularization, clustering without regularization, and regularization without clustering for each neural network. Model selection based on validation set MCC.

| Model | $f$ | $L_{\text{regularization}}$ | $L_V$ | NN$_{S \to X}$ | ES | MCC | Accuracy |
|---|---|---|---|---|---|---|---|
| GCN | None | None | ✗ | None | MCC | $0.000 \pm 0.000$ | $0.000 \pm 0.000$ |
| GCN | None | DMoN | ✗ | None | MCC | $0.001 \pm 0.003$ | $0.027 \pm 0.085$ |
| GCN | None | L2 | ✗ | None | Loss | $0.000 \pm 0.000$ | $\mathbf{0.368 \pm 0.000}$ |
| GCN | DMoN | None | ✓ | Transformer | MCC | $0.010 \pm 0.019$ | $0.119 \pm 0.130$ |
| GCN | DMoN | DMoN | ✗ | None | Loss | $0.000 \pm 0.000$ | $0.368 \pm 0.000$ |
| GCN | DMoN | L2 | ✗ | None | MCC | $-0.003 \pm 0.034$ | $0.269 \pm 0.074$ |
| GCN | NOCD | None | ✗ | None | Loss | $0.028 \pm 0.009$ | $0.331 \pm 0.020$ |
| GCN | NOCD | DMoN | ✗ | None | Loss | $\mathbf{0.029 \pm 0.010}$ | $0.333 \pm 0.020$ |
| GCN | NOCD | L2 | ✗ | None | MCC | $0.003 \pm 0.005$ | $0.108 \pm 0.173$ |
| GCN | Neuromap | None | ✗ | None | Loss | $0.000 \pm 0.000$ | $0.368 \pm 0.000$ |
| GCN | Neuromap | DMoN | ✓ | Transformer | MCC | $-0.002 \pm 0.023$ | $0.154 \pm 0.164$ |
| GCN | Neuromap | L2 | ✗ | None | Loss | $0.000 \pm 0.000$ | $0.368 \pm 0.000$ |
| GCN | SBM$_{\text{NN}}$ | None | ✗ | None | Loss | $0.020 \pm 0.014$ | $0.346 \pm 0.010$ |
| GCN | SBM$_{\text{NN}}$ | DMoN | ✗ | None | Loss | $0.021 \pm 0.014$ | $0.346 \pm 0.006$ |
| GCN | SBM$_{\text{NN}}$ | L2 | ✗ | None | MCC | $-0.000 \pm 0.007$ | $0.050 \pm 0.106$ |
| GraphSAGE | None | None | ✗ | None | Loss | $0.000 \pm 0.000$ | $\mathbf{0.368 \pm 0.000}$ |
| GraphSAGE | None | DMoN | ✗ | None | Loss | $0.000 \pm 0.000$ | $0.368 \pm 0.000$ |
| GraphSAGE | None | L2 | ✗ | None | MCC | $-0.000 \pm 0.001$ | $0.027 \pm 0.085$ |
| GraphSAGE | DMoN | None | ✗ | None | Loss | $0.000 \pm 0.000$ | $0.368 \pm 0.000$ |
| GraphSAGE | DMoN | DMoN | ✗ | None | MCC | $0.000 \pm 0.000$ | $0.027 \pm 0.085$ |
| GraphSAGE | DMoN | L2 | ✓ | MLP | MCC | $0.000 \pm 0.001$ | $0.073 \pm 0.154$ |
| GraphSAGE | NOCD | None | ✗ | None | Loss | $0.000 \pm 0.000$ | $0.368 \pm 0.000$ |
| GraphSAGE | NOCD | DMoN | ✗ | None | Loss | $0.000 \pm 0.000$ | $0.368 \pm 0.000$ |
| GraphSAGE | NOCD | L2 | ✓ | Transformer | MCC | $-0.000 \pm 0.000$ | $0.030 \pm 0.093$ |
| GraphSAGE | Neuromap | None | ✗ | None | Loss | $0.000 \pm 0.000$ | $0.368 \pm 0.000$ |
| GraphSAGE | Neuromap | DMoN | ✓ | MLP | MCC | $-0.001 \pm 0.002$ | $0.005 \pm 0.015$ |
| GraphSAGE | Neuromap | L2 | ✗ | None | Loss | $0.000 \pm 0.000$ | $0.368 \pm 0.000$ |
| GraphSAGE | SBM$_{\text{NN}}$ | None | ✗ | None | Loss | $0.000 \pm 0.000$ | $0.368 \pm 0.000$ |
| GraphSAGE | SBM$_{\text{NN}}$ | DMoN | ✗ | None | Loss | $0.000 \pm 0.000$ | $0.348 \pm 0.042$ |
| GraphSAGE | SBM$_{\text{NN}}$ | L2 | ✗ | None | MCC | $\mathbf{0.000 \pm 0.001}$ | $0.023 \pm 0.074$ |
| Transformer | None | None | N/A | None | Loss | $\mathbf{0.000 \pm 0.000}$ | $0.354 \pm 0.043$ |
| Transformer | None | DMoN | N/A | None | Loss | $0.000 \pm 0.000$ | $0.358 \pm 0.032$ |
| Transformer | None | L2 | N/A | None | Loss | $0.000 \pm 0.000$ | $\mathbf{0.368 \pm 0.000}$ |
| Transformer | DMoN | None | N/A | None | Loss | $0.000 \pm 0.000$ | $0.368 \pm 0.000$ |
| Transformer | DMoN | DMoN | N/A | None | Loss | $0.000 \pm 0.000$ | $0.358 \pm 0.032$ |
| Transformer | DMoN | L2 | N/A | None | Loss | $0.000 \pm 0.000$ | $0.368 \pm 0.000$ |
| Transformer | NOCD | None | N/A | None | Loss | $0.000 \pm 0.000$ | $0.308 \pm 0.052$ |
| Transformer | NOCD | DMoN | N/A | None | Loss | $0.000 \pm 0.000$ | $0.301 \pm 0.060$ |
| Transformer | NOCD | L2 | N/A | None | Loss | $0.000 \pm 0.000$ | $0.368 \pm 0.000$ |
| Transformer | Neuromap | None | N/A | None | Loss | $0.000 \pm 0.000$ | $0.270 \pm 0.122$ |
| Transformer | Neuromap | DMoN | N/A | None | Loss | $0.000 \pm 0.000$ | $0.241 \pm 0.143$ |
| Transformer | Neuromap | L2 | N/A | None | Loss | $0.000 \pm 0.000$ | $0.368 \pm 0.000$ |

Table 71: Comparing graph clustering and regularization objectives for each neural network.

| Model | $f$ | $L_{\text{regularization}}$ | $L_V$ | NN$_{S \to X}$ | ES | MCC | Accuracy |
|---|---|---|---|---|---|---|---|
| Transformer | SBM$_{\text{NN}}$ | None | N/A | None | Loss | $0.000 \pm 0.000$ | $0.284 \pm 0.046$ |
| Transformer | SBM$_{\text{NN}}$ | DMoN | N/A | None | Loss | $0.000 \pm 0.000$ | $0.314 \pm 0.058$ |
| Transformer | SBM$_{\text{NN}}$ | L2 | N/A | None | Loss | $0.000 \pm 0.000$ | $0.368 \pm 0.000$ |
| MLP | None | None | N/A | None | Loss | $0.000 \pm 0.000$ | $\mathbf{0.368 \pm 0.000}$ |
| MLP | None | DMoN | N/A | None | Loss | $0.000 \pm 0.000$ | $0.368 \pm 0.000$ |
| MLP | None | L2 | N/A | None | MCC | $\mathbf{0.002 \pm 0.005}$ | $0.077 \pm 0.124$ |
| MLP | DMoN | None | N/A | None | Loss | $0.000 \pm 0.000$ | $0.368 \pm 0.000$ |
| MLP | DMoN | DMoN | N/A | None | Loss | $0.000 \pm 0.000$ | $0.368 \pm 0.000$ |
| MLP | DMoN | L2 | N/A | None | MCC | $0.000 \pm 0.004$ | $0.036 \pm 0.075$ |
| MLP | NOCD | None | N/A | None | Loss | $0.000 \pm 0.000$ | $0.301 \pm 0.060$ |
| MLP | NOCD | DMoN | N/A | None | MCC | $-0.002 \pm 0.005$ | $0.061 \pm 0.132$ |
| MLP | NOCD | L2 | N/A | None | Loss | $0.000 \pm 0.000$ | $0.368 \pm 0.000$ |
| MLP | Neuromap | None | N/A | None | Loss | $0.000 \pm 0.000$ | $0.306 \pm 0.093$ |
| MLP | Neuromap | DMoN | N/A | None | Loss | $0.000 \pm 0.000$ | $0.314 \pm 0.058$ |
| MLP | Neuromap | L2 | N/A | None | Loss | $0.000 \pm 0.000$ | $0.368 \pm 0.000$ |
| MLP | SBM$_{\text{NN}}$ | None | N/A | None | Loss | $0.000 \pm 0.000$ | $0.288 \pm 0.042$ |
| MLP | SBM$_{\text{NN}}$ | DMoN | N/A | None | Loss | $0.000 \pm 0.000$ | $0.274 \pm 0.035$ |
| MLP | SBM$_{\text{NN}}$ | L2 | N/A | None | Loss | $0.000 \pm 0.000$ | $0.304 \pm 0.056$ |

### F.4.2 AMAZON−RATINGS DATASET, SPARSE SPLIT WITH 2 TRAIN NODES PER CLASS AND 50 VALIDATION NODES.

Table 72: Comparing graph clustering and regularization objectives with an ablation study of regularization. The ablation compares clustering with clusteirng with regularization, clustering without regularization, and regularization without clustering for each neural network. Model selection based on training loss.

| Model | $f$ | $L_{\text{regularization}}$ | $L_V$ | NN$_{S \to X}$ | ES | MCC | Accuracy |
|---|---|---|---|---|---|---|---|
| GCN | None | None | ✗ | None | Loss | $-0.001 \pm 0.001$ | $0.368 \pm 0.000$ |
| GCN | None | DMoN | ✗ | None | Loss | $-0.002 \pm 0.006$ | $0.368 \pm 0.000$ |
| GCN | None | L2 | ✗ | None | Loss | $0.000 \pm 0.000$ | $0.368 \pm 0.000$ |
| GCN | DMoN | None | ✓ | MLP | Loss | $0.000 \pm 0.004$ | $0.368 \pm 0.000$ |
| GCN | DMoN | DMoN | ✓ | MLP | Loss | $0.004 \pm 0.008$ | $\mathbf{0.368 \pm 0.000}$ |
| GCN | DMoN | L2 | ✗ | None | Loss | $0.000 \pm 0.000$ | $0.368 \pm 0.000$ |
| GCN | NOCD | None | ✗ | None | Loss | $\mathbf{0.030 \pm 0.008}$ | $0.339 \pm 0.005$ |
| GCN | NOCD | DMoN | ✗ | None | Loss | $0.028 \pm 0.008$ | $0.338 \pm 0.006$ |
| GCN | NOCD | L2 | ✗ | None | Loss | $0.000 \pm 0.000$ | $0.368 \pm 0.000$ |
| GCN | Neuromap | None | ✗ | None | Loss | $0.000 \pm 0.000$ | $0.368 \pm 0.000$ |
| GCN | Neuromap | DMoN | ✗ | None | Loss | $0.000 \pm 0.000$ | $0.368 \pm 0.000$ |
| GCN | Neuromap | L2 | ✗ | None | Loss | $0.000 \pm 0.000$ | $0.368 \pm 0.000$ |
| GCN | SBM$_{\text{NN}}$ | None | ✗ | None | Loss | $0.021 \pm 0.012$ | $0.352 \pm 0.007$ |
| GCN | SBM$_{\text{NN}}$ | DMoN | ✗ | None | Loss | $0.017 \pm 0.015$ | $0.349 \pm 0.009$ |
| GCN | SBM$_{\text{NN}}$ | L2 | ✗ | None | Loss | $0.000 \pm 0.000$ | $0.368 \pm 0.000$ |
| GraphSAGE | None | None | ✗ | None | Loss | $\mathbf{0.000 \pm 0.000}$ | $\mathbf{0.368 \pm 0.000}$ |
| GraphSAGE | None | DMoN | ✗ | None | Loss | $0.000 \pm 0.000$ | $0.368 \pm 0.000$ |
| GraphSAGE | None | L2 | ✗ | None | Loss | $0.000 \pm 0.000$ | $0.368 \pm 0.000$ |

Table 72: Comparing graph clustering and regularization objectives for each neural network.

| Model | $f$ | $L_{\text{regularization}}$ | $L_V$ | NN$_{S \to X}$ | ES | MCC | Accuracy |
|---|---|---|---|---|---|---|---|
| GraphSAGE | DMoN | None | ✗ | None | Loss | $0.000 \pm 0.000$ | $0.368 \pm 0.000$ |
| GraphSAGE | DMoN | DMoN | ✗ | None | Loss | $0.000 \pm 0.000$ | $0.368 \pm 0.000$ |
| GraphSAGE | DMoN | L2 | ✗ | None | Loss | $0.000 \pm 0.000$ | $0.368 \pm 0.000$ |
| GraphSAGE | NOCD | None | ✗ | None | Loss | $0.000 \pm 0.000$ | $0.368 \pm 0.000$ |
| GraphSAGE | NOCD | DMoN | ✗ | None | Loss | $0.000 \pm 0.000$ | $0.368 \pm 0.000$ |
| GraphSAGE | NOCD | L2 | ✗ | None | Loss | $0.000 \pm 0.000$ | $0.368 \pm 0.000$ |
| GraphSAGE | Neuromap | None | ✗ | None | Loss | $0.000 \pm 0.000$ | $0.368 \pm 0.000$ |
| GraphSAGE | Neuromap | DMoN | ✗ | None | Loss | $0.000 \pm 0.000$ | $0.358 \pm 0.032$ |
| GraphSAGE | Neuromap | L2 | ✗ | None | Loss | $0.000 \pm 0.000$ | $0.368 \pm 0.000$ |
| GraphSAGE | SBM$_{\text{NN}}$ | None | ✓ | Transformer | Loss | $0.000 \pm 0.000$ | $0.368 \pm 0.000$ |
| GraphSAGE | SBM$_{\text{NN}}$ | DMoN | ✓ | Transformer | Loss | $0.000 \pm 0.000$ | $0.368 \pm 0.000$ |
| GraphSAGE | SBM$_{\text{NN}}$ | L2 | ✓ | MLP | Loss | $0.000 \pm 0.000$ | $0.368 \pm 0.000$ |
| Transformer | None | None | N/A | None | Loss | $\mathbf{0.000 \pm 0.000}$ | $0.358 \pm 0.032$ |
| Transformer | None | DMoN | N/A | None | Loss | $0.000 \pm 0.000$ | $\mathbf{0.368 \pm 0.000}$ |
| Transformer | None | L2 | N/A | None | Loss | $0.000 \pm 0.000$ | $0.368 \pm 0.000$ |
| Transformer | DMoN | None | N/A | None | Loss | $0.000 \pm 0.000$ | $0.344 \pm 0.051$ |
| Transformer | DMoN | DMoN | N/A | None | Loss | $0.000 \pm 0.000$ | $0.368 \pm 0.000$ |
| Transformer | DMoN | L2 | N/A | None | Loss | $0.000 \pm 0.000$ | $0.368 \pm 0.000$ |
| Transformer | NOCD | None | N/A | None | Loss | $0.000 \pm 0.000$ | $0.321 \pm 0.062$ |
| Transformer | NOCD | DMoN | N/A | None | Loss | $0.000 \pm 0.000$ | $0.304 \pm 0.056$ |
| Transformer | NOCD | L2 | N/A | None | Loss | $0.000 \pm 0.000$ | $0.368 \pm 0.000$ |
| Transformer | Neuromap | None | N/A | None | Loss | $0.000 \pm 0.000$ | $0.209 \pm 0.150$ |
| Transformer | Neuromap | DMoN | N/A | None | Loss | $0.000 \pm 0.000$ | $0.233 \pm 0.142$ |
| Transformer | Neuromap | L2 | N/A | None | Loss | $0.000 \pm 0.000$ | $0.334 \pm 0.055$ |
| Transformer | SBM$_{\text{NN}}$ | None | N/A | None | Loss | $0.000 \pm 0.000$ | $0.321 \pm 0.062$ |
| Transformer | SBM$_{\text{NN}}$ | DMoN | N/A | None | Loss | $0.000 \pm 0.000$ | $0.318 \pm 0.053$ |
| Transformer | SBM$_{\text{NN}}$ | L2 | N/A | None | Loss | $0.000 \pm 0.000$ | $0.348 \pm 0.042$ |
| MLP | None | None | N/A | None | Loss | $\mathbf{0.000 \pm 0.000}$ | $\mathbf{0.368 \pm 0.000}$ |
| MLP | None | DMoN | N/A | None | Loss | $0.000 \pm 0.000$ | $0.368 \pm 0.000$ |
| MLP | None | L2 | N/A | None | Loss | $0.000 \pm 0.000$ | $0.368 \pm 0.000$ |
| MLP | DMoN | None | N/A | None | Loss | $0.000 \pm 0.000$ | $0.358 \pm 0.032$ |
| MLP | DMoN | DMoN | N/A | None | Loss | $0.000 \pm 0.000$ | $0.368 \pm 0.000$ |
| MLP | DMoN | L2 | N/A | None | Loss | $0.000 \pm 0.000$ | $0.368 \pm 0.000$ |
| MLP | NOCD | None | N/A | None | Loss | $0.000 \pm 0.000$ | $0.314 \pm 0.058$ |
| MLP | NOCD | DMoN | N/A | None | Loss | $0.000 \pm 0.000$ | $0.298 \pm 0.048$ |
| MLP | NOCD | L2 | N/A | None | Loss | $0.000 \pm 0.000$ | $0.368 \pm 0.000$ |
| MLP | Neuromap | None | N/A | None | Loss | $0.000 \pm 0.000$ | $0.242 \pm 0.119$ |
| MLP | Neuromap | DMoN | N/A | None | Loss | $0.000 \pm 0.000$ | $0.296 \pm 0.091$ |
| MLP | Neuromap | L2 | N/A | None | Loss | $0.000 \pm 0.000$ | $0.368 \pm 0.000$ |
| MLP | SBM$_{\text{NN}}$ | None | N/A | None | Loss | $0.000 \pm 0.000$ | $0.294 \pm 0.052$ |
| MLP | SBM$_{\text{NN}}$ | DMoN | N/A | None | Loss | $0.000 \pm 0.000$ | $0.288 \pm 0.042$ |
| MLP | SBM$_{\text{NN}}$ | L2 | N/A | None | Loss | $0.000 \pm 0.000$ | $0.311 \pm 0.062$ |

Table 73: Comparing graph clustering and regularization objectives with an ablation study of regularization. The ablation compares clustering with clusteirng with regularization, clustering without regularization, and regularization without clustering for each neural network. Model selection based on training set MCC.

| Model | $f$ | $L_{\text{regularization}}$ | $L_V$ | $\text{NN}_{\boldsymbol{S}\to\boldsymbol{X}}$ | ES | MCC | Accuracy |
|---|---|---|---|---|---|---|---|
| GCN | None | None | ✗ | None | MCC | $0.000 \pm 0.000$ | $0.000 \pm 0.000$ |
| GCN | None | DMoN | ✗ | None | MCC | $-0.001 \pm 0.004$ | $0.028 \pm 0.088$ |
| GCN | None | L2 | ✗ | None | MCC | $-0.003 \pm 0.030$ | $0.252 \pm 0.093$ |
| GCN | DMoN | None | ✗ | None | MCC | $0.000 \pm 0.001$ | $0.027 \pm 0.085$ |
| GCN | DMoN | DMoN | ✓ | Transformer | MCC | $0.013 \pm 0.026$ | $0.246 \pm 0.122$ |
| GCN | DMoN | L2 | ✗ | None | MCC | $0.002 \pm 0.030$ | $0.264 \pm 0.084$ |
| GCN | NOCD | None | ✓ | MLP | Loss | $\mathbf{0.031 \pm 0.010}$ | $0.344 \pm 0.006$ |
| GCN | NOCD | DMoN | ✗ | None | Loss | $0.028 \pm 0.008$ | $0.338 \pm 0.006$ |
| GCN | NOCD | L2 | ✓ | MLP | MCC | $-0.001 \pm 0.008$ | $0.064 \pm 0.136$ |
| GCN | Neuromap | None | ✓ | Transformer | MCC | $0.002 \pm 0.014$ | $0.137 \pm 0.162$ |
| GCN | Neuromap | DMoN | ✗ | None | Loss | $0.000 \pm 0.000$ | $\mathbf{0.368 \pm 0.000}$ |
| GCN | Neuromap | L2 | ✗ | None | MCC | $0.022 \pm 0.026$ | $0.268 \pm 0.064$ |
| GCN | SBM$_{\text{NN}}$ | None | ✗ | None | Loss | $0.016 \pm 0.022$ | $0.348 \pm 0.010$ |
| GCN | SBM$_{\text{NN}}$ | DMoN | ✓ | MLP | Loss | $0.021 \pm 0.020$ | $0.345 \pm 0.006$ |
| GCN | SBM$_{\text{NN}}$ | L2 | ✓ | MLP | MCC | $0.005 \pm 0.015$ | $0.148 \pm 0.159$ |
| GraphSAGE | None | None | ✗ | None | MCC | $0.001 \pm 0.004$ | $0.023 \pm 0.073$ |
| GraphSAGE | None | DMoN | ✗ | None | Loss | $0.000 \pm 0.000$ | $\mathbf{0.368 \pm 0.000}$ |
| GraphSAGE | None | L2 | ✗ | None | MCC | $\mathbf{0.002 \pm 0.004}$ | $0.032 \pm 0.076$ |
| GraphSAGE | DMoN | None | ✗ | None | Loss | $0.000 \pm 0.000$ | $0.368 \pm 0.000$ |
| GraphSAGE | DMoN | DMoN | ✓ | MLP | MCC | $-0.001 \pm 0.003$ | $0.051 \pm 0.107$ |
| GraphSAGE | DMoN | L2 | ✗ | None | Loss | $0.000 \pm 0.000$ | $0.368 \pm 0.000$ |
| GraphSAGE | NOCD | None | ✗ | None | Loss | $0.000 \pm 0.000$ | $0.368 \pm 0.000$ |
| GraphSAGE | NOCD | DMoN | ✗ | None | Loss | $0.000 \pm 0.000$ | $0.368 \pm 0.000$ |
| GraphSAGE | NOCD | L2 | ✗ | None | MCC | $0.001 \pm 0.005$ | $0.031 \pm 0.085$ |
| GraphSAGE | Neuromap | None | ✓ | MLP | MCC | $0.000 \pm 0.000$ | $0.037 \pm 0.116$ |
| GraphSAGE | Neuromap | DMoN | ✗ | None | Loss | $0.000 \pm 0.000$ | $0.358 \pm 0.032$ |
| GraphSAGE | Neuromap | L2 | ✗ | None | Loss | $0.000 \pm 0.000$ | $0.368 \pm 0.000$ |
| GraphSAGE | SBM$_{\text{NN}}$ | None | ✗ | None | MCC | $-0.001 \pm 0.002$ | $0.026 \pm 0.081$ |
| GraphSAGE | SBM$_{\text{NN}}$ | DMoN | ✗ | None | Loss | $-0.000 \pm 0.000$ | $0.334 \pm 0.055$ |
| GraphSAGE | SBM$_{\text{NN}}$ | L2 | ✗ | None | Loss | $0.000 \pm 0.000$ | $0.368 \pm 0.000$ |
| Transformer | None | None | N/A | None | Loss | $\mathbf{0.000 \pm 0.000}$ | $0.358 \pm 0.032$ |
| Transformer | None | DMoN | N/A | None | Loss | $0.000 \pm 0.000$ | $\mathbf{0.368 \pm 0.000}$ |
| Transformer | None | L2 | N/A | None | Loss | $0.000 \pm 0.000$ | $0.368 \pm 0.000$ |
| Transformer | DMoN | None | N/A | None | Loss | $0.000 \pm 0.000$ | $0.344 \pm 0.051$ |
| Transformer | DMoN | DMoN | N/A | None | Loss | $0.000 \pm 0.000$ | $0.368 \pm 0.000$ |
| Transformer | DMoN | L2 | N/A | None | Loss | $0.000 \pm 0.000$ | $0.368 \pm 0.000$ |
| Transformer | NOCD | None | N/A | None | Loss | $0.000 \pm 0.000$ | $0.321 \pm 0.062$ |
| Transformer | NOCD | DMoN | N/A | None | Loss | $0.000 \pm 0.000$ | $0.304 \pm 0.056$ |
| Transformer | NOCD | L2 | N/A | None | Loss | $0.000 \pm 0.000$ | $0.368 \pm 0.000$ |
| Transformer | Neuromap | None | N/A | None | Loss | $0.000 \pm 0.000$ | $0.209 \pm 0.150$ |
| Transformer | Neuromap | DMoN | N/A | None | Loss | $0.000 \pm 0.000$ | $0.233 \pm 0.142$ |
| Transformer | Neuromap | L2 | N/A | None | Loss | $0.000 \pm 0.000$ | $0.334 \pm 0.055$ |

Table 73: Comparing graph clustering and regularization objectives for each neural network.

| Model | $f$ | $L_{\text{regularization}}$ | $L_V$ | NN$_{S \to X}$ | ES | MCC | Accuracy |
|---|---|---|---|---|---|---|---|
| Transformer | SBM$_{\text{NN}}$ | None | N/A | None | Loss | $0.000 \pm 0.000$ | $0.321 \pm 0.062$ |
| Transformer | SBM$_{\text{NN}}$ | DMoN | N/A | None | Loss | $0.000 \pm 0.000$ | $0.318 \pm 0.053$ |
| Transformer | SBM$_{\text{NN}}$ | L2 | N/A | None | Loss | $0.000 \pm 0.000$ | $0.348 \pm 0.042$ |
| MLP | None | None | N/A | None | Loss | $0.000 \pm 0.000$ | $\mathbf{0.368 \pm 0.000}$ |
| MLP | None | DMoN | N/A | None | Loss | $0.000 \pm 0.000$ | $0.368 \pm 0.000$ |
| MLP | None | L2 | N/A | None | Loss | $0.000 \pm 0.000$ | $0.368 \pm 0.000$ |
| MLP | DMoN | None | N/A | None | Loss | $0.000 \pm 0.000$ | $0.358 \pm 0.032$ |
| MLP | DMoN | DMoN | N/A | None | Loss | $0.000 \pm 0.000$ | $0.368 \pm 0.000$ |
| MLP | DMoN | L2 | N/A | None | Loss | $0.000 \pm 0.000$ | $0.368 \pm 0.000$ |
| MLP | NOCD | None | N/A | None | Loss | $0.000 \pm 0.000$ | $0.314 \pm 0.058$ |
| MLP | NOCD | DMoN | N/A | None | Loss | $0.000 \pm 0.000$ | $0.298 \pm 0.048$ |
| MLP | NOCD | L2 | N/A | None | MCC | $\mathbf{0.000 \pm 0.001}$ | $0.063 \pm 0.134$ |
| MLP | Neuromap | None | N/A | None | Loss | $0.000 \pm 0.000$ | $0.242 \pm 0.119$ |
| MLP | Neuromap | DMoN | N/A | None | Loss | $0.000 \pm 0.000$ | $0.296 \pm 0.091$ |
| MLP | Neuromap | L2 | N/A | None | Loss | $0.000 \pm 0.000$ | $0.368 \pm 0.000$ |
| MLP | SBM$_{\text{NN}}$ | None | N/A | None | Loss | $0.000 \pm 0.000$ | $0.294 \pm 0.052$ |
| MLP | SBM$_{\text{NN}}$ | DMoN | N/A | None | Loss | $0.000 \pm 0.000$ | $0.288 \pm 0.042$ |
| MLP | SBM$_{\text{NN}}$ | L2 | N/A | None | Loss | $0.000 \pm 0.000$ | $0.311 \pm 0.062$ |

Table 74: Comparing graph clustering and regularization objectives with an ablation study of regularization. The ablation compares clustering with clusteirng with regularization, clustering without regularization, and regularization without clustering for each neural network. Model selection based on validation set MCC.

| Model | $f$ | $L_{\text{regularization}}$ | $L_V$ | NN$_{S \to X}$ | ES | MCC | Accuracy |
|---|---|---|---|---|---|---|---|
| GCN | None | None | ✗ | None | MCC | $0.000 \pm 0.000$ | $0.000 \pm 0.000$ |
| GCN | None | DMoN | ✗ | None | Loss | $-0.002 \pm 0.006$ | $0.368 \pm 0.000$ |
| GCN | None | L2 | ✗ | None | MCC | $-0.003 \pm 0.030$ | $0.252 \pm 0.093$ |
| GCN | DMoN | None | ✗ | None | MCC | $0.000 \pm 0.001$ | $0.027 \pm 0.085$ |
| GCN | DMoN | DMoN | ✓ | Transformer | MCC | $0.013 \pm 0.026$ | $0.246 \pm 0.122$ |
| GCN | DMoN | L2 | ✗ | None | MCC | $0.002 \pm 0.030$ | $0.264 \pm 0.084$ |
| GCN | NOCD | None | ✓ | MLP | Loss | $0.031 \pm 0.010$ | $0.344 \pm 0.006$ |
| GCN | NOCD | DMoN | ✓ | MLP | Loss | $\mathbf{0.032 \pm 0.008}$ | $0.331 \pm 0.029$ |
| GCN | NOCD | L2 | ✗ | None | Loss | $0.000 \pm 0.000$ | $\mathbf{0.368 \pm 0.000}$ |
| GCN | Neuromap | None | ✓ | Transformer | MCC | $0.002 \pm 0.014$ | $0.137 \pm 0.162$ |
| GCN | Neuromap | DMoN | ✗ | None | Loss | $0.000 \pm 0.000$ | $0.368 \pm 0.000$ |
| GCN | Neuromap | L2 | ✗ | None | MCC | $0.022 \pm 0.026$ | $0.268 \pm 0.064$ |
| GCN | SBM$_{\text{NN}}$ | None | ✗ | None | Loss | $0.021 \pm 0.012$ | $0.352 \pm 0.007$ |
| GCN | SBM$_{\text{NN}}$ | DMoN | ✗ | None | Loss | $0.017 \pm 0.015$ | $0.349 \pm 0.009$ |
| GCN | SBM$_{\text{NN}}$ | L2 | ✗ | None | MCC | $0.006 \pm 0.014$ | $0.064 \pm 0.136$ |
| GraphSAGE | None | None | ✗ | None | MCC | $0.001 \pm 0.004$ | $0.023 \pm 0.073$ |
| GraphSAGE | None | DMoN | ✗ | None | Loss | $0.000 \pm 0.000$ | $\mathbf{0.368 \pm 0.000}$ |
| GraphSAGE | None | L2 | ✗ | None | MCC | $\mathbf{0.002 \pm 0.004}$ | $0.032 \pm 0.076$ |
| GraphSAGE | DMoN | None | ✗ | None | Loss | $0.000 \pm 0.000$ | $0.368 \pm 0.000$ |
| GraphSAGE | DMoN | DMoN | ✗ | None | MCC | $-0.002 \pm 0.005$ | $0.036 \pm 0.114$ |
| GraphSAGE | DMoN | L2 | ✗ | None | Loss | $0.000 \pm 0.000$ | $0.368 \pm 0.000$ |

Continued on next page

Table 74: Comparing graph clustering and regularization objectives for each neural network.

| Model | $f$ | $L_{\text{regularization}}$ | $L_V$ | NN$_{\boldsymbol{S}\to\boldsymbol{X}}$ | ES | MCC | Accuracy |
|---|---|---|---|---|---|---|---|
| GraphSAGE | NOCD | None | ✗ | None | Loss | $0.000 \pm 0.000$ | $0.368 \pm 0.000$ |
| GraphSAGE | NOCD | DMoN | ✗ | None | Loss | $0.000 \pm 0.000$ | $0.368 \pm 0.000$ |
| GraphSAGE | NOCD | L2 | ✗ | None | MCC | $0.001 \pm 0.005$ | $0.031 \pm 0.085$ |
| GraphSAGE | Neuromap | None | ✓ | MLP | MCC | $0.000 \pm 0.000$ | $0.037 \pm 0.116$ |
| GraphSAGE | Neuromap | DMoN | ✗ | None | Loss | $0.000 \pm 0.000$ | $0.358 \pm 0.032$ |
| GraphSAGE | Neuromap | L2 | ✗ | None | MCC | $0.000 \pm 0.001$ | $0.027 \pm 0.084$ |
| GraphSAGE | SBM$_{\text{NN}}$ | None | ✗ | None | MCC | $-0.001 \pm 0.002$ | $0.026 \pm 0.081$ |
| GraphSAGE | SBM$_{\text{NN}}$ | DMoN | ✗ | None | Loss | $-0.000 \pm 0.000$ | $0.334 \pm 0.055$ |
| GraphSAGE | SBM$_{\text{NN}}$ | L2 | ✗ | None | Loss | $0.000 \pm 0.000$ | $0.368 \pm 0.000$ |
| Transformer | None | None | N/A | None | Loss | $\mathbf{0.000 \pm 0.000}$ | $0.358 \pm 0.032$ |
| Transformer | None | DMoN | N/A | None | Loss | $0.000 \pm 0.000$ | $\mathbf{0.368 \pm 0.000}$ |
| Transformer | None | L2 | N/A | None | Loss | $0.000 \pm 0.000$ | $0.368 \pm 0.000$ |
| Transformer | DMoN | None | N/A | None | Loss | $0.000 \pm 0.000$ | $0.344 \pm 0.051$ |
| Transformer | DMoN | DMoN | N/A | None | Loss | $0.000 \pm 0.000$ | $0.368 \pm 0.000$ |
| Transformer | DMoN | L2 | N/A | None | Loss | $0.000 \pm 0.000$ | $0.368 \pm 0.000$ |
| Transformer | NOCD | None | N/A | None | Loss | $0.000 \pm 0.000$ | $0.321 \pm 0.062$ |
| Transformer | NOCD | DMoN | N/A | None | Loss | $0.000 \pm 0.000$ | $0.304 \pm 0.056$ |
| Transformer | NOCD | L2 | N/A | None | Loss | $0.000 \pm 0.000$ | $0.368 \pm 0.000$ |
| Transformer | Neuromap | None | N/A | None | Loss | $0.000 \pm 0.000$ | $0.209 \pm 0.150$ |
| Transformer | Neuromap | DMoN | N/A | None | Loss | $0.000 \pm 0.000$ | $0.233 \pm 0.142$ |
| Transformer | Neuromap | L2 | N/A | None | Loss | $0.000 \pm 0.000$ | $0.334 \pm 0.055$ |
| Transformer | SBM$_{\text{NN}}$ | None | N/A | None | Loss | $0.000 \pm 0.000$ | $0.321 \pm 0.062$ |
| Transformer | SBM$_{\text{NN}}$ | DMoN | N/A | None | Loss | $0.000 \pm 0.000$ | $0.318 \pm 0.053$ |
| Transformer | SBM$_{\text{NN}}$ | L2 | N/A | None | Loss | $0.000 \pm 0.000$ | $0.348 \pm 0.042$ |
| MLP | None | None | N/A | None | Loss | $0.000 \pm 0.000$ | $\mathbf{0.368 \pm 0.000}$ |
| MLP | None | DMoN | N/A | None | Loss | $0.000 \pm 0.000$ | $0.368 \pm 0.000$ |
| MLP | None | L2 | N/A | None | Loss | $0.000 \pm 0.000$ | $0.368 \pm 0.000$ |
| MLP | DMoN | None | N/A | None | Loss | $0.000 \pm 0.000$ | $0.358 \pm 0.032$ |
| MLP | DMoN | DMoN | N/A | None | Loss | $0.000 \pm 0.000$ | $0.368 \pm 0.000$ |
| MLP | DMoN | L2 | N/A | None | Loss | $0.000 \pm 0.000$ | $0.368 \pm 0.000$ |
| MLP | NOCD | None | N/A | None | Loss | $0.000 \pm 0.000$ | $0.314 \pm 0.058$ |
| MLP | NOCD | DMoN | N/A | None | Loss | $0.000 \pm 0.000$ | $0.298 \pm 0.048$ |
| MLP | NOCD | L2 | N/A | None | MCC | $\mathbf{0.000 \pm 0.001}$ | $0.063 \pm 0.134$ |
| MLP | Neuromap | None | N/A | None | Loss | $0.000 \pm 0.000$ | $0.242 \pm 0.119$ |
| MLP | Neuromap | DMoN | N/A | None | Loss | $0.000 \pm 0.000$ | $0.296 \pm 0.091$ |
| MLP | Neuromap | L2 | N/A | None | Loss | $0.000 \pm 0.000$ | $0.368 \pm 0.000$ |
| MLP | SBM$_{\text{NN}}$ | None | N/A | None | MCC | $-0.001 \pm 0.003$ | $0.025 \pm 0.079$ |
| MLP | SBM$_{\text{NN}}$ | DMoN | N/A | None | Loss | $0.000 \pm 0.000$ | $0.288 \pm 0.042$ |
| MLP | SBM$_{\text{NN}}$ | L2 | N/A | None | Loss | $0.000 \pm 0.000$ | $0.311 \pm 0.062$ |

F.4.3    COAUTHOR CS DATASET, DEFAULT SPLIT WITH 20 TRAIN NODES PER CLASS AND 500 VALIDATION NODES.

Table 75: Comparing graph clustering and regularization objectives with an ablation study of regularization. The ablation compares clustering with clusteirng with regularization, clustering without regularization, and regularization without clustering for each neural network. Model selection based on training loss.

| Model | $f$ | $L_{\text{regularization}}$ | $L_V$ | $\text{NN}_{\boldsymbol{S} \to \boldsymbol{X}}$ | ES | MCC | Accuracy |
|---|---|---|---|---|---|---|---|
| GCN | None | None | ✗ | None | Loss | **0.875 ± 0.014** | **0.889 ± 0.013** |
| GCN | None | DMoN | ✗ | None | Loss | 0.854 ± 0.020 | 0.869 ± 0.019 |
| GCN | None | L2 | ✗ | None | Loss | 0.000 ± 0.000 | 0.038 ± 0.000 |
| GCN | DMoN | None | ✗ | None | Loss | 0.859 ± 0.011 | 0.874 ± 0.011 |
| GCN | DMoN | DMoN | ✗ | None | Loss | 0.867 ± 0.013 | 0.881 ± 0.012 |
| GCN | DMoN | L2 | ✗ | None | Loss | 0.000 ± 0.000 | 0.038 ± 0.000 |
| GCN | NOCD | None | ✓ | Transformer | Loss | 0.422 ± 0.065 | 0.386 ± 0.081 |
| GCN | NOCD | DMoN | ✓ | Transformer | Loss | 0.319 ± 0.129 | 0.262 ± 0.159 |
| GCN | NOCD | L2 | ✗ | None | Loss | 0.000 ± 0.000 | 0.082 ± 0.059 |
| GCN | Neuromap | None | ✓ | Transformer | Loss | 0.005 ± 0.011 | 0.070 ± 0.038 |
| GCN | Neuromap | DMoN | ✓ | Transformer | Loss | 0.011 ± 0.035 | 0.075 ± 0.064 |
| GCN | Neuromap | L2 | ✓ | MLP | Loss | 0.000 ± 0.000 | 0.038 ± 0.000 |
| GCN | $\text{SBM}_{\text{NN}}$ | None | ✓ | Transformer | Loss | 0.270 ± 0.040 | 0.207 ± 0.023 |
| GCN | $\text{SBM}_{\text{NN}}$ | DMoN | ✓ | Transformer | Loss | 0.269 ± 0.031 | 0.205 ± 0.030 |
| GCN | $\text{SBM}_{\text{NN}}$ | L2 | ✗ | None | Loss | 0.000 ± 0.000 | 0.057 ± 0.042 |
| GraphSAGE | None | None | ✗ | None | Loss | **0.887 ± 0.007** | **0.899 ± 0.006** |
| GraphSAGE | None | DMoN | ✗ | None | Loss | 0.810 ± 0.024 | 0.824 ± 0.025 |
| GraphSAGE | None | L2 | ✗ | None | Loss | 0.000 ± 0.000 | 0.051 ± 0.033 |
| GraphSAGE | DMoN | None | ✗ | None | Loss | 0.812 ± 0.035 | 0.825 ± 0.035 |
| GraphSAGE | DMoN | DMoN | ✓ | MLP | Loss | 0.768 ± 0.020 | 0.782 ± 0.021 |
| GraphSAGE | DMoN | L2 | ✗ | None | Loss | 0.000 ± 0.000 | 0.083 ± 0.083 |
| GraphSAGE | NOCD | None | ✓ | Transformer | Loss | 0.566 ± 0.099 | 0.576 ± 0.110 |
| GraphSAGE | NOCD | DMoN | ✗ | None | Loss | 0.217 ± 0.027 | 0.153 ± 0.022 |
| GraphSAGE | NOCD | L2 | ✗ | None | Loss | 0.000 ± 0.000 | 0.074 ± 0.065 |
| GraphSAGE | Neuromap | None | ✓ | MLP | Loss | 0.005 ± 0.017 | 0.008 ± 0.007 |
| GraphSAGE | Neuromap | DMoN | ✗ | None | Loss | 0.000 ± 0.000 | 0.035 ± 0.035 |
| GraphSAGE | Neuromap | L2 | ✗ | None | Loss | 0.000 ± 0.000 | 0.069 ± 0.065 |
| GraphSAGE | $\text{SBM}_{\text{NN}}$ | None | ✓ | MLP | Loss | 0.159 ± 0.058 | 0.181 ± 0.046 |
| GraphSAGE | $\text{SBM}_{\text{NN}}$ | DMoN | ✓ | MLP | Loss | 0.135 ± 0.057 | 0.156 ± 0.046 |
| GraphSAGE | $\text{SBM}_{\text{NN}}$ | L2 | ✗ | None | Loss | 0.000 ± 0.000 | 0.076 ± 0.060 |
| Transformer | None | None | N/A | None | Loss | 0.038 ± 0.042 | 0.100 ± 0.089 |
| Transformer | None | DMoN | N/A | None | Loss | 0.015 ± 0.030 | 0.053 ± 0.048 |
| Transformer | None | L2 | N/A | None | Loss | 0.005 ± 0.009 | 0.053 ± 0.048 |
| Transformer | DMoN | None | N/A | None | Loss | 0.016 ± 0.034 | 0.065 ± 0.064 |
| Transformer | DMoN | DMoN | N/A | None | Loss | 0.014 ± 0.025 | 0.027 ± 0.017 |
| Transformer | DMoN | L2 | N/A | None | Loss | 0.004 ± 0.011 | 0.040 ± 0.028 |
| Transformer | NOCD | None | N/A | None | Loss | 0.000 ± 0.000 | 0.050 ± 0.026 |
| Transformer | NOCD | DMoN | N/A | None | Loss | 0.000 ± 0.000 | 0.036 ± 0.006 |
| Transformer | NOCD | L2 | N/A | None | Loss | **0.089 ± 0.185** | **0.110 ± 0.189** |
| Transformer | Neuromap | None | N/A | None | Loss | 0.000 ± 0.000 | 0.049 ± 0.037 |
| Transformer | Neuromap | DMoN | N/A | None | Loss | 0.000 ± 0.000 | 0.093 ± 0.082 |
| Transformer | Neuromap | L2 | N/A | None | Loss | 0.000 ± 0.000 | 0.053 ± 0.038 |

Table 75: Comparing graph clustering and regularization objectives for each neural network.

| Model | $f$ | $L_{\text{regularization}}$ | $L_V$ | NN$_{S \rightarrow X}$ | ES | MCC | Accuracy |
|---|---|---|---|---|---|---|---|
| Transformer | SBM$_{\text{NN}}$ | None | N/A | None | Loss | $0.000 \pm 0.000$ | $0.037 \pm 0.005$ |
| Transformer | SBM$_{\text{NN}}$ | DMoN | N/A | None | Loss | $0.000 \pm 0.000$ | $0.043 \pm 0.013$ |
| Transformer | SBM$_{\text{NN}}$ | L2 | N/A | None | Loss | $0.081 \pm 0.072$ | $0.107 \pm 0.068$ |
| MLP | None | None | N/A | None | Loss | $\mathbf{0.859 \pm 0.013}$ | $\mathbf{0.874 \pm 0.012}$ |
| MLP | None | DMoN | N/A | None | Loss | $0.348 \pm 0.033$ | $0.284 \pm 0.055$ |
| MLP | None | L2 | N/A | None | Loss | $0.000 \pm 0.000$ | $0.063 \pm 0.069$ |
| MLP | DMoN | None | N/A | None | Loss | $0.420 \pm 0.050$ | $0.365 \pm 0.078$ |
| MLP | DMoN | DMoN | N/A | None | Loss | $0.423 \pm 0.041$ | $0.372 \pm 0.062$ |
| MLP | DMoN | L2 | N/A | None | Loss | $0.000 \pm 0.000$ | $0.090 \pm 0.062$ |
| MLP | NOCD | None | N/A | None | Loss | $0.144 \pm 0.118$ | $0.168 \pm 0.115$ |
| MLP | NOCD | DMoN | N/A | None | Loss | $0.139 \pm 0.084$ | $0.137 \pm 0.065$ |
| MLP | NOCD | L2 | N/A | None | Loss | $0.647 \pm 0.077$ | $0.670 \pm 0.079$ |
| MLP | Neuromap | None | N/A | None | Loss | $0.018 \pm 0.022$ | $0.063 \pm 0.071$ |
| MLP | Neuromap | DMoN | N/A | None | Loss | $0.004 \pm 0.013$ | $0.091 \pm 0.082$ |
| MLP | Neuromap | L2 | N/A | None | Loss | $0.000 \pm 0.000$ | $0.066 \pm 0.062$ |
| MLP | SBM$_{\text{NN}}$ | None | N/A | None | Loss | $0.000 \pm 0.000$ | $0.038 \pm 0.000$ |
| MLP | SBM$_{\text{NN}}$ | DMoN | N/A | None | Loss | $0.009 \pm 0.020$ | $0.039 \pm 0.002$ |
| MLP | SBM$_{\text{NN}}$ | L2 | N/A | None | Loss | $0.012 \pm 0.035$ | $0.069 \pm 0.069$ |

Table 76: Comparing graph clustering and regularization objectives with an ablation study of regularization. The ablation compares clustering with clusteirng with regularization, clustering without regularization, and regularization without clustering for each neural network. Model selection based on training set MCC.

| Model | $f$ | $L_{\text{regularization}}$ | $L_V$ | NN$_{S \rightarrow X}$ | ES | MCC | Accuracy |
|---|---|---|---|---|---|---|---|
| GCN | None | None | ✗ | None | Loss | $\mathbf{0.875 \pm 0.014}$ | $\mathbf{0.889 \pm 0.013}$ |
| GCN | None | DMoN | ✗ | None | Loss | $0.854 \pm 0.020$ | $0.869 \pm 0.019$ |
| GCN | None | L2 | ✗ | None | MCC | $0.120 \pm 0.094$ | $0.192 \pm 0.079$ |
| GCN | DMoN | None | ✗ | None | Loss | $0.859 \pm 0.011$ | $0.874 \pm 0.011$ |
| GCN | DMoN | DMoN | ✗ | None | Loss | $0.867 \pm 0.013$ | $0.881 \pm 0.012$ |
| GCN | DMoN | L2 | ✗ | None | MCC | $0.029 \pm 0.160$ | $0.139 \pm 0.101$ |
| GCN | NOCD | None | ✓ | Transformer | Loss | $0.422 \pm 0.065$ | $0.386 \pm 0.081$ |
| GCN | NOCD | DMoN | ✓ | Transformer | Loss | $0.319 \pm 0.129$ | $0.262 \pm 0.159$ |
| GCN | NOCD | L2 | ✗ | None | MCC | $0.146 \pm 0.066$ | $0.183 \pm 0.082$ |
| GCN | Neuromap | None | ✗ | None | MCC | $0.443 \pm 0.120$ | $0.447 \pm 0.128$ |
| GCN | Neuromap | DMoN | ✗ | None | MCC | $0.405 \pm 0.098$ | $0.375 \pm 0.149$ |
| GCN | Neuromap | L2 | ✗ | None | MCC | $0.101 \pm 0.161$ | $0.171 \pm 0.106$ |
| GCN | SBM$_{\text{NN}}$ | None | ✓ | Transformer | Loss | $0.270 \pm 0.040$ | $0.207 \pm 0.023$ |
| GCN | SBM$_{\text{NN}}$ | DMoN | ✓ | Transformer | Loss | $0.269 \pm 0.031$ | $0.205 \pm 0.030$ |
| GCN | SBM$_{\text{NN}}$ | L2 | ✗ | None | MCC | $0.087 \pm 0.058$ | $0.115 \pm 0.071$ |
| GraphSAGE | None | None | ✗ | None | Loss | $\mathbf{0.887 \pm 0.007}$ | $\mathbf{0.899 \pm 0.006}$ |
| GraphSAGE | None | DMoN | ✗ | None | Loss | $0.810 \pm 0.024$ | $0.824 \pm 0.025$ |
| GraphSAGE | None | L2 | ✗ | None | Loss | $0.000 \pm 0.000$ | $0.051 \pm 0.033$ |
| GraphSAGE | DMoN | None | ✗ | None | Loss | $0.812 \pm 0.035$ | $0.825 \pm 0.035$ |
| GraphSAGE | DMoN | DMoN | ✓ | MLP | Loss | $0.768 \pm 0.020$ | $0.782 \pm 0.021$ |
| GraphSAGE | DMoN | L2 | ✓ | MLP | MCC | $0.004 \pm 0.013$ | $0.024 \pm 0.050$ |

Table 76: Comparing graph clustering and regularization objectives for each neural network.

| Model | $f$ | $L_{\text{regularization}}$ | $L_V$ | $\text{NN}_{\boldsymbol{S}\to\boldsymbol{X}}$ | ES | MCC | Accuracy |
|---|---|---|---|---|---|---|---|
| GraphSAGE | NOCD | None | ✓ | Transformer | Loss | $0.566 \pm 0.099$ | $0.576 \pm 0.110$ |
| GraphSAGE | NOCD | DMoN | ✗ | None | Loss | $0.217 \pm 0.027$ | $0.153 \pm 0.022$ |
| GraphSAGE | NOCD | L2 | ✓ | MLP | MCC | $0.004 \pm 0.015$ | $0.037 \pm 0.051$ |
| GraphSAGE | Neuromap | None | ✓ | MLP | MCC | $0.187 \pm 0.142$ | $0.174 \pm 0.162$ |
| GraphSAGE | Neuromap | DMoN | ✓ | MLP | MCC | $0.146 \pm 0.068$ | $0.089 \pm 0.079$ |
| GraphSAGE | Neuromap | L2 | ✓ | MLP | MCC | $0.003 \pm 0.011$ | $0.017 \pm 0.030$ |
| GraphSAGE | $\text{SBM}_{\text{NN}}$ | None | ✓ | MLP | Loss | $0.159 \pm 0.058$ | $0.181 \pm 0.046$ |
| GraphSAGE | $\text{SBM}_{\text{NN}}$ | DMoN | ✓ | Transformer | MCC | $0.345 \pm 0.134$ | $0.301 \pm 0.149$ |
| GraphSAGE | $\text{SBM}_{\text{NN}}$ | L2 | ✓ | MLP | MCC | $0.007 \pm 0.022$ | $0.031 \pm 0.065$ |
| Transformer | None | None | N/A | None | MCC | $0.103 \pm 0.077$ | $0.117 \pm 0.095$ |
| Transformer | None | DMoN | N/A | None | MCC | $0.123 \pm 0.109$ | $0.133 \pm 0.121$ |
| Transformer | None | L2 | N/A | None | MCC | $\mathbf{0.803 \pm 0.035}$ | $\mathbf{0.823 \pm 0.032}$ |
| Transformer | DMoN | None | N/A | None | MCC | $0.118 \pm 0.147$ | $0.137 \pm 0.172$ |
| Transformer | DMoN | DMoN | N/A | None | MCC | $0.102 \pm 0.061$ | $0.101 \pm 0.080$ |
| Transformer | DMoN | L2 | N/A | None | MCC | $0.783 \pm 0.063$ | $0.805 \pm 0.058$ |
| Transformer | NOCD | None | N/A | None | Loss | $0.000 \pm 0.000$ | $0.050 \pm 0.026$ |
| Transformer | NOCD | DMoN | N/A | None | Loss | $0.000 \pm 0.000$ | $0.036 \pm 0.006$ |
| Transformer | NOCD | L2 | N/A | None | MCC | $0.248 \pm 0.117$ | $0.233 \pm 0.137$ |
| Transformer | Neuromap | None | N/A | None | Loss | $0.000 \pm 0.000$ | $0.049 \pm 0.037$ |
| Transformer | Neuromap | DMoN | N/A | None | Loss | $0.000 \pm 0.000$ | $0.093 \pm 0.082$ |
| Transformer | Neuromap | L2 | N/A | None | MCC | $0.005 \pm 0.026$ | $0.070 \pm 0.066$ |
| Transformer | $\text{SBM}_{\text{NN}}$ | None | N/A | None | Loss | $0.000 \pm 0.000$ | $0.037 \pm 0.005$ |
| Transformer | $\text{SBM}_{\text{NN}}$ | DMoN | N/A | None | Loss | $0.000 \pm 0.000$ | $0.043 \pm 0.013$ |
| Transformer | $\text{SBM}_{\text{NN}}$ | L2 | N/A | None | Loss | $0.081 \pm 0.072$ | $0.107 \pm 0.068$ |
| MLP | None | None | N/A | None | Loss | $\mathbf{0.859 \pm 0.013}$ | $\mathbf{0.874 \pm 0.012}$ |
| MLP | None | DMoN | N/A | None | Loss | $0.348 \pm 0.033$ | $0.284 \pm 0.055$ |
| MLP | None | L2 | N/A | None | MCC | $0.067 \pm 0.030$ | $0.158 \pm 0.045$ |
| MLP | DMoN | None | N/A | None | Loss | $0.420 \pm 0.050$ | $0.365 \pm 0.078$ |
| MLP | DMoN | DMoN | N/A | None | Loss | $0.423 \pm 0.041$ | $0.372 \pm 0.062$ |
| MLP | DMoN | L2 | N/A | None | MCC | $0.059 \pm 0.025$ | $0.152 \pm 0.069$ |
| MLP | NOCD | None | N/A | None | Loss | $0.144 \pm 0.118$ | $0.168 \pm 0.115$ |
| MLP | NOCD | DMoN | N/A | None | Loss | $0.139 \pm 0.084$ | $0.137 \pm 0.065$ |
| MLP | NOCD | L2 | N/A | None | Loss | $0.647 \pm 0.077$ | $0.670 \pm 0.079$ |
| MLP | Neuromap | None | N/A | None | Loss | $0.018 \pm 0.022$ | $0.063 \pm 0.071$ |
| MLP | Neuromap | DMoN | N/A | None | MCC | $0.137 \pm 0.097$ | $0.122 \pm 0.097$ |
| MLP | Neuromap | L2 | N/A | None | Loss | $0.000 \pm 0.000$ | $0.066 \pm 0.062$ |
| MLP | $\text{SBM}_{\text{NN}}$ | None | N/A | None | MCC | $-0.001 \pm 0.010$ | $0.052 \pm 0.048$ |
| MLP | $\text{SBM}_{\text{NN}}$ | DMoN | N/A | None | Loss | $0.009 \pm 0.020$ | $0.039 \pm 0.002$ |
| MLP | $\text{SBM}_{\text{NN}}$ | L2 | N/A | None | Loss | $0.012 \pm 0.035$ | $0.069 \pm 0.069$ |

Table 77: Comparing graph clustering and regularization objectives with an ablation study of regularization. The ablation compares clustering with clusteirng with regularization, clustering without regularization, and regularization without clustering for each neural network. Model selection based on validation set MCC.

| Model | $f$ | $L_{\text{regularization}}$ | $L_V$ | $NN_{\boldsymbol{S} \to \boldsymbol{X}}$ | ES | MCC | Accuracy |
|---|---|---|---|---|---|---|---|
| GCN | None | None | ✗ | None | MCC | **0.876 ± 0.015** | **0.889 ± 0.014** |
| GCN | None | DMoN | ✗ | None | MCC | 0.865 ± 0.023 | 0.879 ± 0.021 |
| GCN | None | L2 | ✗ | None | MCC | 0.120 ± 0.094 | 0.192 ± 0.079 |
| GCN | DMoN | None | ✓ | Transformer | MCC | 0.875 ± 0.013 | 0.889 ± 0.012 |
| GCN | DMoN | DMoN | ✓ | Transformer | MCC | 0.861 ± 0.011 | 0.875 ± 0.011 |
| GCN | DMoN | L2 | ✗ | None | MCC | 0.029 ± 0.160 | 0.139 ± 0.101 |
| GCN | NOCD | None | ✓ | MLP | MCC | 0.752 ± 0.081 | 0.768 ± 0.081 |
| GCN | NOCD | DMoN | ✓ | MLP | MCC | 0.718 ± 0.089 | 0.727 ± 0.099 |
| GCN | NOCD | L2 | ✗ | None | MCC | 0.146 ± 0.066 | 0.183 ± 0.082 |
| GCN | Neuromap | None | ✗ | None | MCC | 0.443 ± 0.120 | 0.447 ± 0.128 |
| GCN | Neuromap | DMoN | ✗ | None | MCC | 0.405 ± 0.098 | 0.375 ± 0.149 |
| GCN | Neuromap | L2 | ✗ | None | MCC | 0.101 ± 0.161 | 0.171 ± 0.106 |
| GCN | SBM$_{\text{NN}}$ | None | ✓ | MLP | MCC | 0.733 ± 0.100 | 0.745 ± 0.110 |
| GCN | SBM$_{\text{NN}}$ | DMoN | ✗ | None | MCC | 0.734 ± 0.070 | 0.750 ± 0.072 |
| GCN | SBM$_{\text{NN}}$ | L2 | ✗ | None | MCC | 0.087 ± 0.058 | 0.115 ± 0.071 |
| GraphSAGE | None | None | ✗ | None | MCC | **0.889 ± 0.008** | **0.901 ± 0.007** |
| GraphSAGE | None | DMoN | ✗ | None | MCC | 0.881 ± 0.017 | 0.893 ± 0.015 |
| GraphSAGE | None | L2 | ✗ | None | Loss | 0.000 ± 0.000 | 0.051 ± 0.033 |
| GraphSAGE | DMoN | None | ✗ | None | MCC | 0.881 ± 0.015 | 0.894 ± 0.014 |
| GraphSAGE | DMoN | DMoN | ✓ | Transformer | MCC | 0.877 ± 0.013 | 0.890 ± 0.012 |
| GraphSAGE | DMoN | L2 | ✓ | Transformer | MCC | 0.004 ± 0.012 | 0.052 ± 0.071 |
| GraphSAGE | NOCD | None | ✓ | MLP | MCC | 0.722 ± 0.111 | 0.736 ± 0.116 |
| GraphSAGE | NOCD | DMoN | ✓ | MLP | MCC | 0.688 ± 0.098 | 0.698 ± 0.116 |
| GraphSAGE | NOCD | L2 | ✓ | MLP | MCC | 0.004 ± 0.015 | 0.037 ± 0.051 |
| GraphSAGE | Neuromap | None | ✓ | MLP | MCC | 0.187 ± 0.142 | 0.174 ± 0.162 |
| GraphSAGE | Neuromap | DMoN | ✓ | MLP | MCC | 0.146 ± 0.068 | 0.089 ± 0.079 |
| GraphSAGE | Neuromap | L2 | ✓ | Transformer | MCC | 0.000 ± 0.001 | 0.001 ± 0.002 |
| GraphSAGE | SBM$_{\text{NN}}$ | None | ✓ | MLP | MCC | 0.387 ± 0.154 | 0.349 ± 0.185 |
| GraphSAGE | SBM$_{\text{NN}}$ | DMoN | ✗ | None | MCC | 0.342 ± 0.131 | 0.304 ± 0.142 |
| GraphSAGE | SBM$_{\text{NN}}$ | L2 | ✓ | MLP | MCC | 0.007 ± 0.022 | 0.031 ± 0.065 |
| Transformer | None | None | N/A | None | MCC | 0.103 ± 0.077 | 0.117 ± 0.095 |
| Transformer | None | DMoN | N/A | None | MCC | 0.123 ± 0.109 | 0.133 ± 0.121 |
| Transformer | None | L2 | N/A | None | MCC | **0.803 ± 0.035** | **0.823 ± 0.032** |
| Transformer | DMoN | None | N/A | None | MCC | 0.118 ± 0.147 | 0.137 ± 0.172 |
| Transformer | DMoN | DMoN | N/A | None | MCC | 0.102 ± 0.061 | 0.101 ± 0.080 |
| Transformer | DMoN | L2 | N/A | None | MCC | 0.783 ± 0.063 | 0.805 ± 0.058 |
| Transformer | NOCD | None | N/A | None | Loss | 0.000 ± 0.000 | 0.050 ± 0.026 |
| Transformer | NOCD | DMoN | N/A | None | Loss | 0.000 ± 0.000 | 0.036 ± 0.006 |
| Transformer | NOCD | L2 | N/A | None | MCC | 0.248 ± 0.117 | 0.233 ± 0.137 |
| Transformer | Neuromap | None | N/A | None | Loss | 0.000 ± 0.000 | 0.049 ± 0.037 |
| Transformer | Neuromap | DMoN | N/A | None | Loss | 0.000 ± 0.000 | 0.093 ± 0.082 |
| Transformer | Neuromap | L2 | N/A | None | MCC | 0.005 ± 0.026 | 0.070 ± 0.066 |

Table 77: Comparing graph clustering and regularization objectives for each neural network.

| Model | $f$ | $L_{\text{regularization}}$ | $L_V$ | $\text{NN}_{S \to X}$ | ES | MCC | Accuracy |
|---|---|---|---|---|---|---|---|
| Transformer | SBM$_{\text{NN}}$ | None | N/A | None | Loss | $0.000 \pm 0.000$ | $0.037 \pm 0.005$ |
| Transformer | SBM$_{\text{NN}}$ | DMoN | N/A | None | Loss | $0.000 \pm 0.000$ | $0.043 \pm 0.013$ |
| Transformer | SBM$_{\text{NN}}$ | L2 | N/A | None | Loss | $0.081 \pm 0.072$ | $0.107 \pm 0.068$ |
| MLP | None | None | N/A | None | MCC | $\mathbf{0.862 \pm 0.011}$ | $\mathbf{0.877 \pm 0.011}$ |
| MLP | None | DMoN | N/A | None | MCC | $0.838 \pm 0.014$ | $0.855 \pm 0.013$ |
| MLP | None | L2 | N/A | None | MCC | $0.067 \pm 0.030$ | $0.158 \pm 0.045$ |
| MLP | DMoN | None | N/A | None | MCC | $0.839 \pm 0.013$ | $0.856 \pm 0.013$ |
| MLP | DMoN | DMoN | N/A | None | MCC | $0.837 \pm 0.012$ | $0.854 \pm 0.011$ |
| MLP | DMoN | L2 | N/A | None | MCC | $0.059 \pm 0.025$ | $0.152 \pm 0.069$ |
| MLP | NOCD | None | N/A | None | MCC | $0.169 \pm 0.097$ | $0.137 \pm 0.069$ |
| MLP | NOCD | DMoN | N/A | None | MCC | $0.209 \pm 0.121$ | $0.161 \pm 0.122$ |
| MLP | NOCD | L2 | N/A | None | Loss | $0.647 \pm 0.077$ | $0.670 \pm 0.079$ |
| MLP | Neuromap | None | N/A | None | MCC | $0.071 \pm 0.092$ | $0.059 \pm 0.079$ |
| MLP | Neuromap | DMoN | N/A | None | MCC | $0.137 \pm 0.097$ | $0.122 \pm 0.097$ |
| MLP | Neuromap | L2 | N/A | None | MCC | $0.002 \pm 0.015$ | $0.069 \pm 0.067$ |
| MLP | SBM$_{\text{NN}}$ | None | N/A | None | Loss | $0.000 \pm 0.000$ | $0.038 \pm 0.000$ |
| MLP | SBM$_{\text{NN}}$ | DMoN | N/A | None | MCC | $0.016 \pm 0.021$ | $0.061 \pm 0.087$ |
| MLP | SBM$_{\text{NN}}$ | L2 | N/A | None | Loss | $0.012 \pm 0.035$ | $0.069 \pm 0.069$ |

F.4.4 COAUTHOR CS DATASET, SPARSE SPLIT WITH 2 TRAIN NODES PER CLASS AND 50 VALIDATION NODES.

Table 78: Comparing graph clustering and regularization objectives with an ablation study of regularization. The ablation compares clustering with clusteirng with regularization, clustering without regularization, and regularization without clustering for each neural network. Model selection based on training loss.

| Model | $f$ | $L_{\text{regularization}}$ | $L_V$ | $\text{NN}_{S \to X}$ | ES | MCC | Accuracy |
|---|---|---|---|---|---|---|---|
| GCN | None | None | ✗ | None | Loss | $\mathbf{0.742 \pm 0.055}$ | $\mathbf{0.763 \pm 0.054}$ |
| GCN | None | DMoN | ✗ | None | Loss | $0.716 \pm 0.053$ | $0.736 \pm 0.049$ |
| GCN | None | L2 | ✗ | None | Loss | $0.000 \pm 0.000$ | $0.039 \pm 0.000$ |
| GCN | DMoN | None | ✓ | MLP | Loss | $0.668 \pm 0.051$ | $0.690 \pm 0.049$ |
| GCN | DMoN | DMoN | ✗ | None | Loss | $0.707 \pm 0.066$ | $0.729 \pm 0.061$ |
| GCN | DMoN | L2 | ✗ | None | Loss | $0.000 \pm 0.000$ | $0.039 \pm 0.000$ |
| GCN | NOCD | None | ✗ | None | Loss | $0.424 \pm 0.086$ | $0.430 \pm 0.089$ |
| GCN | NOCD | DMoN | ✓ | Transformer | Loss | $0.372 \pm 0.081$ | $0.355 \pm 0.079$ |
| GCN | NOCD | L2 | ✗ | None | Loss | $0.000 \pm 0.000$ | $0.063 \pm 0.044$ |
| GCN | Neuromap | None | ✓ | MLP | Loss | $0.041 \pm 0.063$ | $0.049 \pm 0.054$ |
| GCN | Neuromap | DMoN | ✓ | MLP | Loss | $0.019 \pm 0.027$ | $0.047 \pm 0.069$ |
| GCN | Neuromap | L2 | ✗ | None | Loss | $0.000 \pm 0.000$ | $0.039 \pm 0.000$ |
| GCN | SBM$_{\text{NN}}$ | None | ✗ | None | Loss | $0.236 \pm 0.068$ | $0.260 \pm 0.050$ |
| GCN | SBM$_{\text{NN}}$ | DMoN | ✓ | MLP | Loss | $0.276 \pm 0.083$ | $0.291 \pm 0.088$ |
| GCN | SBM$_{\text{NN}}$ | L2 | ✗ | None | Loss | $0.000 \pm 0.000$ | $0.104 \pm 0.091$ |
| GraphSAGE | None | None | ✗ | None | Loss | $\mathbf{0.727 \pm 0.069}$ | $\mathbf{0.748 \pm 0.067}$ |
| GraphSAGE | None | DMoN | ✗ | None | Loss | $0.593 \pm 0.044$ | $0.612 \pm 0.043$ |
| GraphSAGE | None | L2 | ✗ | None | Loss | $0.000 \pm 0.000$ | $0.044 \pm 0.035$ |

Table 78: Comparing graph clustering and regularization objectives for each neural network.

| Model | $f$ | $L_{\text{regularization}}$ | $L_V$ | NN$_{S \to X}$ | ES | MCC | Accuracy |
|---|---|---|---|---|---|---|---|
| GraphSAGE | DMoN | None | ✓ | MLP | Loss | $0.602 \pm 0.034$ | $0.621 \pm 0.031$ |
| GraphSAGE | DMoN | DMoN | ✓ | MLP | Loss | $0.597 \pm 0.042$ | $0.618 \pm 0.039$ |
| GraphSAGE | DMoN | L2 | ✗ | None | Loss | $0.000 \pm 0.000$ | $0.043 \pm 0.029$ |
| GraphSAGE | NOCD | None | ✓ | Transformer | Loss | $0.448 \pm 0.130$ | $0.462 \pm 0.133$ |
| GraphSAGE | NOCD | DMoN | ✓ | Transformer | Loss | $0.296 \pm 0.061$ | $0.287 \pm 0.059$ |
| GraphSAGE | NOCD | L2 | ✗ | None | Loss | $0.000 \pm 0.000$ | $0.058 \pm 0.065$ |
| GraphSAGE | Neuromap | None | ✓ | MLP | Loss | $0.006 \pm 0.018$ | $0.032 \pm 0.069$ |
| GraphSAGE | Neuromap | DMoN | ✓ | MLP | Loss | $0.023 \pm 0.033$ | $0.025 \pm 0.031$ |
| GraphSAGE | Neuromap | L2 | ✗ | None | Loss | $0.000 \pm 0.000$ | $0.062 \pm 0.062$ |
| GraphSAGE | SBM$_{\text{NN}}$ | None | ✓ | MLP | Loss | $0.174 \pm 0.067$ | $0.199 \pm 0.059$ |
| GraphSAGE | SBM$_{\text{NN}}$ | DMoN | ✓ | Transformer | Loss | $0.131 \pm 0.076$ | $0.160 \pm 0.080$ |
| GraphSAGE | SBM$_{\text{NN}}$ | L2 | ✗ | None | Loss | $0.000 \pm 0.000$ | $0.062 \pm 0.069$ |
| Transformer | None | None | N/A | None | Loss | $0.009 \pm 0.028$ | $0.047 \pm 0.049$ |
| Transformer | None | DMoN | N/A | None | Loss | $0.002 \pm 0.005$ | $0.070 \pm 0.065$ |
| Transformer | None | L2 | N/A | None | Loss | $0.095 \pm 0.106$ | $0.144 \pm 0.114$ |
| Transformer | DMoN | None | N/A | None | Loss | $0.008 \pm 0.017$ | $0.080 \pm 0.043$ |
| Transformer | DMoN | DMoN | N/A | None | Loss | $0.013 \pm 0.031$ | $0.075 \pm 0.064$ |
| Transformer | DMoN | L2 | N/A | None | Loss | $0.049 \pm 0.047$ | $0.103 \pm 0.075$ |
| Transformer | NOCD | None | N/A | None | Loss | $0.000 \pm 0.000$ | $0.040 \pm 0.004$ |
| Transformer | NOCD | DMoN | N/A | None | Loss | $0.000 \pm 0.000$ | $0.039 \pm 0.000$ |
| Transformer | NOCD | L2 | N/A | None | Loss | $\mathbf{0.677 \pm 0.241}$ | $\mathbf{0.699 \pm 0.245}$ |
| Transformer | Neuromap | None | N/A | None | Loss | $0.000 \pm 0.000$ | $0.059 \pm 0.036$ |
| Transformer | Neuromap | DMoN | N/A | None | Loss | $0.000 \pm 0.000$ | $0.057 \pm 0.040$ |
| Transformer | Neuromap | L2 | N/A | None | Loss | $0.000 \pm 0.000$ | $0.051 \pm 0.037$ |
| Transformer | SBM$_{\text{NN}}$ | None | N/A | None | Loss | $0.000 \pm 0.000$ | $0.041 \pm 0.014$ |
| Transformer | SBM$_{\text{NN}}$ | DMoN | N/A | None | Loss | $0.000 \pm 0.000$ | $0.039 \pm 0.000$ |
| Transformer | SBM$_{\text{NN}}$ | L2 | N/A | None | Loss | $0.054 \pm 0.074$ | $0.092 \pm 0.079$ |
| MLP | None | None | N/A | None | Loss | $0.566 \pm 0.084$ | $0.589 \pm 0.101$ |
| MLP | None | DMoN | N/A | None | Loss | $0.246 \pm 0.034$ | $0.260 \pm 0.038$ |
| MLP | None | L2 | N/A | None | Loss | $0.000 \pm 0.000$ | $0.045 \pm 0.040$ |
| MLP | DMoN | None | N/A | None | Loss | $0.467 \pm 0.067$ | $0.493 \pm 0.077$ |
| MLP | DMoN | DMoN | N/A | None | Loss | $0.475 \pm 0.079$ | $0.497 \pm 0.084$ |
| MLP | DMoN | L2 | N/A | None | Loss | $0.000 \pm 0.000$ | $0.049 \pm 0.041$ |
| MLP | NOCD | None | N/A | None | Loss | $0.155 \pm 0.155$ | $0.186 \pm 0.156$ |
| MLP | NOCD | DMoN | N/A | None | Loss | $0.126 \pm 0.077$ | $0.146 \pm 0.073$ |
| MLP | NOCD | L2 | N/A | None | Loss | $\mathbf{0.710 \pm 0.046}$ | $\mathbf{0.738 \pm 0.044}$ |
| MLP | Neuromap | None | N/A | None | Loss | $0.029 \pm 0.052$ | $0.079 \pm 0.048$ |
| MLP | Neuromap | DMoN | N/A | None | Loss | $0.008 \pm 0.012$ | $0.059 \pm 0.071$ |
| MLP | Neuromap | L2 | N/A | None | Loss | $0.000 \pm 0.000$ | $0.038 \pm 0.036$ |
| MLP | SBM$_{\text{NN}}$ | None | N/A | None | Loss | $0.003 \pm 0.009$ | $0.048 \pm 0.026$ |
| MLP | SBM$_{\text{NN}}$ | DMoN | N/A | None | Loss | $0.003 \pm 0.007$ | $0.034 \pm 0.007$ |
| MLP | SBM$_{\text{NN}}$ | L2 | N/A | None | Loss | $0.092 \pm 0.140$ | $0.132 \pm 0.135$ |

Table 79: Comparing graph clustering and regularization objectives with an ablation study of regularization. The ablation compares clustering with clusteirng with regularization, clustering without regularization, and regularization without clustering for each neural network. Model selection based on training set MCC.

| Model | $f$ | $L_{\text{regularization}}$ | $L_V$ | NN$_{S \to X}$ | ES | MCC | Accuracy |
|---|---|---|---|---|---|---|---|
| GCN | None | None | ✗ | None | Loss | **0.742 ± 0.055** | **0.763 ± 0.054** |
| GCN | None | DMoN | ✗ | None | Loss | 0.716 ± 0.053 | 0.736 ± 0.049 |
| GCN | None | L2 | ✗ | None | MCC | -0.014 ± 0.084 | 0.084 ± 0.057 |
| GCN | DMoN | None | ✓ | MLP | Loss | 0.668 ± 0.051 | 0.690 ± 0.049 |
| GCN | DMoN | DMoN | ✗ | None | Loss | 0.707 ± 0.066 | 0.729 ± 0.061 |
| GCN | DMoN | L2 | ✗ | None | MCC | -0.004 ± 0.076 | 0.076 ± 0.062 |
| GCN | NOCD | None | ✗ | None | Loss | 0.424 ± 0.086 | 0.430 ± 0.089 |
| GCN | NOCD | DMoN | ✓ | Transformer | MCC | 0.605 ± 0.097 | 0.628 ± 0.101 |
| GCN | NOCD | L2 | ✓ | Transformer | MCC | 0.060 ± 0.073 | 0.123 ± 0.059 |
| GCN | Neuromap | None | ✗ | None | MCC | 0.209 ± 0.174 | 0.206 ± 0.172 |
| GCN | Neuromap | DMoN | ✗ | None | MCC | 0.283 ± 0.157 | 0.282 ± 0.177 |
| GCN | Neuromap | L2 | ✗ | None | MCC | 0.047 ± 0.099 | 0.126 ± 0.079 |
| GCN | SBM$_{NN}$ | None | ✓ | MLP | Loss | 0.247 ± 0.126 | 0.257 ± 0.123 |
| GCN | SBM$_{NN}$ | DMoN | ✓ | Transformer | Loss | 0.262 ± 0.104 | 0.270 ± 0.104 |
| GCN | SBM$_{NN}$ | L2 | ✓ | Transformer | MCC | 0.043 ± 0.053 | 0.083 ± 0.045 |
| GraphSAGE | None | None | ✗ | None | Loss | **0.727 ± 0.069** | **0.748 ± 0.067** |
| GraphSAGE | None | DMoN | ✗ | None | Loss | 0.593 ± 0.044 | 0.612 ± 0.043 |
| GraphSAGE | None | L2 | ✗ | None | MCC | -0.000 ± 0.000 | 0.002 ± 0.007 |
| GraphSAGE | DMoN | None | ✗ | None | Loss | 0.637 ± 0.029 | 0.654 ± 0.027 |
| GraphSAGE | DMoN | DMoN | ✗ | None | Loss | 0.623 ± 0.028 | 0.643 ± 0.026 |
| GraphSAGE | DMoN | L2 | ✓ | MLP | MCC | -0.000 ± 0.000 | 0.002 ± 0.007 |
| GraphSAGE | NOCD | None | ✓ | Transformer | Loss | 0.448 ± 0.130 | 0.462 ± 0.133 |
| GraphSAGE | NOCD | DMoN | ✗ | None | Loss | 0.327 ± 0.042 | 0.307 ± 0.048 |
| GraphSAGE | NOCD | L2 | ✓ | MLP | MCC | 0.002 ± 0.007 | 0.006 ± 0.013 |
| GraphSAGE | Neuromap | None | ✓ | MLP | MCC | 0.111 ± 0.102 | 0.098 ± 0.109 |
| GraphSAGE | Neuromap | DMoN | ✓ | MLP | MCC | 0.097 ± 0.105 | 0.106 ± 0.112 |
| GraphSAGE | Neuromap | L2 | ✗ | None | MCC | 0.003 ± 0.011 | 0.006 ± 0.012 |
| GraphSAGE | SBM$_{NN}$ | None | ✓ | MLP | Loss | 0.174 ± 0.067 | 0.199 ± 0.059 |
| GraphSAGE | SBM$_{NN}$ | DMoN | ✓ | Transformer | Loss | 0.131 ± 0.076 | 0.160 ± 0.080 |
| GraphSAGE | SBM$_{NN}$ | L2 | ✓ | Transformer | MCC | 0.006 ± 0.010 | 0.037 ± 0.073 |
| Transformer | None | None | N/A | None | MCC | 0.042 ± 0.049 | 0.056 ± 0.081 |
| Transformer | None | DMoN | N/A | None | MCC | 0.039 ± 0.046 | 0.059 ± 0.060 |
| Transformer | None | L2 | N/A | None | MCC | 0.476 ± 0.120 | 0.516 ± 0.130 |
| Transformer | DMoN | None | N/A | None | Loss | 0.008 ± 0.017 | 0.080 ± 0.043 |
| Transformer | DMoN | DMoN | N/A | None | Loss | 0.013 ± 0.031 | 0.075 ± 0.064 |
| Transformer | DMoN | L2 | N/A | None | MCC | 0.518 ± 0.104 | 0.552 ± 0.112 |
| Transformer | NOCD | None | N/A | None | Loss | 0.000 ± 0.000 | 0.040 ± 0.004 |
| Transformer | NOCD | DMoN | N/A | None | Loss | 0.000 ± 0.000 | 0.039 ± 0.000 |
| Transformer | NOCD | L2 | N/A | None | Loss | **0.677 ± 0.241** | **0.699 ± 0.245** |
| Transformer | Neuromap | None | N/A | None | Loss | 0.000 ± 0.000 | 0.059 ± 0.036 |
| Transformer | Neuromap | DMoN | N/A | None | Loss | 0.000 ± 0.000 | 0.057 ± 0.040 |
| Transformer | Neuromap | L2 | N/A | None | MCC | 0.001 ± 0.011 | 0.049 ± 0.050 |

Table 79: Comparing graph clustering and regularization objectives for each neural network.

| Model | $f$ | $L_{\text{regularization}}$ | $L_V$ | $\text{NN}_{S \to X}$ | ES | MCC | Accuracy |
|---|---|---|---|---|---|---|---|
| Transformer | $\text{SBM}_{\text{NN}}$ | None | N/A | None | Loss | $0.000 \pm 0.000$ | $0.041 \pm 0.014$ |
| Transformer | $\text{SBM}_{\text{NN}}$ | DMoN | N/A | None | Loss | $0.000 \pm 0.000$ | $0.039 \pm 0.000$ |
| Transformer | $\text{SBM}_{\text{NN}}$ | L2 | N/A | None | Loss | $0.054 \pm 0.074$ | $0.092 \pm 0.079$ |
| MLP | None | None | N/A | None | Loss | $0.566 \pm 0.084$ | $0.589 \pm 0.101$ |
| MLP | None | DMoN | N/A | None | Loss | $0.246 \pm 0.034$ | $0.260 \pm 0.038$ |
| MLP | None | L2 | N/A | None | MCC | $0.016 \pm 0.020$ | $0.092 \pm 0.063$ |
| MLP | DMoN | None | N/A | None | Loss | $0.467 \pm 0.067$ | $0.493 \pm 0.077$ |
| MLP | DMoN | DMoN | N/A | None | Loss | $0.475 \pm 0.079$ | $0.497 \pm 0.084$ |
| MLP | DMoN | L2 | N/A | None | MCC | $0.024 \pm 0.026$ | $0.115 \pm 0.089$ |
| MLP | NOCD | None | N/A | None | Loss | $0.155 \pm 0.155$ | $0.186 \pm 0.156$ |
| MLP | NOCD | DMoN | N/A | None | Loss | $0.126 \pm 0.077$ | $0.146 \pm 0.073$ |
| MLP | NOCD | L2 | N/A | None | Loss | $\mathbf{0.710 \pm 0.046}$ | $\mathbf{0.738 \pm 0.044}$ |
| MLP | Neuromap | None | N/A | None | MCC | $0.030 \pm 0.045$ | $0.041 \pm 0.057$ |
| MLP | Neuromap | DMoN | N/A | None | MCC | $0.044 \pm 0.044$ | $0.053 \pm 0.047$ |
| MLP | Neuromap | L2 | N/A | None | MCC | $-0.002 \pm 0.019$ | $0.078 \pm 0.084$ |
| MLP | $\text{SBM}_{\text{NN}}$ | None | N/A | None | MCC | $0.006 \pm 0.015$ | $0.025 \pm 0.038$ |
| MLP | $\text{SBM}_{\text{NN}}$ | DMoN | N/A | None | MCC | $0.021 \pm 0.035$ | $0.040 \pm 0.040$ |
| MLP | $\text{SBM}_{\text{NN}}$ | L2 | N/A | None | Loss | $0.092 \pm 0.140$ | $0.132 \pm 0.135$ |

Table 80: Comparing graph clustering and regularization objectives with an ablation study of regularization. The ablation compares clustering with clusteirng with regularization, clustering without regularization, and regularization without clustering for each neural network. Model selection based on validation set MCC.

| Model | $f$ | $L_{\text{regularization}}$ | $L_V$ | $\text{NN}_{S \to X}$ | ES | MCC | Accuracy |
|---|---|---|---|---|---|---|---|
| GCN | None | None | ✗ | None | MCC | $\mathbf{0.757 \pm 0.051}$ | $\mathbf{0.779 \pm 0.052}$ |
| GCN | None | DMoN | ✗ | None | MCC | $0.748 \pm 0.052$ | $0.770 \pm 0.050$ |
| GCN | None | L2 | ✗ | None | Loss | $0.000 \pm 0.000$ | $0.039 \pm 0.000$ |
| GCN | DMoN | None | ✗ | None | MCC | $0.756 \pm 0.045$ | $0.778 \pm 0.042$ |
| GCN | DMoN | DMoN | ✓ | Transformer | MCC | $0.754 \pm 0.040$ | $0.776 \pm 0.040$ |
| GCN | DMoN | L2 | ✓ | MLP | MCC | $0.007 \pm 0.022$ | $0.079 \pm 0.060$ |
| GCN | NOCD | None | ✗ | None | MCC | $0.551 \pm 0.118$ | $0.573 \pm 0.117$ |
| GCN | NOCD | DMoN | ✓ | Transformer | MCC | $0.605 \pm 0.097$ | $0.628 \pm 0.101$ |
| GCN | NOCD | L2 | ✗ | None | MCC | $0.052 \pm 0.057$ | $0.085 \pm 0.071$ |
| GCN | Neuromap | None | ✗ | None | MCC | $0.209 \pm 0.174$ | $0.206 \pm 0.172$ |
| GCN | Neuromap | DMoN | ✗ | None | MCC | $0.283 \pm 0.157$ | $0.282 \pm 0.177$ |
| GCN | Neuromap | L2 | ✗ | None | MCC | $0.047 \pm 0.099$ | $0.126 \pm 0.079$ |
| GCN | $\text{SBM}_{\text{NN}}$ | None | ✓ | Transformer | MCC | $0.509 \pm 0.119$ | $0.528 \pm 0.125$ |
| GCN | $\text{SBM}_{\text{NN}}$ | DMoN | ✓ | MLP | MCC | $0.553 \pm 0.142$ | $0.568 \pm 0.154$ |
| GCN | $\text{SBM}_{\text{NN}}$ | L2 | ✓ | Transformer | MCC | $0.043 \pm 0.053$ | $0.083 \pm 0.045$ |
| GraphSAGE | None | None | ✗ | None | MCC | $0.735 \pm 0.069$ | $0.757 \pm 0.066$ |
| GraphSAGE | None | DMoN | ✗ | None | MCC | $0.758 \pm 0.042$ | $0.778 \pm 0.041$ |
| GraphSAGE | None | L2 | ✗ | None | Loss | $0.000 \pm 0.000$ | $0.044 \pm 0.035$ |
| GraphSAGE | DMoN | None | ✗ | None | MCC | $0.758 \pm 0.029$ | $0.778 \pm 0.028$ |
| GraphSAGE | DMoN | DMoN | ✗ | None | MCC | $\mathbf{0.764 \pm 0.037}$ | $\mathbf{0.784 \pm 0.037}$ |
| GraphSAGE | DMoN | L2 | ✓ | Transformer | MCC | $0.003 \pm 0.008$ | $0.019 \pm 0.059$ |

Table 80: Comparing graph clustering and regularization objectives for each neural network.

| Model | $f$ | $L_{\text{regularization}}$ | $L_V$ | NN$_{S \to X}$ | ES | MCC | Accuracy |
|---|---|---|---|---|---|---|---|
| GraphSAGE | NOCD | None | ✓ | MLP | MCC | $0.514 \pm 0.137$ | $0.539 \pm 0.143$ |
| GraphSAGE | NOCD | DMoN | ✓ | Transformer | MCC | $0.589 \pm 0.076$ | $0.620 \pm 0.070$ |
| GraphSAGE | NOCD | L2 | ✓ | MLP | MCC | $0.002 \pm 0.007$ | $0.006 \pm 0.013$ |
| GraphSAGE | Neuromap | None | ✓ | MLP | MCC | $0.111 \pm 0.102$ | $0.098 \pm 0.109$ |
| GraphSAGE | Neuromap | DMoN | ✓ | MLP | MCC | $0.097 \pm 0.105$ | $0.106 \pm 0.112$ |
| GraphSAGE | Neuromap | L2 | ✓ | MLP | MCC | $0.006 \pm 0.018$ | $0.012 \pm 0.039$ |
| GraphSAGE | SBM$_{\text{NN}}$ | None | ✗ | None | MCC | $0.289 \pm 0.105$ | $0.260 \pm 0.106$ |
| GraphSAGE | SBM$_{\text{NN}}$ | DMoN | ✗ | None | MCC | $0.281 \pm 0.091$ | $0.257 \pm 0.104$ |
| GraphSAGE | SBM$_{\text{NN}}$ | L2 | ✗ | None | MCC | $0.002 \pm 0.010$ | $0.017 \pm 0.029$ |
| Transformer | None | None | N/A | None | MCC | $0.042 \pm 0.049$ | $0.056 \pm 0.081$ |
| Transformer | None | DMoN | N/A | None | MCC | $0.039 \pm 0.046$ | $0.059 \pm 0.060$ |
| Transformer | None | L2 | N/A | None | MCC | $0.476 \pm 0.120$ | $0.516 \pm 0.130$ |
| Transformer | DMoN | None | N/A | None | MCC | $0.021 \pm 0.047$ | $0.050 \pm 0.086$ |
| Transformer | DMoN | DMoN | N/A | None | MCC | $0.045 \pm 0.080$ | $0.068 \pm 0.099$ |
| Transformer | DMoN | L2 | N/A | None | MCC | $0.518 \pm 0.104$ | $0.552 \pm 0.112$ |
| Transformer | NOCD | None | N/A | None | Loss | $0.000 \pm 0.000$ | $0.040 \pm 0.004$ |
| Transformer | NOCD | DMoN | N/A | None | Loss | $0.000 \pm 0.000$ | $0.039 \pm 0.000$ |
| Transformer | NOCD | L2 | N/A | None | Loss | $\mathbf{0.677 \pm 0.241}$ | $\mathbf{0.699 \pm 0.245}$ |
| Transformer | Neuromap | None | N/A | None | Loss | $0.000 \pm 0.000$ | $0.059 \pm 0.036$ |
| Transformer | Neuromap | DMoN | N/A | None | Loss | $0.000 \pm 0.000$ | $0.057 \pm 0.040$ |
| Transformer | Neuromap | L2 | N/A | None | MCC | $0.001 \pm 0.011$ | $0.049 \pm 0.050$ |
| Transformer | SBM$_{\text{NN}}$ | None | N/A | None | Loss | $0.000 \pm 0.000$ | $0.041 \pm 0.014$ |
| Transformer | SBM$_{\text{NN}}$ | DMoN | N/A | None | Loss | $0.000 \pm 0.000$ | $0.039 \pm 0.000$ |
| Transformer | SBM$_{\text{NN}}$ | L2 | N/A | None | Loss | $0.054 \pm 0.074$ | $0.092 \pm 0.079$ |
| MLP | None | None | N/A | None | MCC | $0.602 \pm 0.061$ | $0.639 \pm 0.059$ |
| MLP | None | DMoN | N/A | None | MCC | $0.514 \pm 0.061$ | $0.548 \pm 0.066$ |
| MLP | None | L2 | N/A | None | MCC | $0.016 \pm 0.020$ | $0.092 \pm 0.063$ |
| MLP | DMoN | None | N/A | None | MCC | $0.527 \pm 0.048$ | $0.554 \pm 0.058$ |
| MLP | DMoN | DMoN | N/A | None | MCC | $0.510 \pm 0.030$ | $0.544 \pm 0.035$ |
| MLP | DMoN | L2 | N/A | None | MCC | $0.024 \pm 0.026$ | $0.115 \pm 0.089$ |
| MLP | NOCD | None | N/A | None | MCC | $0.166 \pm 0.087$ | $0.133 \pm 0.077$ |
| MLP | NOCD | DMoN | N/A | None | MCC | $0.146 \pm 0.088$ | $0.121 \pm 0.095$ |
| MLP | NOCD | L2 | N/A | None | Loss | $\mathbf{0.710 \pm 0.046}$ | $\mathbf{0.738 \pm 0.044}$ |
| MLP | Neuromap | None | N/A | None | MCC | $0.030 \pm 0.045$ | $0.041 \pm 0.057$ |
| MLP | Neuromap | DMoN | N/A | None | MCC | $0.044 \pm 0.044$ | $0.053 \pm 0.047$ |
| MLP | Neuromap | L2 | N/A | None | Loss | $0.000 \pm 0.000$ | $0.038 \pm 0.036$ |
| MLP | SBM$_{\text{NN}}$ | None | N/A | None | MCC | $0.006 \pm 0.015$ | $0.025 \pm 0.038$ |
| MLP | SBM$_{\text{NN}}$ | DMoN | N/A | None | MCC | $0.021 \pm 0.035$ | $0.040 \pm 0.040$ |
| MLP | SBM$_{\text{NN}}$ | L2 | N/A | None | Loss | $0.092 \pm 0.140$ | $0.132 \pm 0.135$ |

F.4.5   AMAZON COMPUTERS DATASET, DEFAULT SPLIT WITH 20 TRAIN NODES PER CLASS AND 500 VALIDATION NODES.

Table 81: Comparing graph clustering and regularization objectives with an ablation study of regularization. The ablation compares clustering with clusteirng with regularization, clustering without regularization, and regularization without clustering for each neural network. Model selection based on training loss.

| Model | $f$ | $L_{\text{regularization}}$ | $L_V$ | $\text{NN}_{\boldsymbol{S}\rightarrow\boldsymbol{X}}$ | ES | MCC | Accuracy |
|---|---|---|---|---|---|---|---|
| GCN | None | None | ✗ | None | Loss | $0.716 \pm 0.050$ | $0.763 \pm 0.045$ |
| GCN | None | DMoN | ✗ | None | Loss | $\mathbf{0.723 \pm 0.036}$ | $\mathbf{0.770 \pm 0.032}$ |
| GCN | None | L2 | ✗ | None | Loss | $0.000 \pm 0.000$ | $0.031 \pm 0.000$ |
| GCN | DMoN | None | ✓ | Transformer | Loss | $0.706 \pm 0.037$ | $0.754 \pm 0.030$ |
| GCN | DMoN | DMoN | ✗ | None | Loss | $0.709 \pm 0.041$ | $0.756 \pm 0.038$ |
| GCN | DMoN | L2 | ✗ | None | Loss | $0.000 \pm 0.000$ | $0.031 \pm 0.000$ |
| GCN | NOCD | None | ✓ | MLP | Loss | $0.587 \pm 0.069$ | $0.626 \pm 0.082$ |
| GCN | NOCD | DMoN | ✓ | MLP | Loss | $0.578 \pm 0.102$ | $0.606 \pm 0.132$ |
| GCN | NOCD | L2 | ✗ | None | Loss | $0.000 \pm 0.000$ | $0.112 \pm 0.106$ |
| GCN | Neuromap | None | ✓ | Transformer | Loss | $-0.002 \pm 0.004$ | $0.153 \pm 0.161$ |
| GCN | Neuromap | DMoN | ✓ | Transformer | Loss | $-0.001 \pm 0.002$ | $0.056 \pm 0.054$ |
| GCN | Neuromap | L2 | ✗ | None | Loss | $0.000 \pm 0.000$ | $0.031 \pm 0.000$ |
| GCN | $\text{SBM}_{\text{NN}}$ | None | ✗ | None | Loss | $0.469 \pm 0.074$ | $0.489 \pm 0.098$ |
| GCN | $\text{SBM}_{\text{NN}}$ | DMoN | ✓ | MLP | Loss | $0.406 \pm 0.127$ | $0.392 \pm 0.177$ |
| GCN | $\text{SBM}_{\text{NN}}$ | L2 | ✗ | None | Loss | $0.000 \pm 0.000$ | $0.154 \pm 0.161$ |
| GraphSAGE | None | None | ✗ | None | Loss | $\mathbf{0.740 \pm 0.021}$ | $\mathbf{0.781 \pm 0.020}$ |
| GraphSAGE | None | DMoN | ✗ | None | Loss | $0.728 \pm 0.027$ | $0.771 \pm 0.026$ |
| GraphSAGE | None | L2 | ✗ | None | Loss | $0.000 \pm 0.000$ | $0.107 \pm 0.108$ |
| GraphSAGE | DMoN | None | ✗ | None | Loss | $0.728 \pm 0.022$ | $0.772 \pm 0.021$ |
| GraphSAGE | DMoN | DMoN | ✗ | None | Loss | $0.734 \pm 0.027$ | $0.777 \pm 0.025$ |
| GraphSAGE | DMoN | L2 | ✗ | None | Loss | $0.000 \pm 0.000$ | $0.095 \pm 0.109$ |
| GraphSAGE | NOCD | None | ✓ | Transformer | Loss | $0.372 \pm 0.086$ | $0.346 \pm 0.119$ |
| GraphSAGE | NOCD | DMoN | ✗ | None | Loss | $0.290 \pm 0.044$ | $0.225 \pm 0.086$ |
| GraphSAGE | NOCD | L2 | ✗ | None | Loss | $0.000 \pm 0.000$ | $0.103 \pm 0.112$ |
| GraphSAGE | Neuromap | None | ✓ | Transformer | Loss | $0.003 \pm 0.008$ | $0.055 \pm 0.056$ |
| GraphSAGE | Neuromap | DMoN | ✗ | None | Loss | $0.002 \pm 0.007$ | $0.151 \pm 0.135$ |
| GraphSAGE | Neuromap | L2 | ✗ | None | Loss | $0.000 \pm 0.000$ | $0.065 \pm 0.054$ |
| GraphSAGE | $\text{SBM}_{\text{NN}}$ | None | ✓ | Transformer | Loss | $0.180 \pm 0.098$ | $0.177 \pm 0.132$ |
| GraphSAGE | $\text{SBM}_{\text{NN}}$ | DMoN | ✓ | MLP | Loss | $0.095 \pm 0.082$ | $0.094 \pm 0.070$ |
| GraphSAGE | $\text{SBM}_{\text{NN}}$ | L2 | ✓ | MLP | Loss | $0.000 \pm 0.000$ | $0.111 \pm 0.111$ |
| Transformer | None | None | N/A | None | Loss | $0.000 \pm 0.000$ | $0.087 \pm 0.061$ |
| Transformer | None | DMoN | N/A | None | Loss | $0.000 \pm 0.000$ | $0.186 \pm 0.144$ |
| Transformer | None | L2 | N/A | None | Loss | $0.000 \pm 0.000$ | $0.136 \pm 0.137$ |
| Transformer | DMoN | None | N/A | None | Loss | $0.000 \pm 0.000$ | $0.138 \pm 0.136$ |
| Transformer | DMoN | DMoN | N/A | None | Loss | $0.000 \pm 0.000$ | $0.131 \pm 0.140$ |
| Transformer | DMoN | L2 | N/A | None | Loss | $0.000 \pm 0.000$ | $0.127 \pm 0.139$ |
| Transformer | NOCD | None | N/A | None | Loss | $0.000 \pm 0.000$ | $0.030 \pm 0.005$ |
| Transformer | NOCD | DMoN | N/A | None | Loss | $0.000 \pm 0.000$ | $0.055 \pm 0.054$ |
| Transformer | NOCD | L2 | N/A | None | Loss | $0.000 \pm 0.000$ | $0.121 \pm 0.111$ |
| Transformer | Neuromap | None | N/A | None | Loss | $0.000 \pm 0.000$ | $0.105 \pm 0.147$ |
| Transformer | Neuromap | DMoN | N/A | None | Loss | $0.000 \pm 0.000$ | $0.145 \pm 0.135$ |
| Transformer | Neuromap | L2 | N/A | None | Loss | $0.000 \pm 0.000$ | $\mathbf{0.214 \pm 0.176}$ |

Table 81: Comparing graph clustering and regularization objectives for each neural network.

| Model | $f$ | $L_{\text{regularization}}$ | $L_V$ | NN$_{S \rightarrow X}$ | ES | MCC | Accuracy |
|---|---|---|---|---|---|---|---|
| Transformer | SBM$_{\text{NN}}$ | None | N/A | None | Loss | $0.000 \pm 0.000$ | $0.031 \pm 0.001$ |
| Transformer | SBM$_{\text{NN}}$ | DMoN | N/A | None | Loss | $0.000 \pm 0.000$ | $0.030 \pm 0.003$ |
| Transformer | SBM$_{\text{NN}}$ | L2 | N/A | None | Loss | $\mathbf{0.002 \pm 0.005}$ | $0.092 \pm 0.110$ |
| MLP | None | None | N/A | None | Loss | $\mathbf{0.547 \pm 0.031}$ | $\mathbf{0.612 \pm 0.032}$ |
| MLP | None | DMoN | N/A | None | Loss | $0.437 \pm 0.028$ | $0.482 \pm 0.032$ |
| MLP | None | L2 | N/A | None | Loss | $0.000 \pm 0.000$ | $0.081 \pm 0.114$ |
| MLP | DMoN | None | N/A | None | Loss | $0.443 \pm 0.030$ | $0.497 \pm 0.035$ |
| MLP | DMoN | DMoN | N/A | None | Loss | $0.431 \pm 0.023$ | $0.473 \pm 0.029$ |
| MLP | DMoN | L2 | N/A | None | Loss | $0.000 \pm 0.000$ | $0.083 \pm 0.049$ |
| MLP | NOCD | None | N/A | None | Loss | $0.001 \pm 0.002$ | $0.029 \pm 0.004$ |
| MLP | NOCD | DMoN | N/A | None | Loss | $0.000 \pm 0.000$ | $0.042 \pm 0.040$ |
| MLP | NOCD | L2 | N/A | None | Loss | $0.017 \pm 0.037$ | $0.116 \pm 0.107$ |
| MLP | Neuromap | None | N/A | None | Loss | $0.000 \pm 0.001$ | $0.136 \pm 0.136$ |
| MLP | Neuromap | DMoN | N/A | None | Loss | $0.000 \pm 0.000$ | $0.126 \pm 0.103$ |
| MLP | Neuromap | L2 | N/A | None | Loss | $0.000 \pm 0.000$ | $0.079 \pm 0.058$ |
| MLP | SBM$_{\text{NN}}$ | None | N/A | None | Loss | $0.000 \pm 0.001$ | $0.031 \pm 0.000$ |
| MLP | SBM$_{\text{NN}}$ | DMoN | N/A | None | Loss | $0.000 \pm 0.000$ | $0.031 \pm 0.000$ |
| MLP | SBM$_{\text{NN}}$ | L2 | N/A | None | Loss | $0.000 \pm 0.000$ | $0.030 \pm 0.003$ |

Table 82: Comparing graph clustering and regularization objectives with an ablation study of regularization. The ablation compares clustering with clusteirng with regularization, clustering without regularization, and regularization without clustering for each neural network. Model selection based on training set MCC.

| Model | $f$ | $L_{\text{regularization}}$ | $L_V$ | NN$_{S \rightarrow X}$ | ES | MCC | Accuracy |
|---|---|---|---|---|---|---|---|
| GCN | None | None | ✗ | None | Loss | $0.716 \pm 0.050$ | $0.763 \pm 0.045$ |
| GCN | None | DMoN | ✗ | None | Loss | $\mathbf{0.723 \pm 0.036}$ | $\mathbf{0.770 \pm 0.032}$ |
| GCN | None | L2 | ✗ | None | MCC | $0.053 \pm 0.137$ | $0.198 \pm 0.128$ |
| GCN | DMoN | None | ✓ | Transformer | Loss | $0.706 \pm 0.037$ | $0.754 \pm 0.030$ |
| GCN | DMoN | DMoN | ✗ | None | Loss | $0.709 \pm 0.041$ | $0.756 \pm 0.038$ |
| GCN | DMoN | L2 | ✓ | Transformer | MCC | $0.033 \pm 0.073$ | $0.138 \pm 0.091$ |
| GCN | NOCD | None | ✓ | MLP | Loss | $0.587 \pm 0.069$ | $0.626 \pm 0.082$ |
| GCN | NOCD | DMoN | ✓ | MLP | Loss | $0.578 \pm 0.102$ | $0.606 \pm 0.132$ |
| GCN | NOCD | L2 | ✗ | None | MCC | $0.115 \pm 0.039$ | $0.195 \pm 0.074$ |
| GCN | Neuromap | None | ✗ | None | MCC | $0.086 \pm 0.066$ | $0.084 \pm 0.058$ |
| GCN | Neuromap | DMoN | ✗ | None | MCC | $0.113 \pm 0.103$ | $0.118 \pm 0.140$ |
| GCN | Neuromap | L2 | ✗ | None | MCC | $-0.022 \pm 0.149$ | $0.162 \pm 0.109$ |
| GCN | SBM$_{\text{NN}}$ | None | ✗ | None | Loss | $0.469 \pm 0.074$ | $0.489 \pm 0.098$ |
| GCN | SBM$_{\text{NN}}$ | DMoN | ✓ | MLP | Loss | $0.406 \pm 0.127$ | $0.392 \pm 0.177$ |
| GCN | SBM$_{\text{NN}}$ | L2 | ✗ | None | MCC | $0.051 \pm 0.092$ | $0.118 \pm 0.114$ |
| GraphSAGE | None | None | ✗ | None | Loss | $\mathbf{0.740 \pm 0.021}$ | $\mathbf{0.781 \pm 0.020}$ |
| GraphSAGE | None | DMoN | ✗ | None | Loss | $0.728 \pm 0.027$ | $0.771 \pm 0.026$ |
| GraphSAGE | None | L2 | ✗ | None | MCC | $0.002 \pm 0.014$ | $0.074 \pm 0.115$ |
| GraphSAGE | DMoN | None | ✗ | None | Loss | $0.728 \pm 0.022$ | $0.772 \pm 0.021$ |
| GraphSAGE | DMoN | DMoN | ✗ | None | Loss | $0.734 \pm 0.027$ | $0.777 \pm 0.025$ |
| GraphSAGE | DMoN | L2 | ✓ | Transformer | MCC | $0.002 \pm 0.015$ | $0.081 \pm 0.111$ |

Continued on next page

Table 82: Comparing graph clustering and regularization objectives for each neural network.

| Model | $f$ | $L_{\text{regularization}}$ | $L_V$ | $NN_{S \to X}$ | ES | MCC | Accuracy |
|---|---|---|---|---|---|---|---|
| GraphSAGE | NOCD | None | ✓ | Transformer | Loss | $0.372 \pm 0.086$ | $0.346 \pm 0.119$ |
| GraphSAGE | NOCD | DMoN | ✗ | None | Loss | $0.290 \pm 0.044$ | $0.225 \pm 0.086$ |
| GraphSAGE | NOCD | L2 | ✗ | None | MCC | $0.006 \pm 0.012$ | $0.052 \pm 0.060$ |
| GraphSAGE | Neuromap | None | ✓ | MLP | MCC | $0.070 \pm 0.081$ | $0.109 \pm 0.118$ |
| GraphSAGE | Neuromap | DMoN | ✓ | MLP | MCC | $0.029 \pm 0.058$ | $0.060 \pm 0.132$ |
| GraphSAGE | Neuromap | L2 | ✓ | MLP | MCC | $0.003 \pm 0.007$ | $0.040 \pm 0.051$ |
| GraphSAGE | $SBM_{NN}$ | None | ✓ | Transformer | Loss | $0.180 \pm 0.098$ | $0.177 \pm 0.132$ |
| GraphSAGE | $SBM_{NN}$ | DMoN | ✓ | MLP | Loss | $0.095 \pm 0.082$ | $0.094 \pm 0.070$ |
| GraphSAGE | $SBM_{NN}$ | L2 | ✗ | None | MCC | $0.010 \pm 0.032$ | $0.085 \pm 0.112$ |
| Transformer | None | None | N/A | None | Loss | $0.000 \pm 0.000$ | $0.087 \pm 0.061$ |
| Transformer | None | DMoN | N/A | None | MCC | $0.004 \pm 0.013$ | $0.016 \pm 0.052$ |
| Transformer | None | L2 | N/A | None | MCC | $0.031 \pm 0.050$ | $\mathbf{0.234 \pm 0.095}$ |
| Transformer | DMoN | None | N/A | None | Loss | $0.000 \pm 0.000$ | $0.138 \pm 0.136$ |
| Transformer | DMoN | DMoN | N/A | None | Loss | $0.000 \pm 0.000$ | $0.131 \pm 0.140$ |
| Transformer | DMoN | L2 | N/A | None | MCC | $\mathbf{0.048 \pm 0.037}$ | $0.193 \pm 0.106$ |
| Transformer | NOCD | None | N/A | None | Loss | $0.000 \pm 0.000$ | $0.030 \pm 0.005$ |
| Transformer | NOCD | DMoN | N/A | None | Loss | $0.000 \pm 0.000$ | $0.055 \pm 0.054$ |
| Transformer | NOCD | L2 | N/A | None | MCC | $0.030 \pm 0.035$ | $0.133 \pm 0.104$ |
| Transformer | Neuromap | None | N/A | None | Loss | $0.000 \pm 0.000$ | $0.105 \pm 0.147$ |
| Transformer | Neuromap | DMoN | N/A | None | Loss | $0.000 \pm 0.000$ | $0.145 \pm 0.135$ |
| Transformer | Neuromap | L2 | N/A | None | Loss | $0.000 \pm 0.000$ | $0.214 \pm 0.176$ |
| Transformer | $SBM_{NN}$ | None | N/A | None | Loss | $0.000 \pm 0.000$ | $0.031 \pm 0.001$ |
| Transformer | $SBM_{NN}$ | DMoN | N/A | None | Loss | $0.000 \pm 0.000$ | $0.030 \pm 0.003$ |
| Transformer | $SBM_{NN}$ | L2 | N/A | None | Loss | $0.002 \pm 0.005$ | $0.092 \pm 0.110$ |
| MLP | None | None | N/A | None | Loss | $\mathbf{0.547 \pm 0.031}$ | $\mathbf{0.612 \pm 0.032}$ |
| MLP | None | DMoN | N/A | None | Loss | $0.437 \pm 0.028$ | $0.482 \pm 0.032$ |
| MLP | None | L2 | N/A | None | MCC | $-0.017 \pm 0.048$ | $0.188 \pm 0.107$ |
| MLP | DMoN | None | N/A | None | Loss | $0.443 \pm 0.030$ | $0.497 \pm 0.035$ |
| MLP | DMoN | DMoN | N/A | None | Loss | $0.431 \pm 0.023$ | $0.473 \pm 0.029$ |
| MLP | DMoN | L2 | N/A | None | Loss | $0.000 \pm 0.000$ | $0.083 \pm 0.049$ |
| MLP | NOCD | None | N/A | None | MCC | $0.002 \pm 0.010$ | $0.014 \pm 0.026$ |
| MLP | NOCD | DMoN | N/A | None | MCC | $0.001 \pm 0.003$ | $0.041 \pm 0.051$ |
| MLP | NOCD | L2 | N/A | None | Loss | $0.017 \pm 0.037$ | $0.116 \pm 0.107$ |
| MLP | Neuromap | None | N/A | None | MCC | $0.004 \pm 0.010$ | $0.131 \pm 0.147$ |
| MLP | Neuromap | DMoN | N/A | None | MCC | $0.004 \pm 0.007$ | $0.042 \pm 0.052$ |
| MLP | Neuromap | L2 | N/A | None | Loss | $0.000 \pm 0.000$ | $0.079 \pm 0.058$ |
| MLP | $SBM_{NN}$ | None | N/A | None | MCC | $0.009 \pm 0.022$ | $0.089 \pm 0.120$ |
| MLP | $SBM_{NN}$ | DMoN | N/A | None | MCC | $0.001 \pm 0.005$ | $0.052 \pm 0.060$ |
| MLP | $SBM_{NN}$ | L2 | N/A | None | MCC | $0.003 \pm 0.004$ | $0.072 \pm 0.113$ |

Table 83: Comparing graph clustering and regularization objectives with an ablation study of regularization. The ablation compares clustering with clusteirng with regularization, clustering without regularization, and regularization without clustering for each neural network. Model selection based on validation set MCC.

| Model | $f$ | $L_{\text{regularization}}$ | $L_V$ | NN$_{\boldsymbol{S}\to\boldsymbol{X}}$ | ES | MCC | Accuracy |
|---|---|---|---|---|---|---|---|
| GCN | None | None | ✗ | None | Loss | $0.716 \pm 0.050$ | $0.763 \pm 0.045$ |
| GCN | None | DMoN | ✗ | None | Loss | $\mathbf{0.723 \pm 0.036}$ | $\mathbf{0.770 \pm 0.032}$ |
| GCN | None | L2 | ✗ | None | MCC | $0.053 \pm 0.137$ | $0.198 \pm 0.128$ |
| GCN | DMoN | None | ✓ | Transformer | Loss | $0.706 \pm 0.037$ | $0.754 \pm 0.030$ |
| GCN | DMoN | DMoN | ✗ | None | Loss | $0.709 \pm 0.041$ | $0.756 \pm 0.038$ |
| GCN | DMoN | L2 | ✓ | Transformer | MCC | $0.033 \pm 0.073$ | $0.138 \pm 0.091$ |
| GCN | NOCD | None | ✓ | MLP | Loss | $0.587 \pm 0.069$ | $0.626 \pm 0.082$ |
| GCN | NOCD | DMoN | ✓ | Transformer | MCC | $0.576 \pm 0.120$ | $0.630 \pm 0.127$ |
| GCN | NOCD | L2 | ✗ | None | MCC | $0.115 \pm 0.039$ | $0.195 \pm 0.074$ |
| GCN | Neuromap | None | ✗ | None | MCC | $0.086 \pm 0.066$ | $0.084 \pm 0.058$ |
| GCN | Neuromap | DMoN | ✗ | None | MCC | $0.113 \pm 0.103$ | $0.118 \pm 0.140$ |
| GCN | Neuromap | L2 | ✗ | None | Loss | $0.000 \pm 0.000$ | $0.031 \pm 0.000$ |
| GCN | SBM$_{\text{NN}}$ | None | ✗ | None | Loss | $0.469 \pm 0.074$ | $0.489 \pm 0.098$ |
| GCN | SBM$_{\text{NN}}$ | DMoN | ✓ | Transformer | Loss | $0.455 \pm 0.168$ | $0.471 \pm 0.206$ |
| GCN | SBM$_{\text{NN}}$ | L2 | ✗ | None | MCC | $0.072 \pm 0.042$ | $0.148 \pm 0.102$ |
| GraphSAGE | None | None | ✗ | None | Loss | $0.740 \pm 0.021$ | $0.781 \pm 0.020$ |
| GraphSAGE | None | DMoN | ✗ | None | MCC | $\mathbf{0.751 \pm 0.020}$ | $\mathbf{0.794 \pm 0.018}$ |
| GraphSAGE | None | L2 | ✗ | None | MCC | $0.002 \pm 0.014$ | $0.074 \pm 0.115$ |
| GraphSAGE | DMoN | None | ✗ | None | MCC | $0.732 \pm 0.028$ | $0.777 \pm 0.026$ |
| GraphSAGE | DMoN | DMoN | ✗ | None | MCC | $0.726 \pm 0.075$ | $0.765 \pm 0.086$ |
| GraphSAGE | DMoN | L2 | ✗ | None | MCC | $0.003 \pm 0.020$ | $0.079 \pm 0.107$ |
| GraphSAGE | NOCD | None | ✓ | MLP | MCC | $0.555 \pm 0.148$ | $0.574 \pm 0.172$ |
| GraphSAGE | NOCD | DMoN | ✓ | Transformer | MCC | $0.552 \pm 0.132$ | $0.584 \pm 0.166$ |
| GraphSAGE | NOCD | L2 | ✗ | None | MCC | $0.006 \pm 0.012$ | $0.052 \pm 0.060$ |
| GraphSAGE | Neuromap | None | ✓ | MLP | MCC | $0.070 \pm 0.081$ | $0.109 \pm 0.118$ |
| GraphSAGE | Neuromap | DMoN | ✓ | MLP | MCC | $0.029 \pm 0.058$ | $0.060 \pm 0.132$ |
| GraphSAGE | Neuromap | L2 | ✗ | None | MCC | $0.000 \pm 0.039$ | $0.083 \pm 0.095$ |
| GraphSAGE | SBM$_{\text{NN}}$ | None | ✓ | MLP | MCC | $0.212 \pm 0.134$ | $0.237 \pm 0.122$ |
| GraphSAGE | SBM$_{\text{NN}}$ | DMoN | ✗ | None | MCC | $0.231 \pm 0.136$ | $0.229 \pm 0.165$ |
| GraphSAGE | SBM$_{\text{NN}}$ | L2 | ✗ | None | MCC | $0.010 \pm 0.032$ | $0.085 \pm 0.112$ |
| Transformer | None | None | N/A | None | Loss | $0.000 \pm 0.000$ | $0.087 \pm 0.061$ |
| Transformer | None | DMoN | N/A | None | MCC | $0.004 \pm 0.013$ | $0.016 \pm 0.052$ |
| Transformer | None | L2 | N/A | None | MCC | $0.031 \pm 0.050$ | $\mathbf{0.234 \pm 0.095}$ |
| Transformer | DMoN | None | N/A | None | Loss | $0.000 \pm 0.000$ | $0.138 \pm 0.136$ |
| Transformer | DMoN | DMoN | N/A | None | Loss | $0.000 \pm 0.000$ | $0.131 \pm 0.140$ |
| Transformer | DMoN | L2 | N/A | None | MCC | $\mathbf{0.048 \pm 0.037}$ | $0.193 \pm 0.106$ |
| Transformer | NOCD | None | N/A | None | Loss | $0.000 \pm 0.000$ | $0.030 \pm 0.005$ |
| Transformer | NOCD | DMoN | N/A | None | Loss | $0.000 \pm 0.000$ | $0.055 \pm 0.054$ |
| Transformer | NOCD | L2 | N/A | None | MCC | $0.030 \pm 0.035$ | $0.133 \pm 0.104$ |
| Transformer | Neuromap | None | N/A | None | Loss | $0.000 \pm 0.000$ | $0.105 \pm 0.147$ |
| Transformer | Neuromap | DMoN | N/A | None | Loss | $0.000 \pm 0.000$ | $0.145 \pm 0.135$ |
| Transformer | Neuromap | L2 | N/A | None | Loss | $0.000 \pm 0.000$ | $0.214 \pm 0.176$ |

Continued on next page

Table 83: Comparing graph clustering and regularization objectives for each neural network.

| Model | $f$ | $L_{\text{regularization}}$ | $L_V$ | $NN_{S \to X}$ | ES | MCC | Accuracy |
|---|---|---|---|---|---|---|---|
| Transformer | $SBM_{NN}$ | None | N/A | None | Loss | $0.000 \pm 0.000$ | $0.031 \pm 0.001$ |
| Transformer | $SBM_{NN}$ | DMoN | N/A | None | Loss | $0.000 \pm 0.000$ | $0.030 \pm 0.003$ |
| Transformer | $SBM_{NN}$ | L2 | N/A | None | MCC | $0.003 \pm 0.007$ | $0.030 \pm 0.062$ |
| MLP | None | None | N/A | None | MCC | $\mathbf{0.586 \pm 0.026}$ | $\mathbf{0.656 \pm 0.024}$ |
| MLP | None | DMoN | N/A | None | MCC | $0.552 \pm 0.025$ | $0.619 \pm 0.024$ |
| MLP | None | L2 | N/A | None | Loss | $0.000 \pm 0.000$ | $0.081 \pm 0.114$ |
| MLP | DMoN | None | N/A | None | MCC | $0.553 \pm 0.027$ | $0.616 \pm 0.026$ |
| MLP | DMoN | DMoN | N/A | None | MCC | $0.550 \pm 0.033$ | $0.616 \pm 0.039$ |
| MLP | DMoN | L2 | N/A | None | MCC | $0.001 \pm 0.053$ | $0.159 \pm 0.093$ |
| MLP | NOCD | None | N/A | None | Loss | $0.001 \pm 0.002$ | $0.029 \pm 0.004$ |
| MLP | NOCD | DMoN | N/A | None | MCC | $0.001 \pm 0.003$ | $0.041 \pm 0.051$ |
| MLP | NOCD | L2 | N/A | None | Loss | $0.017 \pm 0.037$ | $0.116 \pm 0.107$ |
| MLP | Neuromap | None | N/A | None | Loss | $0.000 \pm 0.001$ | $0.136 \pm 0.136$ |
| MLP | Neuromap | DMoN | N/A | None | MCC | $0.004 \pm 0.007$ | $0.042 \pm 0.052$ |
| MLP | Neuromap | L2 | N/A | None | Loss | $0.000 \pm 0.000$ | $0.079 \pm 0.058$ |
| MLP | $SBM_{NN}$ | None | N/A | None | MCC | $0.009 \pm 0.022$ | $0.089 \pm 0.120$ |
| MLP | $SBM_{NN}$ | DMoN | N/A | None | MCC | $0.001 \pm 0.005$ | $0.052 \pm 0.060$ |
| MLP | $SBM_{NN}$ | L2 | N/A | None | MCC | $0.003 \pm 0.004$ | $0.072 \pm 0.113$ |

### F.4.6 AMAZON COMPUTERS DATASET, SPARSE SPLIT WITH 2 TRAIN NODES PER CLASS AND 50 VALIDATION NODES.

Table 84: Comparing graph clustering and regularization objectives with an ablation study of regularization. The ablation compares clustering with clusteirng with regularization, clustering without regularization, and regularization without clustering for each neural network. Model selection based on training loss.

| Model | $f$ | $L_{\text{regularization}}$ | $L_V$ | $NN_{S \to X}$ | ES | MCC | Accuracy |
|---|---|---|---|---|---|---|---|
| GCN | None | None | ✗ | None | Loss | $0.502 \pm 0.096$ | $0.569 \pm 0.101$ |
| GCN | None | DMoN | ✗ | None | Loss | $\mathbf{0.521 \pm 0.064}$ | $\mathbf{0.580 \pm 0.069}$ |
| GCN | None | L2 | ✗ | None | Loss | $0.000 \pm 0.000$ | $0.032 \pm 0.000$ |
| GCN | DMoN | None | ✗ | None | Loss | $0.476 \pm 0.086$ | $0.529 \pm 0.097$ |
| GCN | DMoN | DMoN | ✓ | MLP | Loss | $0.507 \pm 0.065$ | $0.574 \pm 0.065$ |
| GCN | DMoN | L2 | ✗ | None | Loss | $0.000 \pm 0.000$ | $0.032 \pm 0.000$ |
| GCN | NOCD | None | ✗ | None | Loss | $0.328 \pm 0.094$ | $0.302 \pm 0.147$ |
| GCN | NOCD | DMoN | ✓ | MLP | Loss | $0.297 \pm 0.070$ | $0.271 \pm 0.085$ |
| GCN | NOCD | L2 | ✗ | None | Loss | $0.000 \pm 0.000$ | $0.067 \pm 0.054$ |
| GCN | Neuromap | None | ✓ | MLP | Loss | $0.000 \pm 0.001$ | $0.075 \pm 0.052$ |
| GCN | Neuromap | DMoN | ✓ | Transformer | Loss | $-0.001 \pm 0.002$ | $0.101 \pm 0.114$ |
| GCN | Neuromap | L2 | ✗ | None | Loss | $0.000 \pm 0.000$ | $0.032 \pm 0.000$ |
| GCN | $SBM_{NN}$ | None | ✓ | Transformer | Loss | $0.294 \pm 0.051$ | $0.318 \pm 0.082$ |
| GCN | $SBM_{NN}$ | DMoN | ✓ | MLP | Loss | $0.271 \pm 0.084$ | $0.260 \pm 0.110$ |
| GCN | $SBM_{NN}$ | L2 | ✗ | None | Loss | $0.000 \pm 0.000$ | $0.093 \pm 0.068$ |
| GraphSAGE | None | None | ✗ | None | Loss | $0.427 \pm 0.105$ | $0.470 \pm 0.120$ |
| GraphSAGE | None | DMoN | ✗ | None | Loss | $\mathbf{0.466 \pm 0.105}$ | $\mathbf{0.522 \pm 0.114}$ |
| GraphSAGE | None | L2 | ✗ | None | Loss | $0.000 \pm 0.000$ | $0.063 \pm 0.052$ |

Table 84: Comparing graph clustering and regularization objectives for each neural network.

| Model | $f$ | $L_{\text{regularization}}$ | $L_V$ | NN$_{S \to X}$ | ES | MCC | Accuracy |
|---|---|---|---|---|---|---|---|
| GraphSAGE | DMoN | None | ✗ | None | Loss | $0.389 \pm 0.089$ | $0.434 \pm 0.103$ |
| GraphSAGE | DMoN | DMoN | ✗ | None | Loss | $0.453 \pm 0.092$ | $0.504 \pm 0.100$ |
| GraphSAGE | DMoN | L2 | ✗ | None | Loss | $0.000 \pm 0.000$ | $0.091 \pm 0.113$ |
| GraphSAGE | NOCD | None | ✗ | None | Loss | $0.260 \pm 0.119$ | $0.282 \pm 0.127$ |
| GraphSAGE | NOCD | DMoN | ✓ | Transformer | Loss | $0.210 \pm 0.075$ | $0.162 \pm 0.096$ |
| GraphSAGE | NOCD | L2 | ✗ | None | Loss | $0.000 \pm 0.000$ | $0.073 \pm 0.049$ |
| GraphSAGE | Neuromap | None | ✓ | Transformer | Loss | $0.004 \pm 0.020$ | $0.125 \pm 0.137$ |
| GraphSAGE | Neuromap | DMoN | ✓ | Transformer | Loss | $-0.001 \pm 0.004$ | $0.109 \pm 0.108$ |
| GraphSAGE | Neuromap | L2 | ✗ | None | Loss | $0.000 \pm 0.000$ | $0.117 \pm 0.143$ |
| GraphSAGE | SBM$_{NN}$ | None | ✗ | None | Loss | $0.082 \pm 0.094$ | $0.089 \pm 0.098$ |
| GraphSAGE | SBM$_{NN}$ | DMoN | ✓ | MLP | Loss | $0.046 \pm 0.063$ | $0.054 \pm 0.020$ |
| GraphSAGE | SBM$_{NN}$ | L2 | ✗ | None | Loss | $0.000 \pm 0.000$ | $0.097 \pm 0.107$ |
| Transformer | None | None | N/A | None | Loss | $0.000 \pm 0.000$ | $0.075 \pm 0.051$ |
| Transformer | None | DMoN | N/A | None | Loss | $0.000 \pm 0.000$ | $0.120 \pm 0.107$ |
| Transformer | None | L2 | N/A | None | Loss | $-0.000 \pm 0.000$ | $0.078 \pm 0.059$ |
| Transformer | DMoN | None | N/A | None | Loss | $0.000 \pm 0.000$ | $0.133 \pm 0.138$ |
| Transformer | DMoN | DMoN | N/A | None | Loss | $0.000 \pm 0.000$ | $0.090 \pm 0.109$ |
| Transformer | DMoN | L2 | N/A | None | Loss | $0.000 \pm 0.000$ | $0.156 \pm 0.157$ |
| Transformer | NOCD | None | N/A | None | Loss | $0.000 \pm 0.000$ | $0.032 \pm 0.004$ |
| Transformer | NOCD | DMoN | N/A | None | Loss | $0.000 \pm 0.000$ | $0.050 \pm 0.044$ |
| Transformer | NOCD | L2 | N/A | None | Loss | $\mathbf{0.106 \pm 0.093}$ | $\mathbf{0.177 \pm 0.105}$ |
| Transformer | Neuromap | None | N/A | None | Loss | $0.000 \pm 0.000$ | $0.122 \pm 0.136$ |
| Transformer | Neuromap | DMoN | N/A | None | Loss | $0.000 \pm 0.000$ | $0.073 \pm 0.041$ |
| Transformer | Neuromap | L2 | N/A | None | Loss | $0.000 \pm 0.000$ | $0.071 \pm 0.054$ |
| Transformer | SBM$_{NN}$ | None | N/A | None | Loss | $0.000 \pm 0.000$ | $0.031 \pm 0.003$ |
| Transformer | SBM$_{NN}$ | DMoN | N/A | None | Loss | $0.000 \pm 0.000$ | $0.032 \pm 0.001$ |
| Transformer | SBM$_{NN}$ | L2 | N/A | None | Loss | $0.006 \pm 0.019$ | $0.099 \pm 0.108$ |
| MLP | None | None | N/A | None | Loss | $0.150 \pm 0.053$ | $0.207 \pm 0.057$ |
| MLP | None | DMoN | N/A | None | Loss | $0.192 \pm 0.035$ | $\mathbf{0.278 \pm 0.048}$ |
| MLP | None | L2 | N/A | None | Loss | $0.000 \pm 0.000$ | $0.098 \pm 0.111$ |
| MLP | DMoN | None | N/A | None | Loss | $\mathbf{0.195 \pm 0.065}$ | $0.257 \pm 0.109$ |
| MLP | DMoN | DMoN | N/A | None | Loss | $0.189 \pm 0.042$ | $0.260 \pm 0.070$ |
| MLP | DMoN | L2 | N/A | None | Loss | $0.000 \pm 0.000$ | $0.111 \pm 0.109$ |
| MLP | NOCD | None | N/A | None | Loss | $0.003 \pm 0.005$ | $0.040 \pm 0.024$ |
| MLP | NOCD | DMoN | N/A | None | Loss | $0.002 \pm 0.005$ | $0.040 \pm 0.023$ |
| MLP | NOCD | L2 | N/A | None | Loss | $0.108 \pm 0.076$ | $0.166 \pm 0.123$ |
| MLP | Neuromap | None | N/A | None | Loss | $0.000 \pm 0.000$ | $0.125 \pm 0.105$ |
| MLP | Neuromap | DMoN | N/A | None | Loss | $0.000 \pm 0.000$ | $0.137 \pm 0.136$ |
| MLP | Neuromap | L2 | N/A | None | Loss | $0.000 \pm 0.000$ | $0.092 \pm 0.108$ |
| MLP | SBM$_{NN}$ | None | N/A | None | Loss | $-0.000 \pm 0.000$ | $0.097 \pm 0.110$ |
| MLP | SBM$_{NN}$ | DMoN | N/A | None | Loss | $0.000 \pm 0.000$ | $0.032 \pm 0.000$ |
| MLP | SBM$_{NN}$ | L2 | N/A | None | Loss | $0.006 \pm 0.017$ | $0.057 \pm 0.052$ |

Table 85: Comparing graph clustering and regularization objectives with an ablation study of regularization. The ablation compares clustering with clusteirng with regularization, clustering without regularization, and regularization without clustering for each neural network. Model selection based on training set MCC.

| Model | $f$ | $L_{\text{regularization}}$ | $L_V$ | NN$_{\boldsymbol{S}\to\boldsymbol{X}}$ | ES | MCC | Accuracy |
|---|---|---|---|---|---|---|---|
| GCN | None | None | ✗ | None | Loss | $0.502 \pm 0.096$ | $0.569 \pm 0.101$ |
| GCN | None | DMoN | ✗ | None | Loss | $\mathbf{0.521 \pm 0.064}$ | $\mathbf{0.580 \pm 0.069}$ |
| GCN | None | L2 | ✗ | None | MCC | $0.030 \pm 0.102$ | $0.152 \pm 0.098$ |
| GCN | DMoN | None | ✗ | None | Loss | $0.476 \pm 0.086$ | $0.529 \pm 0.097$ |
| GCN | DMoN | DMoN | ✓ | MLP | Loss | $0.507 \pm 0.065$ | $0.574 \pm 0.065$ |
| GCN | DMoN | L2 | ✓ | MLP | MCC | $0.029 \pm 0.054$ | $0.219 \pm 0.108$ |
| GCN | NOCD | None | ✗ | None | Loss | $0.328 \pm 0.094$ | $0.302 \pm 0.147$ |
| GCN | NOCD | DMoN | ✓ | MLP | Loss | $0.297 \pm 0.070$ | $0.271 \pm 0.085$ |
| GCN | NOCD | L2 | ✓ | MLP | MCC | $0.057 \pm 0.074$ | $0.173 \pm 0.126$ |
| GCN | Neuromap | None | ✗ | None | MCC | $0.055 \pm 0.066$ | $0.086 \pm 0.071$ |
| GCN | Neuromap | DMoN | ✗ | None | MCC | $0.049 \pm 0.063$ | $0.061 \pm 0.070$ |
| GCN | Neuromap | L2 | ✗ | None | MCC | $0.013 \pm 0.099$ | $0.183 \pm 0.107$ |
| GCN | SBM$_{\text{NN}}$ | None | ✓ | Transformer | Loss | $0.294 \pm 0.051$ | $0.318 \pm 0.082$ |
| GCN | SBM$_{\text{NN}}$ | DMoN | ✓ | MLP | Loss | $0.271 \pm 0.084$ | $0.260 \pm 0.110$ |
| GCN | SBM$_{\text{NN}}$ | L2 | ✓ | MLP | MCC | $0.039 \pm 0.063$ | $0.171 \pm 0.114$ |
| GraphSAGE | None | None | ✗ | None | Loss | $0.427 \pm 0.105$ | $0.470 \pm 0.120$ |
| GraphSAGE | None | DMoN | ✗ | None | Loss | $\mathbf{0.466 \pm 0.105}$ | $\mathbf{0.522 \pm 0.114}$ |
| GraphSAGE | None | L2 | ✗ | None | Loss | $0.000 \pm 0.000$ | $0.063 \pm 0.052$ |
| GraphSAGE | DMoN | None | ✗ | None | Loss | $0.389 \pm 0.089$ | $0.434 \pm 0.103$ |
| GraphSAGE | DMoN | DMoN | ✗ | None | Loss | $0.453 \pm 0.092$ | $0.504 \pm 0.100$ |
| GraphSAGE | DMoN | L2 | ✗ | None | Loss | $0.000 \pm 0.000$ | $0.091 \pm 0.113$ |
| GraphSAGE | NOCD | None | ✓ | MLP | Loss | $0.234 \pm 0.134$ | $0.245 \pm 0.156$ |
| GraphSAGE | NOCD | DMoN | ✓ | Transformer | Loss | $0.210 \pm 0.075$ | $0.162 \pm 0.096$ |
| GraphSAGE | NOCD | L2 | ✓ | MLP | MCC | $0.004 \pm 0.022$ | $0.087 \pm 0.106$ |
| GraphSAGE | Neuromap | None | ✓ | MLP | MCC | $0.052 \pm 0.072$ | $0.111 \pm 0.107$ |
| GraphSAGE | Neuromap | DMoN | ✗ | None | MCC | $0.037 \pm 0.059$ | $0.143 \pm 0.145$ |
| GraphSAGE | Neuromap | L2 | ✗ | None | Loss | $0.000 \pm 0.000$ | $0.117 \pm 0.143$ |
| GraphSAGE | SBM$_{\text{NN}}$ | None | ✗ | None | Loss | $0.082 \pm 0.094$ | $0.089 \pm 0.098$ |
| GraphSAGE | SBM$_{\text{NN}}$ | DMoN | ✓ | MLP | Loss | $0.046 \pm 0.063$ | $0.054 \pm 0.020$ |
| GraphSAGE | SBM$_{\text{NN}}$ | L2 | ✓ | MLP | MCC | $0.004 \pm 0.019$ | $0.070 \pm 0.118$ |
| Transformer | None | None | N/A | None | Loss | $0.000 \pm 0.000$ | $0.075 \pm 0.051$ |
| Transformer | None | DMoN | N/A | None | Loss | $0.000 \pm 0.000$ | $0.120 \pm 0.107$ |
| Transformer | None | L2 | N/A | None | MCC | $-0.001 \pm 0.030$ | $0.172 \pm 0.101$ |
| Transformer | DMoN | None | N/A | None | Loss | $0.000 \pm 0.000$ | $0.133 \pm 0.138$ |
| Transformer | DMoN | DMoN | N/A | None | Loss | $0.000 \pm 0.000$ | $0.090 \pm 0.109$ |
| Transformer | DMoN | L2 | N/A | None | MCC | $0.014 \pm 0.028$ | $0.161 \pm 0.110$ |
| Transformer | NOCD | None | N/A | None | Loss | $0.000 \pm 0.000$ | $0.032 \pm 0.004$ |
| Transformer | NOCD | DMoN | N/A | None | Loss | $0.000 \pm 0.000$ | $0.050 \pm 0.044$ |
| Transformer | NOCD | L2 | N/A | None | Loss | $\mathbf{0.106 \pm 0.093}$ | $\mathbf{0.177 \pm 0.105}$ |
| Transformer | Neuromap | None | N/A | None | Loss | $0.000 \pm 0.000$ | $0.122 \pm 0.136$ |
| Transformer | Neuromap | DMoN | N/A | None | Loss | $0.000 \pm 0.000$ | $0.073 \pm 0.041$ |
| Transformer | Neuromap | L2 | N/A | None | MCC | $0.002 \pm 0.011$ | $0.075 \pm 0.060$ |

Table 85: Comparing graph clustering and regularization objectives for each neural network.

| Model | $f$ | $L_{\text{regularization}}$ | $L_V$ | $\text{NN}_{\boldsymbol{S} \to \boldsymbol{X}}$ | ES | MCC | Accuracy |
|---|---|---|---|---|---|---|---|
| Transformer | $\text{SBM}_{\text{NN}}$ | None | N/A | None | Loss | $0.000 \pm 0.000$ | $0.031 \pm 0.003$ |
| Transformer | $\text{SBM}_{\text{NN}}$ | DMoN | N/A | None | Loss | $0.000 \pm 0.000$ | $0.032 \pm 0.001$ |
| Transformer | $\text{SBM}_{\text{NN}}$ | L2 | N/A | None | Loss | $0.006 \pm 0.019$ | $0.099 \pm 0.108$ |
| MLP | None | None | N/A | None | Loss | $0.150 \pm 0.053$ | $0.207 \pm 0.057$ |
| MLP | None | DMoN | N/A | None | Loss | $0.192 \pm 0.035$ | $\mathbf{0.278 \pm 0.048}$ |
| MLP | None | L2 | N/A | None | MCC | $0.003 \pm 0.024$ | $0.125 \pm 0.106$ |
| MLP | DMoN | None | N/A | None | Loss | $\mathbf{0.195 \pm 0.065}$ | $0.257 \pm 0.109$ |
| MLP | DMoN | DMoN | N/A | None | Loss | $0.189 \pm 0.042$ | $0.260 \pm 0.070$ |
| MLP | DMoN | L2 | N/A | None | Loss | $0.000 \pm 0.000$ | $0.111 \pm 0.109$ |
| MLP | NOCD | None | N/A | None | Loss | $0.003 \pm 0.005$ | $0.040 \pm 0.024$ |
| MLP | NOCD | DMoN | N/A | None | Loss | $0.002 \pm 0.005$ | $0.040 \pm 0.023$ |
| MLP | NOCD | L2 | N/A | None | Loss | $0.108 \pm 0.076$ | $0.166 \pm 0.123$ |
| MLP | Neuromap | None | N/A | None | MCC | $0.005 \pm 0.011$ | $0.083 \pm 0.119$ |
| MLP | Neuromap | DMoN | N/A | None | MCC | $0.020 \pm 0.029$ | $0.056 \pm 0.073$ |
| MLP | Neuromap | L2 | N/A | None | Loss | $0.000 \pm 0.000$ | $0.092 \pm 0.108$ |
| MLP | $\text{SBM}_{\text{NN}}$ | None | N/A | None | Loss | $-0.000 \pm 0.000$ | $0.097 \pm 0.110$ |
| MLP | $\text{SBM}_{\text{NN}}$ | DMoN | N/A | None | Loss | $0.000 \pm 0.000$ | $0.032 \pm 0.000$ |
| MLP | $\text{SBM}_{\text{NN}}$ | L2 | N/A | None | Loss | $0.006 \pm 0.017$ | $0.057 \pm 0.052$ |

Table 86: Comparing graph clustering and regularization objectives with an ablation study of regularization. The ablation compares clustering with clusteirng with regularization, clustering without regularization, and regularization without clustering for each neural network. Model selection based on validation set MCC.

| Model | $f$ | $L_{\text{regularization}}$ | $L_V$ | $\text{NN}_{\boldsymbol{S} \to \boldsymbol{X}}$ | ES | MCC | Accuracy |
|---|---|---|---|---|---|---|---|
| GCN | None | None | ✗ | None | Loss | $0.502 \pm 0.096$ | $0.569 \pm 0.101$ |
| GCN | None | DMoN | ✗ | None | Loss | $\mathbf{0.521 \pm 0.064}$ | $\mathbf{0.580 \pm 0.069}$ |
| GCN | None | L2 | ✗ | None | MCC | $0.030 \pm 0.102$ | $0.152 \pm 0.098$ |
| GCN | DMoN | None | ✗ | None | Loss | $0.476 \pm 0.086$ | $0.529 \pm 0.097$ |
| GCN | DMoN | DMoN | ✓ | MLP | Loss | $0.507 \pm 0.065$ | $0.574 \pm 0.065$ |
| GCN | DMoN | L2 | ✓ | MLP | MCC | $0.029 \pm 0.054$ | $0.219 \pm 0.108$ |
| GCN | NOCD | None | ✓ | MLP | MCC | $0.317 \pm 0.182$ | $0.367 \pm 0.201$ |
| GCN | NOCD | DMoN | ✓ | MLP | MCC | $0.373 \pm 0.143$ | $0.414 \pm 0.154$ |
| GCN | NOCD | L2 | ✓ | MLP | MCC | $0.057 \pm 0.074$ | $0.173 \pm 0.126$ |
| GCN | Neuromap | None | ✗ | None | MCC | $0.055 \pm 0.066$ | $0.086 \pm 0.071$ |
| GCN | Neuromap | DMoN | ✓ | MLP | MCC | $0.016 \pm 0.030$ | $0.095 \pm 0.102$ |
| GCN | Neuromap | L2 | ✓ | MLP | MCC | $0.015 \pm 0.020$ | $0.130 \pm 0.099$ |
| GCN | $\text{SBM}_{\text{NN}}$ | None | ✓ | MLP | MCC | $0.275 \pm 0.124$ | $0.326 \pm 0.162$ |
| GCN | $\text{SBM}_{\text{NN}}$ | DMoN | ✓ | Transformer | MCC | $0.326 \pm 0.151$ | $0.383 \pm 0.180$ |
| GCN | $\text{SBM}_{\text{NN}}$ | L2 | ✗ | None | MCC | $0.041 \pm 0.058$ | $0.174 \pm 0.111$ |
| GraphSAGE | None | None | ✗ | None | MCC | $0.373 \pm 0.063$ | $0.430 \pm 0.087$ |
| GraphSAGE | None | DMoN | ✗ | None | Loss | $0.466 \pm 0.105$ | $0.522 \pm 0.114$ |
| GraphSAGE | None | L2 | ✗ | None | Loss | $0.000 \pm 0.000$ | $0.063 \pm 0.052$ |
| GraphSAGE | DMoN | None | ✗ | None | MCC | $\mathbf{0.484 \pm 0.110}$ | $\mathbf{0.550 \pm 0.121}$ |
| GraphSAGE | DMoN | DMoN | ✗ | None | MCC | $0.446 \pm 0.105$ | $0.518 \pm 0.099$ |
| GraphSAGE | DMoN | L2 | ✗ | None | MCC | $-0.003 \pm 0.006$ | $0.071 \pm 0.121$ |

Table 86: Comparing graph clustering and regularization objectives for each neural network.

| Model | $f$ | $L_{\text{regularization}}$ | $L_V$ | $\text{NN}_{S \to X}$ | ES | MCC | Accuracy |
|---|---|---|---|---|---|---|---|
| GraphSAGE | NOCD | None | ✓ | MLP | MCC | $0.338 \pm 0.113$ | $0.355 \pm 0.140$ |
| GraphSAGE | NOCD | DMoN | ✓ | Transformer | MCC | $0.391 \pm 0.122$ | $0.455 \pm 0.139$ |
| GraphSAGE | NOCD | L2 | ✗ | None | Loss | $0.000 \pm 0.000$ | $0.073 \pm 0.049$ |
| GraphSAGE | Neuromap | None | ✓ | MLP | MCC | $0.052 \pm 0.072$ | $0.111 \pm 0.107$ |
| GraphSAGE | Neuromap | DMoN | ✗ | None | MCC | $0.037 \pm 0.059$ | $0.143 \pm 0.145$ |
| GraphSAGE | Neuromap | L2 | ✗ | None | MCC | $0.005 \pm 0.019$ | $0.079 \pm 0.095$ |
| GraphSAGE | $\text{SBM}_{\text{NN}}$ | None | ✗ | None | MCC | $0.127 \pm 0.144$ | $0.178 \pm 0.160$ |
| GraphSAGE | $\text{SBM}_{\text{NN}}$ | DMoN | ✓ | MLP | MCC | $0.094 \pm 0.117$ | $0.140 \pm 0.138$ |
| GraphSAGE | $\text{SBM}_{\text{NN}}$ | L2 | ✓ | Transformer | MCC | $0.002 \pm 0.012$ | $0.118 \pm 0.137$ |
| Transformer | None | None | N/A | None | Loss | $0.000 \pm 0.000$ | $0.075 \pm 0.051$ |
| Transformer | None | DMoN | N/A | None | Loss | $0.000 \pm 0.000$ | $0.120 \pm 0.107$ |
| Transformer | None | L2 | N/A | None | Loss | $-0.000 \pm 0.000$ | $0.078 \pm 0.059$ |
| Transformer | DMoN | None | N/A | None | Loss | $0.000 \pm 0.000$ | $0.133 \pm 0.138$ |
| Transformer | DMoN | DMoN | N/A | None | Loss | $0.000 \pm 0.000$ | $0.090 \pm 0.109$ |
| Transformer | DMoN | L2 | N/A | None | MCC | $0.014 \pm 0.028$ | $0.161 \pm 0.110$ |
| Transformer | NOCD | None | N/A | None | Loss | $0.000 \pm 0.000$ | $0.032 \pm 0.004$ |
| Transformer | NOCD | DMoN | N/A | None | Loss | $0.000 \pm 0.000$ | $0.050 \pm 0.044$ |
| Transformer | NOCD | L2 | N/A | None | Loss | $\mathbf{0.106 \pm 0.093}$ | $\mathbf{0.177 \pm 0.105}$ |
| Transformer | Neuromap | None | N/A | None | Loss | $0.000 \pm 0.000$ | $0.122 \pm 0.136$ |
| Transformer | Neuromap | DMoN | N/A | None | Loss | $0.000 \pm 0.000$ | $0.073 \pm 0.041$ |
| Transformer | Neuromap | L2 | N/A | None | MCC | $0.002 \pm 0.011$ | $0.075 \pm 0.060$ |
| Transformer | $\text{SBM}_{\text{NN}}$ | None | N/A | None | Loss | $0.000 \pm 0.000$ | $0.031 \pm 0.003$ |
| Transformer | $\text{SBM}_{\text{NN}}$ | DMoN | N/A | None | MCC | $-0.001 \pm 0.002$ | $0.036 \pm 0.115$ |
| Transformer | $\text{SBM}_{\text{NN}}$ | L2 | N/A | None | Loss | $0.006 \pm 0.019$ | $0.099 \pm 0.108$ |
| MLP | None | None | N/A | None | MCC | $0.180 \pm 0.067$ | $0.269 \pm 0.099$ |
| MLP | None | DMoN | N/A | None | MCC | $\mathbf{0.222 \pm 0.034}$ | $\mathbf{0.297 \pm 0.041}$ |
| MLP | None | L2 | N/A | None | Loss | $0.000 \pm 0.000$ | $0.098 \pm 0.111$ |
| MLP | DMoN | None | N/A | None | MCC | $0.201 \pm 0.037$ | $0.266 \pm 0.043$ |
| MLP | DMoN | DMoN | N/A | None | MCC | $0.207 \pm 0.058$ | $0.297 \pm 0.066$ |
| MLP | DMoN | L2 | N/A | None | MCC | $0.001 \pm 0.024$ | $0.134 \pm 0.108$ |
| MLP | NOCD | None | N/A | None | Loss | $0.003 \pm 0.005$ | $0.040 \pm 0.024$ |
| MLP | NOCD | DMoN | N/A | None | Loss | $0.002 \pm 0.005$ | $0.040 \pm 0.023$ |
| MLP | NOCD | L2 | N/A | None | Loss | $0.108 \pm 0.076$ | $0.166 \pm 0.123$ |
| MLP | Neuromap | None | N/A | None | MCC | $0.005 \pm 0.011$ | $0.083 \pm 0.119$ |
| MLP | Neuromap | DMoN | N/A | None | MCC | $0.020 \pm 0.029$ | $0.056 \pm 0.073$ |
| MLP | Neuromap | L2 | N/A | None | MCC | $0.002 \pm 0.017$ | $0.082 \pm 0.072$ |
| MLP | $\text{SBM}_{\text{NN}}$ | None | N/A | None | Loss | $-0.000 \pm 0.000$ | $0.097 \pm 0.110$ |
| MLP | $\text{SBM}_{\text{NN}}$ | DMoN | N/A | None | MCC | $-0.000 \pm 0.002$ | $0.009 \pm 0.019$ |
| MLP | $\text{SBM}_{\text{NN}}$ | L2 | N/A | None | MCC | $0.007 \pm 0.015$ | $0.068 \pm 0.073$ |

F.4.7    AMAZON PHOTO DATASET, DEFAULT SPLIT WITH $20$ TRAIN NODES PER CLASS AND $500$
VALIDATION NODES.

Table 87: Comparing graph clustering and regularization objectives with an ablation study of regularization. The ablation compares clustering with clusteirng with regularization, clustering without regularization, and regularization without clustering for each neural network. Model selection based on training loss.

| Model | $f$ | $L_{\text{regularization}}$ | $L_V$ | NN$_{S \to X}$ | ES | MCC | Accuracy |
|---|---|---|---|---|---|---|---|
| GCN | None | None | ✗ | None | Loss | $0.785 \pm 0.081$ | $0.803 \pm 0.088$ |
| GCN | None | DMoN | ✗ | None | Loss | $0.799 \pm 0.075$ | $0.819 \pm 0.081$ |
| GCN | None | L2 | ✗ | None | Loss | $0.000 \pm 0.000$ | $0.047 \pm 0.001$ |
| GCN | DMoN | None | ✓ | MLP | Loss | $0.796 \pm 0.070$ | $0.817 \pm 0.077$ |
| GCN | DMoN | DMoN | ✓ | Transformer | Loss | $\mathbf{0.840 \pm 0.031}$ | $\mathbf{0.862 \pm 0.028}$ |
| GCN | DMoN | L2 | ✗ | None | Loss | $0.000 \pm 0.000$ | $0.047 \pm 0.000$ |
| GCN | NOCD | None | ✓ | MLP | Loss | $0.667 \pm 0.113$ | $0.683 \pm 0.128$ |
| GCN | NOCD | DMoN | ✓ | Transformer | Loss | $0.706 \pm 0.089$ | $0.722 \pm 0.105$ |
| GCN | NOCD | L2 | ✗ | None | Loss | $0.000 \pm 0.000$ | $0.086 \pm 0.031$ |
| GCN | Neuromap | None | ✗ | None | Loss | $-0.002 \pm 0.005$ | $0.101 \pm 0.059$ |
| GCN | Neuromap | DMoN | ✓ | Transformer | Loss | $-0.001 \pm 0.004$ | $0.145 \pm 0.084$ |
| GCN | Neuromap | L2 | ✗ | None | Loss | $0.000 \pm 0.000$ | $0.047 \pm 0.001$ |
| GCN | SBM$_{\text{NN}}$ | None | ✓ | MLP | Loss | $0.504 \pm 0.075$ | $0.482 \pm 0.086$ |
| GCN | SBM$_{\text{NN}}$ | DMoN | ✓ | MLP | Loss | $0.534 \pm 0.089$ | $0.526 \pm 0.093$ |
| GCN | SBM$_{\text{NN}}$ | L2 | ✗ | None | Loss | $0.000 \pm 0.000$ | $0.109 \pm 0.073$ |
| GraphSAGE | None | None | ✗ | None | Loss | $0.830 \pm 0.059$ | $0.850 \pm 0.062$ |
| GraphSAGE | None | DMoN | ✗ | None | Loss | $0.816 \pm 0.052$ | $0.838 \pm 0.055$ |
| GraphSAGE | None | L2 | ✗ | None | Loss | $0.000 \pm 0.000$ | $0.099 \pm 0.053$ |
| GraphSAGE | DMoN | None | ✓ | MLP | Loss | $0.828 \pm 0.030$ | $0.851 \pm 0.027$ |
| GraphSAGE | DMoN | DMoN | ✓ | Transformer | Loss | $\mathbf{0.830 \pm 0.026}$ | $\mathbf{0.853 \pm 0.024}$ |
| GraphSAGE | DMoN | L2 | ✗ | None | Loss | $0.000 \pm 0.000$ | $0.102 \pm 0.059$ |
| GraphSAGE | NOCD | None | ✓ | Transformer | Loss | $0.461 \pm 0.065$ | $0.439 \pm 0.075$ |
| GraphSAGE | NOCD | DMoN | ✓ | Transformer | Loss | $0.314 \pm 0.054$ | $0.251 \pm 0.058$ |
| GraphSAGE | NOCD | L2 | ✗ | None | Loss | $0.000 \pm 0.000$ | $0.141 \pm 0.075$ |
| GraphSAGE | Neuromap | None | ✓ | Transformer | Loss | $0.004 \pm 0.011$ | $0.101 \pm 0.052$ |
| GraphSAGE | Neuromap | DMoN | ✓ | Transformer | Loss | $0.020 \pm 0.047$ | $0.171 \pm 0.082$ |
| GraphSAGE | Neuromap | L2 | ✗ | None | Loss | $0.000 \pm 0.000$ | $0.110 \pm 0.067$ |
| GraphSAGE | SBM$_{\text{NN}}$ | None | ✓ | MLP | Loss | $0.172 \pm 0.108$ | $0.144 \pm 0.089$ |
| GraphSAGE | SBM$_{\text{NN}}$ | DMoN | ✓ | MLP | Loss | $0.184 \pm 0.088$ | $0.151 \pm 0.063$ |
| GraphSAGE | SBM$_{\text{NN}}$ | L2 | ✗ | None | Loss | $0.000 \pm 0.000$ | $0.104 \pm 0.054$ |
| Transformer | None | None | N/A | None | Loss | $0.000 \pm 0.000$ | $0.149 \pm 0.060$ |
| Transformer | None | DMoN | N/A | None | Loss | $0.000 \pm 0.000$ | $0.097 \pm 0.064$ |
| Transformer | None | L2 | N/A | None | Loss | $0.000 \pm 0.000$ | $\mathbf{0.162 \pm 0.068}$ |
| Transformer | DMoN | None | N/A | None | Loss | $0.004 \pm 0.014$ | $0.112 \pm 0.074$ |
| Transformer | DMoN | DMoN | N/A | None | Loss | $0.000 \pm 0.000$ | $0.154 \pm 0.072$ |
| Transformer | DMoN | L2 | N/A | None | Loss | $0.000 \pm 0.000$ | $0.152 \pm 0.062$ |
| Transformer | NOCD | None | N/A | None | Loss | $0.000 \pm 0.000$ | $0.072 \pm 0.066$ |
| Transformer | NOCD | DMoN | N/A | None | Loss | $0.000 \pm 0.000$ | $0.046 \pm 0.002$ |
| Transformer | NOCD | L2 | N/A | None | Loss | $\mathbf{0.071 \pm 0.113}$ | $0.155 \pm 0.110$ |
| Transformer | Neuromap | None | N/A | None | Loss | $0.000 \pm 0.000$ | $0.110 \pm 0.045$ |
| Transformer | Neuromap | DMoN | N/A | None | Loss | $0.000 \pm 0.000$ | $0.114 \pm 0.073$ |
| Transformer | Neuromap | L2 | N/A | None | Loss | $0.000 \pm 0.000$ | $0.094 \pm 0.029$ |

Table 87: Comparing graph clustering and regularization objectives for each neural network.

| Model | $f$ | $L_{\text{regularization}}$ | $L_V$ | NN$_{S \to X}$ | ES | MCC | Accuracy |
|---|---|---|---|---|---|---|---|
| Transformer | SBM$_{\text{NN}}$ | None | N/A | None | Loss | $0.000 \pm 0.000$ | $0.047 \pm 0.001$ |
| Transformer | SBM$_{\text{NN}}$ | DMoN | N/A | None | Loss | $0.000 \pm 0.000$ | $0.047 \pm 0.001$ |
| Transformer | SBM$_{\text{NN}}$ | L2 | N/A | None | Loss | $0.007 \pm 0.019$ | $0.117 \pm 0.072$ |
| MLP | None | None | N/A | None | Loss | $\mathbf{0.680 \pm 0.021}$ | $\mathbf{0.720 \pm 0.020}$ |
| MLP | None | DMoN | N/A | None | Loss | $0.564 \pm 0.055$ | $0.603 \pm 0.051$ |
| MLP | None | L2 | N/A | None | Loss | $0.000 \pm 0.000$ | $0.116 \pm 0.071$ |
| MLP | DMoN | None | N/A | None | Loss | $0.618 \pm 0.048$ | $0.652 \pm 0.052$ |
| MLP | DMoN | DMoN | N/A | None | Loss | $0.585 \pm 0.053$ | $0.623 \pm 0.054$ |
| MLP | DMoN | L2 | N/A | None | Loss | $0.000 \pm 0.000$ | $0.139 \pm 0.078$ |
| MLP | NOCD | None | N/A | None | Loss | $0.018 \pm 0.058$ | $0.058 \pm 0.024$ |
| MLP | NOCD | DMoN | N/A | None | Loss | $0.001 \pm 0.003$ | $0.067 \pm 0.067$ |
| MLP | NOCD | L2 | N/A | None | Loss | $0.147 \pm 0.095$ | $0.185 \pm 0.097$ |
| MLP | Neuromap | None | N/A | None | Loss | $0.000 \pm 0.001$ | $0.148 \pm 0.091$ |
| MLP | Neuromap | DMoN | N/A | None | Loss | $-0.000 \pm 0.000$ | $0.181 \pm 0.079$ |
| MLP | Neuromap | L2 | N/A | None | Loss | $0.000 \pm 0.000$ | $0.098 \pm 0.053$ |
| MLP | SBM$_{\text{NN}}$ | None | N/A | None | Loss | $0.000 \pm 0.000$ | $0.064 \pm 0.056$ |
| MLP | SBM$_{\text{NN}}$ | DMoN | N/A | None | Loss | $0.000 \pm 0.000$ | $0.067 \pm 0.066$ |
| MLP | SBM$_{\text{NN}}$ | L2 | N/A | None | Loss | $0.004 \pm 0.014$ | $0.055 \pm 0.024$ |

Table 88: Comparing graph clustering and regularization objectives with an ablation study of regularization. The ablation compares clustering with clusteirng with regularization, clustering without regularization, and regularization without clustering for each neural network. Model selection based on training set MCC.

| Model | $f$ | $L_{\text{regularization}}$ | $L_V$ | NN$_{S \to X}$ | ES | MCC | Accuracy |
|---|---|---|---|---|---|---|---|
| GCN | None | None | ✗ | None | Loss | $0.785 \pm 0.081$ | $0.803 \pm 0.088$ |
| GCN | None | DMoN | ✗ | None | Loss | $0.799 \pm 0.075$ | $0.819 \pm 0.081$ |
| GCN | None | L2 | ✗ | None | MCC | $0.145 \pm 0.133$ | $0.254 \pm 0.086$ |
| GCN | DMoN | None | ✓ | MLP | Loss | $0.796 \pm 0.070$ | $0.817 \pm 0.077$ |
| GCN | DMoN | DMoN | ✓ | Transformer | Loss | $\mathbf{0.840 \pm 0.031}$ | $\mathbf{0.862 \pm 0.028}$ |
| GCN | DMoN | L2 | ✓ | Transformer | MCC | $0.047 \pm 0.098$ | $0.158 \pm 0.047$ |
| GCN | NOCD | None | ✓ | MLP | Loss | $0.667 \pm 0.113$ | $0.683 \pm 0.128$ |
| GCN | NOCD | DMoN | ✓ | Transformer | Loss | $0.706 \pm 0.089$ | $0.722 \pm 0.105$ |
| GCN | NOCD | L2 | ✗ | None | MCC | $0.168 \pm 0.065$ | $0.216 \pm 0.081$ |
| GCN | Neuromap | None | ✗ | None | MCC | $0.151 \pm 0.125$ | $0.165 \pm 0.134$ |
| GCN | Neuromap | DMoN | ✗ | None | MCC | $0.147 \pm 0.071$ | $0.147 \pm 0.031$ |
| GCN | Neuromap | L2 | ✗ | None | MCC | $0.068 \pm 0.165$ | $0.192 \pm 0.123$ |
| GCN | SBM$_{\text{NN}}$ | None | ✓ | MLP | Loss | $0.504 \pm 0.075$ | $0.482 \pm 0.086$ |
| GCN | SBM$_{\text{NN}}$ | DMoN | ✓ | MLP | Loss | $0.534 \pm 0.089$ | $0.526 \pm 0.093$ |
| GCN | SBM$_{\text{NN}}$ | L2 | ✗ | None | MCC | $0.093 \pm 0.087$ | $0.179 \pm 0.058$ |
| GraphSAGE | None | None | ✗ | None | Loss | $0.830 \pm 0.059$ | $0.850 \pm 0.062$ |
| GraphSAGE | None | DMoN | ✗ | None | Loss | $0.816 \pm 0.052$ | $0.838 \pm 0.055$ |
| GraphSAGE | None | L2 | ✗ | None | MCC | $-0.000 \pm 0.009$ | $0.122 \pm 0.075$ |
| GraphSAGE | DMoN | None | ✓ | MLP | Loss | $0.828 \pm 0.030$ | $0.851 \pm 0.027$ |
| GraphSAGE | DMoN | DMoN | ✓ | Transformer | Loss | $\mathbf{0.830 \pm 0.026}$ | $\mathbf{0.853 \pm 0.024}$ |
| GraphSAGE | DMoN | L2 | ✗ | None | Loss | $0.000 \pm 0.000$ | $0.102 \pm 0.059$ |

Table 88: Comparing graph clustering and regularization objectives for each neural network.

| Model | $f$ | $L_{\text{regularization}}$ | $L_V$ | $\text{NN}_{S \to X}$ | ES | MCC | Accuracy |
|---|---|---|---|---|---|---|---|
| GraphSAGE | NOCD | None | ✓ | Transformer | Loss | $0.461 \pm 0.065$ | $0.439 \pm 0.075$ |
| GraphSAGE | NOCD | DMoN | ✓ | Transformer | Loss | $0.314 \pm 0.054$ | $0.251 \pm 0.058$ |
| GraphSAGE | NOCD | L2 | ✗ | None | Loss | $0.000 \pm 0.000$ | $0.141 \pm 0.075$ |
| GraphSAGE | Neuromap | None | ✗ | None | MCC | $0.054 \pm 0.087$ | $0.122 \pm 0.100$ |
| GraphSAGE | Neuromap | DMoN | ✗ | None | MCC | $0.093 \pm 0.102$ | $0.147 \pm 0.112$ |
| GraphSAGE | Neuromap | L2 | ✗ | None | MCC | $-0.004 \pm 0.019$ | $0.123 \pm 0.087$ |
| GraphSAGE | $\text{SBM}_{\text{NN}}$ | None | ✗ | None | Loss | $0.196 \pm 0.102$ | $0.172 \pm 0.095$ |
| GraphSAGE | $\text{SBM}_{\text{NN}}$ | DMoN | ✗ | None | MCC | $0.324 \pm 0.139$ | $0.285 \pm 0.143$ |
| GraphSAGE | $\text{SBM}_{\text{NN}}$ | L2 | ✗ | None | Loss | $0.000 \pm 0.000$ | $0.101 \pm 0.023$ |
| Transformer | None | None | N/A | None | Loss | $0.000 \pm 0.000$ | $0.149 \pm 0.060$ |
| Transformer | None | DMoN | N/A | None | Loss | $0.000 \pm 0.000$ | $0.097 \pm 0.064$ |
| Transformer | None | L2 | N/A | None | MCC | $\mathbf{0.093 \pm 0.032}$ | $0.198 \pm 0.054$ |
| Transformer | DMoN | None | N/A | None | Loss | $0.004 \pm 0.014$ | $0.112 \pm 0.074$ |
| Transformer | DMoN | DMoN | N/A | None | Loss | $0.000 \pm 0.000$ | $0.154 \pm 0.072$ |
| Transformer | DMoN | L2 | N/A | None | MCC | $0.085 \pm 0.068$ | $\mathbf{0.244 \pm 0.058}$ |
| Transformer | NOCD | None | N/A | None | Loss | $0.000 \pm 0.000$ | $0.072 \pm 0.066$ |
| Transformer | NOCD | DMoN | N/A | None | Loss | $0.000 \pm 0.000$ | $0.046 \pm 0.002$ |
| Transformer | NOCD | L2 | N/A | None | Loss | $0.071 \pm 0.113$ | $0.155 \pm 0.110$ |
| Transformer | Neuromap | None | N/A | None | Loss | $0.000 \pm 0.000$ | $0.110 \pm 0.045$ |
| Transformer | Neuromap | DMoN | N/A | None | Loss | $0.000 \pm 0.000$ | $0.114 \pm 0.073$ |
| Transformer | Neuromap | L2 | N/A | None | MCC | $0.004 \pm 0.016$ | $0.120 \pm 0.077$ |
| Transformer | $\text{SBM}_{\text{NN}}$ | None | N/A | None | Loss | $0.000 \pm 0.000$ | $0.047 \pm 0.001$ |
| Transformer | $\text{SBM}_{\text{NN}}$ | DMoN | N/A | None | Loss | $0.000 \pm 0.000$ | $0.047 \pm 0.001$ |
| Transformer | $\text{SBM}_{\text{NN}}$ | L2 | N/A | None | Loss | $0.007 \pm 0.019$ | $0.117 \pm 0.072$ |
| MLP | None | None | N/A | None | Loss | $\mathbf{0.680 \pm 0.021}$ | $\mathbf{0.720 \pm 0.020}$ |
| MLP | None | DMoN | N/A | None | Loss | $0.564 \pm 0.055$ | $0.603 \pm 0.051$ |
| MLP | None | L2 | N/A | None | MCC | $0.018 \pm 0.063$ | $0.192 \pm 0.045$ |
| MLP | DMoN | None | N/A | None | Loss | $0.618 \pm 0.048$ | $0.652 \pm 0.052$ |
| MLP | DMoN | DMoN | N/A | None | Loss | $0.585 \pm 0.053$ | $0.623 \pm 0.054$ |
| MLP | DMoN | L2 | N/A | None | MCC | $-0.008 \pm 0.065$ | $0.146 \pm 0.057$ |
| MLP | NOCD | None | N/A | None | MCC | $0.010 \pm 0.020$ | $0.069 \pm 0.082$ |
| MLP | NOCD | DMoN | N/A | None | MCC | $0.009 \pm 0.018$ | $0.090 \pm 0.073$ |
| MLP | NOCD | L2 | N/A | None | Loss | $0.147 \pm 0.095$ | $0.185 \pm 0.097$ |
| MLP | Neuromap | None | N/A | None | MCC | $0.013 \pm 0.012$ | $0.075 \pm 0.057$ |
| MLP | Neuromap | DMoN | N/A | None | MCC | $0.026 \pm 0.043$ | $0.092 \pm 0.093$ |
| MLP | Neuromap | L2 | N/A | None | MCC | $0.002 \pm 0.026$ | $0.120 \pm 0.062$ |
| MLP | $\text{SBM}_{\text{NN}}$ | None | N/A | None | MCC | $0.003 \pm 0.004$ | $0.091 \pm 0.105$ |
| MLP | $\text{SBM}_{\text{NN}}$ | DMoN | N/A | None | MCC | $-0.000 \pm 0.012$ | $0.067 \pm 0.078$ |
| MLP | $\text{SBM}_{\text{NN}}$ | L2 | N/A | None | Loss | $0.004 \pm 0.014$ | $0.055 \pm 0.024$ |

Table 89: Comparing graph clustering and regularization objectives with an ablation study of regularization. The ablation compares clustering with clusteirng with regularization, clustering without regularization, and regularization without clustering for each neural network. Model selection based on validation set MCC.

| Model | $f$ | $L_{\text{regularization}}$ | $L_V$ | NN$_{S \to X}$ | ES | MCC | Accuracy |
|---|---|---|---|---|---|---|---|
| GCN | None | None | ✗ | None | MCC | $0.804 \pm 0.069$ | $0.826 \pm 0.071$ |
| GCN | None | DMoN | ✗ | None | MCC | $0.796 \pm 0.073$ | $0.815 \pm 0.081$ |
| GCN | None | L2 | ✗ | None | MCC | $0.145 \pm 0.133$ | $0.254 \pm 0.086$ |
| GCN | DMoN | None | ✓ | MLP | Loss | $0.796 \pm 0.070$ | $0.817 \pm 0.077$ |
| GCN | DMoN | DMoN | ✓ | Transformer | Loss | $\mathbf{0.840 \pm 0.031}$ | $\mathbf{0.862 \pm 0.028}$ |
| GCN | DMoN | L2 | ✓ | Transformer | MCC | $0.047 \pm 0.098$ | $0.158 \pm 0.047$ |
| GCN | NOCD | None | ✓ | MLP | MCC | $0.670 \pm 0.080$ | $0.695 \pm 0.088$ |
| GCN | NOCD | DMoN | ✓ | Transformer | Loss | $0.706 \pm 0.089$ | $0.722 \pm 0.105$ |
| GCN | NOCD | L2 | ✗ | None | MCC | $0.168 \pm 0.065$ | $0.216 \pm 0.081$ |
| GCN | Neuromap | None | ✗ | None | MCC | $0.151 \pm 0.125$ | $0.165 \pm 0.134$ |
| GCN | Neuromap | DMoN | ✗ | None | MCC | $0.147 \pm 0.071$ | $0.147 \pm 0.031$ |
| GCN | Neuromap | L2 | ✗ | None | MCC | $0.068 \pm 0.165$ | $0.192 \pm 0.123$ |
| GCN | SBM$_{NN}$ | None | ✓ | Transformer | MCC | $0.575 \pm 0.144$ | $0.582 \pm 0.163$ |
| GCN | SBM$_{NN}$ | DMoN | ✓ | MLP | Loss | $0.534 \pm 0.089$ | $0.526 \pm 0.093$ |
| GCN | SBM$_{NN}$ | L2 | ✓ | Transformer | MCC | $0.088 \pm 0.080$ | $0.162 \pm 0.088$ |
| GraphSAGE | None | None | ✗ | None | Loss | $0.830 \pm 0.059$ | $0.850 \pm 0.062$ |
| GraphSAGE | None | DMoN | ✗ | None | Loss | $0.816 \pm 0.052$ | $0.838 \pm 0.055$ |
| GraphSAGE | None | L2 | ✗ | None | MCC | $-0.000 \pm 0.009$ | $0.122 \pm 0.075$ |
| GraphSAGE | DMoN | None | ✓ | MLP | Loss | $0.828 \pm 0.030$ | $0.851 \pm 0.027$ |
| GraphSAGE | DMoN | DMoN | ✓ | Transformer | Loss | $\mathbf{0.830 \pm 0.026}$ | $\mathbf{0.853 \pm 0.024}$ |
| GraphSAGE | DMoN | L2 | ✓ | Transformer | MCC | $0.005 \pm 0.014$ | $0.085 \pm 0.075$ |
| GraphSAGE | NOCD | None | ✗ | None | MCC | $0.641 \pm 0.150$ | $0.651 \pm 0.163$ |
| GraphSAGE | NOCD | DMoN | ✓ | MLP | MCC | $0.694 \pm 0.110$ | $0.711 \pm 0.116$ |
| GraphSAGE | NOCD | L2 | ✓ | MLP | MCC | $0.008 \pm 0.018$ | $0.129 \pm 0.088$ |
| GraphSAGE | Neuromap | None | ✓ | MLP | MCC | $0.081 \pm 0.096$ | $0.139 \pm 0.111$ |
| GraphSAGE | Neuromap | DMoN | ✗ | None | MCC | $0.093 \pm 0.102$ | $0.147 \pm 0.112$ |
| GraphSAGE | Neuromap | L2 | ✗ | None | Loss | $0.000 \pm 0.000$ | $0.110 \pm 0.067$ |
| GraphSAGE | SBM$_{NN}$ | None | ✓ | MLP | MCC | $0.302 \pm 0.144$ | $0.258 \pm 0.131$ |
| GraphSAGE | SBM$_{NN}$ | DMoN | ✗ | None | MCC | $0.324 \pm 0.139$ | $0.285 \pm 0.143$ |
| GraphSAGE | SBM$_{NN}$ | L2 | ✓ | MLP | MCC | $-0.001 \pm 0.004$ | $0.068 \pm 0.079$ |
| Transformer | None | None | N/A | None | Loss | $0.000 \pm 0.000$ | $0.149 \pm 0.060$ |
| Transformer | None | DMoN | N/A | None | Loss | $0.000 \pm 0.000$ | $0.097 \pm 0.064$ |
| Transformer | None | L2 | N/A | None | MCC | $\mathbf{0.093 \pm 0.032}$ | $0.198 \pm 0.054$ |
| Transformer | DMoN | None | N/A | None | Loss | $0.004 \pm 0.014$ | $0.112 \pm 0.074$ |
| Transformer | DMoN | DMoN | N/A | None | Loss | $0.000 \pm 0.000$ | $0.154 \pm 0.072$ |
| Transformer | DMoN | L2 | N/A | None | MCC | $0.085 \pm 0.068$ | $\mathbf{0.244 \pm 0.058}$ |
| Transformer | NOCD | None | N/A | None | Loss | $0.000 \pm 0.000$ | $0.072 \pm 0.066$ |
| Transformer | NOCD | DMoN | N/A | None | Loss | $0.000 \pm 0.000$ | $0.046 \pm 0.002$ |
| Transformer | NOCD | L2 | N/A | None | Loss | $0.071 \pm 0.113$ | $0.155 \pm 0.110$ |
| Transformer | Neuromap | None | N/A | None | Loss | $0.000 \pm 0.000$ | $0.110 \pm 0.045$ |
| Transformer | Neuromap | DMoN | N/A | None | Loss | $0.000 \pm 0.000$ | $0.114 \pm 0.073$ |
| Transformer | Neuromap | L2 | N/A | None | Loss | $0.000 \pm 0.000$ | $0.094 \pm 0.029$ |

Table 89: Comparing graph clustering and regularization objectives for each neural network.

| Model | $f$ | $L_{\text{regularization}}$ | $L_V$ | NN$_{S \to X}$ | ES | MCC | Accuracy |
|---|---|---|---|---|---|---|---|
| Transformer | SBM$_{\text{NN}}$ | None | N/A | None | Loss | $0.000 \pm 0.000$ | $0.047 \pm 0.001$ |
| Transformer | SBM$_{\text{NN}}$ | DMoN | N/A | None | Loss | $0.000 \pm 0.000$ | $0.047 \pm 0.001$ |
| Transformer | SBM$_{\text{NN}}$ | L2 | N/A | None | Loss | $0.007 \pm 0.019$ | $0.117 \pm 0.072$ |
| MLP | None | None | N/A | None | MCC | $\mathbf{0.698 \pm 0.024}$ | $0.739 \pm 0.023$ |
| MLP | None | DMoN | N/A | None | MCC | $0.696 \pm 0.025$ | $\mathbf{0.741 \pm 0.021}$ |
| MLP | None | L2 | N/A | None | MCC | $0.018 \pm 0.063$ | $0.192 \pm 0.045$ |
| MLP | DMoN | None | N/A | None | MCC | $0.697 \pm 0.031$ | $0.740 \pm 0.027$ |
| MLP | DMoN | DMoN | N/A | None | MCC | $0.684 \pm 0.023$ | $0.728 \pm 0.021$ |
| MLP | DMoN | L2 | N/A | None | MCC | $-0.008 \pm 0.065$ | $0.146 \pm 0.057$ |
| MLP | NOCD | None | N/A | None | Loss | $0.018 \pm 0.058$ | $0.058 \pm 0.024$ |
| MLP | NOCD | DMoN | N/A | None | MCC | $0.009 \pm 0.018$ | $0.090 \pm 0.073$ |
| MLP | NOCD | L2 | N/A | None | Loss | $0.147 \pm 0.095$ | $0.185 \pm 0.097$ |
| MLP | Neuromap | None | N/A | None | MCC | $0.013 \pm 0.012$ | $0.075 \pm 0.057$ |
| MLP | Neuromap | DMoN | N/A | None | MCC | $0.026 \pm 0.043$ | $0.092 \pm 0.093$ |
| MLP | Neuromap | L2 | N/A | None | Loss | $0.000 \pm 0.000$ | $0.098 \pm 0.053$ |
| MLP | SBM$_{\text{NN}}$ | None | N/A | None | Loss | $0.000 \pm 0.000$ | $0.064 \pm 0.056$ |
| MLP | SBM$_{\text{NN}}$ | DMoN | N/A | None | Loss | $0.000 \pm 0.000$ | $0.067 \pm 0.066$ |
| MLP | SBM$_{\text{NN}}$ | L2 | N/A | None | Loss | $0.004 \pm 0.014$ | $0.055 \pm 0.024$ |

### F.4.8    AMAZON PHOTO DATASET, SPARSE SPLIT WITH 2 TRAIN NODES PER CLASS AND 50 VALIDATION NODES.

Table 90: Comparing graph clustering and regularization objectives with an ablation study of regularization. The ablation compares clustering with clusteirng with regularization, clustering without regularization, and regularization without clustering for each neural network. Model selection based on training loss.

| Model | $f$ | $L_{\text{regularization}}$ | $L_V$ | NN$_{S \to X}$ | ES | MCC | Accuracy |
|---|---|---|---|---|---|---|---|
| GCN | None | None | ✗ | None | Loss | $0.616 \pm 0.099$ | $0.647 \pm 0.112$ |
| GCN | None | DMoN | ✗ | None | Loss | $0.644 \pm 0.088$ | $0.679 \pm 0.088$ |
| GCN | None | L2 | ✗ | None | Loss | $0.000 \pm 0.000$ | $0.048 \pm 0.000$ |
| GCN | DMoN | None | ✗ | None | Loss | $0.608 \pm 0.098$ | $0.635 \pm 0.104$ |
| GCN | DMoN | DMoN | ✗ | None | Loss | $\mathbf{0.700 \pm 0.075}$ | $\mathbf{0.736 \pm 0.068}$ |
| GCN | DMoN | L2 | ✗ | None | Loss | $0.000 \pm 0.000$ | $0.048 \pm 0.000$ |
| GCN | NOCD | None | ✓ | MLP | Loss | $0.361 \pm 0.124$ | $0.355 \pm 0.150$ |
| GCN | NOCD | DMoN | ✓ | MLP | Loss | $0.374 \pm 0.044$ | $0.365 \pm 0.039$ |
| GCN | NOCD | L2 | ✗ | None | Loss | $0.000 \pm 0.000$ | $0.141 \pm 0.073$ |
| GCN | Neuromap | None | ✓ | Transformer | Loss | $-0.000 \pm 0.004$ | $0.159 \pm 0.076$ |
| GCN | Neuromap | DMoN | ✓ | Transformer | Loss | $0.023 \pm 0.072$ | $0.157 \pm 0.083$ |
| GCN | Neuromap | L2 | ✗ | None | Loss | $0.000 \pm 0.000$ | $0.048 \pm 0.000$ |
| GCN | SBM$_{\text{NN}}$ | None | ✗ | None | Loss | $0.218 \pm 0.073$ | $0.203 \pm 0.060$ |
| GCN | SBM$_{\text{NN}}$ | DMoN | ✗ | None | Loss | $0.258 \pm 0.109$ | $0.241 \pm 0.114$ |
| GCN | SBM$_{\text{NN}}$ | L2 | ✗ | None | Loss | $0.000 \pm 0.000$ | $0.134 \pm 0.080$ |
| GraphSAGE | None | None | ✗ | None | Loss | $0.640 \pm 0.068$ | $0.675 \pm 0.072$ |
| GraphSAGE | None | DMoN | ✗ | None | Loss | $0.637 \pm 0.089$ | $0.676 \pm 0.083$ |
| GraphSAGE | None | L2 | ✗ | None | Loss | $0.000 \pm 0.000$ | $0.105 \pm 0.075$ |

Continued on next page

Table 90: Comparing graph clustering and regularization objectives for each neural network.

| Model | $f$ | $L_{\text{regularization}}$ | $L_V$ | NN$_{S \rightarrow X}$ | ES | MCC | Accuracy |
|---|---|---|---|---|---|---|---|
| GraphSAGE | DMoN | None | ✗ | None | Loss | **0.667 ± 0.068** | **0.702 ± 0.070** |
| GraphSAGE | DMoN | DMoN | ✗ | None | Loss | 0.642 ± 0.082 | 0.681 ± 0.080 |
| GraphSAGE | DMoN | L2 | ✗ | None | Loss | 0.000 ± 0.000 | 0.117 ± 0.053 |
| GraphSAGE | NOCD | None | ✓ | Transformer | Loss | 0.389 ± 0.107 | 0.402 ± 0.110 |
| GraphSAGE | NOCD | DMoN | ✗ | None | Loss | 0.252 ± 0.082 | 0.205 ± 0.088 |
| GraphSAGE | NOCD | L2 | ✗ | None | Loss | 0.000 ± 0.000 | 0.098 ± 0.048 |
| GraphSAGE | Neuromap | None | ✓ | MLP | Loss | 0.000 ± 0.000 | 0.184 ± 0.070 |
| GraphSAGE | Neuromap | DMoN | ✗ | None | Loss | -0.002 ± 0.006 | 0.135 ± 0.044 |
| GraphSAGE | Neuromap | L2 | ✗ | None | Loss | 0.000 ± 0.000 | 0.180 ± 0.091 |
| GraphSAGE | SBM$_{\text{NN}}$ | None | ✓ | MLP | Loss | 0.119 ± 0.101 | 0.130 ± 0.064 |
| GraphSAGE | SBM$_{\text{NN}}$ | DMoN | ✗ | None | Loss | 0.098 ± 0.112 | 0.108 ± 0.089 |
| GraphSAGE | SBM$_{\text{NN}}$ | L2 | ✓ | MLP | Loss | 0.000 ± 0.000 | 0.130 ± 0.069 |
| Transformer | None | None | N/A | None | Loss | 0.000 ± 0.000 | 0.159 ± 0.075 |
| Transformer | None | DMoN | N/A | None | Loss | 0.000 ± 0.000 | 0.105 ± 0.078 |
| Transformer | None | L2 | N/A | None | Loss | 0.000 ± 0.000 | 0.145 ± 0.078 |
| Transformer | DMoN | None | N/A | None | Loss | 0.000 ± 0.000 | 0.170 ± 0.070 |
| Transformer | DMoN | DMoN | N/A | None | Loss | 0.000 ± 0.000 | 0.094 ± 0.033 |
| Transformer | DMoN | L2 | N/A | None | Loss | 0.000 ± 0.000 | 0.128 ± 0.077 |
| Transformer | NOCD | None | N/A | None | Loss | 0.000 ± 0.000 | 0.067 ± 0.030 |
| Transformer | NOCD | DMoN | N/A | None | Loss | 0.000 ± 0.000 | 0.096 ± 0.086 |
| Transformer | NOCD | L2 | N/A | None | Loss | **0.155 ± 0.138** | **0.232 ± 0.137** |
| Transformer | Neuromap | None | N/A | None | Loss | 0.000 ± 0.000 | 0.157 ± 0.086 |
| Transformer | Neuromap | DMoN | N/A | None | Loss | 0.000 ± 0.000 | 0.096 ± 0.054 |
| Transformer | Neuromap | L2 | N/A | None | Loss | 0.000 ± 0.000 | 0.116 ± 0.042 |
| Transformer | SBM$_{\text{NN}}$ | None | N/A | None | Loss | 0.000 ± 0.000 | 0.075 ± 0.066 |
| Transformer | SBM$_{\text{NN}}$ | DMoN | N/A | None | Loss | 0.000 ± 0.000 | 0.093 ± 0.086 |
| Transformer | SBM$_{\text{NN}}$ | L2 | N/A | None | Loss | 0.001 ± 0.004 | 0.125 ± 0.066 |
| MLP | None | None | N/A | None | Loss | 0.205 ± 0.099 | 0.276 ± 0.114 |
| MLP | None | DMoN | N/A | None | Loss | 0.235 ± 0.063 | 0.288 ± 0.062 |
| MLP | None | L2 | N/A | None | Loss | 0.000 ± 0.000 | 0.089 ± 0.037 |
| MLP | DMoN | None | N/A | None | Loss | 0.235 ± 0.049 | 0.301 ± 0.061 |
| MLP | DMoN | DMoN | N/A | None | MCC | **0.396 ± 0.038** | **0.469 ± 0.034** |
| MLP | DMoN | L2 | N/A | None | Loss | 0.000 ± 0.000 | 0.154 ± 0.075 |
| MLP | NOCD | None | N/A | None | Loss | 0.037 ± 0.078 | 0.123 ± 0.106 |
| MLP | NOCD | DMoN | N/A | None | Loss | 0.012 ± 0.024 | 0.084 ± 0.066 |
| MLP | NOCD | L2 | N/A | None | Loss | 0.202 ± 0.115 | 0.250 ± 0.115 |
| MLP | Neuromap | None | N/A | None | Loss | 0.004 ± 0.013 | 0.143 ± 0.066 |
| MLP | Neuromap | DMoN | N/A | None | Loss | 0.000 ± 0.000 | 0.134 ± 0.079 |
| MLP | Neuromap | L2 | N/A | None | Loss | 0.000 ± 0.000 | 0.103 ± 0.052 |
| MLP | SBM$_{\text{NN}}$ | None | N/A | None | Loss | 0.000 ± 0.000 | 0.048 ± 0.000 |
| MLP | SBM$_{\text{NN}}$ | DMoN | N/A | None | Loss | 0.000 ± 0.000 | 0.052 ± 0.014 |
| MLP | SBM$_{\text{NN}}$ | L2 | N/A | None | Loss | 0.005 ± 0.017 | 0.073 ± 0.034 |

Table 91: Comparing graph clustering and regularization objectives with an ablation study of regularization. The ablation compares clustering with clusteirng with regularization, clustering without regularization, and regularization without clustering for each neural network. Model selection based on training set MCC.

| Model | $f$ | $L_{\text{regularization}}$ | $L_V$ | $\text{NN}_{\boldsymbol{S}\to\boldsymbol{X}}$ | ES | MCC | Accuracy |
|---|---|---|---|---|---|---|---|
| GCN | None | None | ✗ | None | Loss | $0.616 \pm 0.099$ | $0.647 \pm 0.112$ |
| GCN | None | DMoN | ✗ | None | Loss | $0.644 \pm 0.088$ | $0.679 \pm 0.088$ |
| GCN | None | L2 | ✗ | None | MCC | $0.039 \pm 0.169$ | $0.163 \pm 0.130$ |
| GCN | DMoN | None | ✗ | None | Loss | $0.608 \pm 0.098$ | $0.635 \pm 0.104$ |
| GCN | DMoN | DMoN | ✗ | None | Loss | $\mathbf{0.700 \pm 0.075}$ | $\mathbf{0.736 \pm 0.068}$ |
| GCN | DMoN | L2 | ✗ | None | MCC | $0.033 \pm 0.174$ | $0.184 \pm 0.112$ |
| GCN | NOCD | None | ✓ | MLP | Loss | $0.361 \pm 0.124$ | $0.355 \pm 0.150$ |
| GCN | NOCD | DMoN | ✓ | MLP | Loss | $0.374 \pm 0.044$ | $0.365 \pm 0.039$ |
| GCN | NOCD | L2 | ✗ | None | MCC | $0.107 \pm 0.115$ | $0.201 \pm 0.101$ |
| GCN | Neuromap | None | ✗ | None | MCC | $0.102 \pm 0.110$ | $0.129 \pm 0.142$ |
| GCN | Neuromap | DMoN | ✓ | Transformer | MCC | $0.055 \pm 0.049$ | $0.143 \pm 0.084$ |
| GCN | Neuromap | L2 | ✗ | None | MCC | $0.054 \pm 0.154$ | $0.208 \pm 0.105$ |
| GCN | $\text{SBM}_{\text{NN}}$ | None | ✓ | MLP | Loss | $0.263 \pm 0.086$ | $0.251 \pm 0.088$ |
| GCN | $\text{SBM}_{\text{NN}}$ | DMoN | ✗ | None | Loss | $0.234 \pm 0.097$ | $0.203 \pm 0.094$ |
| GCN | $\text{SBM}_{\text{NN}}$ | L2 | ✗ | None | MCC | $0.045 \pm 0.053$ | $0.144 \pm 0.050$ |
| GraphSAGE | None | None | ✗ | None | Loss | $0.640 \pm 0.068$ | $0.675 \pm 0.072$ |
| GraphSAGE | None | DMoN | ✗ | None | Loss | $0.637 \pm 0.089$ | $0.676 \pm 0.083$ |
| GraphSAGE | None | L2 | ✗ | None | Loss | $0.000 \pm 0.000$ | $0.105 \pm 0.075$ |
| GraphSAGE | DMoN | None | ✗ | None | Loss | $\mathbf{0.667 \pm 0.068}$ | $\mathbf{0.702 \pm 0.070}$ |
| GraphSAGE | DMoN | DMoN | ✗ | None | Loss | $0.642 \pm 0.082$ | $0.681 \pm 0.080$ |
| GraphSAGE | DMoN | L2 | ✓ | Transformer | MCC | $0.001 \pm 0.002$ | $0.058 \pm 0.083$ |
| GraphSAGE | NOCD | None | ✓ | Transformer | Loss | $0.389 \pm 0.107$ | $0.402 \pm 0.110$ |
| GraphSAGE | NOCD | DMoN | ✓ | MLP | MCC | $0.496 \pm 0.103$ | $0.521 \pm 0.116$ |
| GraphSAGE | NOCD | L2 | ✓ | MLP | MCC | $-0.006 \pm 0.018$ | $0.064 \pm 0.089$ |
| GraphSAGE | Neuromap | None | ✓ | MLP | MCC | $0.072 \pm 0.077$ | $0.097 \pm 0.093$ |
| GraphSAGE | Neuromap | DMoN | ✓ | MLP | MCC | $0.040 \pm 0.071$ | $0.148 \pm 0.124$ |
| GraphSAGE | Neuromap | L2 | ✗ | None | Loss | $0.000 \pm 0.000$ | $0.180 \pm 0.091$ |
| GraphSAGE | $\text{SBM}_{\text{NN}}$ | None | ✓ | MLP | Loss | $0.119 \pm 0.101$ | $0.130 \pm 0.064$ |
| GraphSAGE | $\text{SBM}_{\text{NN}}$ | DMoN | ✓ | MLP | MCC | $0.185 \pm 0.149$ | $0.187 \pm 0.133$ |
| GraphSAGE | $\text{SBM}_{\text{NN}}$ | L2 | ✗ | None | Loss | $0.000 \pm 0.000$ | $0.121 \pm 0.060$ |
| Transformer | None | None | N/A | None | Loss | $0.000 \pm 0.000$ | $0.159 \pm 0.075$ |
| Transformer | None | DMoN | N/A | None | Loss | $0.000 \pm 0.000$ | $0.105 \pm 0.078$ |
| Transformer | None | L2 | N/A | None | MCC | $0.016 \pm 0.024$ | $0.130 \pm 0.079$ |
| Transformer | DMoN | None | N/A | None | MCC | $0.001 \pm 0.004$ | $0.011 \pm 0.035$ |
| Transformer | DMoN | DMoN | N/A | None | Loss | $0.000 \pm 0.000$ | $0.094 \pm 0.033$ |
| Transformer | DMoN | L2 | N/A | None | MCC | $0.012 \pm 0.023$ | $0.138 \pm 0.057$ |
| Transformer | NOCD | None | N/A | None | Loss | $0.000 \pm 0.000$ | $0.067 \pm 0.030$ |
| Transformer | NOCD | DMoN | N/A | None | Loss | $0.000 \pm 0.000$ | $0.096 \pm 0.086$ |
| Transformer | NOCD | L2 | N/A | None | Loss | $\mathbf{0.155 \pm 0.138}$ | $\mathbf{0.232 \pm 0.137}$ |
| Transformer | Neuromap | None | N/A | None | Loss | $0.000 \pm 0.000$ | $0.157 \pm 0.086$ |
| Transformer | Neuromap | DMoN | N/A | None | Loss | $0.000 \pm 0.000$ | $0.096 \pm 0.054$ |
| Transformer | Neuromap | L2 | N/A | None | Loss | $0.000 \pm 0.000$ | $0.116 \pm 0.042$ |

Table 91: Comparing graph clustering and regularization objectives for each neural network.

| Model | $f$ | $L_{\text{regularization}}$ | $L_V$ | $\text{NN}_{S \to X}$ | ES | MCC | Accuracy |
|---|---|---|---|---|---|---|---|
| Transformer | $\text{SBM}_{\text{NN}}$ | None | N/A | None | Loss | $0.000 \pm 0.000$ | $0.075 \pm 0.066$ |
| Transformer | $\text{SBM}_{\text{NN}}$ | DMoN | N/A | None | Loss | $0.000 \pm 0.000$ | $0.093 \pm 0.086$ |
| Transformer | $\text{SBM}_{\text{NN}}$ | L2 | N/A | None | Loss | $0.001 \pm 0.004$ | $0.125 \pm 0.066$ |
| MLP | None | None | N/A | None | Loss | $0.205 \pm 0.099$ | $0.276 \pm 0.114$ |
| MLP | None | DMoN | N/A | None | Loss | $0.235 \pm 0.063$ | $0.288 \pm 0.062$ |
| MLP | None | L2 | N/A | None | MCC | $0.008 \pm 0.027$ | $0.157 \pm 0.070$ |
| MLP | DMoN | None | N/A | None | Loss | $0.235 \pm 0.049$ | $0.301 \pm 0.061$ |
| MLP | DMoN | DMoN | N/A | None | MCC | $\mathbf{0.396 \pm 0.038}$ | $\mathbf{0.469 \pm 0.034}$ |
| MLP | DMoN | L2 | N/A | None | MCC | $0.013 \pm 0.034$ | $0.176 \pm 0.077$ |
| MLP | NOCD | None | N/A | None | MCC | $0.011 \pm 0.020$ | $0.063 \pm 0.099$ |
| MLP | NOCD | DMoN | N/A | None | Loss | $0.012 \pm 0.024$ | $0.084 \pm 0.066$ |
| MLP | NOCD | L2 | N/A | None | Loss | $0.202 \pm 0.115$ | $0.250 \pm 0.115$ |
| MLP | Neuromap | None | N/A | None | MCC | $0.017 \pm 0.016$ | $0.099 \pm 0.085$ |
| MLP | Neuromap | DMoN | N/A | None | MCC | $0.002 \pm 0.011$ | $0.031 \pm 0.043$ |
| MLP | Neuromap | L2 | N/A | None | MCC | $0.008 \pm 0.040$ | $0.136 \pm 0.072$ |
| MLP | $\text{SBM}_{\text{NN}}$ | None | N/A | None | Loss | $0.000 \pm 0.000$ | $0.048 \pm 0.000$ |
| MLP | $\text{SBM}_{\text{NN}}$ | DMoN | N/A | None | MCC | $0.001 \pm 0.018$ | $0.063 \pm 0.089$ |
| MLP | $\text{SBM}_{\text{NN}}$ | L2 | N/A | None | Loss | $0.005 \pm 0.017$ | $0.073 \pm 0.034$ |

Table 92: Comparing graph clustering and regularization objectives with an ablation study of regularization. The ablation compares clustering with clusteirng with regularization, clustering without regularization, and regularization without clustering for each neural network. Model selection based on validation set MCC.

| Model | $f$ | $L_{\text{regularization}}$ | $L_V$ | $\text{NN}_{S \to X}$ | ES | MCC | Accuracy |
|---|---|---|---|---|---|---|---|
| GCN | None | None | ✗ | None | MCC | $0.620 \pm 0.078$ | $0.653 \pm 0.087$ |
| GCN | None | DMoN | ✗ | None | Loss | $\mathbf{0.644 \pm 0.088}$ | $0.679 \pm 0.088$ |
| GCN | None | L2 | ✗ | None | MCC | $0.039 \pm 0.169$ | $0.163 \pm 0.130$ |
| GCN | DMoN | None | ✗ | None | MCC | $0.618 \pm 0.093$ | $0.659 \pm 0.098$ |
| GCN | DMoN | DMoN | ✗ | None | MCC | $0.640 \pm 0.074$ | $\mathbf{0.680 \pm 0.076}$ |
| GCN | DMoN | L2 | ✗ | None | MCC | $0.033 \pm 0.174$ | $0.184 \pm 0.112$ |
| GCN | NOCD | None | ✓ | MLP | MCC | $0.489 \pm 0.126$ | $0.522 \pm 0.127$ |
| GCN | NOCD | DMoN | ✓ | Transformer | MCC | $0.502 \pm 0.169$ | $0.541 \pm 0.176$ |
| GCN | NOCD | L2 | ✗ | None | MCC | $0.107 \pm 0.115$ | $0.201 \pm 0.101$ |
| GCN | Neuromap | None | ✗ | None | MCC | $0.102 \pm 0.110$ | $0.129 \pm 0.142$ |
| GCN | Neuromap | DMoN | ✓ | Transformer | MCC | $0.055 \pm 0.049$ | $0.143 \pm 0.084$ |
| GCN | Neuromap | L2 | ✗ | None | MCC | $0.054 \pm 0.154$ | $0.208 \pm 0.105$ |
| GCN | $\text{SBM}_{\text{NN}}$ | None | ✓ | MLP | MCC | $0.437 \pm 0.202$ | $0.471 \pm 0.199$ |
| GCN | $\text{SBM}_{\text{NN}}$ | DMoN | ✗ | None | MCC | $0.433 \pm 0.106$ | $0.453 \pm 0.118$ |
| GCN | $\text{SBM}_{\text{NN}}$ | L2 | ✓ | Transformer | MCC | $0.075 \pm 0.054$ | $0.178 \pm 0.068$ |
| GraphSAGE | None | None | ✗ | None | MCC | $0.603 \pm 0.104$ | $0.641 \pm 0.110$ |
| GraphSAGE | None | DMoN | ✗ | None | MCC | $\mathbf{0.626 \pm 0.089}$ | $\mathbf{0.671 \pm 0.086}$ |
| GraphSAGE | None | L2 | ✗ | None | Loss | $0.000 \pm 0.000$ | $0.105 \pm 0.075$ |
| GraphSAGE | DMoN | None | ✗ | None | MCC | $0.581 \pm 0.048$ | $0.617 \pm 0.052$ |
| GraphSAGE | DMoN | DMoN | ✗ | None | MCC | $0.571 \pm 0.102$ | $0.618 \pm 0.105$ |
| GraphSAGE | DMoN | L2 | ✓ | Transformer | MCC | $0.001 \pm 0.002$ | $0.058 \pm 0.083$ |

Continued on next page

Table 92: Comparing graph clustering and regularization objectives for each neural network.

| Model | $f$ | $L_{\text{regularization}}$ | $L_V$ | NN$_{S \to X}$ | ES | MCC | Accuracy |
|---|---|---|---|---|---|---|---|
| GraphSAGE | NOCD | None | ✓ | MLP | MCC | $0.480 \pm 0.091$ | $0.505 \pm 0.091$ |
| GraphSAGE | NOCD | DMoN | ✓ | MLP | MCC | $0.496 \pm 0.103$ | $0.521 \pm 0.116$ |
| GraphSAGE | NOCD | L2 | ✗ | None | Loss | $0.000 \pm 0.000$ | $0.098 \pm 0.048$ |
| GraphSAGE | Neuromap | None | ✓ | MLP | MCC | $0.072 \pm 0.077$ | $0.097 \pm 0.093$ |
| GraphSAGE | Neuromap | DMoN | ✗ | None | MCC | $0.054 \pm 0.087$ | $0.129 \pm 0.131$ |
| GraphSAGE | Neuromap | L2 | ✓ | MLP | MCC | $0.002 \pm 0.009$ | $0.041 \pm 0.071$ |
| GraphSAGE | SBM$_{\text{NN}}$ | None | ✗ | None | MCC | $0.138 \pm 0.116$ | $0.163 \pm 0.102$ |
| GraphSAGE | SBM$_{\text{NN}}$ | DMoN | ✓ | MLP | MCC | $0.185 \pm 0.149$ | $0.187 \pm 0.133$ |
| GraphSAGE | SBM$_{\text{NN}}$ | L2 | ✗ | None | Loss | $0.000 \pm 0.000$ | $0.121 \pm 0.060$ |
| Transformer | None | None | N/A | None | Loss | $0.000 \pm 0.000$ | $0.159 \pm 0.075$ |
| Transformer | None | DMoN | N/A | None | Loss | $0.000 \pm 0.000$ | $0.105 \pm 0.078$ |
| Transformer | None | L2 | N/A | None | MCC | $0.016 \pm 0.024$ | $0.130 \pm 0.079$ |
| Transformer | DMoN | None | N/A | None | Loss | $0.000 \pm 0.000$ | $0.170 \pm 0.070$ |
| Transformer | DMoN | DMoN | N/A | None | Loss | $0.000 \pm 0.000$ | $0.094 \pm 0.033$ |
| Transformer | DMoN | L2 | N/A | None | MCC | $0.012 \pm 0.023$ | $0.138 \pm 0.057$ |
| Transformer | NOCD | None | N/A | None | Loss | $0.000 \pm 0.000$ | $0.067 \pm 0.030$ |
| Transformer | NOCD | DMoN | N/A | None | Loss | $0.000 \pm 0.000$ | $0.096 \pm 0.086$ |
| Transformer | NOCD | L2 | N/A | None | Loss | $\mathbf{0.155 \pm 0.138}$ | $\mathbf{0.232 \pm 0.137}$ |
| Transformer | Neuromap | None | N/A | None | Loss | $0.000 \pm 0.000$ | $0.157 \pm 0.086$ |
| Transformer | Neuromap | DMoN | N/A | None | Loss | $0.000 \pm 0.000$ | $0.096 \pm 0.054$ |
| Transformer | Neuromap | L2 | N/A | None | MCC | $-0.001 \pm 0.022$ | $0.097 \pm 0.077$ |
| Transformer | SBM$_{\text{NN}}$ | None | N/A | None | Loss | $0.000 \pm 0.000$ | $0.075 \pm 0.066$ |
| Transformer | SBM$_{\text{NN}}$ | DMoN | N/A | None | Loss | $0.000 \pm 0.000$ | $0.093 \pm 0.086$ |
| Transformer | SBM$_{\text{NN}}$ | L2 | N/A | None | Loss | $0.001 \pm 0.004$ | $0.125 \pm 0.066$ |
| MLP | None | None | N/A | None | MCC | $0.269 \pm 0.073$ | $0.362 \pm 0.089$ |
| MLP | None | DMoN | N/A | None | MCC | $\mathbf{0.399 \pm 0.052}$ | $\mathbf{0.474 \pm 0.044}$ |
| MLP | None | L2 | N/A | None | MCC | $0.008 \pm 0.027$ | $0.157 \pm 0.070$ |
| MLP | DMoN | None | N/A | None | MCC | $0.387 \pm 0.019$ | $0.461 \pm 0.016$ |
| MLP | DMoN | DMoN | N/A | None | MCC | $0.396 \pm 0.038$ | $0.469 \pm 0.034$ |
| MLP | DMoN | L2 | N/A | None | MCC | $0.013 \pm 0.034$ | $0.176 \pm 0.077$ |
| MLP | NOCD | None | N/A | None | Loss | $0.037 \pm 0.078$ | $0.123 \pm 0.106$ |
| MLP | NOCD | DMoN | N/A | None | Loss | $0.012 \pm 0.024$ | $0.084 \pm 0.066$ |
| MLP | NOCD | L2 | N/A | None | Loss | $0.202 \pm 0.115$ | $0.250 \pm 0.115$ |
| MLP | Neuromap | None | N/A | None | Loss | $0.004 \pm 0.013$ | $0.143 \pm 0.066$ |
| MLP | Neuromap | DMoN | N/A | None | MCC | $0.002 \pm 0.011$ | $0.031 \pm 0.043$ |
| MLP | Neuromap | L2 | N/A | None | Loss | $0.000 \pm 0.000$ | $0.103 \pm 0.052$ |
| MLP | SBM$_{\text{NN}}$ | None | N/A | None | MCC | $-0.005 \pm 0.020$ | $0.077 \pm 0.092$ |
| MLP | SBM$_{\text{NN}}$ | DMoN | N/A | None | MCC | $0.001 \pm 0.018$ | $0.063 \pm 0.089$ |
| MLP | SBM$_{\text{NN}}$ | L2 | N/A | None | Loss | $0.005 \pm 0.017$ | $0.073 \pm 0.034$ |

F.4.9 COAUTHOR PHYSICS DATASET, DEFAULT SPLIT WITH 20 TRAIN NODES PER CLASS AND 500 VALIDATION NODES.

Table 93: Comparing graph clustering and regularization objectives with an ablation study of regularization. The ablation compares clustering with clusteirng with regularization, clustering without regularization, and regularization without clustering for each neural network. Model selection based on training loss.

| Model | $f$ | $L_{\text{regularization}}$ | $L_V$ | NN$_{S \to X}$ | ES | MCC | Accuracy |
|---|---|---|---|---|---|---|---|
| GCN | None | None | ✗ | None | Loss | **0.891 ± 0.015** | **0.925 ± 0.012** |
| GCN | None | DMoN | ✗ | None | Loss | 0.706 ± 0.019 | 0.752 ± 0.019 |
| GCN | None | L2 | ✗ | None | Loss | 0.000 ± 0.000 | 0.167 ± 0.000 |
| GCN | DMoN | None | ✗ | None | Loss | 0.703 ± 0.031 | 0.750 ± 0.029 |
| GCN | DMoN | DMoN | ✗ | None | Loss | 0.700 ± 0.039 | 0.747 ± 0.038 |
| GCN | DMoN | L2 | ✗ | None | Loss | 0.000 ± 0.000 | 0.167 ± 0.000 |
| GCN | NOCD | None | ✓ | Transformer | Loss | 0.436 ± 0.162 | 0.486 ± 0.184 |
| GCN | NOCD | DMoN | ✓ | Transformer | Loss | 0.298 ± 0.114 | 0.363 ± 0.138 |
| GCN | NOCD | L2 | ✗ | None | Loss | 0.000 ± 0.000 | 0.185 ± 0.171 |
| GCN | Neuromap | None | ✗ | None | Loss | 0.361 ± 0.322 | 0.420 ± 0.333 |
| GCN | Neuromap | DMoN | ✓ | MLP | Loss | 0.134 ± 0.231 | 0.310 ± 0.201 |
| GCN | Neuromap | L2 | ✗ | None | Loss | 0.000 ± 0.000 | 0.167 ± 0.000 |
| GCN | SBM$_{\text{NN}}$ | None | ✗ | None | Loss | 0.350 ± 0.148 | 0.418 ± 0.176 |
| GCN | SBM$_{\text{NN}}$ | DMoN | ✗ | None | Loss | 0.336 ± 0.132 | 0.426 ± 0.137 |
| GCN | SBM$_{\text{NN}}$ | L2 | ✗ | None | Loss | 0.000 ± 0.000 | 0.211 ± 0.159 |
| GraphSAGE | None | None | ✗ | None | Loss | **0.899 ± 0.011** | **0.931 ± 0.008** |
| GraphSAGE | None | DMoN | ✗ | None | Loss | 0.520 ± 0.064 | 0.582 ± 0.048 |
| GraphSAGE | None | L2 | ✗ | None | Loss | 0.000 ± 0.000 | 0.238 ± 0.187 |
| GraphSAGE | DMoN | None | ✓ | MLP | Loss | 0.580 ± 0.027 | 0.628 ± 0.025 |
| GraphSAGE | DMoN | DMoN | ✗ | None | Loss | 0.548 ± 0.041 | 0.605 ± 0.031 |
| GraphSAGE | DMoN | L2 | ✗ | None | Loss | 0.000 ± 0.000 | 0.283 ± 0.194 |
| GraphSAGE | NOCD | None | ✓ | Transformer | Loss | 0.508 ± 0.108 | 0.582 ± 0.130 |
| GraphSAGE | NOCD | DMoN | ✗ | None | Loss | 0.150 ± 0.053 | 0.212 ± 0.024 |
| GraphSAGE | NOCD | L2 | ✗ | None | Loss | 0.000 ± 0.000 | 0.209 ± 0.159 |
| GraphSAGE | Neuromap | None | ✓ | MLP | Loss | 0.123 ± 0.099 | 0.174 ± 0.068 |
| GraphSAGE | Neuromap | DMoN | ✓ | MLP | Loss | 0.124 ± 0.108 | 0.167 ± 0.071 |
| GraphSAGE | Neuromap | L2 | ✗ | None | Loss | 0.000 ± 0.000 | 0.175 ± 0.121 |
| GraphSAGE | SBM$_{\text{NN}}$ | None | ✗ | None | Loss | 0.277 ± 0.141 | 0.349 ± 0.158 |
| GraphSAGE | SBM$_{\text{NN}}$ | DMoN | ✓ | MLP | Loss | 0.332 ± 0.152 | 0.432 ± 0.140 |
| GraphSAGE | SBM$_{\text{NN}}$ | L2 | ✓ | MLP | Loss | 0.000 ± 0.000 | 0.170 ± 0.124 |
| Transformer | None | None | N/A | None | Loss | 0.032 ± 0.100 | 0.243 ± 0.200 |
| Transformer | None | DMoN | N/A | None | Loss | 0.021 ± 0.067 | 0.284 ± 0.200 |
| Transformer | None | L2 | N/A | None | Loss | 0.590 ± 0.106 | 0.697 ± 0.095 |
| Transformer | DMoN | None | N/A | None | Loss | 0.034 ± 0.108 | 0.244 ± 0.194 |
| Transformer | DMoN | DMoN | N/A | None | Loss | 0.016 ± 0.051 | 0.162 ± 0.134 |
| Transformer | DMoN | L2 | N/A | None | Loss | 0.642 ± 0.073 | 0.744 ± 0.062 |
| Transformer | NOCD | None | N/A | None | Loss | 0.000 ± 0.000 | 0.194 ± 0.111 |
| Transformer | NOCD | DMoN | N/A | None | Loss | 0.000 ± 0.000 | 0.158 ± 0.028 |
| Transformer | NOCD | L2 | N/A | None | Loss | **0.689 ± 0.246** | **0.745 ± 0.237** |
| Transformer | Neuromap | None | N/A | None | Loss | 0.000 ± 0.000 | 0.221 ± 0.198 |
| Transformer | Neuromap | DMoN | N/A | None | Loss | 0.000 ± 0.000 | 0.259 ± 0.213 |
| Transformer | Neuromap | L2 | N/A | None | Loss | 0.000 ± 0.000 | 0.200 ± 0.164 |

Table 93: Comparing graph clustering and regularization objectives for each neural network.

| Model | $f$ | $L_{\text{regularization}}$ | $L_V$ | NN$_{S \to X}$ | ES | MCC | Accuracy |
|---|---|---|---|---|---|---|---|
| Transformer | SBM$_{\text{NN}}$ | None | N/A | None | Loss | $0.000 \pm 0.000$ | $0.167 \pm 0.000$ |
| Transformer | SBM$_{\text{NN}}$ | DMoN | N/A | None | Loss | $0.000 \pm 0.000$ | $0.167 \pm 0.000$ |
| Transformer | SBM$_{\text{NN}}$ | L2 | N/A | None | Loss | $0.411 \pm 0.173$ | $0.490 \pm 0.205$ |
| MLP | None | None | N/A | None | Loss | $\mathbf{0.820 \pm 0.026}$ | $\mathbf{0.872 \pm 0.023}$ |
| MLP | None | DMoN | N/A | None | Loss | $0.152 \pm 0.082$ | $0.238 \pm 0.096$ |
| MLP | None | L2 | N/A | None | Loss | $0.000 \pm 0.000$ | $0.236 \pm 0.189$ |
| MLP | DMoN | None | N/A | None | Loss | $0.228 \pm 0.097$ | $0.248 \pm 0.047$ |
| MLP | DMoN | DMoN | N/A | None | Loss | $0.230 \pm 0.072$ | $0.263 \pm 0.076$ |
| MLP | DMoN | L2 | N/A | None | Loss | $0.000 \pm 0.000$ | $0.151 \pm 0.130$ |
| MLP | NOCD | None | N/A | None | Loss | $0.152 \pm 0.138$ | $0.254 \pm 0.101$ |
| MLP | NOCD | DMoN | N/A | None | Loss | $0.121 \pm 0.153$ | $0.244 \pm 0.108$ |
| MLP | NOCD | L2 | N/A | None | Loss | $0.744 \pm 0.198$ | $0.826 \pm 0.103$ |
| MLP | Neuromap | None | N/A | None | Loss | $0.019 \pm 0.053$ | $0.206 \pm 0.161$ |
| MLP | Neuromap | DMoN | N/A | None | Loss | $0.049 \pm 0.089$ | $0.245 \pm 0.198$ |
| MLP | Neuromap | L2 | N/A | None | Loss | $0.000 \pm 0.000$ | $0.219 \pm 0.199$ |
| MLP | SBM$_{\text{NN}}$ | None | N/A | None | Loss | $0.025 \pm 0.070$ | $0.174 \pm 0.021$ |
| MLP | SBM$_{\text{NN}}$ | DMoN | N/A | None | Loss | $0.000 \pm 0.000$ | $0.167 \pm 0.000$ |
| MLP | SBM$_{\text{NN}}$ | L2 | N/A | None | Loss | $0.016 \pm 0.047$ | $0.182 \pm 0.123$ |

Table 94: Comparing graph clustering and regularization objectives with an ablation study of regularization. The ablation compares clustering with clusteirng with regularization, clustering without regularization, and regularization without clustering for each neural network. Model selection based on training set MCC.

| Model | $f$ | $L_{\text{regularization}}$ | $L_V$ | NN$_{S \to X}$ | ES | MCC | Accuracy |
|---|---|---|---|---|---|---|---|
| GCN | None | None | ✗ | None | Loss | $\mathbf{0.891 \pm 0.015}$ | $\mathbf{0.925 \pm 0.012}$ |
| GCN | None | DMoN | ✗ | None | Loss | $0.706 \pm 0.019$ | $0.752 \pm 0.019$ |
| GCN | None | L2 | ✗ | None | MCC | $0.411 \pm 0.078$ | $0.599 \pm 0.046$ |
| GCN | DMoN | None | ✗ | None | Loss | $0.703 \pm 0.031$ | $0.750 \pm 0.029$ |
| GCN | DMoN | DMoN | ✗ | None | Loss | $0.700 \pm 0.039$ | $0.747 \pm 0.038$ |
| GCN | DMoN | L2 | ✗ | None | MCC | $0.081 \pm 0.333$ | $0.275 \pm 0.205$ |
| GCN | NOCD | None | ✓ | Transformer | Loss | $0.436 \pm 0.162$ | $0.486 \pm 0.184$ |
| GCN | NOCD | DMoN | ✓ | Transformer | Loss | $0.298 \pm 0.114$ | $0.363 \pm 0.138$ |
| GCN | NOCD | L2 | ✓ | MLP | MCC | $0.248 \pm 0.125$ | $0.367 \pm 0.167$ |
| GCN | Neuromap | None | ✗ | None | MCC | $0.728 \pm 0.135$ | $0.774 \pm 0.167$ |
| GCN | Neuromap | DMoN | ✗ | None | MCC | $0.734 \pm 0.105$ | $0.781 \pm 0.142$ |
| GCN | Neuromap | L2 | ✗ | None | MCC | $0.228 \pm 0.316$ | $0.425 \pm 0.222$ |
| GCN | SBM$_{\text{NN}}$ | None | ✗ | None | MCC | $0.784 \pm 0.156$ | $0.808 \pm 0.193$ |
| GCN | SBM$_{\text{NN}}$ | DMoN | ✗ | None | Loss | $0.336 \pm 0.132$ | $0.426 \pm 0.137$ |
| GCN | SBM$_{\text{NN}}$ | L2 | ✓ | MLP | MCC | $0.234 \pm 0.177$ | $0.355 \pm 0.210$ |
| GraphSAGE | None | None | ✗ | None | Loss | $\mathbf{0.899 \pm 0.011}$ | $\mathbf{0.931 \pm 0.008}$ |
| GraphSAGE | None | DMoN | ✗ | None | Loss | $0.520 \pm 0.064$ | $0.582 \pm 0.048$ |
| GraphSAGE | None | L2 | ✗ | None | MCC | $0.002 \pm 0.014$ | $0.067 \pm 0.073$ |
| GraphSAGE | DMoN | None | ✗ | None | Loss | $0.565 \pm 0.043$ | $0.615 \pm 0.030$ |
| GraphSAGE | DMoN | DMoN | ✗ | None | Loss | $0.548 \pm 0.041$ | $0.605 \pm 0.031$ |
| GraphSAGE | DMoN | L2 | ✓ | Transformer | MCC | $-0.003 \pm 0.009$ | $0.046 \pm 0.146$ |

Continued on next page

Table 94: Comparing graph clustering and regularization objectives for each neural network.

| Model | $f$ | $L_{\text{regularization}}$ | $L_V$ | NN$_{S \to X}$ | ES | MCC | Accuracy |
|-------|-----|------------------|-------|-----------|-----|-----|----------|
| GraphSAGE | NOCD | None | ✓ | Transformer | Loss | $0.508 \pm 0.108$ | $0.582 \pm 0.130$ |
| GraphSAGE | NOCD | DMoN | ✗ | None | Loss | $0.150 \pm 0.053$ | $0.212 \pm 0.024$ |
| GraphSAGE | NOCD | L2 | ✓ | MLP | MCC | $0.020 \pm 0.037$ | $0.045 \pm 0.077$ |
| GraphSAGE | Neuromap | None | ✓ | MLP | MCC | $0.515 \pm 0.238$ | $0.537 \pm 0.291$ |
| GraphSAGE | Neuromap | DMoN | ✓ | MLP | MCC | $0.496 \pm 0.199$ | $0.489 \pm 0.249$ |
| GraphSAGE | Neuromap | L2 | ✓ | Transformer | MCC | $0.003 \pm 0.026$ | $0.028 \pm 0.045$ |
| GraphSAGE | SBM$_{\text{NN}}$ | None | ✗ | None | Loss | $0.277 \pm 0.141$ | $0.349 \pm 0.158$ |
| GraphSAGE | SBM$_{\text{NN}}$ | DMoN | ✓ | MLP | Loss | $0.332 \pm 0.152$ | $0.432 \pm 0.140$ |
| GraphSAGE | SBM$_{\text{NN}}$ | L2 | ✗ | None | MCC | $0.019 \pm 0.042$ | $0.059 \pm 0.131$ |
| Transformer | None | None | N/A | None | MCC | $0.143 \pm 0.176$ | $0.201 \pm 0.248$ |
| Transformer | None | DMoN | N/A | None | MCC | $0.082 \pm 0.119$ | $0.078 \pm 0.106$ |
| Transformer | None | L2 | N/A | None | MCC | $\mathbf{0.751 \pm 0.092}$ | $0.821 \pm 0.074$ |
| Transformer | DMoN | None | N/A | None | MCC | $0.090 \pm 0.126$ | $0.160 \pm 0.169$ |
| Transformer | DMoN | DMoN | N/A | None | MCC | $0.184 \pm 0.176$ | $0.243 \pm 0.234$ |
| Transformer | DMoN | L2 | N/A | None | MCC | $0.750 \pm 0.073$ | $\mathbf{0.829 \pm 0.051}$ |
| Transformer | NOCD | None | N/A | None | Loss | $0.000 \pm 0.000$ | $0.194 \pm 0.111$ |
| Transformer | NOCD | DMoN | N/A | None | Loss | $0.000 \pm 0.000$ | $0.158 \pm 0.028$ |
| Transformer | NOCD | L2 | N/A | None | Loss | $0.689 \pm 0.246$ | $0.745 \pm 0.237$ |
| Transformer | Neuromap | None | N/A | None | Loss | $0.000 \pm 0.000$ | $0.221 \pm 0.198$ |
| Transformer | Neuromap | DMoN | N/A | None | Loss | $0.000 \pm 0.000$ | $0.259 \pm 0.213$ |
| Transformer | Neuromap | L2 | N/A | None | MCC | $0.013 \pm 0.036$ | $0.097 \pm 0.060$ |
| Transformer | SBM$_{\text{NN}}$ | None | N/A | None | Loss | $0.000 \pm 0.000$ | $0.167 \pm 0.000$ |
| Transformer | SBM$_{\text{NN}}$ | DMoN | N/A | None | Loss | $0.000 \pm 0.000$ | $0.167 \pm 0.000$ |
| Transformer | SBM$_{\text{NN}}$ | L2 | N/A | None | Loss | $0.411 \pm 0.173$ | $0.490 \pm 0.205$ |
| MLP | None | None | N/A | None | Loss | $\mathbf{0.820 \pm 0.026}$ | $\mathbf{0.872 \pm 0.023}$ |
| MLP | None | DMoN | N/A | None | Loss | $0.152 \pm 0.082$ | $0.238 \pm 0.096$ |
| MLP | None | L2 | N/A | None | MCC | $0.364 \pm 0.085$ | $0.531 \pm 0.115$ |
| MLP | DMoN | None | N/A | None | Loss | $0.228 \pm 0.097$ | $0.248 \pm 0.047$ |
| MLP | DMoN | DMoN | N/A | None | Loss | $0.230 \pm 0.072$ | $0.263 \pm 0.076$ |
| MLP | DMoN | L2 | N/A | None | MCC | $0.277 \pm 0.139$ | $0.490 \pm 0.138$ |
| MLP | NOCD | None | N/A | None | Loss | $0.152 \pm 0.138$ | $0.254 \pm 0.101$ |
| MLP | NOCD | DMoN | N/A | None | MCC | $0.303 \pm 0.211$ | $0.343 \pm 0.256$ |
| MLP | NOCD | L2 | N/A | None | Loss | $0.744 \pm 0.198$ | $0.826 \pm 0.103$ |
| MLP | Neuromap | None | N/A | None | MCC | $0.247 \pm 0.154$ | $0.318 \pm 0.239$ |
| MLP | Neuromap | DMoN | N/A | None | MCC | $0.262 \pm 0.159$ | $0.298 \pm 0.232$ |
| MLP | Neuromap | L2 | N/A | None | MCC | $-0.003 \pm 0.051$ | $0.215 \pm 0.161$ |
| MLP | SBM$_{\text{NN}}$ | None | N/A | None | Loss | $0.025 \pm 0.070$ | $0.174 \pm 0.021$ |
| MLP | SBM$_{\text{NN}}$ | DMoN | N/A | None | MCC | $0.020 \pm 0.094$ | $0.046 \pm 0.091$ |
| MLP | SBM$_{\text{NN}}$ | L2 | N/A | None | Loss | $0.016 \pm 0.047$ | $0.182 \pm 0.123$ |

Table 95: Comparing graph clustering and regularization objectives with an ablation study of regularization. The ablation compares clustering with clusteirng with regularization, clustering without regularization, and regularization without clustering for each neural network. Model selection based on validation set MCC.

| Model | $f$ | $L_{\text{regularization}}$ | $L_V$ | NN$_{\boldsymbol{S}\to\boldsymbol{X}}$ | ES | MCC | Accuracy |
|---|---|---|---|---|---|---|---|
| GCN | None | None | ✗ | None | MCC | $0.907 \pm 0.009$ | $0.936 \pm 0.007$ |
| GCN | None | DMoN | ✗ | None | MCC | $\mathbf{0.909 \pm 0.012}$ | $\mathbf{0.938 \pm 0.008}$ |
| GCN | None | L2 | ✗ | None | MCC | $0.411 \pm 0.078$ | $0.599 \pm 0.046$ |
| GCN | DMoN | None | ✗ | None | MCC | $0.907 \pm 0.008$ | $0.936 \pm 0.006$ |
| GCN | DMoN | DMoN | ✗ | None | MCC | $0.902 \pm 0.014$ | $0.933 \pm 0.009$ |
| GCN | DMoN | L2 | ✗ | None | MCC | $0.081 \pm 0.333$ | $0.275 \pm 0.205$ |
| GCN | NOCD | None | ✓ | Transformer | MCC | $0.716 \pm 0.188$ | $0.735 \pm 0.231$ |
| GCN | NOCD | DMoN | ✓ | Transformer | MCC | $0.738 \pm 0.225$ | $0.757 \pm 0.253$ |
| GCN | NOCD | L2 | ✓ | Transformer | MCC | $0.254 \pm 0.135$ | $0.373 \pm 0.175$ |
| GCN | Neuromap | None | ✗ | None | MCC | $0.728 \pm 0.135$ | $0.774 \pm 0.167$ |
| GCN | Neuromap | DMoN | ✗ | None | MCC | $0.734 \pm 0.105$ | $0.781 \pm 0.142$ |
| GCN | Neuromap | L2 | ✗ | None | MCC | $0.228 \pm 0.316$ | $0.425 \pm 0.222$ |
| GCN | SBM$_{\text{NN}}$ | None | ✗ | None | MCC | $0.784 \pm 0.156$ | $0.808 \pm 0.193$ |
| GCN | SBM$_{\text{NN}}$ | DMoN | ✓ | Transformer | MCC | $0.740 \pm 0.213$ | $0.769 \pm 0.252$ |
| GCN | SBM$_{\text{NN}}$ | L2 | ✓ | MLP | MCC | $0.234 \pm 0.177$ | $0.355 \pm 0.210$ |
| GraphSAGE | None | None | ✗ | None | Loss | $0.899 \pm 0.011$ | $0.931 \pm 0.008$ |
| GraphSAGE | None | DMoN | ✗ | None | MCC | $\mathbf{0.905 \pm 0.006}$ | $\mathbf{0.935 \pm 0.005}$ |
| GraphSAGE | None | L2 | ✗ | None | MCC | $0.002 \pm 0.014$ | $0.067 \pm 0.073$ |
| GraphSAGE | DMoN | None | ✗ | None | MCC | $0.904 \pm 0.007$ | $0.934 \pm 0.005$ |
| GraphSAGE | DMoN | DMoN | ✓ | MLP | MCC | $0.899 \pm 0.006$ | $0.931 \pm 0.004$ |
| GraphSAGE | DMoN | L2 | ✓ | Transformer | MCC | $-0.003 \pm 0.009$ | $0.046 \pm 0.146$ |
| GraphSAGE | NOCD | None | ✓ | Transformer | MCC | $0.756 \pm 0.168$ | $0.805 \pm 0.183$ |
| GraphSAGE | NOCD | DMoN | ✓ | Transformer | MCC | $0.766 \pm 0.121$ | $0.816 \pm 0.139$ |
| GraphSAGE | NOCD | L2 | ✗ | None | MCC | $0.029 \pm 0.056$ | $0.087 \pm 0.163$ |
| GraphSAGE | Neuromap | None | ✓ | MLP | MCC | $0.515 \pm 0.238$ | $0.537 \pm 0.291$ |
| GraphSAGE | Neuromap | DMoN | ✓ | MLP | MCC | $0.496 \pm 0.199$ | $0.489 \pm 0.249$ |
| GraphSAGE | Neuromap | L2 | ✓ | Transformer | MCC | $0.003 \pm 0.026$ | $0.028 \pm 0.045$ |
| GraphSAGE | SBM$_{\text{NN}}$ | None | ✓ | Transformer | MCC | $0.576 \pm 0.180$ | $0.622 \pm 0.235$ |
| GraphSAGE | SBM$_{\text{NN}}$ | DMoN | ✗ | None | MCC | $0.514 \pm 0.179$ | $0.541 \pm 0.233$ |
| GraphSAGE | SBM$_{\text{NN}}$ | L2 | ✗ | None | MCC | $0.025 \pm 0.069$ | $0.060 \pm 0.119$ |
| Transformer | None | None | N/A | None | MCC | $0.143 \pm 0.176$ | $0.201 \pm 0.248$ |
| Transformer | None | DMoN | N/A | None | MCC | $0.082 \pm 0.119$ | $0.078 \pm 0.106$ |
| Transformer | None | L2 | N/A | None | MCC | $\mathbf{0.751 \pm 0.092}$ | $0.821 \pm 0.074$ |
| Transformer | DMoN | None | N/A | None | MCC | $0.090 \pm 0.126$ | $0.160 \pm 0.169$ |
| Transformer | DMoN | DMoN | N/A | None | MCC | $0.184 \pm 0.176$ | $0.243 \pm 0.234$ |
| Transformer | DMoN | L2 | N/A | None | MCC | $0.750 \pm 0.073$ | $\mathbf{0.829 \pm 0.051}$ |
| Transformer | NOCD | None | N/A | None | Loss | $0.000 \pm 0.000$ | $0.194 \pm 0.111$ |
| Transformer | NOCD | DMoN | N/A | None | Loss | $0.000 \pm 0.000$ | $0.158 \pm 0.028$ |
| Transformer | NOCD | L2 | N/A | None | Loss | $0.689 \pm 0.246$ | $0.745 \pm 0.237$ |
| Transformer | Neuromap | None | N/A | None | Loss | $0.000 \pm 0.000$ | $0.221 \pm 0.198$ |
| Transformer | Neuromap | DMoN | N/A | None | Loss | $0.000 \pm 0.000$ | $0.259 \pm 0.213$ |
| Transformer | Neuromap | L2 | N/A | None | MCC | $0.013 \pm 0.036$ | $0.097 \pm 0.060$ |

Table 95: Comparing graph clustering and regularization objectives for each neural network.

| Model | $f$ | $L_{\text{regularization}}$ | $L_V$ | $\text{NN}_{S \to X}$ | ES | MCC | Accuracy |
|---|---|---|---|---|---|---|---|
| Transformer | $\text{SBM}_{\text{NN}}$ | None | N/A | None | Loss | $0.000 \pm 0.000$ | $0.167 \pm 0.000$ |
| Transformer | $\text{SBM}_{\text{NN}}$ | DMoN | N/A | None | Loss | $0.000 \pm 0.000$ | $0.167 \pm 0.000$ |
| Transformer | $\text{SBM}_{\text{NN}}$ | L2 | N/A | None | Loss | $0.411 \pm 0.173$ | $0.490 \pm 0.205$ |
| MLP | None | None | N/A | None | MCC | $\mathbf{0.819 \pm 0.023}$ | $\mathbf{0.874 \pm 0.016}$ |
| MLP | None | DMoN | N/A | None | MCC | $0.721 \pm 0.036$ | $0.791 \pm 0.033$ |
| MLP | None | L2 | N/A | None | MCC | $0.364 \pm 0.085$ | $0.531 \pm 0.115$ |
| MLP | DMoN | None | N/A | None | MCC | $0.732 \pm 0.051$ | $0.800 \pm 0.047$ |
| MLP | DMoN | DMoN | N/A | None | MCC | $0.732 \pm 0.064$ | $0.799 \pm 0.056$ |
| MLP | DMoN | L2 | N/A | None | MCC | $0.277 \pm 0.139$ | $0.490 \pm 0.138$ |
| MLP | NOCD | None | N/A | None | MCC | $0.239 \pm 0.229$ | $0.286 \pm 0.276$ |
| MLP | NOCD | DMoN | N/A | None | MCC | $0.303 \pm 0.211$ | $0.343 \pm 0.256$ |
| MLP | NOCD | L2 | N/A | None | Loss | $0.744 \pm 0.198$ | $0.826 \pm 0.103$ |
| MLP | Neuromap | None | N/A | None | MCC | $0.247 \pm 0.154$ | $0.318 \pm 0.239$ |
| MLP | Neuromap | DMoN | N/A | None | MCC | $0.262 \pm 0.159$ | $0.298 \pm 0.232$ |
| MLP | Neuromap | L2 | N/A | None | MCC | $-0.003 \pm 0.051$ | $0.215 \pm 0.161$ |
| MLP | $\text{SBM}_{\text{NN}}$ | None | N/A | None | MCC | $0.042 \pm 0.168$ | $0.123 \pm 0.203$ |
| MLP | $\text{SBM}_{\text{NN}}$ | DMoN | N/A | None | MCC | $0.020 \pm 0.094$ | $0.046 \pm 0.091$ |
| MLP | $\text{SBM}_{\text{NN}}$ | L2 | N/A | None | Loss | $0.016 \pm 0.047$ | $0.182 \pm 0.123$ |

### F.4.10 COAUTHOR PHYSICS DATASET, SPARSE SPLIT WITH 2 TRAIN NODES PER CLASS AND 50 VALIDATION NODES.

Table 96: Comparing graph clustering and regularization objectives with an ablation study of regularization. The ablation compares clustering with clusteirng with regularization, clustering without regularization, and regularization without clustering for each neural network. Model selection based on training loss.

| Model | $f$ | $L_{\text{regularization}}$ | $L_V$ | $\text{NN}_{S \to X}$ | ES | MCC | Accuracy |
|---|---|---|---|---|---|---|---|
| GCN | None | None | ✗ | None | Loss | $\mathbf{0.796 \pm 0.058}$ | $\mathbf{0.859 \pm 0.041}$ |
| GCN | None | DMoN | ✗ | None | Loss | $0.502 \pm 0.072$ | $0.567 \pm 0.060$ |
| GCN | None | L2 | ✗ | None | Loss | $0.000 \pm 0.000$ | $0.167 \pm 0.000$ |
| GCN | DMoN | None | ✗ | None | Loss | $0.487 \pm 0.062$ | $0.552 \pm 0.060$ |
| GCN | DMoN | DMoN | ✗ | None | Loss | $0.497 \pm 0.066$ | $0.564 \pm 0.055$ |
| GCN | DMoN | L2 | ✗ | None | Loss | $0.000 \pm 0.000$ | $0.167 \pm 0.000$ |
| GCN | NOCD | None | ✗ | None | Loss | $0.391 \pm 0.104$ | $0.434 \pm 0.122$ |
| GCN | NOCD | DMoN | ✗ | None | Loss | $0.217 \pm 0.124$ | $0.275 \pm 0.073$ |
| GCN | NOCD | L2 | ✗ | None | Loss | $0.000 \pm 0.000$ | $0.196 \pm 0.166$ |
| GCN | Neuromap | None | ✗ | None | Loss | $0.451 \pm 0.364$ | $0.524 \pm 0.344$ |
| GCN | Neuromap | DMoN | ✓ | MLP | Loss | $0.290 \pm 0.180$ | $0.336 \pm 0.142$ |
| GCN | Neuromap | L2 | ✗ | None | Loss | $0.000 \pm 0.000$ | $0.167 \pm 0.000$ |
| GCN | $\text{SBM}_{\text{NN}}$ | None | ✓ | MLP | Loss | $0.397 \pm 0.117$ | $0.511 \pm 0.126$ |
| GCN | $\text{SBM}_{\text{NN}}$ | DMoN | ✓ | Transformer | Loss | $0.431 \pm 0.126$ | $0.539 \pm 0.131$ |
| GCN | $\text{SBM}_{\text{NN}}$ | L2 | ✗ | None | Loss | $0.000 \pm 0.000$ | $0.215 \pm 0.156$ |
| GraphSAGE | None | None | ✗ | None | Loss | $\mathbf{0.802 \pm 0.047}$ | $\mathbf{0.864 \pm 0.034}$ |
| GraphSAGE | None | DMoN | ✗ | None | Loss | $0.214 \pm 0.085$ | $0.380 \pm 0.097$ |
| GraphSAGE | None | L2 | ✗ | None | Loss | $0.000 \pm 0.000$ | $0.196 \pm 0.165$ |

Table 96: Comparing graph clustering and regularization objectives for each neural network.

| Model | $f$ | $L_{\text{regularization}}$ | $L_V$ | NN$_{S \to X}$ | ES | MCC | Accuracy |
|---|---|---|---|---|---|---|---|
| GraphSAGE | DMoN | None | ✓ | MLP | Loss | $0.298 \pm 0.077$ | $0.422 \pm 0.075$ |
| GraphSAGE | DMoN | DMoN | ✓ | MLP | Loss | $0.280 \pm 0.075$ | $0.386 \pm 0.048$ |
| GraphSAGE | DMoN | L2 | ✗ | None | Loss | $0.000 \pm 0.000$ | $0.122 \pm 0.032$ |
| GraphSAGE | NOCD | None | ✗ | None | Loss | $0.256 \pm 0.150$ | $0.349 \pm 0.120$ |
| GraphSAGE | NOCD | DMoN | ✗ | None | Loss | $0.213 \pm 0.060$ | $0.278 \pm 0.030$ |
| GraphSAGE | NOCD | L2 | ✗ | None | Loss | $0.000 \pm 0.000$ | $0.229 \pm 0.193$ |
| GraphSAGE | Neuromap | None | ✓ | MLP | Loss | $0.281 \pm 0.147$ | $0.307 \pm 0.096$ |
| GraphSAGE | Neuromap | DMoN | ✓ | MLP | Loss | $0.222 \pm 0.113$ | $0.247 \pm 0.098$ |
| GraphSAGE | Neuromap | L2 | ✗ | None | Loss | $0.000 \pm 0.000$ | $0.187 \pm 0.170$ |
| GraphSAGE | SBM$_{\text{NN}}$ | None | ✓ | Transformer | Loss | $0.232 \pm 0.178$ | $0.335 \pm 0.175$ |
| GraphSAGE | SBM$_{\text{NN}}$ | DMoN | ✗ | None | Loss | $0.294 \pm 0.154$ | $0.394 \pm 0.157$ |
| GraphSAGE | SBM$_{\text{NN}}$ | L2 | ✗ | None | Loss | $0.000 \pm 0.000$ | $0.181 \pm 0.118$ |
| Transformer | None | None | N/A | None | Loss | $0.023 \pm 0.081$ | $0.159 \pm 0.124$ |
| Transformer | None | DMoN | N/A | None | Loss | $0.001 \pm 0.002$ | $0.128 \pm 0.034$ |
| Transformer | None | L2 | N/A | None | Loss | $0.356 \pm 0.119$ | $0.443 \pm 0.164$ |
| Transformer | DMoN | None | N/A | None | Loss | $0.023 \pm 0.074$ | $0.199 \pm 0.154$ |
| Transformer | DMoN | DMoN | N/A | None | Loss | $0.000 \pm 0.001$ | $0.191 \pm 0.169$ |
| Transformer | DMoN | L2 | N/A | None | Loss | $0.379 \pm 0.116$ | $0.490 \pm 0.143$ |
| Transformer | NOCD | None | N/A | None | Loss | $0.000 \pm 0.000$ | $0.152 \pm 0.032$ |
| Transformer | NOCD | DMoN | N/A | None | Loss | $0.000 \pm 0.000$ | $0.194 \pm 0.111$ |
| Transformer | NOCD | L2 | N/A | None | Loss | $\mathbf{0.542 \pm 0.124}$ | $\mathbf{0.632 \pm 0.126}$ |
| Transformer | Neuromap | None | N/A | None | Loss | $0.000 \pm 0.000$ | $0.187 \pm 0.169$ |
| Transformer | Neuromap | DMoN | N/A | None | Loss | $0.000 \pm 0.000$ | $0.238 \pm 0.187$ |
| Transformer | Neuromap | L2 | N/A | None | Loss | $0.000 \pm 0.000$ | $0.227 \pm 0.194$ |
| Transformer | SBM$_{\text{NN}}$ | None | N/A | None | Loss | $0.000 \pm 0.000$ | $0.167 \pm 0.000$ |
| Transformer | SBM$_{\text{NN}}$ | DMoN | N/A | None | Loss | $0.000 \pm 0.000$ | $0.167 \pm 0.000$ |
| Transformer | SBM$_{\text{NN}}$ | L2 | N/A | None | Loss | $0.325 \pm 0.181$ | $0.445 \pm 0.200$ |
| MLP | None | None | N/A | None | Loss | $0.488 \pm 0.060$ | $0.604 \pm 0.089$ |
| MLP | None | DMoN | N/A | None | Loss | $0.279 \pm 0.141$ | $0.383 \pm 0.118$ |
| MLP | None | L2 | N/A | None | Loss | $0.000 \pm 0.000$ | $0.240 \pm 0.186$ |
| MLP | DMoN | None | N/A | None | Loss | $0.302 \pm 0.114$ | $0.396 \pm 0.145$ |
| MLP | DMoN | DMoN | N/A | None | Loss | $0.293 \pm 0.106$ | $0.408 \pm 0.163$ |
| MLP | DMoN | L2 | N/A | None | Loss | $0.000 \pm 0.000$ | $0.198 \pm 0.166$ |
| MLP | NOCD | None | N/A | None | Loss | $0.058 \pm 0.128$ | $0.195 \pm 0.102$ |
| MLP | NOCD | DMoN | N/A | None | Loss | $0.114 \pm 0.140$ | $0.258 \pm 0.121$ |
| MLP | NOCD | L2 | N/A | None | Loss | $\mathbf{0.622 \pm 0.090}$ | $\mathbf{0.725 \pm 0.064}$ |
| MLP | Neuromap | None | N/A | None | Loss | $0.020 \pm 0.028$ | $0.308 \pm 0.210$ |
| MLP | Neuromap | DMoN | N/A | None | Loss | $0.055 \pm 0.090$ | $0.223 \pm 0.159$ |
| MLP | Neuromap | L2 | N/A | None | Loss | $0.000 \pm 0.000$ | $0.164 \pm 0.124$ |
| MLP | SBM$_{\text{NN}}$ | None | N/A | None | Loss | $0.033 \pm 0.077$ | $0.183 \pm 0.047$ |
| MLP | SBM$_{\text{NN}}$ | DMoN | N/A | None | Loss | $0.000 \pm 0.000$ | $0.167 \pm 0.000$ |
| MLP | SBM$_{\text{NN}}$ | L2 | N/A | None | Loss | $0.099 \pm 0.193$ | $0.243 \pm 0.189$ |

Table 97: Comparing graph clustering and regularization objectives with an ablation study of regularization. The ablation compares clustering with clusteirng with regularization, clustering without regularization, and regularization without clustering for each neural network. Model selection based on training set MCC.

| Model | $f$ | $L_{\text{regularization}}$ | $L_V$ | NN$_{\boldsymbol{S \to X}}$ | ES | MCC | Accuracy |
|---|---|---|---|---|---|---|---|
| GCN | None | None | ✗ | None | Loss | **0.796 ± 0.058** | **0.859 ± 0.041** |
| GCN | None | DMoN | ✗ | None | Loss | 0.502 ± 0.072 | 0.567 ± 0.060 |
| GCN | None | L2 | ✗ | None | MCC | 0.010 ± 0.199 | 0.309 ± 0.203 |
| GCN | DMoN | None | ✗ | None | Loss | 0.487 ± 0.062 | 0.552 ± 0.060 |
| GCN | DMoN | DMoN | ✗ | None | Loss | 0.497 ± 0.066 | 0.564 ± 0.055 |
| GCN | DMoN | L2 | ✗ | None | MCC | 0.168 ± 0.260 | 0.397 ± 0.250 |
| GCN | NOCD | None | ✗ | None | Loss | 0.391 ± 0.104 | 0.434 ± 0.122 |
| GCN | NOCD | DMoN | ✓ | MLP | Loss | 0.241 ± 0.108 | 0.293 ± 0.068 |
| GCN | NOCD | L2 | ✗ | None | MCC | 0.176 ± 0.147 | 0.313 ± 0.187 |
| GCN | Neuromap | None | ✗ | None | MCC | 0.341 ± 0.204 | 0.413 ± 0.213 |
| GCN | Neuromap | DMoN | ✗ | None | MCC | 0.510 ± 0.141 | 0.615 ± 0.155 |
| GCN | Neuromap | L2 | ✗ | None | MCC | 0.096 ± 0.183 | 0.308 ± 0.185 |
| GCN | SBM$_{\text{NN}}$ | None | ✓ | MLP | Loss | 0.397 ± 0.117 | 0.511 ± 0.126 |
| GCN | SBM$_{\text{NN}}$ | DMoN | ✓ | Transformer | Loss | 0.431 ± 0.126 | 0.539 ± 0.131 |
| GCN | SBM$_{\text{NN}}$ | L2 | ✗ | None | MCC | 0.089 ± 0.128 | 0.283 ± 0.192 |
| GraphSAGE | None | None | ✗ | None | Loss | **0.802 ± 0.047** | **0.864 ± 0.034** |
| GraphSAGE | None | DMoN | ✗ | None | MCC | 0.757 ± 0.068 | 0.820 ± 0.064 |
| GraphSAGE | None | L2 | ✗ | None | MCC | -0.001 ± 0.004 | 0.012 ± 0.039 |
| GraphSAGE | DMoN | None | ✗ | None | Loss | 0.209 ± 0.073 | 0.350 ± 0.054 |
| GraphSAGE | DMoN | DMoN | ✗ | None | Loss | 0.239 ± 0.119 | 0.368 ± 0.088 |
| GraphSAGE | DMoN | L2 | ✗ | None | MCC | 0.018 ± 0.042 | 0.082 ± 0.166 |
| GraphSAGE | NOCD | None | ✓ | MLP | Loss | 0.367 ± 0.121 | 0.454 ± 0.119 |
| GraphSAGE | NOCD | DMoN | ✗ | None | Loss | 0.213 ± 0.060 | 0.278 ± 0.030 |
| GraphSAGE | NOCD | L2 | ✓ | MLP | MCC | 0.026 ± 0.082 | 0.041 ± 0.129 |
| GraphSAGE | Neuromap | None | ✓ | MLP | MCC | 0.350 ± 0.122 | 0.354 ± 0.154 |
| GraphSAGE | Neuromap | DMoN | ✓ | MLP | MCC | 0.394 ± 0.129 | 0.449 ± 0.190 |
| GraphSAGE | Neuromap | L2 | ✗ | None | Loss | 0.000 ± 0.000 | 0.187 ± 0.170 |
| GraphSAGE | SBM$_{\text{NN}}$ | None | ✓ | MLP | MCC | 0.415 ± 0.246 | 0.458 ± 0.287 |
| GraphSAGE | SBM$_{\text{NN}}$ | DMoN | ✗ | None | Loss | 0.294 ± 0.154 | 0.394 ± 0.157 |
| GraphSAGE | SBM$_{\text{NN}}$ | L2 | ✗ | None | MCC | 0.016 ± 0.050 | 0.052 ± 0.165 |
| Transformer | None | None | N/A | None | MCC | 0.027 ± 0.047 | 0.083 ± 0.166 |
| Transformer | None | DMoN | N/A | None | Loss | 0.001 ± 0.002 | 0.128 ± 0.034 |
| Transformer | None | L2 | N/A | None | Loss | 0.356 ± 0.119 | 0.443 ± 0.164 |
| Transformer | DMoN | None | N/A | None | MCC | 0.043 ± 0.074 | 0.091 ± 0.172 |
| Transformer | DMoN | DMoN | N/A | None | MCC | 0.033 ± 0.085 | 0.072 ± 0.179 |
| Transformer | DMoN | L2 | N/A | None | Loss | 0.379 ± 0.116 | 0.490 ± 0.143 |
| Transformer | NOCD | None | N/A | None | Loss | 0.000 ± 0.000 | 0.152 ± 0.032 |
| Transformer | NOCD | DMoN | N/A | None | Loss | 0.000 ± 0.000 | 0.194 ± 0.111 |
| Transformer | NOCD | L2 | N/A | None | Loss | **0.542 ± 0.124** | **0.632 ± 0.126** |
| Transformer | Neuromap | None | N/A | None | Loss | 0.000 ± 0.000 | 0.187 ± 0.169 |
| Transformer | Neuromap | DMoN | N/A | None | Loss | 0.000 ± 0.000 | 0.238 ± 0.187 |
| Transformer | Neuromap | L2 | N/A | None | Loss | 0.000 ± 0.000 | 0.227 ± 0.194 |

Table 97: Comparing graph clustering and regularization objectives for each neural network.

| Model | $f$ | $L_{\text{regularization}}$ | $L_V$ | $NN_{S \to X}$ | ES | MCC | Accuracy |
|---|---|---|---|---|---|---|---|
| Transformer | $SBM_{NN}$ | None | N/A | None | Loss | $0.000 \pm 0.000$ | $0.167 \pm 0.000$ |
| Transformer | $SBM_{NN}$ | DMoN | N/A | None | Loss | $0.000 \pm 0.000$ | $0.167 \pm 0.000$ |
| Transformer | $SBM_{NN}$ | L2 | N/A | None | Loss | $0.325 \pm 0.181$ | $0.445 \pm 0.200$ |
| MLP | None | None | N/A | None | Loss | $0.488 \pm 0.060$ | $0.604 \pm 0.089$ |
| MLP | None | DMoN | N/A | None | Loss | $0.279 \pm 0.141$ | $0.383 \pm 0.118$ |
| MLP | None | L2 | N/A | None | MCC | $0.222 \pm 0.110$ | $0.324 \pm 0.192$ |
| MLP | DMoN | None | N/A | None | Loss | $0.302 \pm 0.114$ | $0.396 \pm 0.145$ |
| MLP | DMoN | DMoN | N/A | None | Loss | $0.293 \pm 0.106$ | $0.408 \pm 0.163$ |
| MLP | DMoN | L2 | N/A | None | MCC | $0.182 \pm 0.142$ | $0.315 \pm 0.192$ |
| MLP | NOCD | None | N/A | None | Loss | $0.058 \pm 0.128$ | $0.195 \pm 0.102$ |
| MLP | NOCD | DMoN | N/A | None | Loss | $0.114 \pm 0.140$ | $0.258 \pm 0.121$ |
| MLP | NOCD | L2 | N/A | None | Loss | $\mathbf{0.622 \pm 0.090}$ | $\mathbf{0.725 \pm 0.064}$ |
| MLP | Neuromap | None | N/A | None | MCC | $0.149 \pm 0.090$ | $0.209 \pm 0.125$ |
| MLP | Neuromap | DMoN | N/A | None | MCC | $0.178 \pm 0.123$ | $0.280 \pm 0.206$ |
| MLP | Neuromap | L2 | N/A | None | MCC | $-0.011 \pm 0.046$ | $0.134 \pm 0.050$ |
| MLP | $SBM_{NN}$ | None | N/A | None | MCC | $0.036 \pm 0.062$ | $0.087 \pm 0.096$ |
| MLP | $SBM_{NN}$ | DMoN | N/A | None | MCC | $0.029 \pm 0.058$ | $0.071 \pm 0.117$ |
| MLP | $SBM_{NN}$ | L2 | N/A | None | Loss | $0.099 \pm 0.193$ | $0.243 \pm 0.189$ |

Table 98: Comparing graph clustering and regularization objectives with an ablation study of regularization. The ablation compares clustering with clusteirng with regularization, clustering without regularization, and regularization without clustering for each neural network. Model selection based on validation set MCC.

| Model | $f$ | $L_{\text{regularization}}$ | $L_V$ | $NN_{S \to X}$ | ES | MCC | Accuracy |
|---|---|---|---|---|---|---|---|
| GCN | None | None | ✗ | None | MCC | $0.816 \pm 0.046$ | $0.870 \pm 0.039$ |
| GCN | None | DMoN | ✗ | None | MCC | $0.757 \pm 0.049$ | $0.827 \pm 0.040$ |
| GCN | None | L2 | ✗ | None | Loss | $0.000 \pm 0.000$ | $0.167 \pm 0.000$ |
| GCN | DMoN | None | ✗ | None | MCC | $\mathbf{0.830 \pm 0.042}$ | $\mathbf{0.882 \pm 0.030}$ |
| GCN | DMoN | DMoN | ✓ | Transformer | MCC | $0.817 \pm 0.054$ | $0.873 \pm 0.038$ |
| GCN | DMoN | L2 | ✗ | None | MCC | $0.168 \pm 0.260$ | $0.397 \pm 0.250$ |
| GCN | NOCD | None | ✓ | MLP | MCC | $0.673 \pm 0.193$ | $0.726 \pm 0.222$ |
| GCN | NOCD | DMoN | ✓ | Transformer | MCC | $0.616 \pm 0.170$ | $0.679 \pm 0.199$ |
| GCN | NOCD | L2 | ✗ | None | MCC | $0.176 \pm 0.147$ | $0.313 \pm 0.187$ |
| GCN | Neuromap | None | ✗ | None | Loss | $0.451 \pm 0.364$ | $0.524 \pm 0.344$ |
| GCN | Neuromap | DMoN | ✗ | None | MCC | $0.510 \pm 0.141$ | $0.615 \pm 0.155$ |
| GCN | Neuromap | L2 | ✗ | None | MCC | $0.096 \pm 0.183$ | $0.308 \pm 0.185$ |
| GCN | $SBM_{NN}$ | None | ✗ | None | MCC | $0.641 \pm 0.162$ | $0.698 \pm 0.203$ |
| GCN | $SBM_{NN}$ | DMoN | ✓ | Transformer | MCC | $0.698 \pm 0.113$ | $0.768 \pm 0.130$ |
| GCN | $SBM_{NN}$ | L2 | ✓ | MLP | MCC | $0.103 \pm 0.152$ | $0.265 \pm 0.174$ |
| GraphSAGE | None | None | ✗ | None | MCC | $\mathbf{0.795 \pm 0.061}$ | $\mathbf{0.856 \pm 0.048}$ |
| GraphSAGE | None | DMoN | ✗ | None | MCC | $0.757 \pm 0.068$ | $0.820 \pm 0.064$ |
| GraphSAGE | None | L2 | ✗ | None | MCC | $-0.001 \pm 0.004$ | $0.012 \pm 0.039$ |
| GraphSAGE | DMoN | None | ✗ | None | MCC | $0.792 \pm 0.047$ | $0.855 \pm 0.034$ |
| GraphSAGE | DMoN | DMoN | ✓ | Transformer | MCC | $0.791 \pm 0.040$ | $0.853 \pm 0.026$ |
| GraphSAGE | DMoN | L2 | ✗ | None | MCC | $0.018 \pm 0.042$ | $0.082 \pm 0.166$ |

Table 98: Comparing graph clustering and regularization objectives for each neural network.

| Model | $f$ | $L_{\text{regularization}}$ | $L_V$ | NN$_{\boldsymbol{S}\rightarrow\boldsymbol{X}}$ | ES | MCC | Accuracy |
|---|---|---|---|---|---|---|---|
| GraphSAGE | NOCD | None | ✗ | None | MCC | $0.635 \pm 0.156$ | $0.690 \pm 0.190$ |
| GraphSAGE | NOCD | DMoN | ✓ | Transformer | MCC | $0.672 \pm 0.171$ | $0.743 \pm 0.180$ |
| GraphSAGE | NOCD | L2 | ✓ | MLP | MCC | $0.026 \pm 0.082$ | $0.041 \pm 0.129$ |
| GraphSAGE | Neuromap | None | ✗ | None | MCC | $0.387 \pm 0.120$ | $0.501 \pm 0.171$ |
| GraphSAGE | Neuromap | DMoN | ✓ | MLP | MCC | $0.394 \pm 0.129$ | $0.449 \pm 0.190$ |
| GraphSAGE | Neuromap | L2 | ✓ | MLP | MCC | $0.005 \pm 0.017$ | $0.014 \pm 0.045$ |
| GraphSAGE | SBM$_{\text{NN}}$ | None | ✗ | None | MCC | $0.462 \pm 0.174$ | $0.504 \pm 0.235$ |
| GraphSAGE | SBM$_{\text{NN}}$ | DMoN | ✗ | None | MCC | $0.474 \pm 0.208$ | $0.546 \pm 0.242$ |
| GraphSAGE | SBM$_{\text{NN}}$ | L2 | ✗ | None | MCC | $0.016 \pm 0.050$ | $0.052 \pm 0.165$ |
| Transformer | None | None | N/A | None | Loss | $0.023 \pm 0.081$ | $0.159 \pm 0.124$ |
| Transformer | None | DMoN | N/A | None | Loss | $0.001 \pm 0.002$ | $0.128 \pm 0.034$ |
| Transformer | None | L2 | N/A | None | MCC | $0.490 \pm 0.107$ | $0.592 \pm 0.134$ |
| Transformer | DMoN | None | N/A | None | MCC | $0.043 \pm 0.074$ | $0.091 \pm 0.172$ |
| Transformer | DMoN | DMoN | N/A | None | MCC | $0.033 \pm 0.085$ | $0.072 \pm 0.179$ |
| Transformer | DMoN | L2 | N/A | None | MCC | $\mathbf{0.497 \pm 0.095}$ | $\mathbf{0.614 \pm 0.115}$ |
| Transformer | NOCD | None | N/A | None | Loss | $0.000 \pm 0.000$ | $0.152 \pm 0.032$ |
| Transformer | NOCD | DMoN | N/A | None | Loss | $0.000 \pm 0.000$ | $0.194 \pm 0.111$ |
| Transformer | NOCD | L2 | N/A | None | MCC | $0.495 \pm 0.265$ | $0.586 \pm 0.310$ |
| Transformer | Neuromap | None | N/A | None | Loss | $0.000 \pm 0.000$ | $0.187 \pm 0.169$ |
| Transformer | Neuromap | DMoN | N/A | None | Loss | $0.000 \pm 0.000$ | $0.238 \pm 0.187$ |
| Transformer | Neuromap | L2 | N/A | None | Loss | $0.000 \pm 0.000$ | $0.227 \pm 0.194$ |
| Transformer | SBM$_{\text{NN}}$ | None | N/A | None | Loss | $0.000 \pm 0.000$ | $0.167 \pm 0.000$ |
| Transformer | SBM$_{\text{NN}}$ | DMoN | N/A | None | Loss | $0.000 \pm 0.000$ | $0.167 \pm 0.000$ |
| Transformer | SBM$_{\text{NN}}$ | L2 | N/A | None | Loss | $0.325 \pm 0.181$ | $0.445 \pm 0.200$ |
| MLP | None | None | N/A | None | MCC | $0.493 \pm 0.120$ | $0.613 \pm 0.146$ |
| MLP | None | DMoN | N/A | None | MCC | $0.446 \pm 0.126$ | $0.551 \pm 0.144$ |
| MLP | None | L2 | N/A | None | MCC | $0.222 \pm 0.110$ | $0.324 \pm 0.192$ |
| MLP | DMoN | None | N/A | None | MCC | $0.533 \pm 0.093$ | $0.641 \pm 0.089$ |
| MLP | DMoN | DMoN | N/A | None | MCC | $0.548 \pm 0.097$ | $0.655 \pm 0.087$ |
| MLP | DMoN | L2 | N/A | None | MCC | $0.182 \pm 0.142$ | $0.315 \pm 0.192$ |
| MLP | NOCD | None | N/A | None | MCC | $0.197 \pm 0.185$ | $0.240 \pm 0.230$ |
| MLP | NOCD | DMoN | N/A | None | MCC | $0.223 \pm 0.185$ | $0.295 \pm 0.236$ |
| MLP | NOCD | L2 | N/A | None | Loss | $\mathbf{0.622 \pm 0.090}$ | $\mathbf{0.725 \pm 0.064}$ |
| MLP | Neuromap | None | N/A | None | MCC | $0.149 \pm 0.090$ | $0.209 \pm 0.125$ |
| MLP | Neuromap | DMoN | N/A | None | MCC | $0.178 \pm 0.123$ | $0.280 \pm 0.206$ |
| MLP | Neuromap | L2 | N/A | None | Loss | $0.000 \pm 0.000$ | $0.164 \pm 0.124$ |
| MLP | SBM$_{\text{NN}}$ | None | N/A | None | Loss | $0.033 \pm 0.077$ | $0.183 \pm 0.047$ |
| MLP | SBM$_{\text{NN}}$ | DMoN | N/A | None | MCC | $0.029 \pm 0.058$ | $0.071 \pm 0.117$ |
| MLP | SBM$_{\text{NN}}$ | L2 | N/A | None | Loss | $0.099 \pm 0.193$ | $0.243 \pm 0.189$ |

F.4.11 ROMAN-EMPIRE DATASET, DEFAULT SPLIT WITH DEFAULT TRAIN NODES PER CLASS AND DEFAULT VALIDATION NODES.

Table 99: Comparing graph clustering and regularization objectives with an ablation study of regularization. The ablation compares clustering with clusteirng with regularization, clustering without regularization, and regularization without clustering for each neural network. Model selection based on training loss.

| Model | $f$ | $L_{\text{regularization}}$ | $L_V$ | $\text{NN}_{S \to X}$ | ES | MCC | Accuracy |
|---|---|---|---|---|---|---|---|
| GCN | None | None | ✗ | None | Loss | $0.087 \pm 0.005$ | $0.191 \pm 0.006$ |
| GCN | None | DMoN | ✗ | None | Loss | $0.087 \pm 0.005$ | $0.192 \pm 0.005$ |
| GCN | None | L2 | ✗ | None | Loss | $0.000 \pm 0.000$ | $0.140 \pm 0.000$ |
| GCN | DMoN | None | ✗ | None | Loss | $\mathbf{0.089 \pm 0.005}$ | $0.193 \pm 0.004$ |
| GCN | DMoN | DMoN | ✗ | None | Loss | $0.088 \pm 0.007$ | $0.193 \pm 0.007$ |
| GCN | DMoN | L2 | ✗ | None | Loss | $0.000 \pm 0.000$ | $0.140 \pm 0.000$ |
| GCN | NOCD | None | ✗ | None | Loss | $0.056 \pm 0.018$ | $0.163 \pm 0.016$ |
| GCN | NOCD | DMoN | ✗ | None | Loss | $0.065 \pm 0.008$ | $0.173 \pm 0.009$ |
| GCN | NOCD | L2 | ✗ | None | Loss | $0.000 \pm 0.000$ | $0.140 \pm 0.000$ |
| GCN | Neuromap | None | ✓ | MLP | Loss | $0.000 \pm 0.000$ | $0.139 \pm 0.001$ |
| GCN | Neuromap | DMoN | ✗ | None | Loss | $0.000 \pm 0.000$ | $0.140 \pm 0.000$ |
| GCN | Neuromap | L2 | ✗ | None | Loss | $0.000 \pm 0.000$ | $0.140 \pm 0.000$ |
| GCN | SBM$_{\text{NN}}$ | None | ✗ | None | Loss | $0.088 \pm 0.018$ | $\mathbf{0.196 \pm 0.017}$ |
| GCN | SBM$_{\text{NN}}$ | DMoN | ✓ | MLP | Loss | $0.074 \pm 0.032$ | $0.183 \pm 0.021$ |
| GCN | SBM$_{\text{NN}}$ | L2 | ✗ | None | Loss | $0.000 \pm 0.000$ | $0.140 \pm 0.000$ |
| GraphSAGE | None | None | ✗ | None | Loss | $0.285 \pm 0.150$ | $0.355 \pm 0.113$ |
| GraphSAGE | None | DMoN | ✗ | None | Loss | $0.143 \pm 0.184$ | $0.248 \pm 0.140$ |
| GraphSAGE | None | L2 | ✗ | None | Loss | $0.000 \pm 0.000$ | $0.139 \pm 0.000$ |
| GraphSAGE | DMoN | None | ✓ | MLP | Loss | $0.318 \pm 0.034$ | $0.381 \pm 0.027$ |
| GraphSAGE | DMoN | DMoN | ✓ | MLP | Loss | $\mathbf{0.323 \pm 0.019}$ | $\mathbf{0.385 \pm 0.014}$ |
| GraphSAGE | DMoN | L2 | ✗ | None | Loss | $0.000 \pm 0.000$ | $0.140 \pm 0.000$ |
| GraphSAGE | NOCD | None | ✓ | MLP | Loss | $0.280 \pm 0.037$ | $0.317 \pm 0.050$ |
| GraphSAGE | NOCD | DMoN | ✓ | MLP | Loss | $0.248 \pm 0.065$ | $0.278 \pm 0.061$ |
| GraphSAGE | NOCD | L2 | ✗ | None | Loss | $0.000 \pm 0.000$ | $0.139 \pm 0.001$ |
| GraphSAGE | Neuromap | None | ✗ | None | Loss | $0.000 \pm 0.000$ | $0.130 \pm 0.014$ |
| GraphSAGE | Neuromap | DMoN | ✗ | None | Loss | $0.000 \pm 0.000$ | $0.136 \pm 0.009$ |
| GraphSAGE | Neuromap | L2 | ✗ | None | Loss | $0.000 \pm 0.000$ | $0.139 \pm 0.001$ |
| GraphSAGE | SBM$_{\text{NN}}$ | None | ✗ | None | Loss | $0.167 \pm 0.123$ | $0.248 \pm 0.084$ |
| GraphSAGE | SBM$_{\text{NN}}$ | DMoN | ✗ | None | Loss | $0.201 \pm 0.112$ | $0.274 \pm 0.081$ |
| GraphSAGE | SBM$_{\text{NN}}$ | L2 | ✗ | None | Loss | $0.000 \pm 0.000$ | $0.139 \pm 0.001$ |
| Transformer | None | None | N/A | None | Loss | $\mathbf{0.000 \pm 0.000}$ | $0.139 \pm 0.000$ |
| Transformer | None | DMoN | N/A | None | Loss | $0.000 \pm 0.000$ | $\mathbf{0.140 \pm 0.000}$ |
| Transformer | None | L2 | N/A | None | Loss | $0.000 \pm 0.000$ | $0.139 \pm 0.000$ |
| Transformer | DMoN | None | N/A | None | Loss | $0.000 \pm 0.000$ | $0.140 \pm 0.000$ |
| Transformer | DMoN | DMoN | N/A | None | Loss | $0.000 \pm 0.000$ | $0.140 \pm 0.000$ |
| Transformer | DMoN | L2 | N/A | None | Loss | $0.000 \pm 0.000$ | $0.139 \pm 0.000$ |
| Transformer | NOCD | None | N/A | None | Loss | $0.000 \pm 0.000$ | $0.126 \pm 0.031$ |
| Transformer | NOCD | DMoN | N/A | None | Loss | $0.000 \pm 0.000$ | $0.132 \pm 0.015$ |
| Transformer | NOCD | L2 | N/A | None | Loss | $0.000 \pm 0.000$ | $0.140 \pm 0.000$ |
| Transformer | Neuromap | None | N/A | None | Loss | $0.000 \pm 0.000$ | $0.066 \pm 0.058$ |
| Transformer | Neuromap | DMoN | N/A | None | Loss | $0.000 \pm 0.000$ | $0.098 \pm 0.040$ |
| Transformer | Neuromap | L2 | N/A | None | Loss | $0.000 \pm 0.000$ | $0.109 \pm 0.046$ |

Table 99: Comparing graph clustering and regularization objectives for each neural network.

| Model | $f$ | $L_{\text{regularization}}$ | $L_V$ | $\text{NN}_{S \to X}$ | ES | MCC | Accuracy |
|---|---|---|---|---|---|---|---|
| Transformer | $\text{SBM}_{\text{NN}}$ | None | N/A | None | Loss | $0.000 \pm 0.000$ | $0.100 \pm 0.050$ |
| Transformer | $\text{SBM}_{\text{NN}}$ | DMoN | N/A | None | Loss | $0.000 \pm 0.000$ | $0.086 \pm 0.048$ |
| Transformer | $\text{SBM}_{\text{NN}}$ | L2 | N/A | None | Loss | $0.000 \pm 0.000$ | $0.139 \pm 0.001$ |
| MLP | None | None | N/A | None | Loss | $0.056 \pm 0.090$ | $0.155 \pm 0.027$ |
| MLP | None | DMoN | N/A | None | Loss | $0.081 \pm 0.102$ | $0.183 \pm 0.062$ |
| MLP | None | L2 | N/A | None | Loss | $0.000 \pm 0.000$ | $0.140 \pm 0.000$ |
| MLP | DMoN | None | N/A | None | Loss | $0.061 \pm 0.099$ | $0.173 \pm 0.065$ |
| MLP | DMoN | DMoN | N/A | None | Loss | $\mathbf{0.120 \pm 0.112}$ | $\mathbf{0.203 \pm 0.071}$ |
| MLP | DMoN | L2 | N/A | None | Loss | $0.000 \pm 0.000$ | $0.140 \pm 0.000$ |
| MLP | NOCD | None | N/A | None | Loss | $0.021 \pm 0.067$ | $0.141 \pm 0.029$ |
| MLP | NOCD | DMoN | N/A | None | Loss | $0.000 \pm 0.000$ | $0.111 \pm 0.039$ |
| MLP | NOCD | L2 | N/A | None | Loss | $0.000 \pm 0.000$ | $0.139 \pm 0.001$ |
| MLP | Neuromap | None | N/A | None | Loss | $0.000 \pm 0.000$ | $0.059 \pm 0.041$ |
| MLP | Neuromap | DMoN | N/A | None | Loss | $0.000 \pm 0.000$ | $0.074 \pm 0.053$ |
| MLP | Neuromap | L2 | N/A | None | Loss | $0.000 \pm 0.000$ | $0.129 \pm 0.032$ |
| MLP | $\text{SBM}_{\text{NN}}$ | None | N/A | None | Loss | $0.000 \pm 0.000$ | $0.098 \pm 0.043$ |
| MLP | $\text{SBM}_{\text{NN}}$ | DMoN | N/A | None | Loss | $0.000 \pm 0.000$ | $0.116 \pm 0.031$ |
| MLP | $\text{SBM}_{\text{NN}}$ | L2 | N/A | None | Loss | $0.000 \pm 0.000$ | $0.126 \pm 0.031$ |

Table 100: Comparing graph clustering and regularization objectives with an ablation study of regularization. The ablation compares clustering with clusteirng with regularization, clustering without regularization, and regularization without clustering for each neural network. Model selection based on training set MCC.

| Model | $f$ | $L_{\text{regularization}}$ | $L_V$ | $\text{NN}_{S \to X}$ | ES | MCC | Accuracy |
|---|---|---|---|---|---|---|---|
| GCN | None | None | ✗ | None | Loss | $0.087 \pm 0.005$ | $0.191 \pm 0.006$ |
| GCN | None | DMoN | ✗ | None | Loss | $0.087 \pm 0.005$ | $0.192 \pm 0.005$ |
| GCN | None | L2 | ✗ | None | Loss | $0.000 \pm 0.000$ | $0.140 \pm 0.000$ |
| GCN | DMoN | None | ✓ | MLP | Loss | $\mathbf{0.089 \pm 0.005}$ | $0.198 \pm 0.004$ |
| GCN | DMoN | DMoN | ✓ | MLP | Loss | $0.089 \pm 0.002$ | $\mathbf{0.199 \pm 0.002}$ |
| GCN | DMoN | L2 | ✓ | MLP | MCC | $0.006 \pm 0.066$ | $0.097 \pm 0.063$ |
| GCN | NOCD | None | ✗ | None | Loss | $0.056 \pm 0.018$ | $0.163 \pm 0.016$ |
| GCN | NOCD | DMoN | ✗ | None | Loss | $0.065 \pm 0.008$ | $0.173 \pm 0.009$ |
| GCN | NOCD | L2 | ✓ | Transformer | MCC | $0.022 \pm 0.035$ | $0.108 \pm 0.078$ |
| GCN | Neuromap | None | ✓ | MLP | MCC | $0.032 \pm 0.052$ | $0.122 \pm 0.060$ |
| GCN | Neuromap | DMoN | ✓ | Transformer | MCC | $0.010 \pm 0.030$ | $0.113 \pm 0.053$ |
| GCN | Neuromap | L2 | ✓ | Transformer | MCC | $0.020 \pm 0.053$ | $0.128 \pm 0.043$ |
| GCN | $\text{SBM}_{\text{NN}}$ | None | ✗ | None | Loss | $0.088 \pm 0.018$ | $0.196 \pm 0.017$ |
| GCN | $\text{SBM}_{\text{NN}}$ | DMoN | ✗ | None | Loss | $0.073 \pm 0.032$ | $0.187 \pm 0.023$ |
| GCN | $\text{SBM}_{\text{NN}}$ | L2 | ✓ | Transformer | MCC | $0.036 \pm 0.028$ | $0.112 \pm 0.079$ |
| GraphSAGE | None | None | ✗ | None | Loss | $0.285 \pm 0.150$ | $0.355 \pm 0.113$ |
| GraphSAGE | None | DMoN | ✗ | None | Loss | $0.143 \pm 0.184$ | $0.248 \pm 0.140$ |
| GraphSAGE | None | L2 | ✗ | None | Loss | $0.000 \pm 0.000$ | $0.139 \pm 0.000$ |
| GraphSAGE | DMoN | None | ✓ | MLP | Loss | $0.318 \pm 0.034$ | $0.381 \pm 0.027$ |
| GraphSAGE | DMoN | DMoN | ✓ | MLP | Loss | $\mathbf{0.323 \pm 0.019}$ | $\mathbf{0.385 \pm 0.014}$ |
| GraphSAGE | DMoN | L2 | ✓ | MLP | MCC | $-0.000 \pm 0.000$ | $0.010 \pm 0.018$ |

Continued on next page

Table 100: Comparing graph clustering and regularization objectives for each neural network.

| Model | $f$ | $L_{\text{regularization}}$ | $L_V$ | NN$_{S \to X}$ | ES | MCC | Accuracy |
|-------|-----|------------------------------|-------|----------------|-----|-----|----------|
| GraphSAGE | NOCD | None | ✓ | MLP | Loss | $0.280 \pm 0.037$ | $0.317 \pm 0.050$ |
| GraphSAGE | NOCD | DMoN | ✓ | MLP | Loss | $0.248 \pm 0.065$ | $0.278 \pm 0.061$ |
| GraphSAGE | NOCD | L2 | ✓ | MLP | MCC | $0.001 \pm 0.006$ | $0.015 \pm 0.043$ |
| GraphSAGE | Neuromap | None | ✗ | None | Loss | $0.000 \pm 0.000$ | $0.130 \pm 0.014$ |
| GraphSAGE | Neuromap | DMoN | ✗ | None | MCC | $0.013 \pm 0.041$ | $0.023 \pm 0.071$ |
| GraphSAGE | Neuromap | L2 | ✓ | Transformer | MCC | $0.000 \pm 0.000$ | $0.002 \pm 0.006$ |
| GraphSAGE | SBM$_{NN}$ | None | ✗ | None | Loss | $0.167 \pm 0.123$ | $0.248 \pm 0.084$ |
| GraphSAGE | SBM$_{NN}$ | DMoN | ✗ | None | Loss | $0.201 \pm 0.112$ | $0.274 \pm 0.081$ |
| GraphSAGE | SBM$_{NN}$ | L2 | ✓ | MLP | MCC | $0.007 \pm 0.021$ | $0.022 \pm 0.041$ |
| Transformer | None | None | N/A | None | Loss | $0.000 \pm 0.000$ | $0.139 \pm 0.000$ |
| Transformer | None | DMoN | N/A | None | Loss | $0.000 \pm 0.000$ | $\mathbf{0.140 \pm 0.000}$ |
| Transformer | None | L2 | N/A | None | MCC | $0.012 \pm 0.038$ | $0.022 \pm 0.068$ |
| Transformer | DMoN | None | N/A | None | Loss | $0.000 \pm 0.000$ | $0.140 \pm 0.000$ |
| Transformer | DMoN | DMoN | N/A | None | Loss | $0.000 \pm 0.000$ | $0.140 \pm 0.000$ |
| Transformer | DMoN | L2 | N/A | None | MCC | $0.000 \pm 0.019$ | $0.020 \pm 0.043$ |
| Transformer | NOCD | None | N/A | None | Loss | $0.000 \pm 0.000$ | $0.126 \pm 0.031$ |
| Transformer | NOCD | DMoN | N/A | None | Loss | $0.000 \pm 0.000$ | $0.132 \pm 0.015$ |
| Transformer | NOCD | L2 | N/A | None | MCC | $\mathbf{0.013 \pm 0.039}$ | $0.030 \pm 0.065$ |
| Transformer | Neuromap | None | N/A | None | Loss | $0.000 \pm 0.000$ | $0.066 \pm 0.058$ |
| Transformer | Neuromap | DMoN | N/A | None | Loss | $0.000 \pm 0.000$ | $0.098 \pm 0.040$ |
| Transformer | Neuromap | L2 | N/A | None | MCC | $0.008 \pm 0.014$ | $0.042 \pm 0.048$ |
| Transformer | SBM$_{NN}$ | None | N/A | None | Loss | $0.000 \pm 0.000$ | $0.100 \pm 0.050$ |
| Transformer | SBM$_{NN}$ | DMoN | N/A | None | Loss | $0.000 \pm 0.000$ | $0.086 \pm 0.048$ |
| Transformer | SBM$_{NN}$ | L2 | N/A | None | Loss | $0.000 \pm 0.000$ | $0.139 \pm 0.001$ |
| MLP | None | None | N/A | None | MCC | $0.226 \pm 0.037$ | $0.251 \pm 0.038$ |
| MLP | None | DMoN | N/A | None | MCC | $0.216 \pm 0.026$ | $0.245 \pm 0.025$ |
| MLP | None | L2 | N/A | None | MCC | $0.004 \pm 0.029$ | $0.049 \pm 0.069$ |
| MLP | DMoN | None | N/A | None | MCC | $0.239 \pm 0.046$ | $0.261 \pm 0.051$ |
| MLP | DMoN | DMoN | N/A | None | MCC | $\mathbf{0.245 \pm 0.043}$ | $\mathbf{0.262 \pm 0.051}$ |
| MLP | DMoN | L2 | N/A | None | Loss | $0.000 \pm 0.000$ | $0.140 \pm 0.000$ |
| MLP | NOCD | None | N/A | None | MCC | $0.055 \pm 0.090$ | $0.065 \pm 0.105$ |
| MLP | NOCD | DMoN | N/A | None | MCC | $0.062 \pm 0.095$ | $0.083 \pm 0.111$ |
| MLP | NOCD | L2 | N/A | None | Loss | $0.000 \pm 0.000$ | $0.139 \pm 0.001$ |
| MLP | Neuromap | None | N/A | None | Loss | $0.000 \pm 0.000$ | $0.059 \pm 0.041$ |
| MLP | Neuromap | DMoN | N/A | None | MCC | $0.001 \pm 0.002$ | $0.037 \pm 0.057$ |
| MLP | Neuromap | L2 | N/A | None | Loss | $0.000 \pm 0.000$ | $0.129 \pm 0.032$ |
| MLP | SBM$_{NN}$ | None | N/A | None | MCC | $0.002 \pm 0.007$ | $0.019 \pm 0.044$ |
| MLP | SBM$_{NN}$ | DMoN | N/A | None | Loss | $0.000 \pm 0.000$ | $0.116 \pm 0.031$ |
| MLP | SBM$_{NN}$ | L2 | N/A | None | Loss | $0.000 \pm 0.000$ | $0.126 \pm 0.031$ |

Table 101: Comparing graph clustering and regularization objectives with an ablation study of regularization. The ablation compares clustering with clusteirng with regularization, clustering without regularization, and regularization without clustering for each neural network. Model selection based on validation set MCC.

| Model | $f$ | $L_{\text{regularization}}$ | $L_V$ | $\text{NN}_{\boldsymbol{S}\to\boldsymbol{X}}$ | ES | MCC | Accuracy |
|---|---|---|---|---|---|---|---|
| GCN | None | None | ✗ | None | Loss | $0.087 \pm 0.005$ | $0.191 \pm 0.006$ |
| GCN | None | DMoN | ✗ | None | Loss | $0.087 \pm 0.005$ | $0.192 \pm 0.005$ |
| GCN | None | L2 | ✗ | None | Loss | $0.000 \pm 0.000$ | $0.140 \pm 0.000$ |
| GCN | DMoN | None | ✗ | None | Loss | $0.089 \pm 0.005$ | $0.193 \pm 0.004$ |
| GCN | DMoN | DMoN | ✓ | MLP | Loss | $\mathbf{0.089} \pm \mathbf{0.002}$ | $\mathbf{0.199} \pm \mathbf{0.002}$ |
| GCN | DMoN | L2 | ✓ | MLP | MCC | $0.006 \pm 0.066$ | $0.097 \pm 0.063$ |
| GCN | NOCD | None | ✗ | None | Loss | $0.056 \pm 0.018$ | $0.163 \pm 0.016$ |
| GCN | NOCD | DMoN | ✗ | None | Loss | $0.065 \pm 0.008$ | $0.173 \pm 0.009$ |
| GCN | NOCD | L2 | ✓ | Transformer | MCC | $0.022 \pm 0.035$ | $0.108 \pm 0.078$ |
| GCN | Neuromap | None | ✓ | MLP | MCC | $0.032 \pm 0.052$ | $0.122 \pm 0.060$ |
| GCN | Neuromap | DMoN | ✓ | Transformer | MCC | $0.010 \pm 0.030$ | $0.113 \pm 0.053$ |
| GCN | Neuromap | L2 | ✓ | Transformer | MCC | $0.020 \pm 0.053$ | $0.128 \pm 0.043$ |
| GCN | $\text{SBM}_{\text{NN}}$ | None | ✗ | None | Loss | $0.088 \pm 0.018$ | $0.196 \pm 0.017$ |
| GCN | $\text{SBM}_{\text{NN}}$ | DMoN | ✓ | MLP | Loss | $0.074 \pm 0.032$ | $0.183 \pm 0.021$ |
| GCN | $\text{SBM}_{\text{NN}}$ | L2 | ✓ | Transformer | MCC | $0.036 \pm 0.028$ | $0.112 \pm 0.079$ |
| GraphSAGE | None | None | ✗ | None | Loss | $0.285 \pm 0.150$ | $0.355 \pm 0.113$ |
| GraphSAGE | None | DMoN | ✗ | None | Loss | $0.143 \pm 0.184$ | $0.248 \pm 0.140$ |
| GraphSAGE | None | L2 | ✗ | None | Loss | $0.000 \pm 0.000$ | $0.139 \pm 0.000$ |
| GraphSAGE | DMoN | None | ✓ | MLP | Loss | $0.318 \pm 0.034$ | $0.381 \pm 0.027$ |
| GraphSAGE | DMoN | DMoN | ✓ | MLP | Loss | $\mathbf{0.323} \pm \mathbf{0.019}$ | $\mathbf{0.385} \pm \mathbf{0.014}$ |
| GraphSAGE | DMoN | L2 | ✓ | Transformer | MCC | $-0.000 \pm 0.007$ | $0.021 \pm 0.025$ |
| GraphSAGE | NOCD | None | ✓ | MLP | Loss | $0.280 \pm 0.037$ | $0.317 \pm 0.050$ |
| GraphSAGE | NOCD | DMoN | ✓ | MLP | Loss | $0.248 \pm 0.065$ | $0.278 \pm 0.061$ |
| GraphSAGE | NOCD | L2 | ✓ | MLP | MCC | $0.001 \pm 0.006$ | $0.015 \pm 0.043$ |
| GraphSAGE | Neuromap | None | ✗ | None | Loss | $0.000 \pm 0.000$ | $0.130 \pm 0.014$ |
| GraphSAGE | Neuromap | DMoN | ✗ | None | MCC | $0.013 \pm 0.041$ | $0.023 \pm 0.071$ |
| GraphSAGE | Neuromap | L2 | ✓ | Transformer | MCC | $0.000 \pm 0.000$ | $0.002 \pm 0.006$ |
| GraphSAGE | $\text{SBM}_{\text{NN}}$ | None | ✗ | None | Loss | $0.167 \pm 0.123$ | $0.248 \pm 0.084$ |
| GraphSAGE | $\text{SBM}_{\text{NN}}$ | DMoN | ✗ | None | Loss | $0.201 \pm 0.112$ | $0.274 \pm 0.081$ |
| GraphSAGE | $\text{SBM}_{\text{NN}}$ | L2 | ✓ | MLP | MCC | $0.007 \pm 0.021$ | $0.022 \pm 0.041$ |
| Transformer | None | None | N/A | None | Loss | $0.000 \pm 0.000$ | $0.139 \pm 0.000$ |
| Transformer | None | DMoN | N/A | None | Loss | $0.000 \pm 0.000$ | $\mathbf{0.140} \pm \mathbf{0.000}$ |
| Transformer | None | L2 | N/A | None | MCC | $0.012 \pm 0.038$ | $0.022 \pm 0.068$ |
| Transformer | DMoN | None | N/A | None | Loss | $0.000 \pm 0.000$ | $0.140 \pm 0.000$ |
| Transformer | DMoN | DMoN | N/A | None | Loss | $0.000 \pm 0.000$ | $0.140 \pm 0.000$ |
| Transformer | DMoN | L2 | N/A | None | MCC | $0.000 \pm 0.019$ | $0.020 \pm 0.043$ |
| Transformer | NOCD | None | N/A | None | Loss | $0.000 \pm 0.000$ | $0.126 \pm 0.031$ |
| Transformer | NOCD | DMoN | N/A | None | Loss | $0.000 \pm 0.000$ | $0.132 \pm 0.015$ |
| Transformer | NOCD | L2 | N/A | None | MCC | $\mathbf{0.013} \pm \mathbf{0.039}$ | $0.030 \pm 0.065$ |
| Transformer | Neuromap | None | N/A | None | Loss | $0.000 \pm 0.000$ | $0.066 \pm 0.058$ |
| Transformer | Neuromap | DMoN | N/A | None | Loss | $0.000 \pm 0.000$ | $0.098 \pm 0.040$ |
| Transformer | Neuromap | L2 | N/A | None | MCC | $0.008 \pm 0.014$ | $0.042 \pm 0.048$ |

Table 101: Comparing graph clustering and regularization objectives for each neural network.

| Model | $f$ | $L_{\text{regularization}}$ | $L_V$ | NN$_{S \to X}$ | ES | MCC | Accuracy |
|---|---|---|---|---|---|---|---|
| Transformer | SBM$_{\text{NN}}$ | None | N/A | None | Loss | $0.000 \pm 0.000$ | $0.100 \pm 0.050$ |
| Transformer | SBM$_{\text{NN}}$ | DMoN | N/A | None | Loss | $0.000 \pm 0.000$ | $0.086 \pm 0.048$ |
| Transformer | SBM$_{\text{NN}}$ | L2 | N/A | None | Loss | $0.000 \pm 0.000$ | $0.139 \pm 0.001$ |
| MLP | None | None | N/A | None | MCC | $0.226 \pm 0.037$ | $0.251 \pm 0.038$ |
| MLP | None | DMoN | N/A | None | MCC | $0.216 \pm 0.026$ | $0.245 \pm 0.025$ |
| MLP | None | L2 | N/A | None | MCC | $0.004 \pm 0.029$ | $0.049 \pm 0.069$ |
| MLP | DMoN | None | N/A | None | MCC | $0.239 \pm 0.046$ | $0.261 \pm 0.051$ |
| MLP | DMoN | DMoN | N/A | None | MCC | $\mathbf{0.245 \pm 0.043}$ | $\mathbf{0.262 \pm 0.051}$ |
| MLP | DMoN | L2 | N/A | None | Loss | $0.000 \pm 0.000$ | $0.140 \pm 0.000$ |
| MLP | NOCD | None | N/A | None | MCC | $0.055 \pm 0.090$ | $0.065 \pm 0.105$ |
| MLP | NOCD | DMoN | N/A | None | MCC | $0.062 \pm 0.095$ | $0.083 \pm 0.111$ |
| MLP | NOCD | L2 | N/A | None | MCC | $-0.000 \pm 0.071$ | $0.059 \pm 0.077$ |
| MLP | Neuromap | None | N/A | None | Loss | $0.000 \pm 0.000$ | $0.059 \pm 0.041$ |
| MLP | Neuromap | DMoN | N/A | None | MCC | $0.001 \pm 0.002$ | $0.037 \pm 0.057$ |
| MLP | Neuromap | L2 | N/A | None | Loss | $0.000 \pm 0.000$ | $0.129 \pm 0.032$ |
| MLP | SBM$_{\text{NN}}$ | None | N/A | None | MCC | $0.002 \pm 0.007$ | $0.019 \pm 0.044$ |
| MLP | SBM$_{\text{NN}}$ | DMoN | N/A | None | Loss | $0.000 \pm 0.000$ | $0.116 \pm 0.031$ |
| MLP | SBM$_{\text{NN}}$ | L2 | N/A | None | Loss | $0.000 \pm 0.000$ | $0.126 \pm 0.031$ |

F.4.12    ROMAN-EMPIRE DATASET, SPARSE SPLIT WITH 2 TRAIN NODES PER CLASS AND 50
VALIDATION NODES.

Table 102: Comparing graph clustering and regularization objectives with an ablation study of regularization. The ablation compares clustering with clusteirng with regularization, clustering without regularization, and regularization without clustering for each neural network. Model selection based on training loss.

| Model | $f$ | $L_{\text{regularization}}$ | $L_V$ | NN$_{S \to X}$ | ES | MCC | Accuracy |
|---|---|---|---|---|---|---|---|
| GCN | None | None | ✗ | None | Loss | $0.089 \pm 0.003$ | $0.195 \pm 0.005$ |
| GCN | None | DMoN | ✗ | None | Loss | $\mathbf{0.090 \pm 0.005}$ | $\mathbf{0.196 \pm 0.006}$ |
| GCN | None | L2 | ✗ | None | Loss | $0.000 \pm 0.000$ | $0.140 \pm 0.000$ |
| GCN | DMoN | None | ✗ | None | Loss | $0.090 \pm 0.005$ | $0.195 \pm 0.005$ |
| GCN | DMoN | DMoN | ✗ | None | Loss | $0.090 \pm 0.006$ | $0.195 \pm 0.006$ |
| GCN | DMoN | L2 | ✗ | None | Loss | $0.000 \pm 0.000$ | $0.140 \pm 0.000$ |
| GCN | NOCD | None | ✗ | None | Loss | $0.057 \pm 0.019$ | $0.166 \pm 0.017$ |
| GCN | NOCD | DMoN | ✗ | None | Loss | $0.049 \pm 0.025$ | $0.161 \pm 0.018$ |
| GCN | NOCD | L2 | ✗ | None | Loss | $0.000 \pm 0.000$ | $0.140 \pm 0.000$ |
| GCN | Neuromap | None | ✗ | None | Loss | $0.000 \pm 0.000$ | $0.139 \pm 0.001$ |
| GCN | Neuromap | DMoN | ✗ | None | Loss | $0.000 \pm 0.000$ | $0.139 \pm 0.001$ |
| GCN | Neuromap | L2 | ✗ | None | Loss | $0.000 \pm 0.000$ | $0.140 \pm 0.000$ |
| GCN | SBM$_{\text{NN}}$ | None | ✓ | MLP | Loss | $0.069 \pm 0.023$ | $0.178 \pm 0.015$ |
| GCN | SBM$_{\text{NN}}$ | DMoN | ✗ | None | Loss | $0.072 \pm 0.023$ | $0.185 \pm 0.018$ |
| GCN | SBM$_{\text{NN}}$ | L2 | ✗ | None | Loss | $0.000 \pm 0.000$ | $0.140 \pm 0.000$ |
| GraphSAGE | None | None | ✗ | None | Loss | $0.141 \pm 0.183$ | $0.247 \pm 0.139$ |
| GraphSAGE | None | DMoN | ✗ | None | Loss | $0.214 \pm 0.184$ | $0.301 \pm 0.139$ |
| GraphSAGE | None | L2 | ✗ | None | Loss | $0.000 \pm 0.000$ | $0.140 \pm 0.000$ |

Continued on next page

Table 102: Comparing graph clustering and regularization objectives for each neural network.

| Model | $f$ | $L_{\text{regularization}}$ | $L_V$ | NN$_{S \to X}$ | ES | MCC | Accuracy |
|---|---|---|---|---|---|---|---|
| GraphSAGE | DMoN | None | ✓ | MLP | Loss | $0.324 \pm 0.035$ | $0.385 \pm 0.026$ |
| GraphSAGE | DMoN | DMoN | ✓ | MLP | Loss | $\mathbf{0.327 \pm 0.019}$ | $\mathbf{0.388 \pm 0.014}$ |
| GraphSAGE | DMoN | L2 | ✗ | None | Loss | $0.000 \pm 0.000$ | $0.139 \pm 0.000$ |
| GraphSAGE | NOCD | None | ✓ | MLP | Loss | $0.272 \pm 0.077$ | $0.316 \pm 0.073$ |
| GraphSAGE | NOCD | DMoN | ✓ | MLP | Loss | $0.280 \pm 0.050$ | $0.312 \pm 0.065$ |
| GraphSAGE | NOCD | L2 | ✗ | None | Loss | $0.000 \pm 0.000$ | $0.139 \pm 0.001$ |
| GraphSAGE | Neuromap | None | ✗ | None | Loss | $0.000 \pm 0.000$ | $0.137 \pm 0.009$ |
| GraphSAGE | Neuromap | DMoN | ✓ | MLP | Loss | $0.000 \pm 0.000$ | $0.129 \pm 0.017$ |
| GraphSAGE | Neuromap | L2 | ✗ | None | Loss | $0.000 \pm 0.000$ | $0.140 \pm 0.000$ |
| GraphSAGE | SBM$_{\text{NN}}$ | None | ✗ | None | Loss | $0.183 \pm 0.101$ | $0.256 \pm 0.066$ |
| GraphSAGE | SBM$_{\text{NN}}$ | DMoN | ✗ | None | Loss | $0.183 \pm 0.130$ | $0.269 \pm 0.093$ |
| GraphSAGE | SBM$_{\text{NN}}$ | L2 | ✗ | None | Loss | $0.000 \pm 0.000$ | $0.139 \pm 0.001$ |
| Transformer | None | None | N/A | None | Loss | $\mathbf{0.000 \pm 0.000}$ | $0.139 \pm 0.000$ |
| Transformer | None | DMoN | N/A | None | Loss | $0.000 \pm 0.000$ | $\mathbf{0.139 \pm 0.000}$ |
| Transformer | None | L2 | N/A | None | Loss | $0.000 \pm 0.000$ | $0.139 \pm 0.001$ |
| Transformer | DMoN | None | N/A | None | Loss | $0.000 \pm 0.000$ | $0.139 \pm 0.001$ |
| Transformer | DMoN | DMoN | N/A | None | Loss | $0.000 \pm 0.000$ | $0.139 \pm 0.001$ |
| Transformer | DMoN | L2 | N/A | None | Loss | $0.000 \pm 0.000$ | $0.139 \pm 0.000$ |
| Transformer | NOCD | None | N/A | None | Loss | $0.000 \pm 0.000$ | $0.117 \pm 0.040$ |
| Transformer | NOCD | DMoN | N/A | None | Loss | $0.000 \pm 0.000$ | $0.112 \pm 0.040$ |
| Transformer | NOCD | L2 | N/A | None | Loss | $0.000 \pm 0.000$ | $0.139 \pm 0.000$ |
| Transformer | Neuromap | None | N/A | None | Loss | $0.000 \pm 0.000$ | $0.108 \pm 0.046$ |
| Transformer | Neuromap | DMoN | N/A | None | Loss | $0.000 \pm 0.000$ | $0.095 \pm 0.031$ |
| Transformer | Neuromap | L2 | N/A | None | Loss | $0.000 \pm 0.000$ | $0.128 \pm 0.027$ |
| Transformer | SBM$_{\text{NN}}$ | None | N/A | None | Loss | $0.000 \pm 0.000$ | $0.116 \pm 0.031$ |
| Transformer | SBM$_{\text{NN}}$ | DMoN | N/A | None | Loss | $0.000 \pm 0.000$ | $0.101 \pm 0.043$ |
| Transformer | SBM$_{\text{NN}}$ | L2 | N/A | None | Loss | $0.000 \pm 0.000$ | $0.139 \pm 0.001$ |
| MLP | None | None | N/A | None | Loss | $0.040 \pm 0.086$ | $0.166 \pm 0.060$ |
| MLP | None | DMoN | N/A | None | Loss | $0.109 \pm 0.098$ | $0.196 \pm 0.058$ |
| MLP | None | L2 | N/A | None | Loss | $0.000 \pm 0.000$ | $0.140 \pm 0.000$ |
| MLP | DMoN | None | N/A | None | Loss | $\mathbf{0.112 \pm 0.081}$ | $\mathbf{0.202 \pm 0.049}$ |
| MLP | DMoN | DMoN | N/A | None | Loss | $0.071 \pm 0.092$ | $0.186 \pm 0.061$ |
| MLP | DMoN | L2 | N/A | None | Loss | $0.000 \pm 0.000$ | $0.140 \pm 0.000$ |
| MLP | NOCD | None | N/A | None | Loss | $0.000 \pm 0.000$ | $0.120 \pm 0.041$ |
| MLP | NOCD | DMoN | N/A | None | Loss | $0.000 \pm 0.000$ | $0.123 \pm 0.031$ |
| MLP | NOCD | L2 | N/A | None | Loss | $0.000 \pm 0.000$ | $0.140 \pm 0.000$ |
| MLP | Neuromap | None | N/A | None | Loss | $0.000 \pm 0.000$ | $0.098 \pm 0.046$ |
| MLP | Neuromap | DMoN | N/A | None | Loss | $0.000 \pm 0.000$ | $0.071 \pm 0.046$ |
| MLP | Neuromap | L2 | N/A | None | Loss | $0.000 \pm 0.000$ | $0.128 \pm 0.026$ |
| MLP | SBM$_{\text{NN}}$ | None | N/A | None | Loss | $0.018 \pm 0.057$ | $0.109 \pm 0.060$ |
| MLP | SBM$_{\text{NN}}$ | DMoN | N/A | None | Loss | $0.000 \pm 0.000$ | $0.114 \pm 0.040$ |
| MLP | SBM$_{\text{NN}}$ | L2 | N/A | None | Loss | $0.000 \pm 0.000$ | $0.129 \pm 0.031$ |

Table 103: Comparing graph clustering and regularization objectives with an ablation study of regularization. The ablation compares clustering with clusteirng with regularization, clustering without regularization, and regularization without clustering for each neural network. Model selection based on training set MCC.

| Model | $f$ | $L_{\text{regularization}}$ | $L_V$ | NN$_{\boldsymbol{S} \rightarrow \boldsymbol{X}}$ | ES | MCC | Accuracy |
|---|---|---|---|---|---|---|---|
| GCN | None | None | ✗ | None | Loss | $0.089 \pm 0.003$ | $0.195 \pm 0.005$ |
| GCN | None | DMoN | ✗ | None | Loss | $\mathbf{0.090 \pm 0.005}$ | $0.196 \pm 0.006$ |
| GCN | None | L2 | ✗ | None | Loss | $0.000 \pm 0.000$ | $0.140 \pm 0.000$ |
| GCN | DMoN | None | ✓ | MLP | Loss | $0.089 \pm 0.005$ | $\mathbf{0.198 \pm 0.004}$ |
| GCN | DMoN | DMoN | ✓ | MLP | Loss | $0.089 \pm 0.003$ | $0.198 \pm 0.002$ |
| GCN | DMoN | L2 | ✓ | Transformer | MCC | $0.033 \pm 0.074$ | $0.149 \pm 0.047$ |
| GCN | NOCD | None | ✗ | None | Loss | $0.057 \pm 0.019$ | $0.166 \pm 0.017$ |
| GCN | NOCD | DMoN | ✓ | Transformer | Loss | $0.050 \pm 0.038$ | $0.164 \pm 0.026$ |
| GCN | NOCD | L2 | ✓ | Transformer | MCC | $0.002 \pm 0.046$ | $0.094 \pm 0.070$ |
| GCN | Neuromap | None | ✗ | None | MCC | $0.018 \pm 0.034$ | $0.066 \pm 0.086$ |
| GCN | Neuromap | DMoN | ✓ | Transformer | MCC | $0.015 \pm 0.033$ | $0.076 \pm 0.063$ |
| GCN | Neuromap | L2 | ✓ | MLP | MCC | $0.022 \pm 0.066$ | $0.123 \pm 0.040$ |
| GCN | SBM$_{\text{NN}}$ | None | ✓ | MLP | Loss | $0.069 \pm 0.023$ | $0.178 \pm 0.015$ |
| GCN | SBM$_{\text{NN}}$ | DMoN | ✓ | MLP | Loss | $0.072 \pm 0.035$ | $0.183 \pm 0.028$ |
| GCN | SBM$_{\text{NN}}$ | L2 | ✓ | MLP | MCC | $0.030 \pm 0.045$ | $0.131 \pm 0.058$ |
| GraphSAGE | None | None | ✗ | None | Loss | $0.141 \pm 0.183$ | $0.247 \pm 0.139$ |
| GraphSAGE | None | DMoN | ✗ | None | Loss | $0.214 \pm 0.184$ | $0.301 \pm 0.139$ |
| GraphSAGE | None | L2 | ✗ | None | MCC | $0.001 \pm 0.005$ | $0.023 \pm 0.045$ |
| GraphSAGE | DMoN | None | ✓ | MLP | Loss | $0.324 \pm 0.035$ | $0.385 \pm 0.026$ |
| GraphSAGE | DMoN | DMoN | ✓ | MLP | Loss | $\mathbf{0.327 \pm 0.019}$ | $\mathbf{0.388 \pm 0.014}$ |
| GraphSAGE | DMoN | L2 | ✗ | None | Loss | $0.000 \pm 0.000$ | $0.139 \pm 0.000$ |
| GraphSAGE | NOCD | None | ✓ | MLP | Loss | $0.272 \pm 0.077$ | $0.316 \pm 0.073$ |
| GraphSAGE | NOCD | DMoN | ✓ | MLP | Loss | $0.280 \pm 0.050$ | $0.312 \pm 0.065$ |
| GraphSAGE | NOCD | L2 | ✗ | None | MCC | $0.001 \pm 0.002$ | $0.011 \pm 0.035$ |
| GraphSAGE | Neuromap | None | ✗ | None | Loss | $0.000 \pm 0.000$ | $0.137 \pm 0.009$ |
| GraphSAGE | Neuromap | DMoN | ✓ | MLP | MCC | $0.000 \pm 0.001$ | $0.011 \pm 0.035$ |
| GraphSAGE | Neuromap | L2 | ✓ | Transformer | MCC | $0.004 \pm 0.019$ | $0.026 \pm 0.044$ |
| GraphSAGE | SBM$_{\text{NN}}$ | None | ✗ | None | Loss | $0.183 \pm 0.101$ | $0.256 \pm 0.066$ |
| GraphSAGE | SBM$_{\text{NN}}$ | DMoN | ✗ | None | Loss | $0.183 \pm 0.130$ | $0.269 \pm 0.093$ |
| GraphSAGE | SBM$_{\text{NN}}$ | L2 | ✓ | Transformer | MCC | $0.000 \pm 0.001$ | $0.003 \pm 0.011$ |
| Transformer | None | None | N/A | None | Loss | $0.000 \pm 0.000$ | $0.139 \pm 0.000$ |
| Transformer | None | DMoN | N/A | None | Loss | $0.000 \pm 0.000$ | $\mathbf{0.139 \pm 0.000}$ |
| Transformer | None | L2 | N/A | None | Loss | $0.000 \pm 0.000$ | $0.139 \pm 0.001$ |
| Transformer | DMoN | None | N/A | None | Loss | $0.000 \pm 0.000$ | $0.139 \pm 0.001$ |
| Transformer | DMoN | DMoN | N/A | None | MCC | $\mathbf{0.004 \pm 0.014}$ | $0.017 \pm 0.053$ |
| Transformer | DMoN | L2 | N/A | None | MCC | $0.001 \pm 0.002$ | $0.028 \pm 0.059$ |
| Transformer | NOCD | None | N/A | None | Loss | $0.000 \pm 0.000$ | $0.117 \pm 0.040$ |
| Transformer | NOCD | DMoN | N/A | None | MCC | $0.004 \pm 0.012$ | $0.013 \pm 0.041$ |
| Transformer | NOCD | L2 | N/A | None | MCC | $0.000 \pm 0.000$ | $0.025 \pm 0.053$ |
| Transformer | Neuromap | None | N/A | None | Loss | $0.000 \pm 0.000$ | $0.108 \pm 0.046$ |
| Transformer | Neuromap | DMoN | N/A | None | Loss | $0.000 \pm 0.000$ | $0.095 \pm 0.031$ |
| Transformer | Neuromap | L2 | N/A | None | Loss | $0.000 \pm 0.000$ | $0.128 \pm 0.027$ |

Table 103: Comparing graph clustering and regularization objectives for each neural network.

| Model | $f$ | $L_{\text{regularization}}$ | $L_V$ | $\text{NN}_{S \rightarrow X}$ | ES | MCC | Accuracy |
|---|---|---|---|---|---|---|---|
| Transformer | $\text{SBM}_{\text{NN}}$ | None | N/A | None | Loss | $0.000 \pm 0.000$ | $0.116 \pm 0.031$ |
| Transformer | $\text{SBM}_{\text{NN}}$ | DMoN | N/A | None | Loss | $0.000 \pm 0.000$ | $0.101 \pm 0.043$ |
| Transformer | $\text{SBM}_{\text{NN}}$ | L2 | N/A | None | Loss | $0.000 \pm 0.000$ | $0.139 \pm 0.001$ |
| MLP | None | None | N/A | None | MCC | $0.207 \pm 0.004$ | $0.233 \pm 0.011$ |
| MLP | None | DMoN | N/A | None | MCC | $0.222 \pm 0.034$ | $0.240 \pm 0.035$ |
| MLP | None | L2 | N/A | None | Loss | $0.000 \pm 0.000$ | $0.140 \pm 0.000$ |
| MLP | DMoN | None | N/A | None | MCC | $0.225 \pm 0.039$ | $0.251 \pm 0.039$ |
| MLP | DMoN | DMoN | N/A | None | MCC | $\mathbf{0.234 \pm 0.043}$ | $\mathbf{0.261 \pm 0.041}$ |
| MLP | DMoN | L2 | N/A | None | Loss | $0.000 \pm 0.000$ | $0.140 \pm 0.000$ |
| MLP | NOCD | None | N/A | None | MCC | $0.039 \pm 0.082$ | $0.046 \pm 0.096$ |
| MLP | NOCD | DMoN | N/A | None | MCC | $0.042 \pm 0.088$ | $0.062 \pm 0.103$ |
| MLP | NOCD | L2 | N/A | None | MCC | $0.000 \pm 0.001$ | $0.025 \pm 0.053$ |
| MLP | Neuromap | None | N/A | None | MCC | $0.001 \pm 0.002$ | $0.020 \pm 0.042$ |
| MLP | Neuromap | DMoN | N/A | None | Loss | $0.000 \pm 0.000$ | $0.071 \pm 0.046$ |
| MLP | Neuromap | L2 | N/A | None | MCC | $0.005 \pm 0.012$ | $0.039 \pm 0.041$ |
| MLP | $\text{SBM}_{\text{NN}}$ | None | N/A | None | Loss | $0.018 \pm 0.057$ | $0.109 \pm 0.060$ |
| MLP | $\text{SBM}_{\text{NN}}$ | DMoN | N/A | None | MCC | $0.021 \pm 0.064$ | $0.035 \pm 0.077$ |
| MLP | $\text{SBM}_{\text{NN}}$ | L2 | N/A | None | Loss | $0.000 \pm 0.000$ | $0.129 \pm 0.031$ |

Table 104: Comparing graph clustering and regularization objectives with an ablation study of regularization. The ablation compares clustering with clusteirng with regularization, clustering without regularization, and regularization without clustering for each neural network. Model selection based on validation set MCC.

| Model | $f$ | $L_{\text{regularization}}$ | $L_V$ | $\text{NN}_{S \rightarrow X}$ | ES | MCC | Accuracy |
|---|---|---|---|---|---|---|---|
| GCN | None | None | ✗ | None | Loss | $0.089 \pm 0.003$ | $0.195 \pm 0.005$ |
| GCN | None | DMoN | ✗ | None | Loss | $\mathbf{0.090 \pm 0.005}$ | $\mathbf{0.196 \pm 0.006}$ |
| GCN | None | L2 | ✗ | None | Loss | $0.000 \pm 0.000$ | $0.140 \pm 0.000$ |
| GCN | DMoN | None | ✗ | None | Loss | $0.090 \pm 0.005$ | $0.195 \pm 0.005$ |
| GCN | DMoN | DMoN | ✗ | None | Loss | $0.090 \pm 0.006$ | $0.195 \pm 0.006$ |
| GCN | DMoN | L2 | ✓ | Transformer | MCC | $0.033 \pm 0.074$ | $0.149 \pm 0.047$ |
| GCN | NOCD | None | ✗ | None | Loss | $0.057 \pm 0.019$ | $0.166 \pm 0.017$ |
| GCN | NOCD | DMoN | ✗ | None | Loss | $0.049 \pm 0.025$ | $0.161 \pm 0.018$ |
| GCN | NOCD | L2 | ✗ | None | MCC | $0.003 \pm 0.035$ | $0.094 \pm 0.068$ |
| GCN | Neuromap | None | ✗ | None | MCC | $0.018 \pm 0.034$ | $0.066 \pm 0.086$ |
| GCN | Neuromap | DMoN | ✓ | Transformer | MCC | $0.015 \pm 0.033$ | $0.076 \pm 0.063$ |
| GCN | Neuromap | L2 | ✓ | MLP | MCC | $0.022 \pm 0.066$ | $0.123 \pm 0.040$ |
| GCN | $\text{SBM}_{\text{NN}}$ | None | ✓ | MLP | Loss | $0.069 \pm 0.023$ | $0.178 \pm 0.015$ |
| GCN | $\text{SBM}_{\text{NN}}$ | DMoN | ✓ | MLP | Loss | $0.072 \pm 0.035$ | $0.183 \pm 0.028$ |
| GCN | $\text{SBM}_{\text{NN}}$ | L2 | ✓ | MLP | MCC | $0.030 \pm 0.045$ | $0.131 \pm 0.058$ |
| GraphSAGE | None | None | ✗ | None | Loss | $0.141 \pm 0.183$ | $0.247 \pm 0.139$ |
| GraphSAGE | None | DMoN | ✗ | None | Loss | $0.214 \pm 0.184$ | $0.301 \pm 0.139$ |
| GraphSAGE | None | L2 | ✗ | None | MCC | $0.001 \pm 0.005$ | $0.023 \pm 0.045$ |
| GraphSAGE | DMoN | None | ✓ | MLP | Loss | $0.324 \pm 0.035$ | $0.385 \pm 0.026$ |
| GraphSAGE | DMoN | DMoN | ✓ | MLP | Loss | $\mathbf{0.327 \pm 0.019}$ | $\mathbf{0.388 \pm 0.014}$ |
| GraphSAGE | DMoN | L2 | ✗ | None | Loss | $0.000 \pm 0.000$ | $0.139 \pm 0.000$ |

Table 104: Comparing graph clustering and regularization objectives for each neural network.

| Model | $f$ | $L_{\text{regularization}}$ | $L_V$ | $NN_{\boldsymbol{S} \rightarrow \boldsymbol{X}}$ | ES | MCC | Accuracy |
|---|---|---|---|---|---|---|---|
| GraphSAGE | NOCD | None | ✓ | MLP | Loss | $0.272 \pm 0.077$ | $0.316 \pm 0.073$ |
| GraphSAGE | NOCD | DMoN | ✓ | MLP | Loss | $0.280 \pm 0.050$ | $0.312 \pm 0.065$ |
| GraphSAGE | NOCD | L2 | ✗ | None | MCC | $0.001 \pm 0.002$ | $0.011 \pm 0.035$ |
| GraphSAGE | Neuromap | None | ✓ | MLP | MCC | $0.000 \pm 0.001$ | $0.012 \pm 0.034$ |
| GraphSAGE | Neuromap | DMoN | ✓ | MLP | MCC | $0.000 \pm 0.001$ | $0.011 \pm 0.035$ |
| GraphSAGE | Neuromap | L2 | ✓ | Transformer | MCC | $0.004 \pm 0.019$ | $0.026 \pm 0.044$ |
| GraphSAGE | $SBM_{NN}$ | None | ✗ | None | Loss | $0.183 \pm 0.101$ | $0.256 \pm 0.066$ |
| GraphSAGE | $SBM_{NN}$ | DMoN | ✗ | None | Loss | $0.183 \pm 0.130$ | $0.269 \pm 0.093$ |
| GraphSAGE | $SBM_{NN}$ | L2 | ✓ | Transformer | MCC | $0.000 \pm 0.001$ | $0.003 \pm 0.011$ |
| Transformer | None | None | N/A | None | Loss | $0.000 \pm 0.000$ | $0.139 \pm 0.000$ |
| Transformer | None | DMoN | N/A | None | Loss | $0.000 \pm 0.000$ | $\mathbf{0.139 \pm 0.000}$ |
| Transformer | None | L2 | N/A | None | MCC | $-0.000 \pm 0.026$ | $0.022 \pm 0.050$ |
| Transformer | DMoN | None | N/A | None | Loss | $0.000 \pm 0.000$ | $0.139 \pm 0.001$ |
| Transformer | DMoN | DMoN | N/A | None | MCC | $\mathbf{0.004 \pm 0.014}$ | $0.017 \pm 0.053$ |
| Transformer | DMoN | L2 | N/A | None | MCC | $0.001 \pm 0.002$ | $0.028 \pm 0.059$ |
| Transformer | NOCD | None | N/A | None | Loss | $0.000 \pm 0.000$ | $0.117 \pm 0.040$ |
| Transformer | NOCD | DMoN | N/A | None | MCC | $0.004 \pm 0.012$ | $0.013 \pm 0.041$ |
| Transformer | NOCD | L2 | N/A | None | MCC | $0.000 \pm 0.000$ | $0.025 \pm 0.053$ |
| Transformer | Neuromap | None | N/A | None | Loss | $0.000 \pm 0.000$ | $0.108 \pm 0.046$ |
| Transformer | Neuromap | DMoN | N/A | None | Loss | $0.000 \pm 0.000$ | $0.095 \pm 0.031$ |
| Transformer | Neuromap | L2 | N/A | None | Loss | $0.000 \pm 0.000$ | $0.128 \pm 0.027$ |
| Transformer | $SBM_{NN}$ | None | N/A | None | Loss | $0.000 \pm 0.000$ | $0.116 \pm 0.031$ |
| Transformer | $SBM_{NN}$ | DMoN | N/A | None | Loss | $0.000 \pm 0.000$ | $0.101 \pm 0.043$ |
| Transformer | $SBM_{NN}$ | L2 | N/A | None | Loss | $0.000 \pm 0.000$ | $0.139 \pm 0.001$ |
| MLP | None | None | N/A | None | MCC | $0.207 \pm 0.004$ | $0.233 \pm 0.011$ |
| MLP | None | DMoN | N/A | None | MCC | $0.222 \pm 0.034$ | $0.240 \pm 0.035$ |
| MLP | None | L2 | N/A | None | MCC | $-0.001 \pm 0.015$ | $0.050 \pm 0.063$ |
| MLP | DMoN | None | N/A | None | MCC | $0.225 \pm 0.039$ | $0.251 \pm 0.039$ |
| MLP | DMoN | DMoN | N/A | None | MCC | $\mathbf{0.234 \pm 0.043}$ | $\mathbf{0.261 \pm 0.041}$ |
| MLP | DMoN | L2 | N/A | None | Loss | $0.000 \pm 0.000$ | $0.140 \pm 0.000$ |
| MLP | NOCD | None | N/A | None | MCC | $0.039 \pm 0.082$ | $0.046 \pm 0.096$ |
| MLP | NOCD | DMoN | N/A | None | MCC | $0.042 \pm 0.088$ | $0.062 \pm 0.103$ |
| MLP | NOCD | L2 | N/A | None | Loss | $0.000 \pm 0.000$ | $0.140 \pm 0.000$ |
| MLP | Neuromap | None | N/A | None | MCC | $0.001 \pm 0.002$ | $0.020 \pm 0.042$ |
| MLP | Neuromap | DMoN | N/A | None | Loss | $0.000 \pm 0.000$ | $0.071 \pm 0.046$ |
| MLP | Neuromap | L2 | N/A | None | MCC | $0.005 \pm 0.012$ | $0.039 \pm 0.041$ |
| MLP | $SBM_{NN}$ | None | N/A | None | Loss | $0.018 \pm 0.057$ | $0.109 \pm 0.060$ |
| MLP | $SBM_{NN}$ | DMoN | N/A | None | MCC | $0.021 \pm 0.064$ | $0.035 \pm 0.077$ |
| MLP | $SBM_{NN}$ | L2 | N/A | None | Loss | $0.000 \pm 0.000$ | $0.129 \pm 0.031$ |

