# OpenReview forum: "A Semi-Supervised Clustering Approach For Graph Learning with Neural Networks"
_ICLR.cc/2025/Conference — Submitted to ICLR 2025_

### Official Review · Reviewer_U6JD · 2024-10-16

**Soundness:** 2
**Presentation:** 3
**Contribution:** 2
**Rating:** 5
**Confidence:** 4

**Summary:**

The paper introduces semi-supervised generative models to combine unsupervised clustering and supervised classification in learning on graphs with vertex attributes.

The authors use Bayesian networks to map out how variables are dependent in current generative model designs. By strategically adding new connections between variables, they suggest improvements to these existing models.

Experimental results on three real-world graph datasets demonstrate that semi-supervised objectives consistently improve performance over purely supervised training.

**Strengths:**

1. The idea of using Bayesian networks to analyse and enhance generative models is novel in this specific context of clustering and classification on graphs.
2. Multiple generative models such as stochastic block models and graph neural models are unified into a single framework.
3. The paper is clearly structured, with a logical progression from background and motivation to the proposed methodology. The use of Bayesian network diagrams adds clarity, helping readers understand the conditional dependencies in the different generative models.

**Weaknesses:**

1. The generative models proposed involve learning multiple dependencies among variables. This complexity presents scalability challenges for very large graphs [1, 2]. Including computational complexity analysis and scalability results, e.g. memory used and training time consumed, would add significance to the contributions.
2. The paper primarily evaluates the approach on a limited number of datasets (Cora, Citeseer, Pubmed). A more detailed discussion on generalisability to other graph types (e.g. heterophilic datasets [3]) would strengthen the contributions.
3. The paper lacks a thorough comparison with state-of-the-art methods in graph clustering beyond the few baseline architectures considered. The methods proposed should be positioned with existing methods in the expanding subfield of deep attributed graph clustering [4, 5, 6].
4. The attribute reconstruction might improve learning but this aspect is only briefly evaluated. A more thorough exploration of the benefits of attribute reconstruction, along with ablation studies to demonstrate its impact, would provide greater insight into the conditions under which this component is most effective.

References:
1. Open Graph Benchmark: Datasets for Machine Learning on Graphs, In NeurIPS 2020,
2. Large Scale Learning on Non-Homophilous Graphs: New Benchmarks and Strong Simple Methods, In NeurIPS 2021,
3. A critical look at the evaluation of GNNs under heterophily: Are we really making progress?, In ICLR 2023,
4. An Overview of Advanced Deep Graph Node Clustering, In IEEE Transactions on Computational Social Systems 2024,
5. A Survey of Deep Graph Clustering: Taxonomy, Challenge, Application, and Open Resource, arXiv:2211.12875, 2022,
6. A survey on semi-supervised graph clustering, Eng. Appl. Artif. Intell. 2024.

**Questions:**

1. Was there a detailed analysis of the computational complexity and scalability of their proposed generative models, particularly for very large graphs?
2. How did the proposed models perform on heterophilic datasets or other types of graphs beyond the citation networks used in the current evaluation?
3. What advantages do the proposed models have compared to recent state-of-the-art approaches in deep attributed graph clustering?
4. How sensitive were the proposed methods to hyperparameter choices, e.g. hidden dimension, learning rate?
5. In equation 14 on line 306, GNN depends only on A. It does not depend on X. What are the node features $X$ used in the GNN? Were they randomly initialised?

---

> ### Author Response · Authors · 2024-11-29
>
> Thank you very much for your thoughtful and constructive review.
>
> We greatly appreciate the positive feedback and are encouraged that you found our framework novel and clearly presented. Your detailed comments and suggestions have been invaluable in helping us refine our work. Below, we address the weaknesses and questions you raised and outline the corresponding clarifications and modifications made in the paper.
>
> ### **Weakness 1:**
> Thank you for highlighting the importance of discussing scalability. We have added a dedicated section on scalability, which elaborates on how our method scales depending on the choice of neural network modules and graph clustering methods. While our work focuses on the full-batch setting, we also discuss potential extensions to mini-batch graph learning methods in future work.
>
> ### **Weakness 2:**
> Thank you very much for pointing us to [1]. In response, we have significantly revised our experiments to include more diverse and up-to-date datasets from [1] and [2].
>
> ### **Weakness 3:**
> We appreciate your suggestion to reference the survey and review papers [3, 4, 5]. Our work does not aim to propose a new state-of-the-art graph clustering method but rather to provide a framework that allows the use of _any_ end-to-end graph clustering method with graph neural networks. We have updated the manuscript to cite these papers, positioning our work more clearly in the context of existing research. Notably, [5] highlights the promising direction of integrating graph neural networks for semi-supervised clustering, which aligns with our contributions.
>
> ### **Weakness 4:**
> Thank you for your comments on the experimental clarity. We have reorganized the experimental results into four main investigations and revised the tables for better readability. Additionally, we added a dedicated section focusing on the ablation of attribute reconstruction.
>
> ### **Question 1:**
> The scalability of our method depends on the neural network modules and graph clustering methods used. For instance, if GCN (a graph neural network) and NOCD (a graph clustering method) are chosen, which scale linearly with the number of edges, our method will also scale linearly. While we focus on the full-batch setting, we acknowledge that memory complexity could be a concern for very large graphs. We propose this as future work and discuss this in the newly added scalability section.
>
> ### **Question 2:**
> We have revised the experiments to include newer, more diverse datasets from [1] and [2].
>
> ### **Question 3:**
> The key advantage of our framework is its flexibility, enabling the use of any end-to-end graph clustering method with any neural network. As new methods are proposed, our framework can seamlessly integrate them.
>
> ### **Question 4:**
> We did not tune hyperparameters, as clarified in the additional experimental details section.
>
> ### **Question 5:**
> Thank you for raising this point. Any positional encoding of \(A\) can be used as a node feature input to the GNN. However, since we did not empirically evaluate this, we have removed this architecture from the revised paper for clarity.
>
> ---
>
> Once again, we sincerely thank you for your valuable feedback. We hope the revisions address your concerns and further clarify the contributions and significance of our work. We look forward to any further feedback you may have.
>
> ---
>
> [1] Platonov, Oleg, Denis Kuznedelev, Michael Diskin, Artem Babenko, and Liudmila Prokhorenkova. "A critical look at the evaluation of GNNs under heterophily: Are we really making progress?." arXiv preprint arXiv:2302.11640 (2023).
>
> [2] Shchur, Oleksandr, Maximilian Mumme, Aleksandar Bojchevski, and Stephan Günnemann. "Pitfalls of graph neural network evaluation." arXiv preprint arXiv:1811.05868 (2018).
>
> [3] Wang, Shiping, Jinbin Yang, Jie Yao, Yang Bai, and William Zhu. "An overview of advanced deep graph node clustering." IEEE Transactions on Computational Social Systems 11, no. 1 (2023): 1302-1314.
>
> [4] Liu, Yue, Jun Xia, Sihang Zhou, Xihong Yang, Ke Liang, Chenchen Fan, Yan Zhuang, Stan Z. Li, Xinwang Liu, and Kunlun He. "A Survey of Deep Graph Clustering: Taxonomy, Challenge, Application, and Open Resource." arXiv preprint arXiv:2211.12875 (2022).
>
> [5] Daneshfar, Fatemeh, Sayvan Soleymanbaigi, Pedram Yamini, and Mohammad Sadra Amini. "A survey on semi-supervised graph clustering." Engineering Applications of Artificial Intelligence 133 (2024): 108215.

---

### Official Review · Reviewer_W1Wx · 2024-11-03

**Soundness:** 2
**Presentation:** 1
**Contribution:** 2
**Rating:** 5
**Confidence:** 4

**Summary:**

The paper proposes a semi-supervised framework for node classification on graphs using neural networks, specifically focusing on transformers and MLPs without the need for encoding. It unifies various clustering and classification objectives under a common perspective, showing how graph neural networks implicitly incorporate both node attributes and structure. Through this approach, the paper demonstrates that semi-supervised objectives enable models to perform effectively, outperforming purely supervised training results.

**Strengths:**

1) The code is shared for reproducibility.
2) Detailed background information is provided for readers who may not be very familiar with the field; however, this may not be ideal for research-focused papers.

**Weaknesses:**

1) The paper’s structure is poorly organized, and the presentation is quite weak.
2) The three pages dedicated to Background and Related Work are excessive. Generally, this section should be shorter in research papers (unless it’s a review), with most of it moved to an appendix.
3) The methodology section reads too much like related work, which diminishes the paper’s contributions and makes them unclear.
4) The presentation of results is poor. Overall, Table 1 could be split into multiple tables to more clearly show the effect of each component.
5) As it stands, the paper reads more like a review paper than a research-focused paper. With that, it should include more recent baselines from top AI/ML conferences addressing graph clustering with neural networks.
6) The paper has very few evaluation metrics and datasets.

**Questions:**

With the current state of the paper, I'm more inclining of rejecting, however, I'm open for discussion. If my concerns (weaknesses and questions) are addressed, I can increase my score.

1) Can you clarify how the structure of the paper supports its contributions? Could you reorganize the paper to improve readability and presentation? Some suggestions include:
- Reducing background information, with most parts moved to the appendix.
- Making the methodology section clearer, focusing on and highlighting the main contributions.
- Improving the presentation of results, such as positioning Table 1 better and splitting it into smaller sections for clarity.
- Making equations easier to read (e.g., Lines 270-278).

2) Could you incorporate more baselines, evaluation metrics (such as modularity, graph conductance), and datasets?

---

> ### Author Response · Authors · 2024-11-29
>
> Thank you very much for your detailed and constructive feedback. We greatly appreciate the time and effort you put into reviewing our paper. Your insights have been invaluable in improving the clarity, organization, and overall quality of our work. Below, we address the specific weaknesses and questions you raised and outline the corresponding changes we have made to the manuscript:
>
> ---
>
> ### **Weaknesses 1, 2, 3, and 4**:
>
> We appreciate your comments on the presentation of the paper and the helpful suggestions for improvement. As you noted, we had structured the paper to provide an accessible introduction to the problem and our proposed method, which was also acknowledged positively by reviewers i9As and U6JD. However, we recognize that there was room to clarify our contributions and the proposed method while maintaining accessibility.
>
> To address this, we have made the following changes:
>
> - **Introduction and Methodology Sections**: We have significantly revised these sections to clearly enumerate the contributions and provide a more precise description of the proposed method.
> - **Experiments Section**: The experiments and tables have been reorganized to align with the four main investigations we conduct. We also provide more detailed descriptions of the experiments and results in the main text to enhance clarity.
>
> ---
>
> ### **Weakness 5**:
>
> While our paper does not aim to contribute a review of the field, we have strived to ensure that our literature review is comprehensive. Our primary goal is not to outperform existing clustering methods but to propose a framework that leverages end-to-end graph clustering methods to enhance classification. Our experiments benchmark recent methods including DMoN (2023) [1] and Neuromap (2023) [2].
>
> ---
>
> ### **Weakness 6**:
>
> Thank you for your feedback regarding the datasets. In response, we have significantly revised our experiments to include additional datasets, particularly newer ones from [1] and [2]. This expansion strengthens the generalizability and relevance of our findings.
>
> ---
>
> ### **Question 1**:
>
> Thank you for your question regarding our choice of presentation and your helpful suggestions for improvement. As mentioned earlier, we had structured the paper to balance accessibility and depth, but we recognize the importance of clearly presenting our contributions and methodology.
>
> In response:
>
> - **Background Information**: Most of the background information has been moved to the appendix to declutter the main text.
> - **Introduction and Methods Sections**: We have revised these sections to highlight our contributions and proposed method more prominently.
> - **Equations**: The equations in the methods section have been reorganized for improved readability.
> - **Experiments**: These have been restructured around the four main investigations, with more detailed descriptions added.

---

> ### Author Response · Authors · 2024-11-29
>
> ### **Question 2**:
>
> Thank you for raising this question! In response, we have included additional datasets from [3] and [4], and we now use the Matthews correlation coefficient (MCC) as an additional evaluation metric. MCC provides both empirical and theoretical advantages, as discussed in the paper, and complements accuracy, which remains the standard metric in multi-class node classification benchmarks [3, 4, 5].
>
> We have also considered your suggestions about evaluation metrics:
>
> - **Unsupervised Clustering Metrics**: Metrics like modularity and graph conductance were not included because they are tailored for unattributed graphs and the unsupervised setting.
> - **Baselines**: We include three unsupervised graph clustering baselines, two of which are from 2023, in addition to a method we propose. These baselines showcase how our framework leverages existing methods to improve classification rather than directly outperforming them.
>
> Finally, we demonstrate the flexibility of our framework by incorporating both standard GNN architectures [5] and alternative models such as MLPs and transformers, showing how our approach enables these models to learn on graphs without positional encodings or architectural modifications.
>
> ---
>
> We hope these revisions address your concerns and enhance the paper’s overall quality. Thank you once again for your thoughtful and constructive feedback. We believe that your comments have significantly strengthened our work, and we look forward to any further feedback you may have.
>
> ---
>
> [1] Tsitsulin, Anton, John Palowitch, Bryan Perozzi, and Emmanuel Müller. "Graph clustering with graph neural networks." Journal of Machine Learning Research 24, no. 127 (2023): 1-21.
>
> [2] Blöcker, Christopher, Chester Tan, Ingo Scholtes, and D. E. Julius-Maximilians-Universiät Würzburg. "The Map Equation Goes Neural: Mapping Network Flows with Graph Neural Networks." arXiv preprint arXiv:2310.01144 (2023).
>
> [3] Shchur, Oleksandr, Maximilian Mumme, Aleksandar Bojchevski, and Stephan Günnemann. "Pitfalls of graph neural network evaluation." arXiv preprint arXiv:1811.05868 (2018).
>
> [4] Platonov, Oleg, Denis Kuznedelev, Michael Diskin, Artem Babenko, and Liudmila Prokhorenkova. "A critical look at the evaluation of GNNs under heterophily: Are we really making progress?." arXiv preprint arXiv:2302.11640 (2023).
>
> [5] Hu, Weihua, Matthias Fey, Marinka Zitnik, Yuxiao Dong, Hongyu Ren, Bowen Liu, Michele Catasta, and Jure Leskovec. "Open graph benchmark: Datasets for machine learning on graphs." Advances in neural information processing systems 33 (2020): 22118-22133.

---

> > ### Comment · Reviewer_W1Wx · 2024-12-01
> > **Thanks for your answer and the effort.**
> >
> > Thank you very much for your detailed response and for addressing my questions and concerns.
> >
> > Here are my responses to your comments:
> >
> > 1) The contribution of the paper to research at ICLR is quite limited. This is why I mentioned that the paper feels more like a review paper, as it includes numerous experiments but offers limited novelty.
> > 2) Unsupervised clustering metrics are essential for evaluating clustering methods, even if the method itself is not unsupervised, because the clustering algorithm's nature is inherently unsupervised. If you wish to focus solely on supervised metrics, I would suggest renaming the algorithm to "node classification" instead of "clustering."
> > 3) As we are close to the review period, I will not request the inclusion of new baselines. However, for future work, you may consider incorporating more recent (2024) baselines, such as [1], into your paper.
> >
> > I appreciate your efforts in addressing my concerns and incorporating structural changes along with additional experimental results. However, I still feel that the paper requires greater novelty and the inclusion of unsupervised metrics for future submissions. I am increasing my score to 5, but I remain inclined toward recommending resubmission.
> >
> > [1] Bhowmick, A., M. Kosan, Z. Huang, A. Singh, and S. Medya, "DGCLUSTER: A Neural Framework for Attributed Graph Clustering via Modularity Maximization", The 38th Annual AAAI Conference on Artificial Intelligence (AAAI'24), 2024.

---

> > > ### Author Response · Authors · 2024-12-01
> > >
> > > Thank you very much for your prompt response and for increasing your score in recognition of the clarifications we provided and the structural changes made to the paper. We appreciate your constructive feedback, which has helped us improve the manuscript significantly.
> > >
> > > We would like to address a few key points in your comments:
> > >
> > > ---
> > >
> > > **Novelty and Contribution:**
> > >
> > > While we have made efforts to enumerate our numerous contributions clearly in the paper, we would greatly appreciate further clarification on why you feel the paper does not represent a significant enough contribution. This feedback would help us better understand and address perceived limitations of our work.
> > >
> > > **Unsupervised Clustering Metrics:**
> > >
> > > We acknowledge your point about the importance of unsupervised clustering metrics. However, the primary reason we did not include such metrics is that they are traditionally designed for unattributed graphs. For example, [1] states that the unsupervised metrics such as modularity and graph conductance they use rely solely on graph structure.
> > > Given the complex interplay of graph structure, node attributes, and node labels in real-world networks, we argue that unsupervised clustering metrics based solely on graph structure does not adequately capture the performance of methods designed for attributed graphs.
> > > Currently, we are unaware of unsupervised clustering metrics specifically tailored for attributed graphs. If you are aware of such metrics, we would be happy to include them in a camera-ready version of the paper if accepted.
> > >
> > > **Incorporation of Recent Baselines:**
> > >
> > > Thank you for bringing [1] to our attention, as it offers valuable insights including an additional end-to-end graph clustering objective and supervised objective that can be readily incorporated into our proposed framework.
> > >
> > > We note that [1] was only published earlier this year. While it is certainly an important and relevant contribution, we believe the absence of baselines published within the same year does not detract from the validity of our work.
> > >
> > > Our work does not aim to outperform every end-to-end graph clustering objective like the one proposed in [1]. Instead, our focus is on proposing a framework that can seamlessly incorporate and complement existing clustering objectives, including those in [1].
> > >
> > > Specifically, while [1] introduces end-to-end modularity-based graph clustering and supervised auxiliary loss objectives, it still faces limitations regarding necessary conditional dependencies for clustering attributed graphs -- an issue our framework seeks to address.
> > > The supervised loss proposed in [1] could also be integrated into our framework in place of an explicit cross-entropy reconstruction objective.
> > >
> > > ---
> > >
> > > We thank you once again for your feedback and your constructive suggestions. Please do not hesitate to let us know if further clarifications are required.
> > >
> > > [1] Bhowmick, Aritra, Mert Kosan, Zexi Huang, Ambuj Singh, and Sourav Medya. "DGCLUSTER: A Neural Framework for Attributed Graph Clustering via Modularity Maximization." In Proceedings of the AAAI Conference on Artificial Intelligence, vol. 38, no. 10, pp. 11069-11077. 2024.

---

> > > > ### Comment · Reviewer_W1Wx · 2024-12-01
> > > > **Follow up.**
> > > >
> > > > Thank you so much for your detailed response to my comments.
> > > >
> > > > Please find my responses to them below:
> > > >
> > > > 1) Thank you for clearly presenting your contributions in the paper. Please note that I am not underestimating your contributions to the field; however, I believe the impact of these contributions is not sophisticated enough for publication in one of the most prestigious machine learning conferences, ICLR; especially considering that you mentioned you are not aiming to outperform the state of the art.
> > > >
> > > > 2) I strongly disagree that unsupervised clustering metrics can only be applied to unattributed graphs. If the output of your algorithm essentially involves clustering items or nodes, it means unsupervised clustering metrics are applicable and relevant. By definition, these metrics do not require the graphs to be unattributed. While I understand that they may only consider the graph structure, metrics such as modularity are essential in clustering. Therefore, I recommend including these metrics in your future endeavors.
> > > >
> > > > 3) My decision was not influenced by the exclusion of this paper, but I do recommend including it in your studies for future versions. According to ICLR's policy, papers published within the last four months are not required to be compared. However, [1] was published approximately seven months prior to the submission. I am including the relevant FAQ section here in case you missed it:
> > > >
> > > > "
> > > > Q: What constitutes concurrent/contemporaneous work, and what is the relevant policy regarding it?
> > > >
> > > > A: We consider papers contemporaneous if they are published within the last four months. That means, since our full paper deadline is October 1, if a paper was published (i.e., at a peer-reviewed venue) on or after July 1, 2024, authors are not required to compare their own work to that paper. Authors are encouraged to cite and discuss all relevant papers, but they may be excused for not knowing about papers not published in peer-reviewed conference proceedings or journals, which includes papers exclusively available on arXiv.
> > > > "
> > > >
> > > > [1] Bhowmick, Aritra, Mert Kosan, Zexi Huang, Ambuj Singh, and Sourav Medya. "DGCLUSTER: A Neural Framework for Attributed Graph Clustering via Modularity Maximization." In Proceedings of the AAAI Conference on Artificial Intelligence, vol. 38, no. 10, pp. 11069-11077. 2024.

---

> > > > > ### Author Response · Authors · 2024-12-01
> > > > >
> > > > > Thank you very much for your prompt yet detailed follow-up response and for taking the time to clarify your feedback further. We greatly appreciate the opportunity to address your comments and provide additional context.
> > > > >
> > > > > **On Sophistication and State-of-the-Art Comparison:**
> > > > >
> > > > > We would like to clarify our earlier statement regarding state-of-the-art methods. Our framework is designed to complement and seamlessly incorporate state-of-the-art clustering methods rather than to independently outperform them. This design choice reflects our goal of building a versatile and extensible framework that can leverage advances in clustering objectives, including those like [1]. We hope this helps contextualize our contributions and demonstrates that the novelty of our work lies in its adaptability and ability to generalize state-of-the-art approaches.
> > > > >
> > > > > **On Unsupervised Clustering Metrics:**
> > > > >
> > > > > We acknowledge your point regarding the applicability of unsupervised clustering metrics like modularity and graph conductance to attributed graphs, and we regret any misunderstanding caused by our earlier response. To clarify, we do not claim these metrics cannot be applied to attributed graphs -- they can indeed be calculated using predicted cluster assignments and graph structure.
> > > > >
> > > > > However, our concern lies with their suitability for attributed graphs with labels. For instance:
> > > > > - In heterophilous graphs, accurate clustering with respect to real labels might yield low modularity or graph conductance scores.
> > > > > - In cases where the graph structure does not correlate with labels but the attributes do, modularity and graph conductance may fail to reflect clustering quality accurately.
> > > > >
> > > > > We appreciate your recommendation to include these metrics in future studies and will consider them carefully, ensuring that their limitations in such contexts are discussed transparently.
> > > > >
> > > > > **On the Inclusion of [1]:**
> > > > >
> > > > > We are aware of ICLR's policy regarding contemporaneous work, and we deeply appreciate your reasonable approach in not basing your decision on the inclusion of [1]. Our intent was not to express reluctance in including this study -- on the contrary, we are genuinely grateful for your recommendation. [1] provides valuable insights and aligns well with the goals of our framework, and we are excited to integrate it into future revisions and a camera-ready version of our paper if accepted.
> > > > >
> > > > > Once again, thank you for your thoughtful comments and constructive feedback. We look forward to hearing your reply and addressing any further clarifications or concerns you may have.
> > > > >
> > > > > [1] Bhowmick, Aritra, Mert Kosan, Zexi Huang, Ambuj Singh, and Sourav Medya. "DGCLUSTER: A Neural Framework for Attributed Graph Clustering via Modularity Maximization." In Proceedings of the AAAI Conference on Artificial Intelligence, vol. 38, no. 10, pp. 11069-11077. 2024.

---

### Official Review · Reviewer_i9As · 2024-11-07

**Soundness:** 1
**Presentation:** 1
**Contribution:** 1
**Rating:** 3
**Confidence:** 4

**Summary:**

This paper combines the loss of supervised and unsupervised learning on graphs for semi-supervised learning. It first unifies SBM and GNN under the framework of the Bayesian network and re-implement the framework with the neural network. The evaluations are conducted on three small networks.

**Strengths:**

The combination of SBM and GNN is interesting.

**Weaknesses:**

- The organization and writing are poor. Most parts are about the background. The organization is confusing.
- The novelty is very limited. The semi-supervised framework is a simple combination of supervised and unsupervised loss. Thus, the contribution is low.
- The unification with the Bayesian network is direct without any novelty. Besides, this unification is not important for the following reimplementation.
- The technique contribution is limited. The reimplementations of the SBM with neural networks are direct. They just use NN and GNN for latent node embedding and membership matrix. Furthermore, there are many existing works on this strategy such as [1].
- The evaluations are not convincing. Only three very small networks are employed.

[1] Liang Yang, Fan Wu, Junhua Gu, Chuan Wang, Xiaochun Cao, Di Jin, Yuanfang Guo: Graph Attention Topic Modeling Network. WWW 2020: 144-154

**Questions:**

Refer to weaknesses.

---

> ### Author Response · Authors · 2024-11-29
>
> Thank you very much for your thoughtful and constructive review. We sincerely appreciate the time and effort you have dedicated to evaluating our work. We are encouraged by your acknowledgment of the potential of our framework in connecting stochastic block models (SBMs) and graph neural networks (GNNs).
>
> To address the weaknesses and questions you have raised, we have made substantial revisions to the paper and would like to outline our responses and improvements below.
>
> ---
>
> ### **Weakness 1:**
>
> Thank you for your feedback and helpful suggestions regarding the presentation of our paper.
>
> While we initially structured the paper to provide an accessible introduction to the problem and our proposed method, as acknowledged by reviewers i9As and U6JD, we understand the importance of balancing accessibility with clarity in presenting our contributions.
>
> To address this, we have significantly revised both the introduction and methodology sections to clearly enumerate the contributions and the proposed method. Additionally, the experiments section and corresponding tables have been reorganized to highlight the four main investigations in our experiments. We have also enhanced the descriptions of experiments and results in the main text for better clarity.
>
> ### **Weakness 2:**
>
> We appreciate your feedback on the novelty of our method.
>
> We are encouraged that you find our framework simple, as simplicity and flexibility were key design goals. Our framework provides a principled way to leverage any existing neural network and end-to-end graph clustering method for semi-supervised graph clustering. This includes enabling pure transformers and MLPs to learn on graphs without requiring positional encodings or architectural modifications—a feature we believe strengthens the framework.
>
> To further clarify our contributions, we have significantly revised the introduction and methodology sections to clearly articulate our novel perspectives and methods.
>
> ### **Weakness 3:**
>
> Thank you for your comments regarding Bayesian networks.
>
> We did not aim to advance the development of Bayesian networks themselves. Instead, we use Bayesian networks to visualize conditional dependencies in graph learning methods, including our framework. We believe this visualization approach makes these dependencies more accessible and provides a novel perspective for understanding and comparing graph learning methods.
>
> ### **Weakness 4:**
>
> We appreciate your observations about the simplicity of our framework.
>
> Rather than designing a complex framework with numerous additional components, we aimed to create a flexible and principled perspective that unifies and connects existing methods.
>
> We are grateful for your suggestion to include [1], which provides a fascinating perspective on interpreting graph attention networks as semi-amortized variational inference of SBMs. We have incorporated this work into our discussion of related literature.
>
> Our approach complements [1] by addressing a different problem and offering a distinct perspective to connect SBMs and (G)NNs. Specifically, we design fully neural versions of two attributed SBMs and introduce a framework that is able to combine any end-to-end unsupervised graph learning objective and any graph neural network. Notably, the NSBM we propose enables pure transformers and MLPs to operate on graphs without requiring positional encodings or architectural changes.
>
> We have revised the introduction and methodology sections to more clearly articulate these contributions.
>
> ### **Weakness 5:**
>
> Thank you for your suggestions regarding the experiments.
>
> In response, we have significantly revised the experiments section and now include more recent and larger datasets, such as those from [1] and [2]. These datasets provide a more comprehensive evaluation and align with current benchmarks in the field.
>
> ---
>
> We hope that these improvements address your concerns and further strengthen the presentation and impact of our work.
>
> We are deeply grateful for your detailed feedback, which has helped us improve the quality of our paper. Please do not hesitate to share any additional comments or suggestions.
>
> ---
>
> [1] Yang, Liang, Fan Wu, Junhua Gu, Chuan Wang, Xiaochun Cao, Di Jin, and Yuanfang Guo. "Graph attention topic modeling network." In Proceedings of the web conference 2020, pp. 144-154. 2020.
>
> [2] Shchur, Oleksandr, Maximilian Mumme, Aleksandar Bojchevski, and Stephan Günnemann. "Pitfalls of graph neural network evaluation." arXiv preprint arXiv:1811.05868 (2018).
>
> [3] Platonov, Oleg, Denis Kuznedelev, Michael Diskin, Artem Babenko, and Liudmila Prokhorenkova. "A critical look at the evaluation of GNNs under heterophily: Are we really making progress?." arXiv preprint arXiv:2302.11640 (2023).

---

### Official Review · Reviewer_Din7 · 2024-11-08

**Soundness:** 3
**Presentation:** 3
**Contribution:** 3
**Rating:** 6
**Confidence:** 3

**Summary:**

The paper focuses on a semi-supervised method for a traditional graph clustering task. It studies generative models to formulate the semi-supervised graph clustering for attributed graphs with cluster structure. The common framework of generative models analyzed in the paper covers graph neural networks, graph autoencoders, and proposed stochastic block models. The paper conducts experiments on three common attributed graph datasets and shows the superior performance of semi-supervised objectives.

**Strengths:**

1. The paper proposes a generative model framework for semi-supervised graph clustering.
2. The proposed method can utilize different neural network architectures including pure transformers and MLPs.
3. The paper gives a new perspective to a unified framework of generative models for node clustering and classification tasks.

**Weaknesses:**

1. The paper studies a traditional semi-supervised graph clustering task.
2. Only three small attributed graph datasets are used in the experiments. Larger graph datasets might be also used to show the scalability of the method.
3. The paper only experiment on one kind of GNNs (i.e., GCN2). More GNN models can be considered in the experiments.

**Questions:**

1. Could the idea of generative models be used for other graph machine learning tasks?
2. What is the time complexity of the model?
3. Do the authors try recent GNN models and datasets?

---

> ### Author Response · Authors · 2024-11-29
>
> Thank you very much for your thorough review and your recommendation to accept our paper. We are greatly encouraged by your positive feedback on the soundness, presentation, and contributions of our work.
>
> We sincerely appreciate the time you took to provide constructive feedback. To address the weaknesses and questions raised in your review, we have made the following clarifications and improvements to the paper:
>
> ---
>
> ### **Weakness 1**:
> We clarify that we focus on semi-supervised graph clustering on real-world attributed graphs -- a challenging and modern problem where the interplay between node attributes and graph structure in determining cluster formation is not well understood.
>
> We acknowledge that the discussion on how our task relates to semi-supervised graph clustering in general could be improved. To address this, we have added a more detailed discussion in the revised manuscript, highlighting how our method is positioned in this broader context.
>
> ### **Weakness 2**:
> Thank you for your suggestion regarding larger datasets. In response, we have significantly revised the experimental section and now include evaluations on larger datasets sourced from [1] and [2].
>
> ### **Weakness 3**:
> We appreciate your observation regarding the use of a single GNN architecture. To provide a more comprehensive evaluation, we have expanded our experiments to include GCN [3] and GraphSAGE [4] -- GNN baselines that continue to be used as standard benchmarks such as in OGB [5].
>
> ---
>
> ### **Question 1**:
> Thank you for your thought-provoking question. We intentionally designed our framework to be amenable to deep graph generation, which we hope will inspire future work in this direction.
>
> Additionally, our framework can be extended to edge- and graph-level tasks, such as link prediction, graph classification, and regression. We have expanded the discussion of these extensions in the “Extending the Framework” section.
>
> ### **Question 2**:
> Thank you for highlighting the importance of scalability. In response, we have added a dedicated section discussing the scalability of our method. This section elaborates on how the scalability depends on the choice of neural network modules and graph clustering techniques employed.
>
> ### **Question 3**:
> We have now incorporated significantly newer datasets from [1] and [2] into our experiments. Additionally, we chose GNN baselines that continue to be used as standard benchmarks in recent literature [5].
>
> ---
>
> We hope these clarifications and revisions address your concerns and further enhance the quality of our paper. Once again, we are deeply grateful for your thoughtful feedback, which has been invaluable in improving our work. Please do not hesitate to share any additional comments or suggestions.
>
> ---
>
> [1] Shchur, Oleksandr, Maximilian Mumme, Aleksandar Bojchevski, and Stephan Günnemann. "Pitfalls of graph neural network evaluation." arXiv preprint arXiv:1811.05868 (2018).
>
> [2] Platonov, Oleg, Denis Kuznedelev, Michael Diskin, Artem Babenko, and Liudmila Prokhorenkova. "A critical look at the evaluation of GNNs under heterophily: Are we really making progress?." arXiv preprint arXiv:2302.11640 (2023).
>
> [3] Kipf, Thomas N., and Max Welling. "Semi-supervised classification with graph convolutional networks." arXiv preprint arXiv:1609.02907 (2016).
>
> [4] Hamilton, Will, Zhitao Ying, and Jure Leskovec. "Inductive representation learning on large graphs." Advances in neural information processing systems 30 (2017).
>
> [5] Hu, Weihua, Matthias Fey, Marinka Zitnik, Yuxiao Dong, Hongyu Ren, Bowen Liu, Michele Catasta, and Jure Leskovec. "Open graph benchmark: Datasets for machine learning on graphs." Advances in neural information processing systems 33 (2020): 22118-22133.

---

### Official Review · Reviewer_7cPT · 2024-11-11

**Soundness:** 2
**Presentation:** 1
**Contribution:** 2
**Rating:** 3
**Confidence:** 5

**Summary:**

## Summary

The paper proposes a combination of unsupervised clustering objectives with supervised node classification tasks via generative models to improve the performance of node classification in sparse labeled settings. The key insight from the paper is that clustering objectives can incorporate node attributes for a more holistic clustering and the authors achieve this empirically by proposing a positional-encoding free alternative using transformers and MLPs. Experimental results are provided on three real world graphs to show that semi-supervised clustering objectives outperform purely supervised approaches in sparse label regime.

**Strengths:**

## Strengths

1. Some of the insights into the nature of unsupervised clustering w.r.t the labels of the nodes are useful for the graph learning community for building better methods for these tasks.

2. Viewing node clustering and classification approaches into a single view of generative models is interesting.

**Weaknesses:**

## Weaknesses

1. The motivation of the paper and the setup of the authors’ contribution is not standard where they look at the problem of semi-supervised classification using clustering objectives by incorporating the role of both node features and adjacency matrix via generative models. The assumption that node attributes and/or a combination of nodes attributes and graph adjacency matrices haven’t been explored before (L066, L411-413) is also incorrect, since there are several lines of work in the graph learning literature that precisely solve this problem - MVGRL [1], BGRL [2], S3GC[3], GRACE [4] to name a few. Even the features learnt from Deep Graph Infomax [5] style learning methods have been shown to be fairly effective for both clustering as well as the classification task. While I understand that the paper attempts to provide an alternative to graph convolutions, the merit for this setup and evaluation would have been justified if the authors compared their method to several of these other works and their objectives, demonstrating that these methods perform worse for classification, however the mention of these papers or these comparisons are missing from both the motivation and the experiments.

2. The datasets used for experimentation - Cora, Citeseer and Pubmed are of extremely small scale with a maximum number of nodes as 20,000, making the experimental evaluation extremely weak. It has been shown in several works in the graph learning literature [6] that such small datasets are unreliable for measurements of any reasonable accuracy improvements. When there are several reliable benchmarks datasets available in the graph learning community, such as OGB [7], I would have expected the authors to use these datasets for a reliable and convincing evaluation.

3. In the current form, the paper is hard to read and there are several issues with the presentation and organization of different sections of the paper which need re-writing for clarity and coherence.

-  a. The introduction section does not adequately set up the motivation for incorporating a clustering objective into the semi-supervised classification setup, and why the comparison is only with fully supervised methods. This needs to be supported with more citations, prior work, and concrete hypotheses which will be verified through a solid evaluation.

- b. Section 3.1 from L261 - L300 is confusing and does not clearly enumerate the proposed methodology in the paper.

- c. Section 4 does not include any details of the experimental setup, the datasets, the metric used for measurements, the baselines used for comparison, or even a crisp conclusion that is drawn from the experimental results highlighting the appropriate results. While it is understandable that the detailed description about these can be deferred to the Appendix, it is expected that this section would have at least enough details to be able to interpret the numbers and experimental results in Table 1, which the paper fails to do so in the current format. The main table of results is on the last page of the paper, after the conclusions and future work section, which needs some re-organization.

- d. The more experimental results in Appendix D merely enumerate the results in Tables from page 24 - 59, without any conclusion or interpretation of the results. The authors should focus on insights from their experimentation that will be valuable to the graph learning community, and detail them in the Appendix, if not in the main paper.

[1] Contrastive multi-view representation learning on graphs, Hassani et. al, ICML 2020

[2] Large-scale representation learning on graphs via bootstrapping, Thakoor et. al, ICLR 2021

[3] S3GC: Scalable Self Supervised Graph Clustering, Fnu Devvrit et. al, NeurIPS 2022


[4] Deep graph contrastive representation learning, Zhu et. al, arxiv:2006.04131 2020


[5] Deep graph Infomax, Velickovic et. al, ICLR 2019

[6] Pitfalls of graph neural network evaluation, Shchur et. al, R2L NeurIPS 2018

[7] Open graph Benchmark: Datasets for Machine Learning on Graphs, Hu et. al, 2021

**Questions:**

## Questions

1. The conclusion from Figure 1 is unclear to me. From the supervised clustering figure (which is the proposed method), the nodes in the cluster seem to be more heterogeneous and mixed as compared to the ground truth labels, which brings into question the effectiveness of the method and the practicality of the MCC metric that is measured.

---

> ### Author Response · Authors · 2024-11-29
>
> Thank you very much for your thoughtful review and valuable feedback.
>
> We are encouraged by your positive remarks and are delighted that you found the framework interesting and potentially useful for the community. We also deeply appreciate your constructive feedback, which has helped us refine and improve our work. Below, we address the weaknesses and questions you raised, highlighting the corresponding modifications made to the paper:
>
> ---
>
> ### **Weakness 1**
>
> We sincerely thank you for pointing us to self-supervised graph representation learning methods for graph clustering and suggesting we discuss their relationship to our method.
>
> 1. **Addressing the relationship with self-supervised methods**:
>
>    - We have added a discussion in the _Other Related Work_ section, contextualizing the provided references and exploring how they relate to our work.
>    - Additionally, we discuss how our framework could be extended to leverage mini-batch self-supervised graph representation learning methods to improve scalability.
>
> 2. **Clarifying our assumptions**:
>
>    - We appreciate the opportunity to clarify that our work does not assume the combination of node attributes and adjacencies is novel in deep graph learning or attributed graph clustering. We explicitly reference previous work and highlight that graph neural networks’ ability to leverage both attributes and adjacencies is a foundational aspect of our framework.
>    - Our focus is on end-to-end graph clustering methods, and we provide a perspective on utilizing these methods (e.g., NOCD, DMoN, Neuromap) within our framework for semi-supervised attributed graph clustering.
>
> 3. **Positioning of our framework**:
>    - We emphasize that our framework is not designed to outperform existing clustering methods directly. Instead, it provides a flexible mechanism to enhance classification by integrating these methods. Our experiments demonstrate this capability rather than benchmarking raw clustering performance.
>
> ---
>
> ### **Weakness 2**
>
> Thank you for pointing out the limitations of the Planetoid datasets and recommending newer datasets.
>
> 1. **Dataset revisions**:
>    - We revised the experimental setup to include the datasets proposed in [1] and [2], encompassing both homophilous and heterophilous graphs.
>    - While we acknowledge the relevance of OGB datasets [3], we did not include them as they are primarily designed for mini-batch graph learning. Our work focuses on full-batch graph learning but we highlight the potential for extending our framework to mini-batch settings in future work.
>
> ---
>
> ### **Weakness 3**
>
> We are grateful for your feedback on the presentation and have made the following improvements:
>
> 1. **Introduction and methodology**:
>
>    - We have revised these sections to more clearly articulate the contributions and the proposed method, ensuring clarity while retaining accessibility.
>
> 2. **Tables and experimental results**:
>    - We reorganized the experiments section, explicitly aligning the results with the four main investigations conducted.
>    - Detailed descriptions of experiments and results have been added to the main text for better contextual understanding.
>
> ---
>
> ### **Question 1**
>
> Thank you for identifying the error in our visualizations and MCC values. Upon investigation, we found this was caused by a bug in the code used to compile and visualize results. This has been corrected, and both the visualizations and MCC values now accurately reflect the methods’ performance.
>
> ---
>
> We hope these clarifications and revisions address your concerns and improve the overall quality of the paper. Once again, thank you for your insightful comments and for helping us refine our work. Please do not hesitate to share any additional comments or suggestions.
>
> ---
>
> [1] Shchur, Oleksandr, Maximilian Mumme, Aleksandar Bojchevski, and Stephan Günnemann. "Pitfalls of graph neural network evaluation." arXiv preprint arXiv:1811.05868 (2018).
>
> [2] Platonov, Oleg, Denis Kuznedelev, Michael Diskin, Artem Babenko, and Liudmila Prokhorenkova. "A critical look at the evaluation of GNNs under heterophily: Are we really making progress?." arXiv preprint arXiv:2302.11640 (2023).
>
> [3] Hu, Weihua, Matthias Fey, Marinka Zitnik, Yuxiao Dong, Hongyu Ren, Bowen Liu, Michele Catasta, and Jure Leskovec. "Open graph benchmark: Datasets for machine learning on graphs." Advances in neural information processing systems 33 (2020): 22118-22133.

---

### Author Response · Authors · 2024-11-28

We would like to sincerely thank all reviewers for their reviews.

Your positive feedback has been very encouraging and the constructive criticism has been invaluable in helping us improve the quality of our work.

Over the past two weeks, we have carefully considered all your suggestions and have worked hard to revise the manuscript to the best of our ability. We have uploaded a new version of the paper that we believe addresses all of the concerns raised.

At the moment, we are in the process of finalizing detailed responses to each of your comments. We kindly ask for your understanding as we complete this step.

Thank you once again for your time and thoughtful input. We greatly appreciate your support in improving this work.

---

### Author Response · Authors · 2024-11-29

We would like to sincerely thank all reviewers once again for your thoughtful reviews and for your understanding and patience as we worked to finalize our responses. We deeply appreciate the time and effort you dedicated to providing constructive feedback and valuable suggestions, all of which have significantly contributed to improving our work.

We have now carefully addressed each of your comments, and we hope our responses meet your expectations. If there is anything we may have overlooked or if further clarification is needed, please do not hesitate to let us know. We look forward to any additional feedback you may have and truly value your insights.

Thank you once again for your time and effort.

---

### Meta-Review · Area_Chair_HBEX · 2024-12-10

**Metareview:**

This paper tends to interpret many graph learning models under the framework of the Bayesian network. However, the contributions are very limited since they lack rigorous theoretical justification. This paper has some serious issues. Firstly, the organization is bad, and it is very hard to read. Secondly, the evaluations are not convincing since the datasets are small and compared baselines are old. Thirdly, the combination of the supervised and unsupervised losses is direct without remarkable contributions. Therefore, it needs comprehensive improvements.

**Additional Comments On Reviewer Discussion:**

Since the author's feedback is very near the end of the rebuttal (29th November), all reviewers do not respond to the rebuttal. Both of the reviewers providing clear negative ratings possess high confidence. By checking the feedback, I believe most of concerns are not alleviated.

---

### Decision · Program_Chairs · 2025-01-22

Reject